# GIFT-Eval: A Benchmark for General Time Series Forecasting Model Evaluation

## Abstract

Time series foundation models excel in zero-shot forecasting, handling diverse tasks without explicit training. However, the advancement of these models has been hindered by the lack of comprehensive benchmarks. To address this gap, we introduce the **G**eneral T**I**me Series **F**orecas**T**ing Model **Eval**uation, GIFT-Eval, a pioneering benchmark aimed at promoting evaluation across diverse datasets. GIFT-Eval encompasses 23 datasets over 144,000 time series and 177 million data points, spanning seven domains, 10 frequencies, multivariate inputs, and prediction lengths ranging from short to long-term forecasts. To facilitate the effective pretraining and evaluation of foundation models, we also provide a non-leaking pretraining dataset containing approximately 230 billion data points. Additionally, we provide a comprehensive analysis of R: [20] baselines, which includes statistical models, deep learning models, and foundation models. We discuss each model in the context of various benchmark characteristics and offer a qualitative analysis that spans both deep learning and foundation models. We believe the insights from this analysis, along with access to this new standard zero-shot time series forecasting benchmark, will guide future developments in time series foundation models.

## 1 Introduction

The success of foundation model pretraining in language and vision modalities has catalyzed similar progress in time series forecasting. By pretraining on extensive time series datasets, a universal forecasting model can be developed, equipped to address varied downstream forecasting tasks across multiple domains, frequencies, prediction lengths, and number of variates in a zero-shot manner (Woo et al., 2024; Rasul et al., 2023b; Ansari et al., 2024).

A critical aspect of foundation model research is creating a high-quality benchmark that includes large, diverse evaluation data, and preferably non-leaking pretraining data to fairly evaluate models and identify their weaknesses. Research in Natural Language Processing (NLP) has produced key benchmarks such as GLUE, MMLU, *etc.* (Wang et al., 2018; Hendrycks et al., 2020; Srivastava et al., 2022; Chen et al., 2021), which are crucial for developing high-quality models.

Unlike NLP, time series foundation models lack a unified, diverse benchmark for fair comparison. For instance, Woo et al. (2024) introduces LOTSA, which remains the largest collection of time series forecasting pre-training data to date. However, the proposed architecture, `Moirai`, is evaluated on existing benchmarks that are tailored to specific forecasting tasks, such as the LSF (Zhou et al., 2020) dataset for long-term forecast, and the Monash (Godahewa et al., 2021) dataset for univariate forecasts. Both datasets lack sufficient diversity in time series characteristics and forecasting tasks, making it challenging to evaluate the zero-shot capabilities of foundation models in handling broad and generalized forecasting scenarios. This limitation remains in the recent empirical evaluations of other foundation models, including those featured in benchmarks such as TimesFM, Chronos, and Lag-Llama (Das et al., 2023b; Ansari et al., 2024; Rasul et al., 2023b). Furthermore, the inconsistency in pretraining, training, and test splits across various foundation models complicates comparisons and poses a risk of data leakage during in-domain and out-of-domain evaluations. To accelerate the advancement for research on time series model, it is essential to establish a high-quality and diverse benchmark that supports universal forecasting evaluation.

To fill identified gaps, we introduce the **G**eneral T**I**me Series **F**orecas**T**ing Model **Eval**uation (GIFT-Eval), consisting of distinct pretraining and train/test components. The pretraining component

Table 1: Property comparisons of various forecasting benchmarks.

| Property
Benchmark | Data | | | Forecasting Task | | Evaluation | |
|---|---|---|---|---|---|---|---|
| | Freq. Range | Num. of Domain | Pretraining data | Num. of var. | Pred. Len. | Benchmark Methods | Prob. Forecasting |
| Monash (Godahewa et al., 2021) | Secondly ∼ Yearly | 7 | No | Uni | Short | Stat./DL | No |
| TFB (Qiu et al., 2024) | Minutely ∼ Yearly | 6 | No | Uni/Multi | Short | Stat./DL | No |
| LTSF (Zeng et al., 2022) | Minutely ∼ Weekly | 5 | No | Multi | Long | Stat./DL | No |
| BasicTS+ (Shao et al., 2023) | Minutely ∼ Daily | 3 | No | Multi | Short/Long | Stat./DL | No |
| R: [ProbTS (Zhang et al., 2023)] | Minutely ∼ Weekly | 5 | No | Multi | Short/Long | Stat./DL/FM | Yes |
| GIFT-Eval (our work) | Secondly ∼ Yearly | 7 | Yes | Uni/Multi | Short/Long | Stat./DL/FM | Yes |

features 88 datasets including 240 billion data points (Appendix E lists more details on pretraining data). The train/test component features 23 datasets encompassing 144,000 time series and 177 million data points across seven domains and 10 frequencies, with prediction lengths ranging from short to long-term, as well as univariate and multivariate forecasting settings. Prior to our work, Qiu et al. (2024) introduced TFB, a comprehensive dataset for time series forecasting. While it offered diversity in the number of variates and domains, it lacks the evaluation of foundation models and accompanying pretraining data without leakage. Our benchmark fills these gaps and it also includes a broader range of frequencies, a more diverse taxonomy, and a wider span of prediction lengths. We compare GIFT-Eval with other similar benchmarks in Table 1. Our contributions are three-fold:

- **GIFT-Eval:** We introduce a general time series forecasting benchmark that evaluates the zero-shot and universal forecasting capabilities of foundation models. We provide pretraining and train-test components that ensure diversity across multiple characteristics and time series features.

- **Comprehensive Benchmarking:** We design diverse forecasting tasks and evaluate R: [20 baselines] that encompass statistical, deep learning, and foundational models on GIFT-Eval.

- **Detailed Analysis:** We provide insights into the strengths of different models on all aspects of GIFT-Eval including domains, frequencies, prediction lengths, and the number of variates R: [and also among 6 time series features]. We further provide a qualitative analysis showing failure cases of both deep learning and foundation models. We believe these insights will contribute to the future development of foundation models.

## 2 RELATED WORK

**Forecasting Methods** Time series forecasters can be broadly categorized into statistical models, deep learning models, and, more recently, foundation models. Statistical models rely solely on historical data statistics to predict future values. Among these, ARIMA (Box & Pierce, 1970), ETS (Hyndman et al., 2008), Theta (Garza et al., 2022), and VAR (Godahewa et al., 2021) are some of the most widely used ones. With the advent of deep learning technologies, models that apply these techniques to time series forecasting have emerged. Examples include DeepAR (Flunkert et al., 2017), N-BEATS (Oreshkin et al., 2019), and DLinear (Zeng et al., 2022), which utilize pre-transformer architectures. Additionally, transformer-based models such as PatchTST (Nie et al., 2022), Autoformer (Wu et al., 2021), and Crossformer (Zhang & Yan, 2023) have been developed. R: [Another line of important work are probablistic forecasting models. TimeGrad (Rasul et al., 2021) is an autoregressive probabilistic forecasting model utilizing diffusion probabilistic methods, CSDI (Tashiro et al., 2021) is a time series imputation approach that leverages score-based diffusion models conditioned on observed data. Their predecessor GRU NVP (Rasul et al., 2020) on the other hand, models multivariate temporal dynamics in time series forecasting using an autoregressive deep learning model combined with conditioned normalizing flows.] In the last few years, foundation models have been proposed, inspired by their success in other modalities like language and vision. The multivariate Moirai (Woo et al., 2024) forecaster, for instance, is based on an encoder-decoder architecture pretrained on a large dataset. Conversely, Chronos (Ansari et al., 2024) and TimesFM (Das et al., 2023b), R: [and Lag-Llama (Rasul et al., 2023a) are univariate forecasters trained using a decoder-only model. Following these other foundation models have also been proposed Timer (Liu et al., 2024), UniTS(Gao et al., 2024), TTM (Ekambaram et al., 2024), Moment (Goswami et al., 2024), and VisionTS (Chen et al., 2024).] However, the main bottleneck in building and evaluating these foundation models is the lack of a diverse and large benchmark dataset.

**Forecasting Benchmarks** To address this challenge, several efforts have been made to develop extensive time series benchmarks. Woo et al. (2024) introduced LOTSA, which holds the title for

the largest collection of open time series datasets, encompassing 231 billion data points across nine domains. Despite its vast size, the evaluation datasets reuse existing benchmarks from the time series forecasting community and still lack sufficient variety in terms of time series data characteristics and forecasting tasks, which our benchmark aims to augment. Ansari et al. (2024) developed a dataset specifically structured for pretraining, in-domain evaluation, and zero-shot evaluation splits. However, their work is constrained by a limited range of prediction lengths (from 6 to 56), which excludes long-term forecasts, and it restricts the data to univariate forecasting. In contrast, our benchmark encompasses extensive multivariate scenarios and evaluates diverse data across various domains and frequencies. The corpus by Rasul et al. (2023b) presents a diverse array of domains, yet it comprises only univariate datasets totaling 8,000 time series. In contrast, GIFT-Eval dramatically expands this scope with 144,000 time series, enhancing the breadth and depth of the dataset. The benchmark by Qiu et al. (2024) is closely aligned with our work in its aim to curate a diverse and comprehensive set of data. However, it lacks pretraining data, does not evaluate foundation models, and limits the taxonomy to time series features only. Our benchmark not only includes pretraining data (with zero-shot evaluation support) but also provides evaluations for foundation models and offers a taxonomy over both characteristics and time series properties. In summary, our benchmark, GIFT-Eval, builds upon and seeks to address the gaps identified in existing time series forecasting benchmarks. We provide a wider comparison with more benchmarks in Table 1. By providing a more diverse and extensive dataset, we aim to facilitate the development and evaluation of foundation models in time series forecasting.

**Forecasting Tools**   R: [Apart from raw benchmarks, there are also frameworks that provide access to a range of time series datasets. Prophet (Taylor & Letham, 2018), sktime (Löning et al., 2019), TSlib (Wu et al., 2022) are primarily implemented for point forecasting. On the other hand PyTorchTS (Rasul, 2021), GluonTS (Alexandrov et al., 2020b), NeuralForecast[1] are Python packages for probabilistic time series forecasting, with PyTorchTS including more advanced probabilistic models based on deep generative models. ProbTS (Zhang et al., 2023), on the other hand, differs from these by providing insights into point vs. probabilistic forecasts and short- vs. long-term forecasting. It is the most relevant to our work, as it includes both classical and foundation models. However, our benchmark is larger in scale, as ProbTS incorporates only 12 multivariate datasets, with no univariate datasets included.]

## 3   GIFT-EVAL

In this section, we first provide a background on time series forecasting tasks and define key characteristics and features of time series data. We then outline the design decisions behind the development of GIFT-Eval, concluding with an analysis that highlights the key features of its final distribution.

### 3.1   BACKGROUND

We start by defining univariate and multivariate forecasting tasks. After that, we outline the fundamental characteristics of time series datasets which also influenced our data collection process, including domain, frequency, number of variates, and prediction length. We also introduce time series features as part of our data analysis.

#### 3.1.1   TIME SERIES FORECASTING

Time series forecasting is a task of predicting future values over one (univariate) or more (multivariate) variates given historical (most commonly real-valued) data which is sampled at regular time intervals. Suppose $D = (Y^i, Z^i)_{i=1}^{N}$ is a dataset of N time series where $Y^i = (y_1^i, y_2^i, \ldots, y_{T_i}^i) \in \mathbb{R}^{d_{y_i} \times T_i}$ is the target time series with $d_{y_i}$ variates and $T_i$ time steps and $Z^i = (z_1^i, z_2^i, \ldots, z_{T_i}^i) \in \mathbb{R}^{d_{z_i} \times T_i}$ are the set of covaraites with $d_{z_i}$ variates. Then the forecasting task can be modeled as the predictive distribution: $p(Y_{t:t+h}|Y_{t-l:t}, Z_{t-l:t+h})$ where $l$ is the context length, and $h$ is the forecast horizon. Univariate forecasting is a special case where the target series is univariate (*i.e.,* $d_{y_i} = 1$), no

---

[1] https://github.com/Nixtla/neuralforecast

covariates are used (*i.e.,* $Z = \emptyset$), and only the historical values of the target time series are utilized for prediction.

### 3.1.2 TIME SERIES CHARACTERISTICS AND FEATURES

**Characteristics**    Time series datasets possess inherent characteristics that define their structure, and common patterns observed in the data and even choices of modelling techniques. We believe a universal forecasting model should be able to perform irrespective of the domain from which the data is sourced, the granularity at which it was sampled, the length of the forecast horizon and whether it is univariate or multivariate. Thus in our study, we focus on these four characteristics: ($i$) *Domain*, i.e., the field or industry from which the time series data originates, such as finance, healthcare or meteorology. The domain often has a direct effect on the nature of patterns. Another crucial aspect is ($ii$) the *frequency* of observations, indicating the time intervals at which the data points are recorded – such as hourly, daily, monthly or annually. ($iii$) *Prediction length*, or forecast horizon, is the number of future time steps for which predictions are expected. Lastly, ($iv$) the *number of variates* pertains to the dimensionality of the time series data. A *univariate* time series consists of observations of a single variable over time, whereas a *multivariate* time series involves multiple interrelated variables. The number of variates adds complexity to the modelling process, as models need to account for dependencies among multiple time series. By ensuring diversity across these specific characteristics in our benchmark, we aim to encompass a wide array of real-life scenarios.

**Features**    Time series features [2] are statistical properties that capture essential characteristics of the data. We have selected six such properties to analyze our benchmark, grouped into three categories based on the aspects they assess, *c.f.* Appendix B for a detailed explanation and formula of each feature. First, we chose two metrics for assessing the temporal attributes of each time series:($i$) *Trend* refers to the progression of the time series, indicating whether the data shows an overall increase, decrease or stability over time, where higher values indicate stronger trends.($ii$) *Seasonal strength* measures the extent to which regular, repeating patterns occur at specific intervals, such as daily cycles in energy consumption, or annual peaks in finance. The higher the value the more repeating patterns the data exhibits. Second, to assess the forecastability of the time series, we included two metrics:($iii$) *Entropy* measures the "forecastability" of a time series, where low values indicate a high signal-to-noise ratio and high values occur when a series is difficult to forecast. ($iv$) *Hurst exponent* quantifies the long-term memory or persistence in a time series. It indicates whether future values are likely to be influenced by past trends, revert to the mean, or behave randomly, where higher values indicate more persistence. Lastly, to understand the regularity and variability within the time series, we selected two metrics: ($v$) *Stability* assesses the inconsistency of the mean of the time series. In simpler terms, it can be defined as the variance of the means. Note that, unlike what the name suggests, lower values indicate more stable data. Finally, ($vi$) *Lumpiness* quantifies the variability of the variance across different segments of the time series. A high value of lumpiness indicates significant fluctuations in variability, which can be challenging to model due to the inconsistent behavior of the data.

### 3.2 DATASETS

To evaluate and advance universal time series forecasting methods, we have curated a comprehensive collection of datasets. Our compilation spans a wide array of domains with varying frequencies, numbers of variates, and prediction lengths. This diversity is crucial for assessing the generalization capabilities of forecasting models across different types of time series data. In the following sections, we provide detailed descriptions of GIFT-Eval and its unique splits, outlining their sources, and key properties. We also conduct a detailed analysis on the test data to gain a better understanding of the datasets' characteristics and the distribution of time series features.

**Train/Test Data**    We curated the train/test portion of GIFT-Eval with 15 univariate and eight multivariate datasets, spanning seven domains and 10 frequencies, totaling 144,000 time series and 177 million data points. We adhere to established prediction lengths for well-known datasets like M4 (Makridakis et al., 2018). For other datasets, we establish three prediction settings—short, medium, and long—based on frequency and domain, with medium and long settings extending the

---

[2]We use the python implementation of tsfeatures library (Garza et al., 2024) to calculate each feature.

short-term length by factors of 10 and 15, respectively. To support models without multivariate forecasting, our framework flattens multivariate datasets for broader compatibility. Data is stored in the Arrow format (Richardson et al., 2023), ensuring efficient integration into deep learning pipelines. Our benchmark features 97 unique triplets of dataset, frequency, and length, with aggregated results for each model reported across these configurations. The sources of each dataset used in train/test split can be found in Appendix D.

We structure the evaluation component of our benchmark by dedicating the final 10% of each dataset in train/test portion to testing, with the rest allocated for training. A non-overlapping rolling evaluation method is employed, setting a predetermined number of windows in the test split, each equal to the dataset's prediction length. The final window of the training data serves as validation for tuning deep learning model hyperparameters.

**Analysis over test data** We analyze GIFT-Eval to understand the distribution and characteristics of the time series features across various datasets, for more granular information see Appendix D. Figure 1 illustrates the mean values of each time series features across different dataset characteristics. These heatmaps provide valuable insights into how metrics such as trend, seasonal strength, Hurst exponent, stability, and lumpiness vary across datasets with different domains, frequencies, prediction lengths, and numbers of variates. This visualization aids in identifying patterns and potential biases within the data, ensuring that the benchmark captures a diverse range of time series behaviors. It also facilitates fine-grained analysis of model performance across varying dataset characteristics, offering a comprehensive comparison.

**Number of variates:** Figure 1(a) depicts that multivariate data exhibit higher stability and lumpiness values, suggesting more fluctuation in variance across different segments, indicating multivariate time series are more complex and potentially more challenging to model. Conversely, univariate series show stronger seasonal strength, reflecting more pronounced and regular repeating patterns, making them more predictable over certain periods. Note that the metrics on multivariate time series are calculated individually for each variate and aggregated for each dataset.

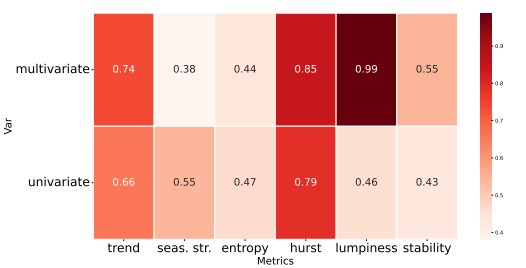

(a) Mean values of TS features across univariate and multivariate datasets.

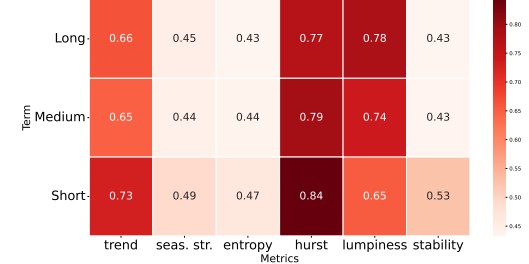

(b) Mean values of TS features across different prediction lengths.

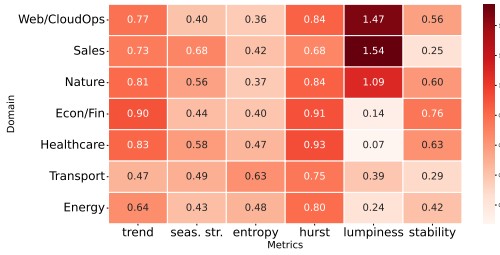

(c) Mean values of TS features across different domains.

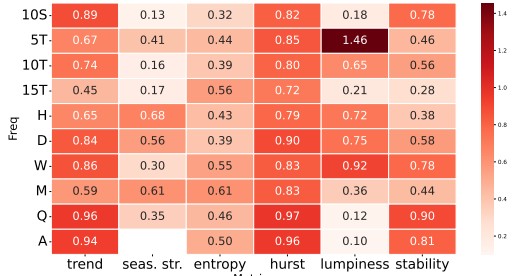

(d) Mean values of TS features across different frequencies.

Figure 1: Heatmaps depicting mean values of six time series features across different characteristics.

**Prediction Length:** Figure 1(b) shows that shorter prediction lengths have higher values for both trend and Hurst metrics. This suggests that time series with shorter forecast horizons exhibit stronger directional movements and greater persistence in their trends, making them potentially easier to

predict. As the prediction length increases, the trend and hurst values tend to decrease significantly which makes forcasting harder. Notably, the stability values decrease from short to long indicating higher steadiness in long term while lumpiness increases suggesting higher fluctuations in different sections of the data.

**Domain:** Figure 1(c) reveals distinct patterns in the metrics. The Web/CloudOps, Sales, and Nature domains exhibit notably high lumpiness, indicating significant fluctuations in variance. This may reflect the volatile nature of online operations, sales dynamics and weather predictions. On the other hand, Transport shows the highest entropy and lowest trend values, indicating less predictability, likely due to the variable nature of transportation data influenced by numerous external factors. The Econ/Fin domain shows the highest trend values, indicating strong directional movements that may imply clear market trends or economic cycles. Finally, healthcare exhibits the highest Hurst and lowest lumpiness values, suggesting persistence in the data, possibly due to consistent patient trends or medical outcomes over time.

**Frequency:** Figure 1(d) lists frequencties from highest to lowest. Data with very short intervals, such as secondly (S) and minutely (T) exhibit the lowest seasonal strengths and poor steadiness, indicative of the erratic and volatile nature typical at these granular levels. There is a noticeable increase in seasonal strength progressing from secondly and minutely data to hourly (H) and daily (D). Finally, yearly (A) and quarterly (Q) data demonstrate the strongest trends and Hurst values, with notably low lumpiness, suggesting increased persistence and high predictability. Notably, the yearly data lack seasonal strength measurements due to the tsfeatures library not providing seasonal strength for excessively long time series, a limitation commonly observed in low-frequency datasets.

**Summary:** These observations confirm that our benchmark is rich and diverse, representing a broad range of real-life time series scenarios. Our dataset encompasses various characteristics—such as differing levels of trend, seasonality, persistence, volatility, and complexity—across multiple domains and frequencies. For instance, we include data from domains with high volatility and significant fluctuations, as well as data exhibiting strong persistence and stability. We also cover a wide spectrum of frequencies, from high-frequency data with erratic patterns to low-frequency data with strong trends and greater predictability. In a similar manner, our metrics show diversity across variate types and prediction lengths. This diversity ensures that models are tested across various temporal behaviors, making our benchmark a robust platform for evaluating the general capabilities of unified models, particularly foundation models in time series forecasting.

**Pretraining Dataset** We have also curated a pretraining dataset aligned with GIFT-Eval that has 71 univariate and 17 multivariate datasets, spanning seven domains and 13 frequencies, totaling 4.5 million time series and 230 billion data points. Notably this collection of data has no leakage issue with the train/test split and can be used to pretrain foundation models that can be fairly evaluated on GIFT-Eval. Further details on pretraining dataset can be found in Appendix E.

## 4 EXPERIMENTS

In this section, we present the experimental evaluation of GIFT-Eval across various models.

**Models** Time series forecasting training and inference may take different forms for different families of models. Statistical models make predictions by directly analyzing patterns in the historical data without a separate training phase. We incorporate five statistical models in our benchmark: `Naive`, `Seasonal Naive` (Hyndman & Athanasopoulos, 2018), `Auto_Arima`, `Auto_ETS`, and `Auto_Theta` (Garza et al., 2022) methods. Deep learning models require training a specific model instance for each dataset. Representing deep learning, we select 8 models: `DeepAR` (Flunkert et al., 2017), `TFT` (Lim et al., 2019), `TiDE` (Das et al., 2023a), `N-BEATS` (Oreshkin et al., 2019), `PatchTST` (Nie et al., 2022), `DLinear` (Zeng et al., 2022), `Crossformer` (Zhang & Yan, 2023) and `iTransformer` (Liu et al., 2023b). To obtain both point and probabilistic forecasts, we either adapt models using gluonts (Alexandrov et al., 2020b) with a small probabilistic head or implement our own modifications. We conduct an extensive hyperparameter search for each deep learning model, see Appendix A for details. We evaluate four foundation models on our benchmark: `TimesFM` (Das et al., 2023b), `Chronos` (Ansari et al., 2024) available in tiny, small, and base sizes, `Moirai` (Woo et al., 2024) available in small, base, and large sizes and, R: [ `Lag-Llama` (Ra-

Table 2: Results on GIFT-Eval aggregated by domain. The best results across each row are **bolded**, while the second best results are underlined.

| Domain | Metric | Nv. | S.Nv. | A.Ar. | A.Th. | D.AR | TFT | TiDE | N-B. | P.TST | iTr. | T.FM | V.TS | Chr.$_S$ | Chr.$_B$ | Chr.$_L$ | Moi.$_S$ | Moi.$_B$ | Moi.$_L$ | Best |
|---|---|---|---|---|---|---|---|---|---|---|---|---|---|---|---|---|---|---|---|---|
| Econ/Fin | MASE | 1.43 | 1.00 | $8.66e^{-1}$ | $9.83e^{-1}$ | 1.54 | 1.03 | 1.51 | $8.61e^{-1}$ | $9.08e^{-1}$ | $9.89e^{-1}$ | $8.24e^{-1}$ | $9.31e^{-1}$ | $7.97e^{-1}$ | $7.83e^{-1}$ | **$7.83e^{-1}$** | 1.04 | $9.27e^{-1}$ | $9.63e^{-1}$ | **Chr.$_L$** |
| | CRPS | 1.17 | 1.00 | $8.21e^{-1}$ | $8.41e^{-1}$ | 1.22 | $8.41e^{-1}$ | 1.08 | $9.67e^{-1}$ | $9.08e^{-1}$ | $8.48e^{-1}$ | **$7.16e^{-1}$** | 1.05 | $7.63e^{-1}$ | $7.51e^{-1}$ | $7.58e^{-1}$ | $7.96e^{-1}$ | $8.16e^{-1}$ | $8.47e^{-1}$ | **T.FM** |
| | Rank | $1.90e^{1}$ | $1.88e^{1}$ | 9.83 | $1.07e^{1}$ | $1.88e^{1}$ | $1.10e^{1}$ | $2.12e^{1}$ | $1.62e^{1}$ | 9.17 | $1.15e^{1}$ | 6.67 | $2.03e^{1}$ | 9.50 | 8.67 | 9.00 | $1.00e^{1}$ | 7.00 | 6.50 | **Moi.$_L$** |
| Energy | MASE | 1.56 | 1.00 | 1.01 | 1.36 | 1.78 | 1.01 | 1.17 | 1.18 | $9.83e^{-1}$ | 1.11 | 1.02 | $9.93e^{-1}$ | $9.47e^{-1}$ | $9.24e^{-1}$ | **$9.19e^{-1}$** | 1.04 | $9.87e^{-1}$ | 1.03 | **Chr.$_L$** |
| | CRPS | 1.53 | 1.00 | $8.33e^{-1}$ | 1.70 | 1.07 | $6.30e^{-1}$ | $7.51e^{-1}$ | $9.35e^{-1}$ | **$6.12e^{-1}$** | $6.95e^{-1}$ | $6.73e^{-1}$ | $7.82e^{-1}$ | $6.48e^{-1}$ | $6.31e^{-1}$ | $6.28e^{-1}$ | $6.68e^{-1}$ | $6.15e^{-1}$ | $6.27e^{-1}$ | **P.TST** |
| | Rank | $2.51e^{1}$ | $2.13e^{1}$ | $1.67e^{1}$ | $2.32e^{1}$ | $2.00e^{1}$ | 9.56 | $1.41e^{1}$ | $2.01e^{1}$ | 7.69 | 9.44 | $1.09e^{1}$ | $1.75e^{1}$ | $1.12e^{1}$ | 9.28 | 9.19 | 9.71 | 6.66 | 7.56 | **Moi.$_B$** |
| Healthcare | MASE | 1.16 | 1.00 | $7.84e^{-1}$ | $9.51e^{-1}$ | $7.65e^{-1}$ | $6.60e^{-1}$ | $8.03e^{-1}$ | $6.91e^{-1}$ | $6.86e^{-1}$ | $7.74e^{-1}$ | $6.98e^{-1}$ | $7.49e^{-1}$ | $6.07e^{-1}$ | $6.45e^{-1}$ | **$5.99e^{-1}$** | $9.51e^{-1}$ | $6.75e^{-1}$ | $6.91e^{-1}$ | **Chr.$_L$** |
| | CRPS | 1.19 | 1.00 | $5.70e^{-1}$ | $8.03e^{-1}$ | $7.23e^{-1}$ | $5.12e^{-1}$ | $5.76e^{-1}$ | $7.13e^{-1}$ | $5.76e^{-1}$ | $6.28e^{-1}$ | $6.52e^{-1}$ | $6.81e^{-1}$ | $4.96e^{-1}$ | $4.85e^{-1}$ | **$4.46e^{-1}$** | $7.72e^{-1}$ | $5.14e^{-1}$ | $5.28e^{-1}$ | **Chr.$_L$** |
| | Rank | $2.26e^{1}$ | $1.98e^{1}$ | 9.60 | $1.52e^{1}$ | $1.26e^{1}$ | 9.40 | $1.74e^{1}$ | $1.72e^{1}$ | $1.06e^{1}$ | $1.26e^{1}$ | 9.60 | $1.60e^{1}$ | 7.00 | 6.00 | 4.60 | $1.63e^{1}$ | 5.80 | 7.20 | **Chr.$_L$** |
| Nature | MASE | $9.62e^{-1}$ | 1.00 | 1.02 | 1.06 | 1.64 | $8.71e^{-1}$ | 1.37 | $9.33e^{-1}$ | $9.16e^{-1}$ | $8.51e^{-1}$ | $8.80e^{-1}$ | $8.60e^{-1}$ | $8.51e^{-1}$ | $8.23e^{-1}$ | $8.13e^{-1}$ | $7.97e^{-1}$ | $7.80e^{-1}$ | **$7.56e^{-1}$** | **Moi.$_L$** |
| | CRPS | 1.33 | 1.00 | $6.58e^{-1}$ | $9.10e^{-1}$ | $5.35e^{-1}$ | $3.48e^{-1}$ | $5.61e^{-1}$ | $5.32e^{-1}$ | $3.47e^{-1}$ | $3.42e^{-1}$ | $3.33e^{-1}$ | $4.06e^{-1}$ | $3.83e^{-1}$ | $3.66e^{-1}$ | $3.64e^{-1}$ | $3.73e^{-1}$ | $3.15e^{-1}$ | **$3.11e^{-1}$** | **Moi.$_L$** |
| | Rank | $2.75e^{1}$ | $2.67e^{1}$ | $2.11e^{1}$ | $2.37e^{1}$ | $1.73e^{1}$ | $1.05e^{1}$ | $1.93e^{1}$ | $2.13e^{1}$ | $1.03e^{1}$ | 8.93 | 8.13 | $1.65e^{1}$ | $1.40e^{1}$ | $1.29e^{1}$ | $1.23e^{1}$ | 9.21 | **5.20** | 5.27 | **Moi.$_B$** |
| Sales | MASE | 1.00 | 1.00 | $8.13e^{-1}$ | $8.73e^{-1}$ | $7.07e^{-1}$ | $7.16e^{-1}$ | $9.81e^{-1}$ | $7.04e^{-1}$ | **$6.90e^{-1}$** | $6.99e^{-1}$ | $7.00e^{-1}$ | $8.17e^{-1}$ | $7.33e^{-1}$ | $7.26e^{-1}$ | $7.24e^{-1}$ | $7.31e^{-1}$ | $6.95e^{-1}$ | $7.10e^{-1}$ | **P.TST** |
| | CRPS | $8.96e^{-1}$ | 1.00 | $4.58e^{-1}$ | $4.80e^{-1}$ | $3.52e^{-1}$ | $3.52e^{-1}$ | $4.84e^{-1}$ | $4.14e^{-1}$ | $3.48e^{-1}$ | $3.48e^{-1}$ | **$3.44e^{-1}$** | $4.92e^{-1}$ | $3.66e^{-1}$ | $3.63e^{-1}$ | $3.62e^{-1}$ | $3.61e^{-1}$ | $3.47e^{-1}$ | $3.63e^{-1}$ | **T.FM** |
| | Rank | $2.80e^{1}$ | $2.80e^{1}$ | $1.98e^{1}$ | $2.10e^{1}$ | 8.75 | $1.10e^{1}$ | $2.05e^{1}$ | $1.45e^{1}$ | 5.00 | 7.00 | **3.00** | $2.15e^{1}$ | $1.22e^{1}$ | $1.05e^{1}$ | $1.00e^{1}$ | $1.00e^{1}$ | 3.25 | 6.75 | **T.FM** |
| Transport | MASE | 1.26 | 1.00 | $9.74e^{-1}$ | 1.08 | $7.45e^{-1}$ | $6.79e^{-1}$ | $7.90e^{-1}$ | $7.31e^{-1}$ | $7.09e^{-1}$ | $7.07e^{-1}$ | $7.41e^{-1}$ | $7.39e^{-1}$ | $7.37e^{-1}$ | $7.12e^{-1}$ | $7.14e^{-1}$ | $7.26e^{-1}$ | $6.34e^{-1}$ | **$6.07e^{-1}$** | **Moi.$_L$** |
| | CRPS | 2.07 | 1.00 | $7.63e^{-1}$ | 1.33 | $4.84e^{-1}$ | $4.43e^{-1}$ | $5.31e^{-1}$ | $5.93e^{-1}$ | $4.61e^{-1}$ | $4.60e^{-1}$ | $5.10e^{-1}$ | $6.01e^{-1}$ | $5.30e^{-1}$ | $5.12e^{-1}$ | $5.12e^{-1}$ | $4.98e^{-1}$ | $4.12e^{-1}$ | **$3.93e^{-1}$** | **Moi.$_L$** |
| | Rank | $2.84e^{1}$ | $2.43e^{1}$ | $2.18e^{1}$ | $2.61e^{1}$ | 8.73 | 6.60 | $1.39e^{1}$ | $1.74e^{1}$ | 8.07 | 7.93 | $1.06e^{1}$ | $1.18e^{1}$ | $1.39e^{1}$ | $1.08e^{1}$ | $1.11e^{1}$ | $1.07e^{1}$ | **5.40** | 5.67 | **Moi.$_B$** |
| Web/CloudOps | MASE | 1.13 | 1.00 | $9.57e^{-1}$ | $5.21e^{-1}$ | $8.50e^{-1}$ | $6.62e^{-1}$ | $6.23e^{-1}$ | $5.43e^{-1}$ | **$4.62e^{-1}$** | $4.88e^{-1}$ | 1.42 | $4.72e^{-1}$ | $6.78e^{-1}$ | $6.76e^{-1}$ | $6.75e^{-1}$ | $7.73e^{-1}$ | $7.62e^{-1}$ | $6.79e^{-1}$ | **P.TST** |
| | CRPS | 1.07 | 1.00 | $9.04e^{-1}$ | $6.08e^{-1}$ | $6.33e^{-1}$ | $5.03e^{-1}$ | $5.68e^{-1}$ | $5.70e^{-1}$ | **$4.37e^{-1}$** | $4.54e^{-1}$ | $7.39e^{-1}$ | $6.03e^{-1}$ | $6.29e^{-1}$ | $6.51e^{-1}$ | $6.47e^{-1}$ | $6.49e^{-1}$ | $6.28e^{-1}$ | $6.19e^{-1}$ | **P.TST** |
| | Rank | $2.19e^{1}$ | $2.18e^{1}$ | $1.99e^{1}$ | $1.66e^{1}$ | $1.48e^{1}$ | 6.95 | $1.22e^{1}$ | $1.29e^{1}$ | **4.75** | 5.85 | $1.84e^{1}$ | $1.35e^{1}$ | $1.29e^{1}$ | $1.45e^{1}$ | $1.48e^{1}$ | $1.35e^{1}$ | $1.22e^{1}$ | $1.13e^{1}$ | **P.TST** |

sul et al., 2023a), `Timer` (Liu et al., 2024), `TTM` (Ekambaram et al., 2024)], `VisionTS` (Chen et al., 2024). These models all provide publicly accessible model parameters for direct use. However, it is important to note that pre-training datasets of `TimesFM`, `Chronos`, and `Moirai` exhibit partial data leakage issues for GIFT-Eval. To keep comparison across models fair, in the main paper we report results with public checkpoints for each model. However, since `Moirai` provides pretraining code, here we pretrain a series of `Moirai` models using GIFT-Eval's pretraining split to demonstrate its utility. We empirically investigate the impact of data leakage in Appendix F.3. Further details on model-specific hyperparameters and tuning can be found in Appendix A.

For readability concerns, we omit results from `Auto_ETS`, `DLinear` and `Crossformer` models in the main tables, however, the reader may refer to Appendix F for results with all models available. For the same space concerns, we use abbreviations to replace each model in the tables. Here is a list of model→abbreviation pairs for reference: `Naive`: **Nv.**, `Seasonal Naive`: **S.Nv.**, `Auto_Arima`: **A.Ar.**, `Auto_Theta`: **A.Th.**, `Auto_ETS`: **A.ETS**, `DeepAR`: **D.AR**, `TFT`: **TFT**, `TiDE`: **TiDE**, `N-BEATS`: **N-B.**, `PatchTST`: **P.TST**, `iTransformer`: **iTr.**, `DLinear`: **DLin.**, `Crossformer`: **C.former**, R: [`Lag-Llama`: **L-Llama**, `Timer`: **Timer**, `TTM`: **TTM**,] `TimesFM`: **T.FM**, `VisionTS`: **V.TS**, `Chronos`: **Chr.**, `Chronos`$_{Small}$: **Chr.$_S$**, `Chronos`$_{Base}$: **Chr.$_B$**, `Chronos`$_{Large}$: **Chr.$_L$**, `Moirai`: **Moi.**, `Moirai`$_{Small}$: **Moi.$_S$**, `Moirai`$_{Base}$: **Moi.$_B$**, `Moirai`$_{Large}$: **Moi.$_L$**.

**Evaluation setting** Performance is assessed using two metrics: the Mean Absolute Scaled Error (MASE) for point forecasts and the Continuous Ranked Probability Score (CRPS) (Gneiting & Raftery, 2007) for probabilistic forecasts (definition of both metrics are in Appendix C), see Appendix F.2 for results with more metrics. To standardize comparison across benchmarks, both metrics are normalized against the `Seasonal Naive` baseline. To avoid skew from any single dataset, we employ a 'Rank' metric that assigns a numerical ranking to each model across all 97 configurations judging by their CRPS score. The average of these ranks is then reported as the final Rank for each model.

## 4.1 RESULTS

We present results across five distinct parts. The first four parts aggregate the results by the key characteristics that guided the development of our benchmark: domain, prediction length, frequency, and number of variates, then conclude the section with aggregation of results across all configurations. For results on all datasets, frequency and prediction length combinations see Tables 22 to 24.

**Domain | Table 2** The results across various domains demonstrate that foundation models consistently outperform both statistical and deep learning models. Notably, the foundation models achieve top performance in most areas, except in the Web/CloudOps domain. As discussed in Section 3.2 Web/CloudOps is one of the domains to exhibit the highest lumpiness. This pattern suggests that foundation models may struggle with time series possessing such characteristics. In contrast, deep learning models like `PatchTST` and `iTransformer` excel in these challenging domains, possibly indicating a shortfall of the training data used for foundation models in these areas. The comparison of different foundation models yields inconsistent conclusions across various domains. We believe

Table 3: Results on GIFT-Eval aggregated by Prediction Length. The best results across each row are **bolded**, while the second best results are underlined.

| Pred. Len. | Metric | Nv. | S.Nv. | A.Ar. | A.Th. | D.AR | TFT | TiDE | N-B. | P.TST | iTr. | T.FM | V.TS | Chr.$_S$ | Chr.$_B$ | Chr.$_L$ | Moi.$_S$ | Moi.$_B$ | Moi.$_L$ | Best |
|---|---|---|---|---|---|---|---|---|---|---|---|---|---|---|---|---|---|---|---|---|
| Long | MASE | 1.40 | 1.00 | $9.85e^{-1}$ | $8.69e^{-1}$ | 1.10 | $5.89e^{-1}$ | $6.55e^{-1}$ | $6.44e^{-1}$ | $5.37e^{-1}$ | $5.66e^{-1}$ | $9.90e^{-1}$ | $\mathbf{5.22e^{-1}}$ | $6.58e^{-1}$ | $6.34e^{-1}$ | $6.32e^{-1}$ | $6.44e^{-1}$ | $6.25e^{-1}$ | $6.04e^{-1}$ | V.TS |
| | CRPS | 1.89 | 1.00 | $8.05e^{-1}$ | 1.40 | $6.28e^{-1}$ | $3.79e^{-1}$ | $4.48e^{-1}$ | $5.65e^{-1}$ | $\mathbf{3.68e^{-1}}$ | $3.91e^{-1}$ | $5.18e^{-1}$ | $4.56e^{-1}$ | $5.22e^{-1}$ | $5.04e^{-1}$ | $5.02e^{-1}$ | $4.45e^{-1}$ | $4.23e^{-1}$ | $4.22e^{-1}$ | P.TST |
| | Rank | $2.72e^1$ | $2.31e^1$ | $2.09e^1$ | $2.43e^1$ | $1.72e^1$ | 6.48 | $1.16e^1$ | $1.61e^1$ | 6.00 | 7.19 | $1.51e^1$ | $1.26e^1$ | $1.56e^1$ | $1.40e^1$ | $1.44e^1$ | 9.29 | 8.24 | 8.19 | P.TST |
| Medium | MASE | 1.46 | 1.00 | 1.02 | 1.17 | 1.33 | $9.49e^{-1}$ | $9.86e^{-1}$ | 1.03 | $8.56e^{-1}$ | $8.67e^{-1}$ | 1.44 | $\mathbf{8.47e^{-1}}$ | 1.04 | 1.04 | 1.03 | 1.03 | 1.03 | $9.72e^{-1}$ | P.TST |
| | CRPS | 1.87 | 1.00 | $8.33e^{-1}$ | 1.53 | $6.40e^{-1}$ | $4.68e^{-1}$ | $5.63e^{-1}$ | $6.78e^{-1}$ | $\mathbf{4.61e^{-1}}$ | $4.70e^{-1}$ | $6.30e^{-1}$ | $5.83e^{-1}$ | $6.25e^{-1}$ | $6.30e^{-1}$ | $6.22e^{-1}$ | $5.55e^{-1}$ | $5.35e^{-1}$ | $5.23e^{-1}$ | P.TST |
| | Rank | $2.62e^1$ | $2.16e^1$ | $1.99e^1$ | $2.43e^1$ | $1.36e^1$ | 5.90 | $1.24e^1$ | $1.70e^1$ | 5.14 | 5.71 | $1.41e^1$ | $1.41e^1$ | $1.50e^1$ | $1.50e^1$ | $1.42e^1$ | $1.00e^1$ | 8.86 | 8.62 | P.TST |
| Short | MASE | 1.14 | 1.00 | $9.35e^{-1}$ | $9.55e^{-1}$ | 1.20 | $8.83e^{-1}$ | 1.14 | $8.62e^{-1}$ | $8.32e^{-1}$ | $8.89e^{-1}$ | $8.23e^{-1}$ | $8.71e^{-1}$ | $7.79e^{-1}$ | $7.68e^{-1}$ | $\mathbf{7.61e^{-1}}$ | $8.97e^{-1}$ | $8.19e^{-1}$ | $8.21e^{-1}$ | Chr.$_L$ |
| | CRPS | 1.09 | 1.00 | $7.35e^{-1}$ | $8.16e^{-1}$ | $7.95e^{-1}$ | $5.92e^{-1}$ | $7.95e^{-1}$ | $7.48e^{-1}$ | $5.71e^{-1}$ | $6.11e^{-1}$ | $5.77e^{-1}$ | $7.51e^{-1}$ | $5.52e^{-1}$ | $5.42e^{-1}$ | $\mathbf{5.38e^{-1}}$ | $6.09e^{-1}$ | $5.48e^{-1}$ | $5.53e^{-1}$ | Chr.$_L$ |
| | Rank | $2.36e^1$ | $2.31e^1$ | $1.64e^1$ | $1.86e^1$ | $1.62e^1$ | $1.09e^1$ | $1.79e^1$ | $1.87e^1$ | 9.27 | $1.02e^1$ | 8.80 | $1.96e^1$ | 9.65 | 8.33 | 8.33 | $1.14e^1$ | 6.18 | 6.93 | Moi.$_B$ |

Table 4: Results on GIFT-Eval aggregated by frequency. The best results across each row are **bolded**, while second best results are underlined.

| Freq. | Metric | Nv. | S.Nv. | A.Ar. | A.Th. | D.AR | TFT | TiDE | N-B. | P.TST | iTr. | T.FM | V.TS | Chr.$_S$ | Chr.$_B$ | Chr.$_L$ | Moi.$_S$ | Moi.$_B$ | Moi.$_L$ | Best |
|---|---|---|---|---|---|---|---|---|---|---|---|---|---|---|---|---|---|---|---|---|
| 10S | MASE | 1.98 | 1.00 | 1.00 | $\mathbf{1.59e^{-1}}$ | $3.76e^{-1}$ | $5.37e^{-1}$ | $3.23e^{-1}$ | $2.71e^{-1}$ | $2.24e^{-1}$ | $2.35e^{-1}$ | $7.87e^{-1}$ | $2.16e^{-1}$ | $5.23e^{-1}$ | $5.23e^{-1}$ | $5.06e^{-1}$ | $7.95e^{-1}$ | $8.41e^{-1}$ | $5.72e^{-1}$ | A.Th. |
| | CRPS | 1.44 | 1.00 | 1.00 | $\mathbf{3.15e^{-1}}$ | $7.54e^{-1}$ | $6.72e^{-1}$ | $7.05e^{-1}$ | $5.98e^{-1}$ | $5.36e^{-1}$ | $5.10e^{-1}$ | 1.30 | $6.91e^{-1}$ | $7.93e^{-1}$ | $8.59e^{-1}$ | $8.18e^{-1}$ | 1.24 | 1.06 | 1.02 | A.Th. |
| | Rank | $1.93e^1$ | $1.13e^1$ | $1.03e^1$ | 1.00 | $1.23e^1$ | 8.83 | $1.12e^1$ | 7.17 | 5.00 | 2.50 | $2.53e^1$ | $1.08e^1$ | $1.12e^1$ | $1.33e^1$ | $1.23e^1$ | $2.26e^1$ | $1.95e^1$ | $1.78e^1$ | A.Th. |
| 5T | MASE | $9.42e^{-1}$ | 1.00 | 1.00 | $9.84e^{-1}$ | 1.40 | $8.36e^{-1}$ | $9.61e^{-1}$ | $8.84e^{-1}$ | $7.87e^{-1}$ | $7.73e^{-1}$ | 2.38 | $8.19e^{-1}$ | $8.72e^{-1}$ | $8.62e^{-1}$ | $8.69e^{-1}$ | $7.39e^{-1}$ | $6.89e^{-1}$ | $\mathbf{6.69e^{-1}}$ | Moi.$_L$ |
| | CRPS | 1.19 | 1.00 | 1.00 | $9.48e^{-1}$ | $7.49e^{-1}$ | $5.36e^{-1}$ | $6.31e^{-1}$ | $6.99e^{-1}$ | $5.22e^{-1}$ | $5.22e^{-1}$ | $6.73e^{-1}$ | $7.02e^{-1}$ | $6.82e^{-1}$ | $6.83e^{-1}$ | $6.87e^{-1}$ | $4.96e^{-1}$ | $4.84e^{-1}$ | $\mathbf{4.61e^{-1}}$ | Moi.$_L$ |
| | Rank | $2.34e^1$ | $2.39e^1$ | $2.24e^1$ | $2.28e^1$ | $1.77e^1$ | 6.58 | $1.33e^1$ | $1.64e^1$ | 6.75 | 7.75 | $1.52e^1$ | $1.63e^1$ | $1.48e^1$ | $1.51e^1$ | $1.58e^1$ | 7.44 | 6.42 | 4.58 | Moi.$_L$ |
| 10T | MASE | 1.28 | 1.00 | 1.00 | 1.62 | 1.55 | $9.42e^{-1}$ | 1.27 | 1.21 | 1.19 | 1.09 | 1.27 | $\mathbf{9.12e^{-1}}$ | 1.20 | 1.09 | 1.07 | 1.00 | 1.15 | 1.13 | V.TS |
| | CRPS | 2.08 | 1.00 | 1.00 | 2.51 | $5.37e^{-1}$ | $\mathbf{3.64e^{-1}}$ | $5.68e^{-1}$ | $6.88e^{-1}$ | $4.34e^{-1}$ | $4.43e^{-1}$ | $4.59e^{-1}$ | $4.42e^{-1}$ | $5.47e^{-1}$ | $4.75e^{-1}$ | $4.71e^{-1}$ | $4.91e^{-1}$ | $5.04e^{-1}$ | $5.14e^{-1}$ | TFT |
| | Rank | $2.67e^1$ | $2.22e^1$ | $2.12e^1$ | $2.80e^1$ | $1.47e^1$ | 5.67 | $1.65e^1$ | $1.82e^1$ | 9.50 | 8.00 | $1.00e^1$ | 9.33 | $1.55e^1$ | $1.05e^1$ | 9.67 | $1.10e^1$ | $1.28e^1$ | $1.30e^1$ | TFT |
| 15T | MASE | 1.52 | 1.00 | $9.78e^{-1}$ | 1.03 | 1.76 | $9.66e^{-1}$ | 1.02 | 1.02 | $\mathbf{8.77e^{-1}}$ | $8.78e^{-1}$ | $9.56e^{-1}$ | $9.05e^{-1}$ | $9.20e^{-1}$ | $8.87e^{-1}$ | $8.85e^{-1}$ | $9.49e^{-1}$ | $9.25e^{-1}$ | $9.77e^{-1}$ | P.TST |
| | CRPS | 2.20 | 1.00 | $9.52e^{-1}$ | 1.51 | 1.26 | $7.08e^{-1}$ | $7.92e^{-1}$ | $9.63e^{-1}$ | $6.55e^{-1}$ | $\mathbf{6.51e^{-1}}$ | $7.68e^{-1}$ | $8.56e^{-1}$ | $7.73e^{-1}$ | $7.49e^{-1}$ | $7.46e^{-1}$ | $7.39e^{-1}$ | $6.91e^{-1}$ | $7.20e^{-1}$ | iTr. |
| | Rank | $2.73e^1$ | $2.03e^1$ | $1.91e^1$ | $2.38e^1$ | $1.97e^1$ | 8.67 | $1.37e^1$ | $2.00e^1$ | 5.00 | 4.67 | $1.07e^1$ | $1.73e^1$ | $1.29e^1$ | $1.08e^1$ | $1.06e^1$ | 9.00 | 6.17 | 9.58 | iTr. |
| H | MASE | 1.46 | 1.00 | 1.02 | 1.28 | 1.31 | $8.25e^{-1}$ | $9.59e^{-1}$ | $8.72e^{-1}$ | $7.74e^{-1}$ | $8.05e^{-1}$ | $8.24e^{-1}$ | $7.70e^{-1}$ | $7.73e^{-1}$ | $\mathbf{7.63e^{-1}}$ | $8.05e^{-1}$ | $8.92e^{-1}$ | $7.78e^{-1}$ | $7.70e^{-1}$ | Chr.$_B$ |
| | CRPS | 1.67 | 1.00 | $7.43e^{-1}$ | 1.57 | $6.23e^{-1}$ | $4.28e^{-1}$ | $5.11e^{-1}$ | $6.00e^{-1}$ | $\mathbf{4.07e^{-1}}$ | $4.24e^{-1}$ | $4.69e^{-1}$ | $5.25e^{-1}$ | $4.68e^{-1}$ | $4.62e^{-1}$ | $4.64e^{-1}$ | $5.13e^{-1}$ | $4.13e^{-1}$ | $4.07e^{-1}$ | P.TST |
| | Rank | $2.75e^1$ | $2.48e^1$ | $2.20e^1$ | $2.66e^1$ | $1.52e^1$ | 8.77 | $1.44e^1$ | $1.85e^1$ | 6.97 | 8.32 | $1.16e^1$ | $1.64e^1$ | $1.18e^1$ | $1.10e^1$ | $1.12e^1$ | $1.13e^1$ | 5.42 | 5.23 | Moi.$_L$ |
| D | MASE | 1.00 | 1.00 | $8.82e^{-1}$ | $9.36e^{-1}$ | $9.06e^{-1}$ | $7.25e^{-1}$ | 1.15 | $7.75e^{-1}$ | $7.49e^{-1}$ | $8.31e^{-1}$ | $7.46e^{-1}$ | $8.22e^{-1}$ | $7.37e^{-1}$ | $\mathbf{7.14e^{-1}}$ | $7.16e^{-1}$ | $7.83e^{-1}$ | $7.47e^{-1}$ | $7.66e^{-1}$ | Chr.$_B$ |
| | CRPS | $7.94e^{-1}$ | 1.00 | $4.69e^{-1}$ | $5.43e^{-1}$ | $4.91e^{-1}$ | $\mathbf{3.70e^{-1}}$ | $6.10e^{-1}$ | $5.24e^{-1}$ | $3.92e^{-1}$ | $4.38e^{-1}$ | $4.13e^{-1}$ | $5.04e^{-1}$ | $3.97e^{-1}$ | $3.78e^{-1}$ | $3.77e^{-1}$ | $3.97e^{-1}$ | $3.86e^{-1}$ | $3.96e^{-1}$ | TFT |
| | Rank | $2.48e^1$ | $2.67e^1$ | $1.45e^1$ | $1.91e^1$ | $1.49e^1$ | 8.87 | $1.82e^1$ | $1.94e^1$ | 9.73 | $1.18e^1$ | 7.47 | $1.97e^1$ | $1.14e^1$ | 9.20 | 9.07 | 9.10 | 7.13 | 8.27 | Moi.$_B$ |
| W | MASE | 1.00 | 1.00 | $9.46e^{-1}$ | 1.03 | 1.46 | $9.21e^{-1}$ | 1.29 | 1.08 | $9.29e^{-1}$ | 1.25 | $8.47e^{-1}$ | $7.62e^{-1}$ | $7.45e^{-1}$ | $7.62e^{-1}$ | $\mathbf{7.37e^{-1}}$ | 1.00 | $9.01e^{-1}$ | $9.31e^{-1}$ | Chr.$_L$ |
| | CRPS | $8.74e^{-1}$ | 1.00 | $7.31e^{-1}$ | $7.87e^{-1}$ | $9.94e^{-1}$ | $7.26e^{-1}$ | $9.56e^{-1}$ | $9.71e^{-1}$ | $6.66e^{-1}$ | $9.56e^{-1}$ | $6.02e^{-1}$ | $9.43e^{-1}$ | $5.36e^{-1}$ | $5.42e^{-1}$ | $\mathbf{5.29e^{-1}}$ | $6.95e^{-1}$ | $6.37e^{-1}$ | $6.34e^{-1}$ | Chr.$_L$ |
| | Rank | $1.81e^1$ | $2.20e^1$ | $1.32e^1$ | $1.60e^1$ | $1.69e^1$ | $1.44e^1$ | $1.70e^1$ | $2.00e^1$ | $1.02e^1$ | $1.62e^1$ | 6.12 | $2.10e^1$ | 6.75 | 6.00 | 5.62 | $1.12e^1$ | 6.88 | 6.88 | Chr.$_L$ |
| M | MASE | 1.20 | 1.00 | $\mathbf{7.59e^{-1}}$ | $9.32e^{-1}$ | 1.22 | $9.01e^{-1}$ | 1.10 | $8.51e^{-1}$ | $8.59e^{-1}$ | $9.07e^{-1}$ | $8.00e^{-1}$ | $9.15e^{-1}$ | $8.27e^{-1}$ | $8.57e^{-1}$ | $8.12e^{-1}$ | 1.04 | $8.07e^{-1}$ | $8.17e^{-1}$ | A.Ar. |
| | CRPS | 1.52 | 1.00 | $7.59e^{-1}$ | $8.73e^{-1}$ | 1.03 | $8.40e^{-1}$ | 1.16 | $9.62e^{-1}$ | $8.32e^{-1}$ | $8.03e^{-1}$ | $\mathbf{7.33e^{-1}}$ | 1.03 | $8.18e^{-1}$ | $8.49e^{-1}$ | $8.07e^{-1}$ | $9.93e^{-1}$ | $7.51e^{-1}$ | $7.75e^{-1}$ | T.FM |
| | Rank | $2.52e^1$ | $1.80e^1$ | 8.60 | $1.16e^1$ | $1.56e^1$ | $1.02e^1$ | $2.00e^1$ | $1.44e^1$ | $1.00e^1$ | 7.40 | 4.80 | $1.90e^1$ | $1.06e^1$ | $1.16e^1$ | $1.04e^1$ | $1.67e^1$ | 4.20 | 7.00 | Moi.$_B$ |
| Q | MASE | $9.25e^{-1}$ | 1.00 | $8.00e^{-1}$ | $7.44e^{-1}$ | $9.00e^{-1}$ | $8.12e^{-1}$ | 1.05 | $7.56e^{-1}$ | $8.25e^{-1}$ | $7.69e^{-1}$ | $8.75e^{-1}$ | $8.50e^{-1}$ | $7.75e^{-1}$ | $7.69e^{-1}$ | $7.69e^{-1}$ | $7.76e^{-1}$ | $7.11e^{-1}$ | $\mathbf{7.11e^{-1}}$ | Moi.$_B$ |
| | CRPS | $9.51e^{-1}$ | 1.00 | $8.23e^{-1}$ | $7.97e^{-1}$ | $8.41e^{-1}$ | $8.37e^{-1}$ | 1.02 | $9.72e^{-1}$ | $8.35e^{-1}$ | $7.97e^{-1}$ | $8.53e^{-1}$ | 1.05 | $8.46e^{-1}$ | $8.40e^{-1}$ | $8.40e^{-1}$ | $7.94e^{-1}$ | $7.40e^{-1}$ | $\mathbf{7.40e^{-1}}$ | Moi.$_B$ |
| | Rank | $1.80e^1$ | $2.00e^1$ | 9.00 | 6.00 | $1.40e^1$ | $1.20e^1$ | $2.10e^1$ | $1.90e^1$ | $1.00e^1$ | 7.00 | $1.60e^1$ | $2.20e^1$ | $1.30e^1$ | 4.50 | 1.00 | $\mathbf{2.00}$ | | | Moi.$_B$ |
| A | MASE | 1.00 | 1.00 | $9.35e^{-1}$ | $7.83e^{-1}$ | $8.56e^{-1}$ | $7.78e^{-1}$ | 1.26 | $7.93e^{-1}$ | $8.29e^{-1}$ | $8.49e^{-1}$ | $8.44e^{-1}$ | $9.65e^{-1}$ | $9.42e^{-1}$ | $9.17e^{-1}$ | $9.17e^{-1}$ | $7.51e^{-1}$ | $7.58e^{-1}$ | $\mathbf{7.49e^{-1}}$ | Moi.$_L$ |
| | CRPS | $9.93e^{-1}$ | 1.00 | $9.42e^{-1}$ | $8.33e^{-1}$ | $8.19e^{-1}$ | $7.97e^{-1}$ | 1.12 | $9.71e^{-1}$ | $8.48e^{-1}$ | $8.48e^{-1}$ | $8.48e^{-1}$ | 1.15 | 1.01 | $9.78e^{-1}$ | $9.78e^{-1}$ | $7.64e^{-1}$ | $7.62e^{-1}$ | $\mathbf{7.57e^{-1}}$ | Moi.$_L$ |
| | Rank | $1.90e^1$ | $2.00e^1$ | $1.40e^1$ | $1.00e^1$ | 8.00 | 6.00 | $2.20e^1$ | $1.60e^1$ | $1.20e^1$ | $1.10e^1$ | $1.30e^1$ | $2.30e^1$ | $2.10e^1$ | $1.70e^1$ | $1.80e^1$ | 3.50 | 2.00 | 1.00 | Moi.$_L$ |

this inconsistency is related to the pre-training datasets utilized by the different foundation models and the varying contribution ratios associated with them.

**Prediction length | Table 3**   The analysis by prediction length reveals marked differences in performance across various forecast horizons. For short-term forecasts, foundation models, particularly the `Moirai` variants, consistently outperform other models, underscoring their robustness in handling immediate trends and fluctuations. As the prediction length extends to medium and long terms, the performance of `TimesFM` and `Chronos` have significantly declined. This is because the decoder-only architecture adopts recursive multi-step forecast strategy, leading to severe accumulation error issue. Meanwhile, deep learning models such as `PatchTST` demonstrate superior performance on medium and long terms forecast. This trend indicates that the fine-tuning of deeplearning models effectively captures longer-term dependencies, which are crucial for accurate predictions over extended periods. Thus despite the progress in foundational time series research, there remains a notable performance gap between deep learning and foundation models for medium to long-term predictions, suggesting an area ripe for further research.

**Frequency | Table 4**   In the analysis of model performance by data frequency, the highest frequency (secondly) is predominantly led by a statistical baseline, `Auto_Theta` model, indicating its strength in rapid, granular trend capture. As the frequency decreases, deep learning models like `iTransformer` and `TFT` begin to assert dominance, particularly for minutely and hourly data granularities. However, from daily to yearly frequencies, foundation models consistently outperform other approaches, securing the best results across these settings. This pattern suggests that foundation models, with their extensive pretraining, are better equipped to leverage the broader patterns and slower dynamics typical of lower frequency data, enhancing their forecasting accuracy. Conversely, higher frequency data, which often contains more noise and less discernible patterns, poses challenges that foundation models are less suited to address, as evidenced by their performance relative to more specialized deep learning and statistical models.

**Number of variates | Table 5**   In multivariate settings, deep learning models, particularly `PatchTST` (best) and `iTransformer` (second best), stand out by delivering the best scores

Table 5: Results on GIFT-Eval aggregated by number of variates. The best results across each row are **bolded**, while the second best results are underlined.

| Num. Var. | Metric | Nv. | S.Nv. | A.Ar. | A.Th. | D.AR | TFT | TiDE | N-B. | P.TST | iTr. | T.FM | V.TS | Chr.S | Chr.B | Chr.L | Moi.S | Moi.B | Moi.L | Best |
|---|---|---|---|---|---|---|---|---|---|---|---|---|---|---|---|---|---|---|---|---|
| Multivariate | MASE | 1.15 | 1.00 | 1.03 | $8.01e^{-1}$ | 1.50 | $8.40e^{-1}$ | 1.01 | $7.82e^{-1}$ | $7.11e^{-1}$ | $7.37e^{-1}$ | 1.17 | **$6.95e^{-1}$** | $8.04e^{-1}$ | $7.94e^{-1}$ | $7.88e^{-1}$ | $8.44e^{-1}$ | $8.31e^{-1}$ | $8.11e^{-1}$ | V.TS |
| | CRPS | 1.26 | 1.00 | $8.37e^{-1}$ | $9.26e^{-1}$ | $8.02e^{-1}$ | $4.95e^{-1}$ | $6.59e^{-1}$ | $6.41e^{-1}$ | **$4.51e^{-1}$** | $4.78e^{-1}$ | $5.82e^{-1}$ | $5.85e^{-1}$ | $5.55e^{-1}$ | $5.55e^{-1}$ | $5.52e^{-1}$ | $5.44e^{-1}$ | $5.15e^{-1}$ | $5.25e^{-1}$ | P.TST |
| | Rank | $2.40e^{1}$ | $2.26e^{1}$ | $1.95e^{1}$ | $2.08e^{1}$ | $1.90e^{1}$ | 8.95 | $1.55e^{1}$ | $1.69e^{1}$ | **6.56** | 7.05 | $1.37e^{1}$ | $1.53e^{1}$ | $1.24e^{1}$ | $1.23e^{1}$ | $1.25e^{1}$ | 9.94 | 8.63 | 8.91 | P.TST |
| Univariate | MASE | 1.36 | 1.00 | $9.12e^{-1}$ | 1.15 | 1.02 | $8.08e^{-1}$ | $9.59e^{-1}$ | $8.92e^{-1}$ | $8.05e^{-1}$ | $8.57e^{-1}$ | $8.29e^{-1}$ | $8.45e^{-1}$ | $7.97e^{-1}$ | $7.80e^{-1}$ | **$7.75e^{-1}$** | $8.95e^{-1}$ | $7.96e^{-1}$ | $7.86e^{-1}$ | Chr.L |
| | CRPS | 1.49 | 1.00 | $7.21e^{-1}$ | 1.16 | $6.62e^{-1}$ | $5.24e^{-1}$ | $6.46e^{-1}$ | $7.30e^{-1}$ | $5.35e^{-1}$ | $5.64e^{-1}$ | $5.69e^{-1}$ | $6.83e^{-1}$ | $5.64e^{-1}$ | $5.47e^{-1}$ | $5.43e^{-1}$ | $5.98e^{-1}$ | $5.16e^{-1}$ | **$5.08e^{-1}$** | Moi.L |
| | Rank | $2.56e^{1}$ | $2.29e^{1}$ | $1.70e^{1}$ | $2.13e^{1}$ | $1.34e^{1}$ | 8.76 | $1.52e^{1}$ | $1.85e^{1}$ | 8.56 | 9.80 | 9.46 | $1.81e^{1}$ | $1.19e^{1}$ | 9.94 | 9.69 | $1.17e^{1}$ | **6.07** | 6.50 | Moi.B |

Table 6: Results on GIFT-Eval aggregated by all results. The best results across each row are **bolded**, while the second best results are underlined.

| Metric | Nv. | S.Nv. | A.Ar. | A.Th. | D.AR | TFT | TiDE | N-B. | P.TST | iTr. | T.FM | V.TS | Chr.S | Chr.B | Chr.L | Moi.S | Moi.B | Moi.L | Best |
|---|---|---|---|---|---|---|---|---|---|---|---|---|---|---|---|---|---|---|---|
| MASE | 1.26 | 1.00 | $9.64e^{-1}$ | $9.78e^{-1}$ | 1.21 | $8.22e^{-1}$ | $9.80e^{-1}$ | $8.42e^{-1}$ | **$7.62e^{-1}$** | $8.02e^{-1}$ | $9.67e^{-1}$ | $7.75e^{-1}$ | $8.00e^{-1}$ | $7.86e^{-1}$ | $7.81e^{-1}$ | $8.74e^{-1}$ | $8.11e^{-1}$ | $7.97e^{-1}$ | P.TST |
| CRPS | 1.38 | 1.00 | $7.70e^{-1}$ | 1.05 | $7.21e^{-1}$ | $5.11e^{-1}$ | $6.52e^{-1}$ | $6.89e^{-1}$ | **$4.96e^{-1}$** | $5.24e^{-1}$ | $5.75e^{-1}$ | $6.38e^{-1}$ | $5.60e^{-1}$ | $5.51e^{-1}$ | $5.47e^{-1}$ | $5.76e^{-1}$ | $5.16e^{-1}$ | $5.15e^{-1}$ | P.TST |
| Rank | $2.49e^{1}$ | $2.28e^{1}$ | $1.81e^{1}$ | $2.11e^{1}$ | $1.59e^{1}$ | 8.85 | $1.53e^{1}$ | $1.78e^{1}$ | 7.67 | 8.58 | $1.13e^{1}$ | $1.69e^{1}$ | $1.21e^{1}$ | $1.10e^{1}$ | $1.09e^{1}$ | $1.10e^{1}$ | **7.21** | 7.57 | Moi.B |

across all evaluated metrics. `Moirai` outperforms other foundation models, as it is the only model that supports multivariate forecasting. On the other hand, in univariate scenarios, foundation models, especially the large variant of `Moirai`, demonstrate superior performance over their deep learning counterparts. This suggests that foundation models, with their broader pretraining on diverse data sets, are particularly adept at extracting and leveraging predictive signals from single streams of data.

**General | Table 6** The comprehensive aggregation of results across the entire benchmark offers insightful performance distinctions. `PatchTST` emerges as the most dominant model for MASE and CRPS metrics, with $\text{Moirai}_{\text{Large}}$ securing the first place within the Rank metric. We also present the number of times each model achieves the best or second best results in Table 7. $\text{Moirai}_{\text{Large}}$ appears most frequently as the best performer, and as the model that appears in top 2 most frequently. The discrepancy between the RANK and MASE or CRPS metrics suggests that certain datasets may disproportionately influence the metric-based results, which is not captured by the ranking-based outcomes. Thus `PatchTST` offers reliable results across diverse datasets, making it a strong generalist. In contrast, $\text{Moirai}_{\text{Large}}$ delivers better performance on particular cases.

Some recent works (Shi et al., 2024a;b; Ansari et al., 2024) have verified the scaling law in time series foundation models (*i.e.*, larger model performs better), however, GIFT-Eval does not consistently support this conclusion.

## 4.2 QUALITATIVE RESULTS / FAILURE CASES

In addition to the quantitative results discussed earlier, we present qualitative analyses by sharing forecasting samples across various datasets using both deep learning and foundation models. For the deep learning models, we selected four representatives: `PatchTST` and `iTransformer`, from recent transformer-based architectures, and `DeepAR` and `N-BEATS`, which are more traditional deep learning approaches. Regarding foundational models, we included `Moirai` to represent encoder-decoder architectures, `Chronos` as a decoder-only model, and `VisionTS` due to its unique method of representing the time series through image modality. By examining how these models perform on different datasets, we aim to provide deeper insights into their forecasting behaviors, strengths, and limitations.

The plots in Figures 2(a) and 2(b) show forecasts by deep learning models on the multivariate *Bizitobs_l2c* dataset (hourly, medium-term) and the univariate *Solar* dataset (ten-minutely, medium-term). In Figure 2(a) the irregular patterns challenge the models, with only `PatchTST` getting close to capturing some of the regular spikes accurately. `DeepAR` and `N-BEATS` perform reasonably but miss key periodic spikes, while `iTransformer`, despite its multivariate capability, oversimplifies the data into a sinusoidal pattern. In Figure 2(b), traditional models handle seasonal data better but

Table 7: Best and second best counts for each model across GIFT-Eval dataset configurations (97) according to the Rank metric. The best results across each row are **bolded**.

| | Moi.L | Moi.B | P.TST | iTr. | C.former | TFT | T.FM | Chr.L | Moi.S | Chr.B | A.Th. | D.AR | A.Ar. | A.ETS | Chr.S | N-B. | TiDE | Nv. | S.Nv. | DLin. | Timer | TTM | L-Llama | V.TS |
|---|---|---|---|---|---|---|---|---|---|---|---|---|---|---|---|---|---|---|---|---|---|---|---|---|
| Best | **16** | 12 | 8 | 7 | 15 | 11 | 8 | 7 | 3 | 2 | 6 | 1 | 1 | 0 | 0 | 0 | 0 | 0 | 0 | 0 | 0 | 0 | 0 | 0 |
| Second Best | 14 | 14 | 13 | 13 | 2 | 3 | 3 | 4 | 7 | 7 | 0 | 5 | 3 | 3 | 3 | 2 | 1 | 0 | 0 | 0 | 0 | 0 | 0 | 0 |
| Total | **30** | 26 | 21 | 20 | 17 | 14 | 11 | 11 | 10 | 9 | 6 | 6 | 4 | 3 | 3 | 2 | 1 | 0 | 0 | 0 | 0 | 0 | 0 | 0 |

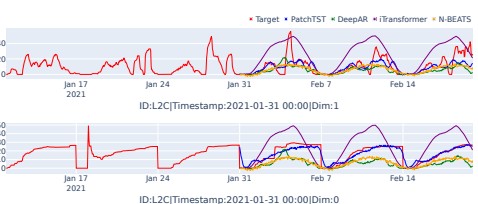

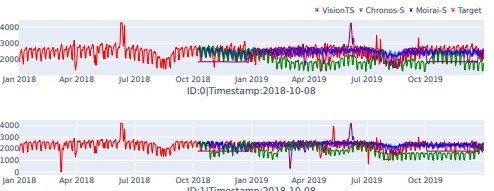

(a) Deep learning forecasts sampled on *Bizitobs_l2c*, hourly dataset with medium prediction length.

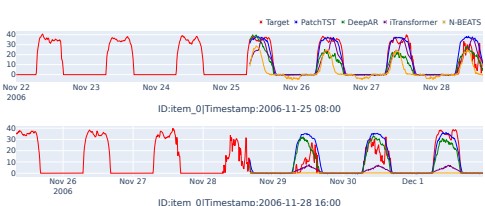

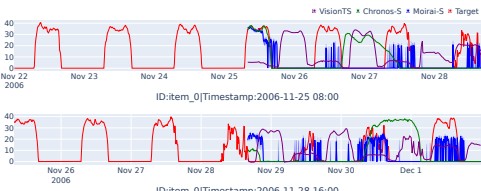

(b) Deep learning forecasts sampled on *Solar* ten–minutely dataset with medium prediction length.

(c) Foundation model forecasts sampled on *M_DENSE* daily dataset with long prediction length.

(d) Foundation model forecasts sampled on *Solar* ten–minutely dataset with medium prediction length.

Figure 2: Qualitative plots showing forecasts from various deep learning and foundation models on several time series forecasting datasets.

still tend to underpredict, with `N-BEATS` producing a flat forecast in the second plot. `PatchTST` consistently outperforms others in both instances, showing robustness with both regular and irregular series, while `iTransformer` continues to underperform.

The plots in Figures 2(c) and 2(d) show forecasts by foundation models on two univariate datasets: *M_DENSE* (daily, long-term) and *Solar* (ten-minutely, medium-term). Figure 2(c) displays varying performance among the foundation models. `Chronos` shows a clear degradation in performance as the prediction horizon extends, struggling to maintain accuracy over time, while `VisionTS` captures spikes but misaligns them. `Moirai` offers smoother, more conservative forecasts, which may result in less sensitivity to extreme events but provide more consistent alignment with the general trend. In Figure 2(d) `VisionTS` predicts seasonal peaks but with timing shifts. On the other hand, both `Moirai` and `Chronos` struggle to capture the well-spaced regularity of the data, missing key trends altogether. These poor results across all foundation models (see Figure 2(b) vs Figure 2(d)) mirror the quantitative findings in Section 4.1, *i.e.* deep learning models outperform foundation models at higher frequencies. For more qualitative examples see Appendix F.4

## 5 CONCLUSION

We introduce GIFT-Eval, a benchmark designed to evaluate time series forecasting models with diversity across four key characteristics: domain, frequency, number of variates, and prediction length. We ensure additional diversity by verifying six statistical features across temporal attributes, forecastability, and regularity. In addition to the train/test dataset, we also provide a pretraining dataset with no leakage into our evaluation set. With this, we aim to provide the necessary ground for fairly comparing different families of models, including foundation models, across a diverse benchmark. We conduct comprehensive experiments with R: [20 baselines] encompassing statistical, deep learning, and foundation models. Leveraging our detailed taxonomy, we provide insights into each model's strengths relative to different characteristics. We also conduct a qualitative analysis highlighting failure cases in both deep learning and foundation models. GIFT-Eval is a comprehensive benchmark with fine-grained taxonomy that we hope will accelerate the development of new foundation time series forecasting models.

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

## A  EXPERIMENTAL SETUP DETAILS

**Statistical models**  We utilize the statsforecast (Garza et al., 2022) library to implement all five statistical baselines: `Naive`, `Seasonal Naive`, `Auto_Theta`, `Auto_ETS` and `Auto_Arima`. Inference is performed on a CPU server equipped with 96 cores. For each dataset, a time limit of

one day is set for the statistical model to complete its run. For any model that times out we halt it and replace its results with those from the `Seasonal Naive` model as a fallback. Given that some datasets in our benchmark are particularly long, we impose a maximum size constraint on each statistical baseline (set to 1000 with our time constraints), truncating the time series to this maximum size.

Table 8: Hyperparameter search range for deep learning baselines.

| | TiDE | | | | | | |
|---|---|---|---|---|---|---|---|
| **Parameters** | num_layers_encoder | num_layers_decoder | hidden_dim | temporal_hidden_dim | decoder_output_dim | dropout_rate | lr |
| **Search Range** | [1,2] | [1,2] | [256,512,1024] | [64,128] | [8,16,32] | [0.0, 0.5] | [1e-5:1e-1] |
| | **N-BEATS** | | | | | **PatchTST** | |
| **Parameters** | loss_function | hidden_layer_units | share_weights_in_stack | nb_blocks_per_stack | lr | d_model | num_encoder_layers | lr |
| **Search Range** | ["mase", "mape", "smape"] | [256, 512, 1024, 2048] | [True, False] | [3, 4] | [1e-5:1e-1] | [128, 256, 512] | [2, 3, 4] | [1e-5:1e-1] |
| | **iTransformer** | | | **DeepAR** | | | **DLinear** |
| **Parameters** | d_model | num_encoder_layers | lr | hidden_size | num_layers | lr | lr |
| **Search Range** | [128, 256, 512] | [2, 3, 4] | [1e-5:1e-1] | [20,25,...,80] | [1,2,3,4] | [1e-5:1e-1] | [1e-5:1e-1] |
| | **Crossformer** | | | **TFT** | | | |
| **Parameters** | d_model | n_heads | lr | num_heads | hidden_dim | lr | |
| **Search Range** | [64,128,256] | [2,4,8] | [1e-5:5e-3] | [2,4,8] | [16,32,64] | [1e-5:1e-1] | |

**Deep learning models**   For all deeplearning models we either used models readily available in gluonts library (Alexandrov et al., 2020b) or we write our own wrappers. Where feasible we also add a probabilistic forecasting head to the models. Where direct probabilistic outputs are not feasible, we generate probabilistic evaluations by converting point forecasts into sample forecasts using a single sample. To identify the optimal hyperparameters, we conducted a comprehensive search across all 97 runs included in GIFT-Eval. We employed the ray library (Moritz et al., 2017) to parallelize the search on a single GPU and used the optuna (Akiba et al., 2019) library to extend this parallelization across multiple GPU servers. We search for 15 trials for each deep learning model per each of the 97 runs. Table 8 lists the range of parameters we search for each model. On top of the listed parameters for each model, we also search for weight decay on all runs in the range: $[1e-8:1e-2]$, R: [and for context length in range $[1, 2, 4, 8] \times prediction\_length$.] For the `Crossformer` model on the long term setting of *Jena Weather* dataset with both ten–minutely and hourly frequencies, we had to limit the search for `d_model` and `n_heads`, fixing them at 32 and 1, respectively. This adjustment was necessary because the model's attention mechanism operates across multiple variates, leading to an OOM (Out of Memory) error due to the high number of variates present in this dataset.

**Foundation models**   For all foundation models we use their public versions available online and conduct zero-shot evaluation on our benchmark's test-split. Since Moirai (Woo et al., 2024) provides multi-patch size projections and varying context lengths. We adopt the similar approach by defining a frequency-to-patch size mapping as follows:

- Yearly, Quarterly: 8

- Monthly: 8

- Weekly, Daily: 16

- Hourly: 32

- Minute-level: 32

- Second-level: 64

R: [We set context length to 4000]. We used the public available `Moirai` models from the corresponding HuggingFace repos, i.e., $\text{Moirai}_{\text{Small}}$ - https://huggingface.co/Salesforce/moirai-1.1-R-small, $\text{Moirai}_{\text{Base}}$ - https://huggingface.co/Salesforce/moirai-1.1-R-base, $\text{Moirai}_{\text{Large}}$ - https://huggingface.co/Salesforce/moirai-1.1-R-large.

For `Chronos`, we mainly follow their official implementation[3] for evaluation: with the number of samples as 20. The models are loaded from the corresponding HuggingFace repos, e.g., $\text{Chronos}_{\text{Tiny}}$ - https://huggingface.co/amazon/chronos-t5-tiny, $\text{Chronos}_{\text{Small}}$ - https://huggingface.co/amazon/chronos-t5-small, $\text{Chronos}_{\text{Base}}$ - https://huggingface.co/amazon/chronos-t5-base.

---

[3]https://github.com/amazon-science/chronos-forecasting/blob/main/scripts/evaluation/evaluate.py

For `TimesFM`, we follow their official implementation[4] for evaluation. We set the context length for evaluation as 512 as mentioned in their paper since the maximum context length in training is 512. Following their default setting in their example, we keep the input patch length as 32, the output patch length as 128, the number of layers as 20, and the model dimension as 1280. `TimesFM` comes with only one model size, i.e., timesfm-1.0-200m, and we load the model from `https://huggingface.co/google/timesfm-1.0-200m`.

For `VisionTS`, we follow their official implementation[5] for evaluation. We set the context length as 2000, the norm constant as 0.4, the alignment constant as 0.4 according to their default settings. We use their implementation for seasonality detection to generate a candidate list and search an optimal seasonality parameter with the validation data.

**Additional parameters and computational resources.**   All experiments are conducted on eight NVIDIA A100 GPUs. For models that has gone through training the loss function and optimizer are set following their original implementation. Additionally we set the batch size to 128 and, number of batches per epoch to 100, and finally number of epochs to 50.

## B   DETAILS OF TIME SERIES FEATURES

This section gives a detailed view of the time series features we used to analyze the test portion of our data in Section 3.2. We use tsfeatures library (Garza et al., 2024) to calculate each metric. Given the scale of our dataset, we limit each time series history to the most recent 500 data points before computing the respective features. The prediction length remains faithful to the original values specified for the dataset and is not clipped. R: [ Table 9 shows specific time series features of each dataset where we computed specific we classified them based on whether each feature (e.g., trend, seasonality, entropy) was lower or higher than the median value across all datasets].

We also acknowledge that for some overly short time series, `tsfeatures` may output `NaN` (Not a Number) values for certain features—for example, the seasonal strength of some yearly time series data. In such cases, we exclude these `NaN` values during aggregation. Below we provide specific details for each feature used:

**Trend**   Using the STL (Seasonal and Trend decomposition using Loess) method, a time series $x_t$ is decomposed into a trend component $f_t$, multiple seasonal components $s_{i,t}$ for $i = 1, \ldots, M$, and a remainder component $e_t$:

$$x_t = f_t + s_{1,t} + \cdots + s_{M,t} + e_t,$$

where $M$ is the number of seasonal periods. The strength of the trend is quantified by comparing the variance of the remainder component $e_t$ to the combined variance of the trend and remainder components. Specifically, the strength of the trend is defined as:

$$\text{trend} = 1 - \frac{\text{Var}(e_t)}{\text{Var}(f_t + e_t)}.$$

If the calculated value of trend is less than 0, it is set to 0; if it is greater than 1, it is set to 1. This measure indicates the proportion of the variability in the time series that is explained by the trend component, with values closer to 1 signifying a stronger trend.

**Seasonal Strength**   Following the same decomposition above the strength of each seasonal component is quantified by comparing the variance of the remainder $e_t$ to the combined variance of the seasonal component $s_{i,t}$ and the remainder.

For each seasonal component $s_{i,t}$, the strength of seasonality is defined as:

---

[4]`https://github.com/google-research/timesfm/blob/master/experiments/long_horizon_benchmarks/run_eval.py`

[5]`https://github.com/Keytoyze/VisionTS/blob/main/eval_gluonts/run.py`

Table 9: R: [Time Series features classification across all datasets in GIFT-Eval.]

| dataset | frequency | trend | seas. str. | entropy | hurst | lumpiness | stability |
|---|---|---|---|---|---|---|---|
| m4_yearly | A | high | low | high | high | low | high |
| bitbrains_fast_storage | 5T | high | high | low | high | high | low |
| bitbrains_fast_storage | H | low | low | high | low | high | low |
| bitbrains_rnd | 5T | high | high | low | low | low | low |
| bitbrains_rnd | H | high | high | low | low | high | low |
| bizitobs_application | 10S | high | low | low | high | low | high |
| bizitobs_l2c | 5T | high | low | low | low | low | high |
| bizitobs_l2c | H | low | low | high | low | high | high |
| bizitobs_service | 10S | low | low | high | low | low | high |
| car_parts | M | low | low | high | low | high | low |
| covid_deaths | D | high | low | low | high | low | high |
| electricity | 15T | low | high | low | high | low | low |
| electricity | D | high | high | low | high | low | high |
| electricity | H | high | high | low | high | low | low |
| electricity | W | high | high | low | low | low | high |
| ett1 | 15T | low | low | high | low | low | high |
| ett1 | D | low | low | high | low | high | high |
| ett1 | H | low | high | high | low | high | low |
| ett1 | W | low | low | high | low | high | high |
| ett2 | 15T | high | low | low | high | low | high |
| ett2 | D | high | low | high | low | high | high |
| ett2 | H | high | low | low | high | low | high |
| ett2 | W | high | low | high | low | high | high |
| hierarchical_sales | D | high | high | low | low | low | low |
| hierarchical_sales | W | low | low | high | low | high | low |
| hospital | M | low | low | high | low | low | low |
| jena_weather | 10T | high | high | low | low | low | low |
| jena_weather | D | low | low | high | high | high | high |
| jena_weather | H | high | high | low | low | low | low |
| kdd_cup_2018 | D | high | high | low | low | high | low |
| kdd_cup_2018 | H | high | high | low | low | low | low |
| loop_seattle | 5T | low | low | high | low | high | low |
| loop_seattle | D | low | high | high | low | high | low |
| loop_seattle | H | low | high | high | low | high | low |
| m_dense | D | low | high | high | low | high | low |
| m_dense | H | low | high | high | low | low | low |
| m4_daily | D | high | low | low | high | low | high |
| m4_hourly | H | low | high | low | low | low | low |
| m4_monthly | M | low | low | high | high | low | high |
| m4_quarterly | Q | high | low | high | high | low | high |
| m4_weekly | W | high | low | high | high | high | high |
| restaurant | D | high | high | low | low | high | low |
| saugeen | D | high | low | high | low | high | high |
| saugeen | M | low | high | high | low | high | low |
| saugeen | W | low | low | high | low | high | high |
| solar | 10T | low | low | low | low | high | low |
| solar | D | low | low | high | high | high | low |
| solar | H | low | high | low | low | low | low |
| solar | W | low | low | high | high | low | high |
| sz_taxi | 15T | low | low | high | high | high | low |
| sz_taxi | H | low | low | high | high | high | low |
| temperature_rain | D | high | high | low | high | high | high |
| us_births | D | high | high | high | high | low | low |
| us_births | M | high | high | high | high | low | high |
| us_births | W | high | low | low | high | low | high |

$$\text{seasonal\_strength}_i = 1 - \frac{\text{Var}(e_t)}{\text{Var}(s_{i,t} + e_t)}.$$

If the calculated value of seasonal_strength$_i$ is less than 0, it is set to 0; if it is greater than 1, it is set to 1. For non-seasonal time series, seasonal_strength $= 0$. This measure indicates the proportion of the variability in the time series that is explained by the $i$-th seasonal component, with values closer to 1 signifying stronger seasonality for that component.

**Entropy** Entropy is defined as the Shannon entropy of the normalized spectral density estimate $\hat{f}(\lambda)$:

$$\text{Entropy} = -\int_{-\pi}^{\pi} \hat{f}(\lambda) \log \hat{f}(\lambda) \, d\lambda,$$

where $\hat{f}(\lambda)$ is an estimate of the spectral density of the data. A lower spectral entropy indicates a higher signal-to-noise ratio, meaning the time series has more predictable patterns and is easier to forecast. Conversely, a higher spectral entropy suggests that the series is more complex and difficult to predict.

**Hurst Exponent** The *Hurst exponent* (*hurst*) is computed as $0.5$ plus the maximum likelihood estimate of the fractional differencing order $d$ by Haslett & Raftery (1989). The addition of $0.5$ ensures consistency with the traditional Hurst coefficient. The values of the Hurst exponent vary between 0 and 1, with higher values indicating a smoother trend, less volatility, and less roughness.

**Stability** Stability measures the variability of the mean values across all tiles. It is defined as the variance of the means of the tiled windows. If the time series is divided into $N$ tiles and $\bar{x}_i$ represents the mean of the $i$-th tile, then the stability is calculated as:

$$\text{Stability} = \text{Var}\left(\bar{x}_1, \bar{x}_2, \ldots, \bar{x}_N\right).$$

A lower stability indicates that the means are consistent across tiles, suggesting a stable time series. A higher stability implies significant differences in means, indicating potential shifts or trends in the data.

**Lumpiness** Lumpiness assesses the variability of the variances across all tiles. It is defined as the variance of the variances of the tiled windows. Let $s_i^2$ denote the variance of the $i$-th tile. Lumpiness is then computed as:

$$\text{Lumpiness} = \text{Var}\left(s_1^2, s_2^2, \ldots, s_N^2\right).$$

A higher lumpiness suggests that the variability within the tiles differs significantly, indicating that the time series may have periods of high and low volatility. A lower lumpiness means the variances are similar across tiles, pointing to a more homogeneous time series in terms of variability.

## C  EVALUATION METRICS

We use two metrics to evaluate performance of forecasters: Mean Absolute Scaled Error (MASE) for point forecasting ability and Continous Ranked Probability Score (CRPS) for probabilistic forecasting. For both metrics we use gluonts library implementation to calculate final values (Alexandrov et al., 2020a).

**MASE** R: [ MASE (Mean Absolute Scaled Error) is an evaluation metric commonly used in time series analysis to assess forecast accuracy. Unlike metrics such as MAPE, MASE addresses issues of scale dependence and sensitivity to outliers. It is defined as the mean absolute error of the forecast $\hat{Y}_t$,

scaled by the mean absolute error of a naïve benchmark forecast, typically a one-step-ahead lag of the actual values. The formula for MASE is: ]

$$\text{MASE} = \frac{\frac{1}{n}\sum_{t=1}^{n}\left|Y_t - \hat{Y}_t\right|}{\frac{1}{n-1}\sum_{t=2}^{n}|Y_t - Y_{t-1}|},$$

where:

- $Y_t$ is the actual value at time $t$,

- $\hat{Y}_t$ is the forecasted value at time $t$,

- $n$ is the number of observations.

MASE is scale-independent, making it suitable for comparing forecast accuracy across different time series. A MASE value less than 1 indicates that the forecast performs better than the naïve benchmark, while a value greater than 1 indicates worse performance. It is particularly useful in scenarios with varying scales or when evaluating the effectiveness of forecasts relative to a simple baseline.

**CRPS**   The *Continuous Ranked Probability Score* (CRPS) is a metric used in probabilistic forecasting to evaluate the accuracy of predicted cumulative distribution functions (CDFs) against observed values. Given a predicted distribution with CDF $F$ and a ground truth value $y$, the CRPS is defined as:

$$\text{CRPS}(F, y) = \int_0^1 2\Lambda_\alpha(F^{-1}(\alpha), y)\, d\alpha,$$

where the quantile loss $\Lambda_\alpha(q, y)$ is defined as:

$$\Lambda_\alpha(q, y) = (\alpha - \mathbf{1}\{y < q\})(y - q).$$

In practice, computing the CRPS integral can be computationally intensive. To address this, we approximate the CRPS using a discrete sum over a finite set of quantile levels. This approximation, often referred to as the mean weighted quantile loss (Park et al., 2021), is given by:

$$\text{CRPS} \approx \frac{1}{K}\sum_{k=1}^{K} \text{wQL}[\alpha_k],$$

where $K$ is the number of quantile levels, and $\{\alpha_1, \alpha_2, \ldots, \alpha_K\}$ are the selected quantile levels (e.g., $\alpha_k = 0.1k$ for $k = 1, 2, \ldots, 9$ when $K = 9$).

The weighted quantile loss $\text{wQL}[\alpha]$ for each quantile level $\alpha$ is calculated as:

$$\text{wQL}[\alpha] = 2\frac{\sum_t \Lambda_\alpha(\hat{q}_t(\alpha), y_t)}{\sum_t |y_t|},$$

where:

- $\hat{q}_t(\alpha)$ is the predicted $\alpha$-quantile at time step $t$,

- $y_t$ is the actual observed value at time $t$,

- $\Lambda_\alpha(\hat{q}_t(\alpha), y_t)$ is the quantile loss at time $t$ for quantile level $\alpha$.

# D GIFT-EVAL TEST DATASETS

In this section we provide comprehensive list of datasets used in test portion of GIFT-Eval along with original sources, for details regarding the pretraining portion see Appendix E. We utilize 10 open domain sources to curate the benchmark, here we list each one along with its properties in detail. We incorporate Jena Weather[6] dataset following **Autoformer** (Wu et al., 2021). We process BizITObs Application, Service, and L2C[7] following the pipeline in **AutoMixer** (Palaskar et al., 2024).These datasets consist of business and IT observability data, fusing both business KPIs and IT event channels into multivariate time series data. Within the same domain we also process Bitbrains datasets from **Grid Workloads Archive** (Shen et al., 2015). The Restaurant data is borrowed from **Recruit Restaurant Forecasting Competition** (Howard et al., 2017), The task associated with this dataset is to use reservation and visitation data to predict the total number of visitors to a restaurant for future dates. From **Informer** (Zhou et al., 2021) we utilize ETT1 and ETT2 datasets, which denote electricity transformer temperature and serve as an indicator used in the electricity power long-term deployment. Datasets for Transport domain are extracted from **LibCity** (Wang et al., 2023a), which provides a collection of urban time series datasets. We utilize the solar dataset from **LSTNet** (Lai et al., 2017) where the task is to predict solar plant energy outputs. The second dataset for Sales data is by Mancuso et al. (2020). **Monash** (Godahewa et al., 2021) is a large collection of diverse time series datasets across many domains, we choose a subset of these datasets making sure there is no leak from pretrain to test split. Finally, from **UCI ML Archive** (Trindade, 2015) we use the electricity dataset which contains electricity consumption of 370 individual clients. Table 14 lists all datasets, along with their source, frequency, prediction length and number of variates setup and presents various statistics from number of series, to series length, and also number of observations. We use last 10% of each timeseries in the test portion of our data for testing and keep the rest for training.

Tables 10 to 13 present detailed statistics on the number of time series and total observations within each characteristic category of the test benchmark. Specifically, these tables break down the data by domain (Table 11), frequency (Table 12), prediction length (Table 10), and variate count (Table 13), offering a quantitative overview of the dataset's composition.

Table 10: GIFT-Eval Test data statistics aggregated by prediction length.

| Pred. Length | 6 | 8 | 12 | 13 | 14 | 18 | 30 | 48 | 60 | 480 | 600 | 720 | 900 |
|---|---|---|---|---|---|---|---|---|---|---|---|---|---|
| # Series | 22,974 | 24,629 | 3,443 | 359 | 4,227 | 48,000 | 34,398 | 6,194 | 22 | 3,874 | 22 | 3,874 | 22 |
| # Obs | 845,109 | 2,525,512 | 201,042 | 371,579 | 10,023,836 | 11,246,411 | 1,447,848 | 131,125,706 | 194,369 | 129,375,020 | 194,369 | 129,375,020 | 194,369 |

Table 11: GIFT-Eval Test data statistics aggregated by domain.

| Domain | Econ/Fin | Energy | Healthcare | Nature | Sales | Transport | Web/CloudOps | Grand Total |
|---|---|---|---|---|---|---|---|---|
| # Series | 99,974 | 2,036 | 1,036 | 32,618 | 3,717 | 1,341 | 3,524 | 144,246 |
| # Obs | 25,266,415 | 74,119,755 | 129,408 | 3,154,921 | 671,707 | 38,028,955 | 16,610,251 | 157,981,412 |

Table 12: GIFT-Eval Test data statistics aggregated by frequency.

| Frequency | 10S | 10T | 15T | 5T | A | D | H | M | Q | W | Grand Total |
|---|---|---|---|---|---|---|---|---|---|---|---|
| # Series | 22 | 138 | 528 | 2,074 | 22,974 | 38,625 | 3,454 | 51,443 | 24,000 | 988 | 144,246 |
| # Obs | 194,369 | 7,253,424 | 52,498,336 | 49,105,728 | 845,109 | 11,471,684 | 22,268,218 | 11,447,453 | 2,406,108 | 490,983 | 157,981,412 |

Table 13: GIFT-Eval Test data statistics aggregated by number of variates.

| # Variates | 1 | 2 | 7 | 21 | Grand Total |
|---|---|---|---|---|---|
| # Series | 140,711 | 3,522 | 10 | 3 | 144,246 |
| # Obs | 141,133,451 | 16,575,619 | 210,488 | 61,854 | 157,981,412 |

---

[6] https://www.bgc-jena.mpg.de/wetter/
[7] https://github.com/BizITObs/BizITObservabilityData/tree/main

Table 14: R: [Individual statistics of GIFT-Eval benchmark across all datasets.]

| Dataset | Source | Domain | Frequency | # Series | Series Length Avg | Min | Max | # Obs | Target Variables | Past Dynamic | Short-term Pred Length(S) | Windows | Med-term Pred Length(M) | Windows | Long-term Pred Length(L) | Windows |
|---|---|---|---|---|---|---|---|---|---|---|---|---|---|---|---|---|
| Jena Weather | Autoformer (https://www.bgc-jena.mpg.de/wetter/) | Nature | 10T | 1 | 52,704 | 52,704 | 52,704 | 52,704 | 21 | 0 | 48 | 20 | 480 | 11 | 720 | 8 |
| Jena Weather | Autoformer (https://www.bgc-jena.mpg.de/wetter/) | Nature | H | 1 | 8,784 | 8,784 | 8,784 | 8,784 | 21 | 0 | 48 | 19 | 480 | 2 | 720 | 2 |
| Jena Weather | Autoformer (https://www.bgc-jena.mpg.de/wetter/) | Nature | D | 1 | 366 | 366 | 366 | 366 | 21 | 0 | 30 | 2 | | | | |
| BizITObs - Application | Automixer | Web/CloudOps | 10S | 1 | 8,834 | 8,834 | 8,834 | 8,834 | 2 | 35 | 60 | 15 | 600 | 2 | 900 | 1 |
| BizITObs - Service | Automixer | Web/CloudOps | 10S | 21 | 8,835 | 8,835 | 8,835 | 185,535 | 2 | 34 | 60 | 15 | 600 | 2 | 900 | 1 |
| BizITObs - L2C | Automixer | Web/CloudOps | 5T | 1 | 31,968 | 31,968 | 31,968 | 31,968 | 7 | 2 | 48 | 20 | 480 | 7 | 720 | 5 |
| BizITObs - L2C | Automixer | Web/CloudOps | H | 1 | 2,664 | 2,664 | 2,664 | 2,664 | 7 | 2 | 48 | 6 | 480 | 1 | 720 | 1 |
| Bitbrains - Fast Storage | Grid Workloads Archive | Web/CloudOps | 5T | 1,250 | 8,640 | 8,640 | 8,640 | 10,800,000 | 2 | 5 | 48 | 18 | 480 | 2 | 720 | 2 |
| Bitbrains - Fast Storage | Grid Workloads Archive | Web/CloudOps | H | 1,250 | 721 | 721 | 721 | 901,250 | 2 | 5 | 48 | 2 | | | | |
| Bitbrains - rnd | Grid Workloads Archive | Web/CloudOps | 5T | 500 | 8,640 | 8,640 | 8,640 | 4,320,000 | 2 | 5 | 48 | 18 | 480 | 2 | 720 | 2 |
| Bitbrains - rnd | Grid Workloads Archive | Web/CloudOps | H | 500 | 720 | 720 | 720 | 360,000 | 2 | 5 | 48 | 2 | | | | |
| Restaurant | https://www.kaggle.com/competitions/recruit-restaurant-visitor-forecasting/overview | Sales | D | 807 | 358 | 67 | 478 | 289,303 | 1 | 0 | 30 | 1 | | | | |
| ETT1 | Informer | Energy | 15T | 1 | 69,680 | 69,680 | 69,680 | 69,680 | 7 | 0 | 48 | 20 | 480 | 15 | 720 | 10 |
| ETT1 | Informer | Energy | H | 1 | 17,420 | 17,420 | 17,420 | 17,420 | 7 | 0 | 48 | 20 | 480 | 4 | 720 | 3 |
| ETT1 | Informer | Energy | D | 1 | 725 | 725 | 725 | 725 | 7 | 0 | 30 | 3 | | | | |
| ETT1 | Informer | Energy | W-THU | 1 | 103 | 103 | 103 | 103 | 7 | 0 | 8 | 2 | | | | |
| ETT2 | Informer | Energy | 15T | 1 | 69,680 | 69,680 | 69,680 | 69,680 | 7 | 0 | 48 | 20 | 480 | 15 | 720 | 10 |
| ETT2 | Informer | Energy | H | 1 | 17,420 | 17,420 | 17,420 | 17,420 | 7 | 0 | 48 | 20 | 480 | 4 | 720 | 3 |
| ETT2 | Informer | Energy | D | 1 | 725 | 725 | 725 | 725 | 7 | 0 | 30 | 3 | | | | |
| ETT2 | Informer | Energy | W-THU | 1 | 103 | 103 | 103 | 103 | 7 | 0 | 8 | 2 | | | | |
| Loop Seattle | LibCity | Transport | 5T | 323 | 105,120 | 105,120 | 105,120 | 33,953,760 | 1 | 0 | 48 | 20 | 480 | 20 | 720 | 15 |
| Loop Seattle | LibCity | Transport | H | 323 | 8,760 | 8,760 | 8,760 | 2,829,480 | 1 | 0 | 48 | 19 | 480 | 2 | 720 | 2 |
| Loop Seattle | LibCity | Transport | D | 323 | 365 | 365 | 365 | 117,895 | 1 | 0 | 30 | 2 | | | | |
| SZ-Taxi | LibCity | Transport | 15T | 156 | 2,976 | 2,976 | 2,976 | 464,256 | 1 | 0 | 48 | 7 | 480 | 1 | 720 | 1 |
| SZ-Taxi | LibCity | Transport | H | 156 | 744 | 744 | 744 | 116,064 | 1 | 0 | 48 | 2 | | | | |
| M_DENSE | LibCity | Transport | H | 30 | 17,520 | 17,520 | 17,520 | 525,600 | 1 | 7 | 48 | 20 | 480 | 4 | 720 | 3 |
| M_DENSE | LibCity | Transport | D | 30 | 730 | 730 | 730 | 21,900 | 1 | 7 | 30 | 3 | | | | |
| Solar | LSTNet | Energy | 10T | 137 | 52,560 | 52,560 | 52,560 | 7,200,720 | 1 | 0 | 48 | 20 | 480 | 11 | 720 | 8 |
| Solar | LSTNet | Energy | H | 137 | 8,760 | 8,760 | 8,760 | 1,200,120 | 1 | 0 | 48 | 19 | 480 | 2 | 720 | 2 |
| Solar | LSTNet | Energy | D | 137 | 365 | 365 | 365 | 50,005 | 1 | 0 | 30 | 2 | | | | |
| Solar | LSTNet | Energy | W-FRI | 137 | 52 | 52 | 52 | 7,124 | 1 | 0 | 8 | 1 | | | | |
| Hierarchical Sales | Mancuso et al. | Sales | D | 118 | 1,825 | 1,825 | 1,825 | 215,350 | 1 | 0 | 30 | 7 | | | | |
| Hierarchical Sales | Mancuso et al. | Sales | W-WED | 118 | 260 | 260 | 260 | 30,680 | 1 | 0 | 8 | 4 | | | | |
| M4 Yearly | Monash | Econ/Fin | A-DEC | 22,974 | 37 | 19 | 284 | 845,109 | 1 | 0 | 6 | 1 | | | | |
| M4 Quarterly | Monash | Econ/Fin | Q-DEC | 24,000 | 100 | 24 | 874 | 2,406,108 | 1 | 0 | 8 | 1 | | | | |
| M4 Monthly | Monash | Econ/Fin | M | 48,000 | 234 | 60 | 2,812 | 11,246,411 | 1 | 0 | 18 | 1 | | | | |
| M4 Weekly | Monash | Econ/Fin | W-SUN | 359 | 1,035 | 93 | 2,610 | 371,579 | 1 | 0 | 13 | 1 | | | | |
| M4 Daily | Monash | Econ/Fin | D | 4,227 | 2,371 | 107 | 9,933 | 10,023,836 | 1 | 0 | 14 | 1 | | | | |
| M4 Hourly | Monash | Econ/Fin | H | 414 | 902 | 748 | 1,008 | 373,372 | 1 | 0 | 48 | 2 | | | | |
| Hospital | Monash | Healthcare | M | 767 | 84 | 84 | 84 | 64,428 | 1 | 0 | 12 | 1 | | | | |
| COVID Deaths | Monash | Healthcare | D | 266 | 212 | 212 | 212 | 56,392 | 1 | 0 | 30 | 1 | | | | |
| US Births | Monash | Healthcare | D | 1 | 7,305 | 7,305 | 7,305 | 7,305 | 1 | 0 | 30 | 20 | | | | |
| US Births | Monash | Healthcare | W-TUE | 1 | 1,043 | 1,043 | 1,043 | 1,043 | 1 | 0 | 8 | 14 | | | | |
| US Births | Monash | Healthcare | M | 1 | 240 | 240 | 240 | 240 | 1 | 0 | 12 | 2 | | | | |
| Saugeen | Monash | Nature | D | 1 | 23,741 | 23,741 | 23,741 | 23,741 | 1 | 0 | 30 | 20 | | | | |
| Saugeen | Monash | Nature | M | 1 | 780 | 780 | 780 | 780 | 1 | 0 | 12 | 7 | | | | |
| Temperature Rain | Monash | Nature | D | 32,072 | 725 | 725 | 725 | 780 | 1 | 0 | 30 | 3 | | | | |
| KDD Cup 2018 | Monash | Nature | H | 270 | 10,898 | 9,504 | 10,920 | 2,942,364 | 1 | 0 | 48 | 20 | 480 | 2 | 720 | 2 |
| KDD Cup 2018 | Monash | Nature | D | 270 | 455 | 396 | 455 | 122,791 | 1 | 0 | 30 | 2 | | | | |
| Car Parts | Monash | Sales | M | 2,674 | 51 | 51 | 51 | 136,374 | 1 | 0 | 12 | 1 | | | | |
| Electricity | UCI ML Archive | Energy | 15T | 370 | 140,256 | 140,256 | 140,256 | 51,894,720 | 1 | 0 | 48 | 20 | 480 | 20 | 720 | 20 |
| Electricity | UCI ML Archive | Energy | H | 370 | 35,064 | 35,064 | 35,064 | 12,973,680 | 1 | 0 | 48 | 20 | 480 | 8 | 720 | 5 |
| Electricity | UCI ML Archive | Energy | D | 370 | 1,461 | 1,461 | 1,461 | 540,570 | 1 | 0 | 30 | 5 | | | | |
| Electricity | UCI ML Archive | Energy | W-FRI | 370 | 208 | 208 | 208 | 76,960 | 1 | 0 | 8 | 3 | | | | |

# E  GIFT-EVAL PRE-TRAINING DATASETS

The pre-training split of GIFT-Eval is constructed based on LOTSA (Woo et al., 2024), and we excluded certain datasets from it to form part of the evaluation set, making it more diverse and balanced.

Here is a brief discussion on each of the used sources: **BuildingsBench** (Emami et al., 2023) compiled datasets on residential and commercial building energy consumption. **ClimateLearn** (Nguyen et al., 2023) offered time series of various climate-related variables, including temperature, humidity, and multiple pressure levels. **CloudOps TSF** (Woo et al., 2023) introduced large-scale CloudOps time series datasets that capture key variables such as CPU and memory utilization. **GluonTS** (Alexandrov et al., 2020a) provided a variety of datasets commonly used in time series forecasting. **LargeST** (Liu et al., 2023a) sourced from the California Department of Transportation Performance Measurement System (PeMS) to date, which is widely used for traffic forecasting. **LibCity** (Wang et al., 2023a) provided a collection urban spatio-temporal datasets. **SubseasonalClimateUSA** (Mouatadid et al., 2023) provided climate time series data at daily level. **ProEnFo** (Wang et al., 2023b) introduced a range of datasets for load forecasting which include various covariates such as temperature, humidity, and wind speed. **Monash** (Godahewa et al., 2021) is a large collection of diverse time series datasets, the most popular source for building time series foundation models. **LOTSA_Others** (Woo et al., 2024) are complementary datasets collected by LOTSA to enhance the diversity.

The complete list of pre-training datasets and their respective sources, key properties are provided in Table 15.

# F  FINEGRAINED RESULTS

## F.1  R: [RESULTS AGGREGATED BY TIME SERIES FEATURES]

R: [ Table 16 shows results aggregated by time series features. We use the determined features of each dataset as depicted in Table 9 and aggregated across both high and low property of each feature in order get these results. The Table reveals a few key insights about model performance under varying features:]

R: [**Temporal Strength (Trend and Seasonality):** When datasets exhibit low trend or low seasonality, deeplearning models such as `PatchTST`, `TFT`, and `iTransformer` generally perform better compared against foundation models. This may suggest that deep learning models are effective in handling scenarios with less pronounced temporal dynamics. In contrast, at higher levels of trend or seasonality, foundation models like `Moirai`$_{Large}$ often show better performance, potentially indicating their strength in capturing more regular patterns.]

Table 15: Pretraining datasets and their key properties.

| Dataset | Source | Domain | Frequency | # Time Series | # Targets | # Covariates | # Obs. |
|---|---|---|---|---|---|---|---|
| BDG-2 Panther | BuildingsBench (Emami et al., 2023) | Energy | H | 105 | 1 | 0 | 919,800 |
| BDG-2 Fox | BuildingsBench (Emami et al., 2023) | Energy | H | 135 | 1 | 0 | 2,324,568 |
| BDG-2 Rat | BuildingsBench (Emami et al., 2023) | Energy | H | 280 | 1 | 0 | 4,728,288 |
| BDG-2 Bear | BuildingsBench (Emami et al., 2023) | Energy | H | 91 | 1 | 0 | 1,482,312 |
| Low Carbon London | BuildingsBench (Emami et al., 2023) | Energy | H | 713 | 1 | 0 | 9,543,348 |
| SMART | BuildingsBench (Emami et al., 2023) | Energy | H | 5 | 1 | 0 | 95,709 |
| IDEAL | BuildingsBench (Emami et al., 2023) | Energy | H | 219 | 1 | 0 | 1,265,672 |
| Sceaux | BuildingsBench (Emami et al., 2023) | Energy | H | 1 | 1 | 0 | 34,223 |
| Borealis | BuildingsBench (Emami et al., 2023) | Energy | H | 15 | 1 | 0 | 83,269 |
| Buildings900K | BuildingsBench (Emami et al., 2023) | Energy | H | 1,792,328 | 1 | 0 | 15,702,590,000 |
| CMIP6 | ClimateLearn (Nguyen et al., 2023) | Climate | 6H | 1,351,680 | 53 | 0 | 1,973,453,000 |
| ERA5 | ClimateLearn (Nguyen et al., 2023) | Climate | H | 245,760 | 45 | 0 | 2,146,959,000 |
| Azure VM Traces 2017 | CloudOpsTSF (Woo et al., 2023) | CloudOps | 5T | 159,472 | 1 | 2 | 885,522,908 |
| Borg Cluster Data 2011 | CloudOpsTSF (Woo et al., 2023) | CloudOps | 5T | 143,386 | 2 | 5 | 537,552,854 |
| Alibaba Cluster Trace 2018 | CloudOpsTSF (Woo et al., 2023) | CloudOps | 5T | 58,409 | 2 | 6 | 95,192,530 |
| Taxi | GluonTS (Alexandrov et al., 2020a) | Transport | 30T | 67,984 | 1 | 0 | 54,999,060 |
| Uber TLC Daily | GluonTS (Alexandrov et al., 2020a) | Transport | D | 262 | 1 | 0 | 47,087 |
| Uber TLC Hourly | GluonTS (Alexandrov et al., 2020a) | Transport | H | 262 | 1 | 0 | 1,129,444 |
| Wiki-Rolling | GluonTS (Alexandrov et al., 2020a) | Web | D | 47,675 | 1 | 0 | 40,619,100 |
| M5 | GluonTS (Alexandrov et al., 2020a) | Sales | D | 30,490 | 1 | 0 | 58,327,370 |
| LargeST | LargeST (Liu et al., 2023a) | Transport | 5T | 42,333 | 1 | 0 | 4,452,510,528 |
| PEMS03 | LibCity (Wang et al., 2023a) | Transport | 5T | 358 | 1 | 0 | 9,382,464 |
| PEMS04 | LibCity (Wang et al., 2023a) | Transport | 5T | 307 | 3 | 0 | 5,216,544 |
| PEMS07 | LibCity (Wang et al., 2023a) | Transport | 5T | 883 | 1 | 0 | 24,921,792 |
| PEMS08 | LibCity (Wang et al., 2023a) | Transport | 5T | 170 | 3 | 0 | 3,035,520 |
| PEMS Bay | LibCity (Wang et al., 2023a) | Transport | 5T | 325 | 1 | 0 | 16,937,700 |
| Los-Loop | LibCity (Wang et al., 2023a) | Transport | 5T | 207 | 1 | 0 | 7,094,304 |
| Beijing Subway | LibCity (Wang et al., 2023a) | Transport | 30T | 276 | 2 | 11 | 248,400 |
| SHMetro | LibCity (Wang et al., 2023a) | Transport | 15T | 288 | 2 | 0 | 1,934,208 |
| HZMetro | LibCity (Wang et al., 2023a) | Transport | 15T | 80 | 2 | 0 | 146,000 |
| Q-Traffic | LibCity (Wang et al., 2023a) | Transport | 15T | 45,148 | 1 | 0 | 264,386,688 |
| Subseasonal | SubseasonalClimateUSA (Mouatadid et al., 2023) | Climate | D | 862 | 4 | 0 | 14,097,148 |
| Subseasonal Precipitation | SubseasonalClimateUSA (Mouatadid et al., 2023) | Climate | D | 862 | 1 | 0 | 9,760,426 |
| Covid19 Energy | ProEnFo (Wang et al., 2023b) | Energy | H | 1 | 1 | 6 | 31,912 |
| GEF12 | ProEnFo (Wang et al., 2023b) | Energy | H | 20 | 1 | 1 | 788,280 |
| GEF14 | ProEnFo (Wang et al., 2023b) | Energy | H | 1 | 1 | 1 | 17,520 |
| GEF17 | ProEnFo (Wang et al., 2023b) | Energy | H | 8 | 1 | 1 | 140,352 |
| PDB | ProEnFo (Wang et al., 2023b) | Energy | H | 1 | 1 | 1 | 17,520 |
| Spanish | ProEnFo (Wang et al., 2023b) | Energy | H | 1 | 1 | 1 | 35,064 |
| BDG-2 Hog | ProEnFo (Wang et al., 2023b) | Energy | H | 24 | 1 | 5 | 421,056 |
| BDG-2 Bull | ProEnFo (Wang et al., 2023b) | Energy | H | 41 | 1 | 3 | 719,304 |
| BDG-2 Cockatoo | ProEnFo (Wang et al., 2023b) | Energy | H | 1 | 1 | 5 | 17,544 |
| ELF | ProEnFo (Wang et al., 2023b) | Energy | H | 1 | 1 | 0 | 21,792 |
| London Smart Meters | Monash (Godahewa et al., 2021) | Energy | 30T | 5,520 | 1 | 0 | 166,238,880 |
| Wind Farms | Monash (Godahewa et al., 2021) | Energy | T | 337 | 1 | 0 | 172,165,370 |
| Wind Power | Monash (Godahewa et al., 2021) | Energy | 4S | 1 | 1 | 0 | 7,397,147 |
| Solar Power | Monash (Godahewa et al., 2021) | Energy | 4S | 1 | 1 | 0 | 7,397,222 |
| Oikolab Weather | Monash (Godahewa et al., 2021) | Climate | H | 8 | 1 | 0 | 800,456 |
| Elecdemand | Monash (Godahewa et al., 2021) | Energy | 30T | 1 | 1 | 0 | 17,520 |
| Covid Mobility | Monash (Godahewa et al., 2021) | Transport | D | 362 | 1 | 0 | 148,602 |
| Kaggle Web Traffic Weekly | Monash (Godahewa et al., 2021) | Web | W | 145,063 | 1 | 0 | 16,537,182 |
| Extended Web Traffic | Monash (Godahewa et al., 2021) | Web | D | 145,063 | 1 | 0 | 370,926,091 |
| M1 Yearly | Monash (Godahewa et al., 2021) | Econ/Fin | Y | 106 | 1 | 0 | 3,136 |
| M1 Quarterly | Monash (Godahewa et al., 2021) | Econ/Fin | Q | 198 | 1 | 0 | 9,854 |
| M1 Monthly | Monash (Godahewa et al., 2021) | Econ/Fin | M | 617 | 1 | 0 | 44,892 |
| M3 Yearly | Monash (Godahewa et al., 2021) | Econ/Fin | Y | 645 | 1 | 0 | 18,319 |
| M3 Quarterly | Monash (Godahewa et al., 2021) | Econ/Fin | Q | 756 | 1 | 0 | 37,004 |
| M3 Monthly | Monash (Godahewa et al., 2021) | Econ/Fin | M | 1,428 | 1 | 0 | 141,858 |
| M3 Other | Monash (Godahewa et al., 2021) | Econ/Fin | Q | 174 | 1 | 0 | 11,933 |
| NN5 Daily | Monash (Godahewa et al., 2021) | Econ/Fin | D | 111 | 1 | 0 | 81,585 |
| NN5 Weekly | Monash (Godahewa et al., 2021) | Econ/Fin | W | 111 | 1 | 0 | 11,655 |
| Tourism Yearly | Monash (Godahewa et al., 2021) | Econ/Fin | Y | 419 | 1 | 0 | 11,198 |
| Tourism Quarterly | Monash (Godahewa et al., 2021) | Econ/Fin | Q | 427 | 1 | 0 | 39,128 |
| Tourism Monthly | Monash (Godahewa et al., 2021) | Econ/Fin | M | 366 | 1 | 0 | 100,496 |
| CIF 2016 | Monash (Godahewa et al., 2021) | Econ/Fin | M | 72 | 1 | 0 | 6,334 |
| Traffic Weekly | Monash (Godahewa et al., 2021) | Transport | W | 862 | 1 | 0 | 82,752 |
| Traffic Hourly | Monash (Godahewa et al., 2021) | Transport | H | 862 | 1 | 0 | 14,978,112 |
| Australian Electricity Demand | Monash (Godahewa et al., 2021) | Energy | 30T | 5 | 1 | 0 | 1,153,584 |
| Rideshare | Monash (Godahewa et al., 2021) | Transport | H | 2,304 | 1 | 0 | 859,392 |
| Sunspot | Monash (Godahewa et al., 2021) | Nature | D | 1 | 1 | 0 | 73,894 |
| Vehicle Trips | Monash (Godahewa et al., 2021) | Transport | D | 329 | 1 | 0 | 32,512 |
| Weather | Monash (Godahewa et al., 2021) | Climate | D | 3,010 | 1 | 0 | 42,941,700 |
| FRED MD | Monash (Godahewa et al., 2021) | Econ/Fin | M | 107 | 1 | 0 | 76,612 |
| Pedestrian Counts | Monash (Godahewa et al., 2021) | Transport | H | 66 | 1 | 0 | 3,130,762 |
| Bitcoin | Monash (Godahewa et al., 2021) | Econ/Fin | D | 18 | 1 | 0 | 74,824 |
| KDD Cup 2022 | LOTSA_Others (Woo et al., 2024) | Energy | 10T | 134 | 1 | 9 | 4,727,519 |
| GoDaddy | LOTSA_Others (Woo et al., 2024) | Econ/Fin | M | 3,135 | 2 | 0 | 128,535 |
| Favorita Sales | LOTSA_Others (Woo et al., 2024) | Sales | D | 111,840 | 1 | 0 | 139,179,538 |
| Favorita Transactions | LOTSA_Others (Woo et al., 2024) | Sales | D | 54 | 1 | 0 | 84,408 |
| China Air Quality | LOTSA_Others (Woo et al., 2024) | Nature | H | 437 | 6 | 0 | 5,739,234 |
| Beijing Air Quality | LOTSA_Others (Woo et al., 2024) | Nature | H | 12 | 11 | 0 | 420,768 |
| Residential Load Power | LOTSA_Others (Woo et al., 2024) | Energy | T | 271 | 3 | 0 | 145,994,559 |
| Residential PV Power | LOTSA_Others (Woo et al., 2024) | Energy | T | 233 | 3 | 0 | 125,338,950 |
| CDC Fluview ILINet | LOTSA_Others (Woo et al., 2024) | Healthcare | W | 75 | 5 | 0 | 63,903 |
| CDC Fluview WHO NREVSS | LOTSA_Others (Woo et al., 2024) | Healthcare | W | 74 | 4 | 0 | 41,760 |
| Project Tycho | LOTSA_Others (Woo et al., 2024) | Healthcare | W | 1,258 | 1 | 0 | 1,377,707 |

R: [**Entropy and Hurst:** High entropy (suggesting greater forecasting difficulty) and low Hurst (indicating stochasticity thus again high difficulty) appear to favor Transformer-based models (except for `Moirai` variants). On the other hand, for lower entropy or higher Hurst values, foundation models, achieve better results on average, suggesting they may handle simpler temporal structures more effectively.]

Table 16: R: [Results on GIFT-Eval aggregated by time series features. Best results are **bolded**, second best results are underlined.]

| Trend | Metric | Nv. | S.Nv. | A.Ar. | A.Th. | A.ETS | D.AR | TFT | TiDE | N-B. | P.TST | iTr. | DLin. | C.former | Timer | TTM | L-Llama | T.FM | V.TS | Chr.$_S$ | Chr.$_B$ | Chr.$_L$ | Moi.$_S$ | Moi.$_B$ | Moi.$_L$ | Best |
|---|---|---|---|---|---|---|---|---|---|---|---|---|---|---|---|---|---|---|---|---|---|---|---|---|---|---|
| High | MASE | 1.08 | 1.00 | $9.91e^{-1}$ | $8.15e^{-1}$ | $8.75e^{-1}$ | 1.29 | $8.03e^{-1}$ | $9.47e^{-1}$ | $7.60e^{-1}$ | $7.22e^{-1}$ | $7.27e^{-1}$ | $9.72e^{-1}$ | 3.03 | $9.06e^{-1}$ | $8.78e^{-1}$ | 1.03 | $9.86e^{-1}$ | $7.28e^{-1}$ | $7.21e^{-1}$ | $7.11e^{-1}$ | $7.00e^{-1}$ | $8.04e^{-1}$ | $7.39e^{-1}$ | $7.31e^{-1}$ | Chr.$_L$ |
| | CRPS | 1.21 | 1.00 | $7.88e^{-1}$ | $8.23e^{-1}$ | 6.40 | $7.52e^{-1}$ | $4.89e^{-1}$ | $6.53e^{-1}$ | $6.36e^{-1}$ | $4.69e^{-1}$ | $4.86e^{-1}$ | $7.04e^{-1}$ | 1.68 | $7.39e^{-1}$ | $6.69e^{-1}$ | $6.98e^{-1}$ | $5.20e^{-1}$ | $5.94e^{-1}$ | $5.01e^{-1}$ | $4.97e^{-1}$ | $4.88e^{-1}$ | $5.32e^{-1}$ | $4.70e^{-1}$ | $4.75e^{-1}$ | P.TST |
| | Rank | $2.40e^1$ | $2.36e^1$ | $1.92e^1$ | $2.02e^1$ | $2.15e^1$ | $1.82e^1$ | 9.83 | $1.71e^1$ | $1.85e^1$ | 8.27 | 8.60 | $2.08e^1$ | $2.07e^1$ | $2.11e^1$ | $1.89e^1$ | $1.83e^1$ | $1.08e^1$ | $1.73e^1$ | $1.12e^1$ | $1.08e^1$ | $1.05e^1$ | $1.03e^1$ | 6.44 | 6.85 | Moi.$_B$ |
| Low | MASE | 1.46 | 1.00 | $9.38e^{-1}$ | 1.17 | 1.35 | 1.13 | $8.40e^{-1}$ | 1.01 | $9.30e^{-1}$ | $8.02e^{-1}$ | $8.83e^{-1}$ | $9.33e^{-1}$ | 1.77 | 1.14 | 1.07 | 1.18 | $9.48e^{-1}$ | $8.24e^{-1}$ | $8.87e^{-1}$ | $8.68e^{-1}$ | $8.68e^{-1}$ | $9.50e^{-1}$ | $8.89e^{-1}$ | $8.67e^{-1}$ | P.TST |
| | CRPS | 1.57 | 1.00 | $7.53e^{-1}$ | 1.33 | 6.26 | $6.91e^{-1}$ | $5.34e^{-1}$ | $6.50e^{-1}$ | $7.46e^{-1}$ | $5.24e^{-1}$ | $5.65e^{-1}$ | $7.25e^{-1}$ | 1.14 | $9.08e^{-1}$ | $8.46e^{-1}$ | $7.92e^{-1}$ | $6.34e^{-1}$ | $6.84e^{-1}$ | $6.25e^{-1}$ | $6.10e^{-1}$ | $6.11e^{-1}$ | $6.23e^{-1}$ | $5.64e^{-1}$ | $5.58e^{-1}$ | P.TST |
| | Rank | $2.59e^1$ | $2.19e^1$ | $1.71e^1$ | $2.19e^1$ | $2.25e^1$ | $1.36e^1$ | 7.88 | $1.36e^1$ | $1.71e^1$ | 7.08 | 8.55 | $1.76e^1$ | $1.69e^1$ | $2.15e^1$ | $2.08e^1$ | $1.77e^1$ | $1.19e^1$ | $1.65e^1$ | $1.30e^1$ | $1.12e^1$ | $1.13e^1$ | $1.16e^1$ | 7.96 | 8.27 | P.TST |
| **Seas. Str.** | | | | | | | | | | | | | | | | | | | | | | | | | | | |
| High | MASE | 1.36 | 1.00 | $9.83e^{-1}$ | 1.19 | 1.15 | 1.27 | $8.44e^{-1}$ | 1.02 | $9.22e^{-1}$ | $8.30e^{-1}$ | $8.18e^{-1}$ | 1.08 | 3.05 | 1.03 | $9.71e^{-1}$ | 1.03 | 1.08 | $8.16e^{-1}$ | $7.76e^{-1}$ | $7.55e^{-1}$ | $7.49e^{-1}$ | $8.70e^{-1}$ | $7.83e^{-1}$ | $7.64e^{-1}$ | Chr.$_L$ |
| | CRPS | 1.62 | 1.00 | $7.36e^{-1}$ | 1.27 | 9.45 | $5.70e^{-1}$ | $4.31e^{-1}$ | $5.36e^{-1}$ | $6.15e^{-1}$ | $4.26e^{-1}$ | $4.27e^{-1}$ | $6.46e^{-1}$ | $9.82e^{-1}$ | $6.84e^{-1}$ | $5.91e^{-1}$ | $5.75e^{-1}$ | $4.45e^{-1}$ | $5.35e^{-1}$ | $4.49e^{-1}$ | $4.35e^{-1}$ | $4.33e^{-1}$ | $4.90e^{-1}$ | $4.10e^{-1}$ | $3.98e^{-1}$ | Moi.$_L$ |
| | Rank | $2.75e^1$ | $2.47e^1$ | $2.08e^1$ | $2.47e^1$ | $2.35e^1$ | $1.44e^1$ | 9.20 | $1.52e^1$ | $1.94e^1$ | 7.80 | 8.20 | $2.08e^1$ | $1.92e^1$ | 9.91 | $1.82e^1$ | $1.67e^1$ | $1.14e^1$ | $1.70e^1$ | $1.14e^1$ | 9.78 | 9.59 | $1.13e^1$ | 5.72 | 5.13 | Moi.$_L$ |
| Low | MASE | 1.17 | 1.00 | $9.47e^{-1}$ | $8.23e^{-1}$ | 1.03 | 1.15 | $8.02e^{-1}$ | $9.46e^{-1}$ | $7.76e^{-1}$ | $7.05e^{-1}$ | $7.87e^{-1}$ | $8.47e^{-1}$ | 1.80 | 1.01 | $9.66e^{-1}$ | 1.17 | $8.78e^{-1}$ | $7.39e^{-1}$ | $8.23e^{-1}$ | $8.15e^{-1}$ | $8.10e^{-1}$ | $8.77e^{-1}$ | $8.38e^{-1}$ | $8.28e^{-1}$ | P.TST |
| | CRPS | 1.20 | 1.00 | $8.02e^{-1}$ | $8.84e^{-1}$ | 4.40 | $8.91e^{-1}$ | $5.96e^{-1}$ | $7.77e^{-1}$ | $7.64e^{-1}$ | $5.68e^{-1}$ | $6.30e^{-1}$ | $7.83e^{-1}$ | 1.88 | $9.66e^{-1}$ | $9.37e^{-1}$ | $9.38e^{-1}$ | $7.24e^{-1}$ | $7.47e^{-1}$ | $6.81e^{-1}$ | $6.75e^{-1}$ | $6.75e^{-1}$ | $6.61e^{-1}$ | $6.33e^{-1}$ | $6.50e^{-1}$ | P.TST |
| | Rank | $2.26e^1$ | $2.10e^1$ | $1.58e^1$ | $1.78e^1$ | $2.03e^1$ | $1.72e^1$ | 8.53 | $1.55e^1$ | $1.62e^1$ | 7.55 | 8.92 | $1.78e^1$ | $1.84e^1$ | $2.12e^1$ | $2.13e^1$ | $1.92e^1$ | $1.26e^1$ | $1.68e^1$ | $1.27e^1$ | $1.21e^1$ | $1.21e^1$ | $1.07e^1$ | 8.55 | 9.76 | P.TST |
| **Entropy** | | | | | | | | | | | | | | | | | | | | | | | | | | | |
| High | MASE | 1.24 | 1.00 | $9.30e^{-1}$ | 1.01 | 1.18 | 1.12 | $7.92e^{-1}$ | 1.03 | $8.48e^{-1}$ | $7.76e^{-1}$ | $8.66e^{-1}$ | $9.34e^{-1}$ | 1.82 | 1.09 | 1.07 | 1.18 | $8.86e^{-1}$ | $8.20e^{-1}$ | $8.35e^{-1}$ | $8.30e^{-1}$ | $8.24e^{-1}$ | $9.06e^{-1}$ | $8.19e^{-1}$ | $8.10e^{-1}$ | P.TST |
| | CRPS | 1.34 | 1.00 | $7.29e^{-1}$ | 1.02 | 5.75 | $7.03e^{-1}$ | $5.36e^{-1}$ | $6.96e^{-1}$ | $7.08e^{-1}$ | $5.27e^{-1}$ | $5.83e^{-1}$ | $7.41e^{-1}$ | 1.58 | $9.05e^{-1}$ | $8.88e^{-1}$ | $8.05e^{-1}$ | $6.10e^{-1}$ | $7.06e^{-1}$ | $6.08e^{-1}$ | $6.03e^{-1}$ | $5.99e^{-1}$ | $6.21e^{-1}$ | $5.46e^{-1}$ | $5.47e^{-1}$ | P.TST |
| | Rank | $2.43e^1$ | $2.24e^1$ | $1.64e^1$ | $2.02e^1$ | $2.09e^1$ | $1.40e^1$ | 8.06 | $1.49e^1$ | $1.66e^1$ | 7.55 | 8.85 | $1.85e^1$ | $1.82e^1$ | $2.15e^1$ | $2.19e^1$ | $1.80e^1$ | $1.10e^1$ | $1.77e^1$ | $1.26e^1$ | $1.13e^1$ | $1.15e^1$ | $1.12e^1$ | 7.13 | 7.81 | Moi.$_B$ |
| Low | MASE | 1.28 | 1.00 | $9.97e^{-1}$ | $9.50e^{-1}$ | 1.00 | 1.29 | $8.51e^{-1}$ | $9.35e^{-1}$ | $8.36e^{-1}$ | $7.49e^{-1}$ | $7.45e^{-1}$ | $9.70e^{-1}$ | 2.89 | $9.56e^{-1}$ | $8.85e^{-1}$ | 1.03 | 1.05 | $7.35e^{-1}$ | $7.69e^{-1}$ | $7.47e^{-1}$ | $7.42e^{-1}$ | $8.41e^{-1}$ | $8.04e^{-1}$ | $7.84e^{-1}$ | V.TS |
| | CRPS | 1.42 | 1.00 | $8.11e^{-1}$ | 1.08 | 6.92 | $7.38e^{-1}$ | $4.89e^{-1}$ | $6.13e^{-1}$ | $6.72e^{-1}$ | $4.68e^{-1}$ | $4.74e^{-1}$ | $6.90e^{-1}$ | 1.22 | $7.48e^{-1}$ | $6.45e^{-1}$ | $6.91e^{-1}$ | $5.43e^{-1}$ | $5.80e^{-1}$ | $5.19e^{-1}$ | $5.06e^{-1}$ | $5.02e^{-1}$ | $5.31e^{-1}$ | $4.88e^{-1}$ | $4.87e^{-1}$ | P.TST |
| | Rank | $2.50e^1$ | $2.30e^1$ | $1.98e^1$ | $2.19e^1$ | $2.30e^1$ | $1.76e^1$ | 9.58 | $1.58e^1$ | $1.88e^1$ | 7.78 | 8.32 | $1.98e^1$ | $1.93e^1$ | $2.11e^1$ | $1.79e^1$ | $1.80e^1$ | $1.16e^1$ | $1.61e^1$ | $1.17e^1$ | $1.07e^1$ | $1.04e^1$ | $1.07e^1$ | 7.28 | 7.34 | Moi.$_B$ |
| **Hurst** | | | | | | | | | | | | | | | | | | | | | | | | | | | |
| High | MASE | 1.29 | 1.00 | $9.28e^{-1}$ | $7.53e^{-1}$ | $8.95e^{-1}$ | 1.12 | $7.79e^{-1}$ | $8.61e^{-1}$ | $7.31e^{-1}$ | $6.63e^{-1}$ | $7.10e^{-1}$ | $8.82e^{-1}$ | 4.11 | $8.62e^{-1}$ | $9.07e^{-1}$ | 1.10 | $8.62e^{-1}$ | $7.10e^{-1}$ | $6.72e^{-1}$ | $6.70e^{-1}$ | $6.59e^{-1}$ | $8.29e^{-1}$ | $7.44e^{-1}$ | $7.29e^{-1}$ | Chr.$_L$ |
| | CRPS | 1.45 | 1.00 | $7.85e^{-1}$ | $8.13e^{-1}$ | $2.02e^1$ | $8.38e^{-1}$ | $5.75e^{-1}$ | $7.21e^{-1}$ | $7.54e^{-1}$ | $5.47e^{-1}$ | $5.85e^{-1}$ | $8.13e^{-1}$ | 2.96 | $9.38e^{-1}$ | $8.64e^{-1}$ | $9.16e^{-1}$ | $6.24e^{-1}$ | $7.21e^{-1}$ | $5.72e^{-1}$ | $5.74e^{-1}$ | $5.63e^{-1}$ | $6.40e^{-1}$ | $5.78e^{-1}$ | $5.87e^{-1}$ | P.TST |
| | Rank | $2.46e^1$ | $2.21e^1$ | $1.64e^1$ | $1.81e^1$ | $1.95e^1$ | $1.82e^1$ | 9.39 | $1.61e^1$ | $1.83e^1$ | 7.83 | 9.11 | $2.05e^1$ | $2.20e^1$ | $2.15e^1$ | $2.04e^1$ | $1.93e^1$ | $1.07e^1$ | $1.82e^1$ | 9.89 | 9.39 | 8.97 | $1.12e^1$ | 7.28 | 8.39 | Moi.$_B$ |
| Low | MASE | 1.25 | 1.00 | $9.85e^{-1}$ | 1.14 | 1.22 | 1.26 | $8.48e^{-1}$ | 1.06 | $9.15e^{-1}$ | $8.27e^{-1}$ | $8.62e^{-1}$ | $9.96e^{-1}$ | 1.64 | 1.07 | 1.01 | 1.10 | 1.03 | $8.16e^{-1}$ | $8.88e^{-1}$ | $8.64e^{-1}$ | $8.62e^{-1}$ | $9.03e^{-1}$ | $8.54e^{-1}$ | $8.40e^{-1}$ | V.TS |
| | CRPS | 1.35 | 1.00 | $7.62e^{-1}$ | 1.22 | 3.19 | $6.59e^{-1}$ | $4.77e^{-1}$ | $6.14e^{-1}$ | $6.54e^{-1}$ | $4.68e^{-1}$ | $4.91e^{-1}$ | $6.62e^{-1}$ | $8.82e^{-1}$ | $7.57e^{-1}$ | $6.94e^{-1}$ | $6.58e^{-1}$ | $5.47e^{-1}$ | $5.93e^{-1}$ | $5.53e^{-1}$ | $5.38e^{-1}$ | $5.38e^{-1}$ | $5.39e^{-1}$ | $4.82e^{-1}$ | $4.77e^{-1}$ | Moi.$_L$ |
| | Rank | $2.52e^1$ | $2.32e^1$ | $1.92e^1$ | $2.29e^1$ | $2.35e^1$ | $1.45e^1$ | 8.52 | $1.49e^1$ | $1.74e^1$ | 7.57 | 8.26 | $1.84e^1$ | $1.69e^1$ | $2.12e^1$ | $1.95e^1$ | $1.72e^1$ | $1.17e^1$ | $1.61e^1$ | $1.34e^1$ | $1.20e^1$ | $1.21e^1$ | $1.08e^1$ | 7.16 | 7.08 | Moi.$_L$ |
| **Lumpiness** | | | | | | | | | | | | | | | | | | | | | | | | | | | |
| High | MASE | 1.14 | 1.00 | $9.54e^{-1}$ | 1.12 | 1.22 | 1.18 | $8.27e^{-1}$ | 1.07 | $8.84e^{-1}$ | $8.18e^{-1}$ | $9.15e^{-1}$ | 1.01 | 1.65 | 1.00 | $9.78e^{-1}$ | 1.04 | 1.01 | $8.48e^{-1}$ | $8.66e^{-1}$ | $8.36e^{-1}$ | $8.35e^{-1}$ | $8.66e^{-1}$ | $8.06e^{-1}$ | $8.06e^{-1}$ | Moi.$_L$ |
| | CRPS | 1.24 | 1.00 | $7.05e^{-1}$ | 1.03 | 8.64 | $6.31e^{-1}$ | $4.94e^{-1}$ | $6.28e^{-1}$ | $6.55e^{-1}$ | $4.85e^{-1}$ | $5.53e^{-1}$ | $6.79e^{-1}$ | 1.09 | $7.54e^{-1}$ | $7.13e^{-1}$ | $6.53e^{-1}$ | $5.52e^{-1}$ | $6.38e^{-1}$ | $5.71e^{-1}$ | $5.55e^{-1}$ | $5.56e^{-1}$ | $5.20e^{-1}$ | $4.93e^{-1}$ | $4.91e^{-1}$ | P.TST |
| | Rank | $2.61e^1$ | $2.47e^1$ | $1.85e^1$ | $2.33e^1$ | $2.33e^1$ | $1.41e^1$ | 8.27 | $1.48e^1$ | $1.73e^1$ | 7.95 | 9.80 | $1.91e^1$ | $1.86e^1$ | $2.10e^1$ | $1.98e^1$ | $1.64e^1$ | $1.07e^1$ | $1.79e^1$ | $1.35e^1$ | $1.17e^1$ | $1.20e^1$ | 9.63 | 6.95 | 7.49 | Moi.$_B$ |
| Low | MASE | 1.36 | 1.00 | $9.71e^{-1}$ | $8.89e^{-1}$ | 1.00 | 1.23 | $8.18e^{-1}$ | $9.16e^{-1}$ | $8.12e^{-1}$ | $7.23e^{-1}$ | $7.28e^{-1}$ | $9.13e^{-1}$ | 2.96 | 1.03 | $9.62e^{-1}$ | 1.15 | $9.33e^{-1}$ | $7.25e^{-1}$ | $7.56e^{-1}$ | $7.52e^{-1}$ | $7.43e^{-1}$ | $8.80e^{-1}$ | $8.15e^{-1}$ | $7.90e^{-1}$ | P.TST |
| | CRPS | 1.50 | 1.00 | $8.22e^{-1}$ | 1.06 | 5.04 | $7.95e^{-1}$ | $5.23e^{-1}$ | $6.70e^{-1}$ | $7.16e^{-1}$ | $5.04e^{-1}$ | $5.04e^{-1}$ | $7.42e^{-1}$ | 1.65 | $8.71e^{-1}$ | $7.84e^{-1}$ | $8.19e^{-1}$ | $5.92e^{-1}$ | $6.37e^{-1}$ | $5.52e^{-1}$ | $5.48e^{-1}$ | $5.40e^{-1}$ | $6.25e^{-1}$ | $5.33e^{-1}$ | $5.34e^{-1}$ | P.TST |
| | Rank | $2.41e^1$ | $2.13e^1$ | $1.79e^1$ | $1.94e^1$ | $2.11e^1$ | $1.71e^1$ | 9.27 | $1.57e^1$ | $1.81e^1$ | 7.46 | 7.68 | $1.92e^1$ | $1.89e^1$ | $2.21e^1$ | $1.98e^1$ | $1.91e^1$ | $1.18e^1$ | $1.62e^1$ | $1.11e^1$ | $1.05e^1$ | $1.01e^1$ | $1.20e^1$ | 7.39 | 7.62 | Moi.$_B$ |
| **Stability** | | | | | | | | | | | | | | | | | | | | | | | | | | | |
| High | MASE | 1.15 | 1.00 | $9.39e^{-1}$ | $7.18e^{-1}$ | $8.87e^{-1}$ | 1.13 | $7.85e^{-1}$ | $9.18e^{-1}$ | $7.29e^{-1}$ | $6.53e^{-1}$ | $7.20e^{-1}$ | $8.14e^{-1}$ | 1.99 | $9.73e^{-1}$ | $9.55e^{-1}$ | 1.19 | $8.38e^{-1}$ | $6.83e^{-1}$ | $7.53e^{-1}$ | $7.60e^{-1}$ | $7.48e^{-1}$ | $8.78e^{-1}$ | $8.28e^{-1}$ | $8.18e^{-1}$ | P.TST |
| | CRPS | 1.04 | 1.00 | $7.84e^{-1}$ | $7.46e^{-1}$ | $9.63e^{-1}$ | $9.41e^{-1}$ | $6.22e^{-1}$ | $8.08e^{-1}$ | $7.70e^{-1}$ | $5.67e^{-1}$ | $6.24e^{-1}$ | $8.00e^{-1}$ | 2.18 | $9.71e^{-1}$ | $9.73e^{-1}$ | 1.00 | $7.80e^{-1}$ | $7.36e^{-1}$ | $6.54e^{-1}$ | $6.60e^{-1}$ | $6.48e^{-1}$ | $6.91e^{-1}$ | $6.55e^{-1}$ | $6.74e^{-1}$ | P.TST |
| | Rank | $2.10e^1$ | $2.05e^1$ | $1.48e^1$ | $1.59e^1$ | $1.87e^1$ | $1.82e^1$ | 9.95 | $1.57e^1$ | $1.65e^1$ | 7.62 | 9.00 | $1.80e^1$ | $1.90e^1$ | $2.13e^1$ | $2.20e^1$ | $2.03e^1$ | $1.29e^1$ | $1.65e^1$ | $1.14e^1$ | $1.15e^1$ | $1.12e^1$ | $1.19e^1$ | 9.03 | 9.95 | P.TST |
| Low | MASE | 1.35 | 1.00 | $9.81e^{-1}$ | 1.22 | 1.26 | 1.27 | $8.48e^{-1}$ | 1.03 | $9.31e^{-1}$ | $8.49e^{-1}$ | $8.65e^{-1}$ | 1.06 | 2.56 | 1.05 | $9.78e^{-1}$ | 1.05 | 1.07 | $8.47e^{-1}$ | $8.36e^{-1}$ | $8.05e^{-1}$ | $8.04e^{-1}$ | $8.70e^{-1}$ | $8.06e^{-1}$ | $7.82e^{-1}$ | Moi.$_L$ |
| | CRPS | 1.70 | 1.00 | $7.60e^{-1}$ | 1.34 | $2.37e^1$ | $5.98e^{-1}$ | $4.45e^{-1}$ | $5.61e^{-1}$ | $6.38e^{-1}$ | $4.52e^{-1}$ | $4.64e^{-1}$ | $6.60e^{-1}$ | 1.00 | $7.28e^{-1}$ | $6.29e^{-1}$ | $6.04e^{-1}$ | $4.86e^{-1}$ | $5.77e^{-1}$ | $5.02e^{-1}$ | $4.85e^{-1}$ | $4.86e^{-1}$ | $5.00e^{-1}$ | $4.36e^{-1}$ | $4.27e^{-1}$ | Moi.$_L$ |
| | Rank | $2.77e^1$ | $2.43e^1$ | $2.05e^1$ | $2.47e^1$ | $2.44e^1$ | $1.42e^1$ | 8.07 | $1.51e^1$ | $1.86e^1$ | 7.70 | 8.28 | $2.00e^1$ | $1.86e^1$ | $2.13e^1$ | $1.83e^1$ | $1.63e^1$ | $1.02e^1$ | $1.71e^1$ | $1.26e^1$ | $1.07e^1$ | $1.07e^1$ | $1.09e^1$ | 5.93 | 5.89 | Moi.$_L$ |

R: [**Stability:** When stability is high (i.e. high variance), deep learning models seem to perform better, while for lower stability (lower variance), foundation models like Moirai$_{Large}$ tend to have an edge. This may indicate that deep learning models excel in highly variable conditions, while foundation models can handle less complex variance better.]

R: [Overall, these observations suggest that deep learning models, particularly Transformers like PatchTST, may perform better in more challenging, less predictable scenarios. In contrast, foundation models like Moirai$_{Large}$ seem to excel when temporal patterns are simpler and more predictable. This aligns with the aggregated results across datasets, where PatchTST shows strong generalist performance, while Moirai$_{Large}$ ranks highly in specific, presumably in easier-to-forecast cases. It is also likely that the supervised learning setup and hyperparameter tuning play a role in these differences, potentially giving an advantage to deep learning models in more complex scenarios.]

## F.2 RESULTS WITH ALL MODELS

In this section, we present results for all models, including those omitted from the main paper in section Section 4.1 due to space constraints. The results are displayed in the same aggregated form through Tables 17 to 21. Furthermore, we provide non-aggregated results across all dataset, term and frequency combinations in Tables 22 to 24.

## F.3 DATA LEAKAGE EFFECT IN FOUNDATION MODELS

Due to the use of different pre-training datasets by various foundation models, there is a partial data leakage issue when these models are evaluated against our GIFT-Eval. To ensure a fair comparison, we pre-trained a new series of Moirai models on our GIFT-Eval pre-training data and report the results from these models in the main paper. To further examine the impact of data leakage on foundation models, we also include the performance of the original Moirai models on our benchmark, both to facilitate replicability and to demonstrate the misleading effects that leakage can introduce. We call the original model Moirai-Leakage and use the abbreviation **Moi Leak.**. The results for all datasets affected by leakage are provided in Table 25. In most cases, data leakage from training sets can boost performance on the corresponding test sets, with this effect becoming more pronounced as prediction length increases. This finding highlights the capacity of foundation models to memorize training data and underscores the critical importance of preventing data leakage when comparing time series foundation models on public benchmarks.

Table 17: Results on GIFT-Eval with all models aggregated by domain. The best results across each row are **bolded**, while second best results are underlined.

| Domain | Metric | Nv. | S.Nv. | A.Ar. | A.Th. | A.ETS | D.AR | TFT | TiDE | N-B. | P.TST | iTr. | DLin. | C.former | Timer | TTM | L-Llama | T.FM | V.TS | Chr.S | Chr.B | Chr.L | Moi.S | Moi.B | Moi.L | Best |
|---|---|---|---|---|---|---|---|---|---|---|---|---|---|---|---|---|---|---|---|---|---|---|---|---|---|---|
| Econ/Fin | sMAPE | 1.18 | 1.00 | $9.51e^{-1}$ | $9.99e^{-1}$ | $9.41e^{-1}$ | 1.12 | 1.01 | 1.11 | $8.53e^{-1}$ | $8.92e^{-1}$ | $9.62e^{-1}$ | 1.02 | 7.42 | 1.48 | 1.11 | 1.77 | $8.40e^{-1}$ | $9.52e^{-1}$ | $8.10e^{-1}$ | $8.02e^{-1}$ | **$8.01e^{-1}$** | $9.19e^{-1}$ | $9.06e^{-1}$ | $9.39e^{-1}$ | Chr.L |
| | MASE | 1.43 | 1.00 | $8.66e^{-1}$ | $9.83e^{-1}$ | $8.99e^{-1}$ | 1.54 | 1.03 | 1.51 | $8.61e^{-1}$ | $9.08e^{-1}$ | $9.89e^{-1}$ | 1.13 | $2.93e^{1}$ | 1.81 | 1.30 | 2.91 | $8.24e^{-1}$ | $9.31e^{-1}$ | $7.97e^{-1}$ | $7.83e^{-1}$ | **$7.83e^{-1}$** | 1.04 | $9.27e^{-1}$ | $9.63e^{-1}$ | Chr.L |
| | ND | 1.20 | 1.00 | $8.99e^{-1}$ | $9.14e^{-1}$ | $9.73e^{-1}$ | 1.34 | $9.44e^{-1}$ | 1.10 | $8.74e^{-1}$ | $8.98e^{-1}$ | $9.53e^{-1}$ | 1.02 | $8.09e^{1}$ | 1.33 | 1.03 | 2.00 | **$8.11e^{-1}$** | $9.46e^{-1}$ | $8.36e^{-1}$ | $8.22e^{-1}$ | $8.27e^{-1}$ | $9.22e^{-1}$ | $9.42e^{-1}$ | T.FM |
| | MSE | 1.55 | 1.00 | $8.23e^{-1}$ | $8.32e^{-1}$ | 1.02 | 1.56 | $9.22e^{-1}$ | 1.07 | $7.80e^{-1}$ | $8.43e^{-1}$ | $9.08e^{-1}$ | $9.53e^{-1}$ | $4.78e^{1}$ | 1.43 | $8.31e^{-1}$ | 3.20 | **$5.94e^{-1}$** | $8.05e^{-1}$ | $7.20e^{-1}$ | $7.31e^{-1}$ | $7.36e^{-1}$ | $8.96e^{-1}$ | $8.02e^{-1}$ | $8.14e^{-1}$ | T.FM |
| | MAE | 1.20 | 1.00 | $8.99e^{-1}$ | $9.15e^{-1}$ | $9.73e^{-1}$ | 1.34 | $9.46e^{-1}$ | 1.10 | $8.76e^{-1}$ | $8.98e^{-1}$ | $9.54e^{-1}$ | 1.02 | $1.10e^{1}$ | 1.33 | 1.03 | 2.00 | **$8.10e^{-1}$** | $9.46e^{-1}$ | $8.36e^{-1}$ | $8.23e^{-1}$ | $8.28e^{-1}$ | $8.97e^{-1}$ | $9.23e^{-1}$ | $9.42e^{-1}$ | T.FM |
| | CRPS | 1.17 | 1.00 | $8.21e^{-1}$ | $8.41e^{-1}$ | $9.40e^{-1}$ | 1.22 | $8.41e^{-1}$ | 1.08 | $9.67e^{-1}$ | $8.03e^{-1}$ | $8.48e^{-1}$ | 1.12 | $1.09e^{2}$ | 1.48 | 1.14 | 1.84 | **$7.16e^{-1}$** | $9.46e^{-1}$ | $7.63e^{-1}$ | $7.51e^{-1}$ | $7.58e^{-1}$ | $7.96e^{-1}$ | $8.16e^{-1}$ | $8.47e^{-1}$ | T.FM |
| | Rank | $1.90e^{1}$ | $1.88e^{1}$ | 9.83 | $1.07e^{1}$ | $1.25e^{1}$ | $1.88e^{1}$ | $1.10e^{1}$ | $2.12e^{1}$ | $1.62e^{1}$ | 9.17 | $1.15e^{1}$ | $2.17e^{1}$ | $3.00e^{1}$ | $2.43e^{1}$ | $2.18e^{1}$ | $2.62e^{1}$ | 6.67 | $2.03e^{1}$ | 9.50 | 8.67 | 9.00 | $1.00e^{1}$ | 7.00 | **6.50** | Moi.L |
| Energy | sMAPE | 1.57 | 1.00 | 1.08 | 1.33 | 1.39 | 2.02 | 1.20 | 1.38 | 1.34 | 1.16 | 1.28 | 1.33 | 2.14 | 1.37 | 1.25 | 1.48 | 1.13 | 1.18 | 1.12 | 1.10 | 1.09 | 1.17 | 1.16 | 1.19 | S.Nv. |
| | MASE | 1.56 | 1.00 | 1.01 | 1.36 | 1.48 | 1.78 | 1.01 | 1.18 | 1.18 | $9.83e^{-1}$ | 1.11 | 1.15 | 2.19 | 1.29 | 1.06 | 1.39 | 1.02 | $9.93e^{-1}$ | $9.47e^{-1}$ | $9.24e^{-1}$ | **$9.19e^{-1}$** | 1.04 | $9.76e^{-1}$ | 1.03 | Chr.L |
| | ND | 1.55 | 1.00 | $9.90e^{-1}$ | 1.37 | 1.39 | 1.66 | $9.97e^{-1}$ | 1.14 | 1.18 | $9.72e^{-1}$ | 1.10 | 1.11 | 1.55 | 1.29 | 1.07 | 1.43 | 1.01 | $9.88e^{-1}$ | $9.38e^{-1}$ | $9.10e^{-1}$ | **$9.04e^{-1}$** | 1.02 | $9.76e^{-1}$ | $9.98e^{-1}$ | Chr.L |
| | MSE | 2.00 | 1.00 | $9.42e^{-1}$ | 1.42 | 1.84 | 2.12 | $8.79e^{-1}$ | 1.11 | 1.23 | $8.33e^{-1}$ | 1.06 | $9.68e^{-1}$ | 1.27 | 1.30 | $9.37e^{-1}$ | 1.84 | $9.50e^{-1}$ | 1.01 | $8.56e^{-1}$ | $8.06e^{-1}$ | **$7.99e^{-1}$** | $9.03e^{-1}$ | $8.83e^{-1}$ | $8.98e^{-1}$ | Chr.L |
| | MAE | 1.55 | 1.00 | $9.90e^{-1}$ | 1.37 | 1.39 | 1.66 | $9.97e^{-1}$ | 1.14 | 1.18 | $9.71e^{-1}$ | 1.10 | 1.11 | 1.56 | 1.29 | 1.07 | 1.44 | 1.01 | $9.88e^{-1}$ | $9.38e^{-1}$ | $9.09e^{-1}$ | **$9.04e^{-1}$** | 1.02 | $9.76e^{-1}$ | $9.97e^{-1}$ | Chr.L |
| | CRPS | 1.53 | 1.00 | $8.33e^{-1}$ | 1.70 | $1.01e^{1}$ | 1.07 | $6.30e^{-1}$ | $7.51e^{-1}$ | $9.35e^{-1}$ | **$6.12e^{-1}$** | $6.95e^{-1}$ | $8.79e^{-1}$ | 1.22 | 1.02 | $8.47e^{-1}$ | $9.23e^{-1}$ | $6.73e^{-1}$ | $7.82e^{-1}$ | $6.48e^{-1}$ | $6.31e^{-1}$ | $6.28e^{-1}$ | $6.68e^{-1}$ | $6.15e^{-1}$ | $6.66e^{-1}$ | P.TST |
| | Rank | $2.51e^{1}$ | $2.13e^{1}$ | $1.67e^{1}$ | $2.32e^{1}$ | $2.19e^{1}$ | $2.00e^{1}$ | 9.56 | $1.41e^{1}$ | $2.01e^{1}$ | 7.69 | 9.44 | $1.96e^{1}$ | $1.95e^{1}$ | $2.22e^{1}$ | $1.87e^{1}$ | $1.87e^{1}$ | $1.09e^{1}$ | $1.75e^{1}$ | $1.12e^{1}$ | 9.28 | 9.19 | 9.71 | 6.66 | **7.56** | Moi.B |
| Healthcare | sMAPE | 1.15 | 1.00 | $7.64e^{-1}$ | $9.57e^{-1}$ | $7.94e^{-1}$ | $8.86e^{-1}$ | $8.08e^{-1}$ | $9.22e^{-1}$ | $8.48e^{-1}$ | $8.25e^{-1}$ | $9.50e^{-1}$ | $9.69e^{-1}$ | 2.19 | 1.57 | 1.34 | 1.75 | $8.02e^{-1}$ | $8.85e^{-1}$ | $7.25e^{-1}$ | $7.70e^{-1}$ | **$7.14e^{-1}$** | 1.14 | $8.25e^{-1}$ | $8.37e^{-1}$ | Chr.L |
| | MASE | 1.16 | 1.00 | $7.84e^{-1}$ | $9.51e^{-1}$ | $7.97e^{-1}$ | $7.65e^{-1}$ | $6.60e^{-1}$ | $8.03e^{-1}$ | $6.91e^{-1}$ | $6.86e^{-1}$ | $7.74e^{-1}$ | $7.92e^{-1}$ | 8.53 | 1.39 | 1.15 | 1.62 | $6.98e^{-1}$ | $7.49e^{-1}$ | $6.02e^{-1}$ | $6.15e^{-1}$ | **$5.99e^{-1}$** | $9.51e^{-1}$ | $6.75e^{-1}$ | $6.91e^{-1}$ | Chr.L |
| | ND | 1.10 | 1.00 | $6.46e^{-1}$ | $9.31e^{-1}$ | $6.81e^{-1}$ | $8.65e^{-1}$ | $5.76e^{-1}$ | $8.74e^{-1}$ | $6.69e^{-1}$ | $6.83e^{-1}$ | $7.18e^{-1}$ | $7.57e^{-1}$ | 4.86 | 1.60 | 1.15 | 1.79 | $7.69e^{-1}$ | $6.39e^{-1}$ | $5.54e^{-1}$ | **$5.08e^{-1}$** | $2.61e^{-1}$ | $7.41e^{-1}$ | $3.64e^{-1}$ | $3.83e^{-1}$ | Chr.L |
| | MSE | 1.43 | 1.00 | $3.69e^{-1}$ | $7.39e^{-1}$ | $4.26e^{-1}$ | $7.35e^{-1}$ | $3.47e^{-1}$ | $7.26e^{-1}$ | $4.80e^{-1}$ | $4.43e^{-1}$ | $5.56e^{-1}$ | $5.75e^{-1}$ | 7.85 | 2.52 | 1.21 | 3.20 | $6.28e^{-1}$ | $3.83e^{-1}$ | $3.17e^{-1}$ | $3.10e^{-1}$ | **$2.61e^{-1}$** | $7.41e^{-1}$ | $3.64e^{-1}$ | $3.83e^{-1}$ | Chr.L |
| | MAE | 1.19 | 1.00 | $6.45e^{-1}$ | $9.30e^{-1}$ | $6.80e^{-1}$ | $8.64e^{-1}$ | $5.76e^{-1}$ | $8.73e^{-1}$ | $6.69e^{-1}$ | $6.83e^{-1}$ | $7.18e^{-1}$ | $7.57e^{-1}$ | 3.17 | 1.60 | 1.15 | 1.79 | $7.69e^{-1}$ | $6.39e^{-1}$ | $5.54e^{-1}$ | $5.56e^{-1}$ | **$5.07e^{-1}$** | $8.89e^{-1}$ | $6.02e^{-1}$ | $6.16e^{-1}$ | Chr.L |
| | CRPS | 1.19 | 1.00 | $5.70e^{-1}$ | $8.03e^{-1}$ | $5.86e^{-1}$ | $7.23e^{-1}$ | $5.12e^{-1}$ | $9.12e^{-1}$ | $7.13e^{-1}$ | $5.76e^{-1}$ | $6.28e^{-1}$ | $6.06e^{-1}$ | 5.18 | 1.70 | 1.23 | 1.60 | $6.52e^{-1}$ | $6.81e^{-1}$ | $4.96e^{-1}$ | $4.85e^{-1}$ | **$4.46e^{-1}$** | $7.72e^{-1}$ | $5.14e^{-1}$ | $5.28e^{-1}$ | Chr.L |
| | Rank | $2.26e^{1}$ | $1.98e^{1}$ | 9.60 | $1.52e^{1}$ | 9.00 | $1.26e^{1}$ | 9.40 | $1.74e^{1}$ | $1.72e^{1}$ | $1.06e^{1}$ | $1.26e^{1}$ | $1.96e^{1}$ | $2.24e^{1}$ | $2.58e^{1}$ | $2.42e^{1}$ | $2.56e^{1}$ | 9.60 | $1.60e^{1}$ | 7.00 | 6.00 | 4.60 | $1.63e^{1}$ | **5.80** | 7.20 | Chr.L |
| Nature | sMAPE | 1.09 | 1.00 | 1.10 | 1.31 | 1.23 | 1.54 | 1.17 | 1.30 | 1.22 | 1.14 | 1.14 | 1.18 | 1.60 | 1.17 | 1.16 | 1.14 | 1.14 | 1.15 | 1.15 | 1.10 | **1.07** | | | | Chr.L |
| | MASE | $9.62e^{-1}$ | 1.00 | 1.02 | 1.06 | 1.12 | 1.64 | $8.71e^{-1}$ | 1.37 | $9.33e^{-1}$ | $9.16e^{-1}$ | $8.51e^{-1}$ | 1.12 | 2.98 | $8.81e^{-1}$ | $8.45e^{-1}$ | $9.32e^{-1}$ | $8.80e^{-1}$ | $8.60e^{-1}$ | $8.51e^{-1}$ | $8.23e^{-1}$ | $8.13e^{-1}$ | $7.97e^{-1}$ | $7.80e^{-1}$ | **$7.56e^{-1}$** | Moi.L |
| | ND | 1.05 | 1.00 | 1.14 | 1.06 | 1.03 | 1.61 | $8.35e^{-1}$ | 1.34 | 1.22 | $8.17e^{-1}$ | $9.01e^{-1}$ | 1.09 | 1.52 | $9.79e^{-1}$ | $8.90e^{-1}$ | $9.92e^{-1}$ | $8.64e^{-1}$ | $9.54e^{-1}$ | $9.44e^{-1}$ | $8.96e^{-1}$ | $8.90e^{-1}$ | $8.66e^{-1}$ | $8.42e^{-1}$ | **$8.26e^{-1}$** | TTM |
| | MSE | 1.10 | 1.00 | 1.23 | 1.38 | 1.03 | 1.61 | $8.35e^{-1}$ | 1.34 | 1.22 | $8.17e^{-1}$ | $9.01e^{-1}$ | $8.50e^{-1}$ | 1.04 | $8.14e^{-1}$ | $7.05e^{-1}$ | $9.77e^{-1}$ | $7.73e^{-1}$ | $7.47e^{-1}$ | $9.53e^{-1}$ | $8.89e^{-1}$ | $8.88e^{-1}$ | $9.30e^{-1}$ | $8.28e^{-1}$ | $7.68e^{-1}$ | TTM |
| | MAE | 1.05 | 1.00 | 1.10 | 1.34 | 1.06 | 1.44 | $9.27e^{-1}$ | 1.32 | 1.17 | $9.10e^{-1}$ | $9.09e^{-1}$ | 1.09 | 1.44 | $9.79e^{-1}$ | $8.90e^{-1}$ | $9.92e^{-1}$ | $8.64e^{-1}$ | $8.95e^{-1}$ | $9.44e^{-1}$ | $8.97e^{-1}$ | $8.89e^{-1}$ | $8.66e^{-1}$ | $8.42e^{-1}$ | **$8.26e^{-1}$** | Moi.L |
| | CRPS | 1.33 | 1.00 | $6.58e^{-1}$ | $9.10e^{-1}$ | 7.02 | $5.35e^{-1}$ | $3.48e^{-1}$ | $5.61e^{-1}$ | $5.32e^{-1}$ | $3.47e^{-1}$ | 3.42 | $4.96e^{-1}$ | $6.90e^{-1}$ | $4.45e^{-1}$ | $4.04e^{-1}$ | $3.85e^{-1}$ | 3.33 | $4.06e^{-1}$ | $3.83e^{-1}$ | $3.66e^{-1}$ | $3.64e^{-1}$ | $3.73e^{-1}$ | $3.15e^{-1}$ | **3.11e-1** | Moi.L |
| | Rank | $2.75e^{1}$ | $2.67e^{1}$ | $2.11e^{1}$ | $2.37e^{1}$ | $2.41e^{1}$ | $1.73e^{1}$ | $1.05e^{1}$ | $1.93e^{1}$ | $2.13e^{1}$ | $1.03e^{1}$ | 8.93 | $2.11e^{1}$ | $1.82e^{1}$ | $1.91e^{1}$ | $1.57e^{1}$ | $1.39e^{1}$ | 8.13 | $1.65e^{1}$ | $1.40e^{1}$ | $1.29e^{1}$ | $1.23e^{1}$ | 9.21 | 5.20 | **5.27** | Moi.L |
| Sales | sMAPE | $9.97e^{-1}$ | 1.00 | 1.04 | $9.42e^{-1}$ | $9.28e^{-1}$ | $9.00e^{-1}$ | $9.16e^{-1}$ | 1.16 | $9.04e^{-1}$ | $8.88e^{-1}$ | $8.94e^{-1}$ | $9.32e^{-1}$ | 1.18 | $9.13e^{-1}$ | $9.09e^{-1}$ | 1.17 | **$8.62e^{-1}$** | $9.09e^{-1}$ | $8.92e^{-1}$ | $8.51e^{-1}$ | $8.81e^{-1}$ | $8.84e^{-1}$ | $8.77e^{-1}$ | $8.71e^{-1}$ | T.FM |
| | MASE | 1.00 | 1.00 | $8.13e^{-1}$ | $8.73e^{-1}$ | $8.87e^{-1}$ | $7.07e^{-1}$ | $7.16e^{-1}$ | $9.81e^{-1}$ | $7.04e^{-1}$ | **$6.90e^{-1}$** | $6.99e^{-1}$ | $8.08e^{-1}$ | 1.59 | $7.75e^{-1}$ | $8.78e^{-1}$ | $8.41e^{-1}$ | $7.00e^{-1}$ | $8.17e^{-1}$ | $7.33e^{-1}$ | $7.26e^{-1}$ | $7.24e^{-1}$ | $7.31e^{-1}$ | $6.95e^{-1}$ | $7.10e^{-1}$ | P.TST |
| | ND | 1.01 | 1.00 | $8.17e^{-1}$ | $8.69e^{-1}$ | $9.06e^{-1}$ | $7.00e^{-1}$ | $6.96e^{-1}$ | $9.75e^{-1}$ | $6.81e^{-1}$ | **$6.66e^{-1}$** | $6.76e^{-1}$ | $7.92e^{-1}$ | 9.49 | $7.69e^{-1}$ | $8.88e^{-1}$ | $8.28e^{-1}$ | $6.90e^{-1}$ | $8.09e^{-1}$ | $7.33e^{-1}$ | $7.25e^{-1}$ | $7.24e^{-1}$ | $7.23e^{-1}$ | $6.83e^{-1}$ | $7.01e^{-1}$ | P.TST |
| | MSE | 1.05 | 1.00 | $6.21e^{-1}$ | $6.42e^{-1}$ | $7.30e^{-1}$ | $5.13e^{-1}$ | $5.40e^{-1}$ | $7.37e^{-1}$ | $5.15e^{-1}$ | $5.06e^{-1}$ | $5.10e^{-1}$ | $5.38e^{-1}$ | 1.09 | $5.53e^{-1}$ | $6.09e^{-1}$ | $7.00e^{-1}$ | **$4.96e^{-1}$** | $5.83e^{-1}$ | $5.96e^{-1}$ | $5.46e^{-1}$ | $5.36e^{-1}$ | $5.21e^{-1}$ | $5.12e^{-1}$ | $5.05e^{-1}$ | T.FM |
| | MAE | 1.01 | 1.00 | $8.16e^{-1}$ | $8.68e^{-1}$ | $9.04e^{-1}$ | $7.01e^{-1}$ | $6.96e^{-1}$ | $9.75e^{-1}$ | $6.80e^{-1}$ | **$6.65e^{-1}$** | $6.74e^{-1}$ | $7.91e^{-1}$ | 1.28 | $7.69e^{-1}$ | $8.87e^{-1}$ | $8.28e^{-1}$ | $6.89e^{-1}$ | $8.08e^{-1}$ | $7.32e^{-1}$ | $7.25e^{-1}$ | $7.23e^{-1}$ | $7.22e^{-1}$ | $6.82e^{-1}$ | $7.00e^{-1}$ | P.TST |
| | CRPS | $8.96e^{-1}$ | 1.00 | $4.58e^{-1}$ | $4.80e^{-1}$ | 2.20 | $3.52e^{-1}$ | $3.52e^{-1}$ | $4.84e^{-1}$ | $4.14e^{-1}$ | $3.48e^{-1}$ | $3.51e^{-1}$ | $4.81e^{-1}$ | 5.77 | $4.68e^{-1}$ | $5.40e^{-1}$ | $4.42e^{-1}$ | **$3.44e^{-1}$** | $4.92e^{-1}$ | $3.66e^{-1}$ | $3.63e^{-1}$ | $3.62e^{-1}$ | $3.61e^{-1}$ | $3.47e^{-1}$ | $3.63e^{-1}$ | T.FM |
| | Rank | $2.80e^{1}$ | $2.80e^{1}$ | $1.96e^{1}$ | $2.10e^{1}$ | $2.58e^{1}$ | 8.75 | $1.10e^{1}$ | $2.05e^{1}$ | $1.45e^{1}$ | 5.00 | 7.00 | $2.02e^{1}$ | $2.89e^{1}$ | $1.98e^{1}$ | $2.28e^{1}$ | $1.70e^{1}$ | 3.00 | $2.15e^{1}$ | $1.22e^{1}$ | $1.05e^{1}$ | $1.10e^{1}$ | $1.28e^{1}$ | 5.20 | **6.75** | T.FM |
| Transport | sMAPE | 1.23 | 1.00 | $9.94e^{-1}$ | 1.14 | 1.07 | $7.53e^{-1}$ | $7.00e^{-1}$ | $8.09e^{-1}$ | $7.46e^{-1}$ | $7.20e^{-1}$ | $7.13e^{-1}$ | $8.43e^{-1}$ | 1.30 | $9.13e^{-1}$ | $9.04e^{-1}$ | 1.14 | $8.39e^{-1}$ | $7.59e^{-1}$ | $7.51e^{-1}$ | $7.64e^{-1}$ | $7.28e^{-1}$ | $7.26e^{-1}$ | $7.20e^{-1}$ | **$6.39e^{-1}$** | Moi.L |
| | MASE | 1.26 | 1.00 | $9.74e^{-1}$ | 1.08 | 1.20 | $7.45e^{-1}$ | $6.79e^{-1}$ | $7.90e^{-1}$ | $7.31e^{-1}$ | $7.09e^{-1}$ | $7.07e^{-1}$ | $8.08e^{-1}$ | 1.77 | $8.93e^{-1}$ | $8.91e^{-1}$ | $8.42e^{-1}$ | $7.41e^{-1}$ | $7.38e^{-1}$ | $7.37e^{-1}$ | $7.12e^{-1}$ | $7.14e^{-1}$ | $7.26e^{-1}$ | $6.34e^{-1}$ | **$6.07e^{-1}$** | Moi.L |
| | ND | 1.28 | 1.00 | $9.73e^{-1}$ | 1.11 | 1.24 | $7.24e^{-1}$ | $6.62e^{-1}$ | $7.80e^{-1}$ | $7.25e^{-1}$ | $6.98e^{-1}$ | $6.99e^{-1}$ | $8.00e^{-1}$ | $9.90e^{-1}$ | $9.07e^{-1}$ | $8.92e^{-1}$ | $8.52e^{-1}$ | $7.37e^{-1}$ | $7.34e^{-1}$ | $7.29e^{-1}$ | $7.03e^{-1}$ | $7.05e^{-1}$ | $7.22e^{-1}$ | $6.27e^{-1}$ | **$5.97e^{-1}$** | Moi.L |
| | MSE | 1.44 | 1.00 | $9.56e^{-1}$ | 1.03 | 1.65 | $5.31e^{-1}$ | $4.75e^{-1}$ | $5.70e^{-1}$ | $5.42e^{-1}$ | $4.98e^{-1}$ | $5.03e^{-1}$ | $5.81e^{-1}$ | 1.47 | $7.11e^{-1}$ | $6.50e^{-1}$ | $6.93e^{-1}$ | $5.40e^{-1}$ | $5.49e^{-1}$ | $5.91e^{-1}$ | $5.58e^{-1}$ | $5.61e^{-1}$ | $5.36e^{-1}$ | $4.27e^{-1}$ | **$3.98e^{-1}$** | Moi.L |
| | MAE | 1.28 | 1.00 | $9.73e^{-1}$ | 1.11 | 1.23 | $7.24e^{-1}$ | $6.62e^{-1}$ | $7.80e^{-1}$ | $7.26e^{-1}$ | $6.98e^{-1}$ | $7.00e^{-1}$ | $7.99e^{-1}$ | 1.45 | $9.07e^{-1}$ | $8.92e^{-1}$ | $8.53e^{-1}$ | $7.37e^{-1}$ | $7.34e^{-1}$ | $7.29e^{-1}$ | $7.03e^{-1}$ | $7.05e^{-1}$ | $7.22e^{-1}$ | $6.27e^{-1}$ | **$5.97e^{-1}$** | Moi.L |
| | CRPS | 2.07 | 1.00 | $7.63e^{-1}$ | 1.33 | 4.24 | $4.84e^{-1}$ | $4.43e^{-1}$ | $5.31e^{-1}$ | $5.93e^{-1}$ | $4.61e^{-1}$ | $4.60e^{-1}$ | $6.54e^{-1}$ | $8.11e^{-1}$ | $7.43e^{-1}$ | $7.30e^{-1}$ | $5.72e^{-1}$ | $5.10e^{-1}$ | $6.01e^{-1}$ | $5.30e^{-1}$ | $5.12e^{-1}$ | $5.12e^{-1}$ | $4.98e^{-1}$ | $4.12e^{-1}$ | **$3.93e^{-1}$** | Moi.L |
| | Rank | $2.84e^{1}$ | $2.43e^{1}$ | $2.18e^{1}$ | $2.61e^{1}$ | $2.55e^{1}$ | 8.73 | 6.60 | $1.39e^{1}$ | $1.74e^{1}$ | 8.07 | 7.93 | $1.97e^{1}$ | $1.49e^{1}$ | $2.22e^{1}$ | $2.13e^{1}$ | $1.54e^{1}$ | $1.06e^{1}$ | $1.81e^{1}$ | $1.39e^{1}$ | $1.08e^{1}$ | $1.11e^{1}$ | $1.07e^{1}$ | 5.40 | **5.67** | Moi.L |
| Web/CloudOps | sMAPE | 1.05 | 1.00 | 1.10 | $9.65e^{-1}$ | 1.01 | 1.34 | 1.24 | 1.13 | 1.00 | **$9.37e^{-1}$** | $9.77e^{-1}$ | 1.22 | 1.36 | 1.26 | 1.35 | 1.28 | $9.87e^{-1}$ | 1.21 | 1.21 | 1.20 | 1.37 | 1.24 | | | P.TST |
| | MASE | 1.13 | 1.00 | $9.57e^{-1}$ | $5.21e^{-1}$ | $7.18e^{-1}$ | $8.50e^{-1}$ | $6.62e^{-1}$ | $6.23e^{-1}$ | $5.43e^{-1}$ | $4.62e^{-1}$ | $4.88e^{-1}$ | $7.24e^{-1}$ | $9.20e^{-1}$ | $7.11e^{-1}$ | $8.67e^{-1}$ | $7.54e^{-1}$ | 1.42 | $4.72e^{-1}$ | $6.78e^{-1}$ | $6.76e^{-1}$ | $6.75e^{-1}$ | $7.73e^{-1}$ | $7.62e^{-1}$ | $6.79e^{-1}$ | P.TST |
| | ND | $8.84e^{-1}$ | 1.00 | $9.48e^{-1}$ | $6.36e^{-1}$ | $8.73e^{-1}$ | $7.42e^{-1}$ | $6.19e^{-1}$ | $6.99e^{-1}$ | $5.99e^{-1}$ | **$5.38e^{-1}$** | $5.61e^{-1}$ | $9.59e^{-1}$ | $6.46e^{-1}$ | $8.10e^{-1}$ | $8.93e^{-1}$ | $8.96e^{-1}$ | $8.79e^{-1}$ | $6.34e^{-1}$ | $7.16e^{-1}$ | $7.41e^{-1}$ | $7.36e^{-1}$ | $7.98e^{-1}$ | $8.15e^{-1}$ | $7.97e^{-1}$ | P.TST |
| | MSE | $8.27e^{-1}$ | 1.00 | $9.28e^{-1}$ | $3.83e^{-1}$ | $8.28e^{-1}$ | $5.68e^{-1}$ | $4.17e^{-1}$ | $4.47e^{-1}$ | $3.81e^{-1}$ | $3.48e^{-1}$ | **3.34e-1** | $4.05e^{-1}$ | $3.52e^{-1}$ | $5.40e^{-1}$ | $6.05e^{-1}$ | $7.51e^{-1}$ | $7.09e^{-1}$ | $4.25e^{-1}$ | $5.92e^{-1}$ | $6.12e^{-1}$ | $6.06e^{-1}$ | $7.31e^{-1}$ | $7.22e^{-1}$ | $7.18e^{-1}$ | iTr. |
| | MAE | $8.84e^{-1}$ | 1.00 | $9.48e^{-1}$ | $6.37e^{-1}$ | $8.73e^{-1}$ | $7.41e^{-1}$ | $6.18e^{-1}$ | $6.99e^{-1}$ | $5.98e^{-1}$ | **$5.38e^{-1}$** | $5.61e^{-1}$ | $9.32e^{-1}$ | $6.46e^{-1}$ | $8.09e^{-1}$ | $8.93e^{-1}$ | $8.95e^{-1}$ | $8.79e^{-1}$ | $6.33e^{-1}$ | $7.16e^{-1}$ | $7.41e^{-1}$ | $7.37e^{-1}$ | $7.98e^{-1}$ | $8.15e^{-1}$ | $7.96e^{-1}$ | P.TST |
| | CRPS | 1.07 | 1.00 | $9.04e^{-1}$ | $6.08e^{-1}$ | 2.64 | $6.33e^{-1}$ | $5.03e^{-1}$ | $5.68e^{-1}$ | $5.70e^{-1}$ | **$4.37e^{-1}$** | $4.54e^{-1}$ | $6.60e^{-1}$ | $6.15e^{-1}$ | $7.71e^{-1}$ | $8.51e^{-1}$ | $7.34e^{-1}$ | $7.39e^{-1}$ | $6.03e^{-1}$ | $6.29e^{-1}$ | $6.51e^{-1}$ | $6.47e^{-1}$ | $6.49e^{-1}$ | $6.28e^{-1}$ | $6.19e^{-1}$ | P.TST |
| | Rank | $2.19e^{1}$ | $2.18e^{1}$ | $1.99e^{1}$ | $1.66e^{1}$ | $2.34e^{1}$ | 4.44 | $1.18e^{1}$ | $1.30e^{1}$ | $1.32e^{1}$ | 6.95 | $1.22e^{1}$ | $1.29e^{1}$ | 4.75 | **5.85** | $1.57e^{1}$ | $1.44e^{1}$ | $1.91e^{1}$ | $2.13e^{1}$ | $1.76e^{1}$ | $1.84e^{1}$ | $1.35e^{1}$ | $1.29e^{1}$ | $1.45e^{1}$ | $1.48e^{1}$ | P.TST |

Table 18: Results on GIFT-Eval with all models aggregated by term length. The best results across each row are **bolded**, while second best results are underlined.

| Pred. Len. | Metric | Nv. | S.Nv. | A.Ar. | A.Th. | A.ETS | D.AR | TFT | TiDE | N-B. | P.TST | iTr. | DLin. | C.former | Timer | TTM | L-Llama | T.FM | V.TS | Chr.S | Chr.B | Chr.L | Moi.S | Moi.B | Moi.L | Best |
|---|---|---|---|---|---|---|---|---|---|---|---|---|---|---|---|---|---|---|---|---|---|---|---|---|---|---|
| Long | sMAPE | 1.40 | 1.00 | 1.11 | 1.18 | 1.19 | 1.48 | 1.03 | 1.07 | 1.05 | **$9.25e^{-1}$** | $9.79e^{-1}$ | 1.09 | 1.43 | 1.18 | 1.15 | 1.17 | 1.19 | $9.37e^{-1}$ | 1.08 | 1.06 | 1.05 | 1.07 | 1.09 | 1.00 | P.TST |
| | MASE | 1.40 | 1.00 | $9.85e^{-1}$ | $8.69e^{-1}$ | $9.56e^{-1}$ | 1.10 | $5.89e^{-1}$ | $6.55e^{-1}$ | $6.44e^{-1}$ | $5.37e^{-1}$ | $5.66e^{-1}$ | $7.00e^{-1}$ | $9.21e^{-1}$ | $7.53e^{-1}$ | $7.31e^{-1}$ | $7.24e^{-1}$ | $9.90e^{-1}$ | **$5.22e^{-1}$** | $6.58e^{-1}$ | $6.34e^{-1}$ | $6.32e^{-1}$ | $6.44e^{-1}$ | $6.25e^{-1}$ | $6.04e^{-1}$ | V.TS |
| | ND | 1.37 | 1.00 | 1.01 | 1.30 | 1.17 | 1.20 | $7.36e^{-1}$ | $8.66e^{-1}$ | $8.96e^{-1}$ | $7.15e^{-1}$ | $7.69e^{-1}$ | $8.97e^{-1}$ | **$4.05e^{-1}$** | 1.03 | $9.44e^{-1}$ | 1.02 | $9.07e^{-1}$ | $7.22e^{-1}$ | $8.68e^{-1}$ | $8.34e^{-1}$ | $8.75e^{-1}$ | $8.27e^{-1}$ | $8.31e^{-1}$ | $6.87e^{-1}$ | C.former |
| | MSE | 1.65 | 1.00 | 1.02 | 1.41 | 1.45 | 1.12 | $5.17e^{-1}$ | $6.21e^{-1}$ | $7.38e^{-1}$ | $5.01e^{-1}$ | $5.69e^{-1}$ | $6.18e^{-1}$ | **$4.44e^{-1}$** | $8.77e^{-1}$ | $7.04e^{-1}$ | $9.60e^{-1}$ | $7.21e^{-1}$ | $5.12e^{-1}$ | $7.43e^{-1}$ | $7.14e^{-1}$ | $7.12e^{-1}$ | $7.79e^{-1}$ | $7.11e^{-1}$ | $6.87e^{-1}$ | C.former |
| | MAE | 1.37 | 1.00 | 1.01 | 1.30 | 1.17 | 1.20 | $7.35e^{-1}$ | $8.66e^{-1}$ | $8.95e^{-1}$ | $7.14e^{-1}$ | $7.68e^{-1}$ | $8.96e^{-1}$ | $8.74e^{-1}$ | 1.03 | $9.44e^{-1}$ | 1.02 | $9.06e^{-1}$ | $7.22e^{-1}$ | $8.68e^{-1}$ | $8.37e^{-1}$ | $8.44e^{-1}$ | $8.75e^{-1}$ | $8.26e^{-1}$ | **$8.30e^{-1}$** | P.TST |
| | CRPS | 1.89 | 1.00 | $8.05e^{-1}$ | 1.40 | $3.69e^{-1}$ | $6.26e^{-1}$ | $3.79e^{-1}$ | $4.88e^{-1}$ | $5.65e^{-1}$ | $3.68e^{-1}$ | $3.91e^{-1}$ | $5.66e^{-1}$ | **2.55e-1** | $6.50e^{-1}$ | $5.96e^{-1}$ | $5.36e^{-1}$ | $5.18e^{-1}$ | $4.56e^{-1}$ | $5.22e^{-1}$ | $5.04e^{-1}$ | $5.05e^{-1}$ | $4.45e^{-1}$ | $4.23e^{-1}$ | $4.22e^{-1}$ | C.former |
| | Rank | $2.72e^{1}$ | $2.31e^{1}$ | $2.09e^{1}$ | $2.43e^{1}$ | $2.61e^{1}$ | $1.72e^{1}$ | 6.48 | $1.16e^{1}$ | $1.61e^{1}$ | 6.00 | 7.19 | $1.75e^{1}$ | $1.09e^{1}$ | $1.98e^{1}$ | $1.80e^{1}$ | $1.52e^{1}$ | $1.51e^{1}$ | $1.26e^{1}$ | $1.56e^{1}$ | $1.40e^{1}$ | $1.44e^{1}$ | 9.29 | 8.24 | **8.19** | P.TST |
| Medium | sMAPE | 1.48 | 1.00 | 1.10 | 1.34 | 1.41 | 1.51 | 1.21 | 1.22 | 1.25 | **1.08** | 1.11 | 1.28 | 1.69 | 1.38 | 1.38 | 1.38 | 1.27 | 1.11 | 1.27 | 1.26 | 1.25 | 1.24 | 1.24 | 1.17 | S.Nv. |
| | MASE | 1.46 | 1.00 | 1.02 | 1.17 | 1.61 | 1.33 | $9.48e^{-1}$ | $9.86e^{-1}$ | 1.03 | $8.56e^{-1}$ | $8.67e^{-1}$ | 1.09 | 1.85 | 1.22 | 1.20 | 1.18 | 1.44 | **$8.47e^{-1}$** | 1.04 | 1.04 | 1.03 | 1.03 | 1.03 | $9.72e^{-1}$ | V.TS |
| | ND | 1.40 | 1.00 | 1.04 | 1.22 | 1.51 | 1.11 | $8.38e^{-1}$ | $9.35e^{-1}$ | $9.86e^{-1}$ | $8.21e^{-1}$ | $8.35e^{-1}$ | $9.95e^{-1}$ | **$6.70e^{-1}$** | 1.21 | 1.10 | 1.16 | 1.04 | $8.47e^{-1}$ | $9.75e^{-1}$ | $9.81e^{-1}$ | $9.69e^{-1}$ | $9.94e^{-1}$ | $9.59e^{-1}$ | $9.36e^{-1}$ | C.former |
| | MSE | 1.75 | 1.00 | 1.07 | 1.12 | 2.51 | 1.07 | $6.81e^{-1}$ | $7.43e^{-1}$ | $9.06e^{-1}$ | **$6.56e^{-1}$** | $6.74e^{-1}$ | $7.68e^{-1}$ | $8.26e^{-1}$ | 1.13 | $9.19e^{-1}$ | 1.23 | $9.50e^{-1}$ | $7.03e^{-1}$ | $9.39e^{-1}$ | $9.29e^{-1}$ | $9.08e^{-1}$ | $9.58e^{-1}$ | $8.89e^{-1}$ | $8.62e^{-1}$ | P.TST |
| | MAE | 1.40 | 1.00 | 1.04 | 1.22 | 1.51 | 1.11 | $8.39e^{-1}$ | $9.34e^{-1}$ | $9.86e^{-1}$ | **$8.20e^{-1}$** | $8.36e^{-1}$ | $9.95e^{-1}$ | 1.20 | 1.21 | 1.10 | 1.16 | 1.04 | $8.47e^{-1}$ | $9.75e^{-1}$ | $9.81e^{-1}$ | $9.68e^{-1}$ | $9.94e^{-1}$ | $9.59e^{-1}$ | $9.36e^{-1}$ | P.TST |
| | CRPS | 1.87 | 1.00 | $8.30e^{-1}$ | 1.53 | 6.23 | $8.09e^{-1}$ | $6.14e^{-1}$ | $7.16e^{-1}$ | $8.20e^{-1}$ | **$6.56e^{-1}$**... | $4.61e^{-1}$ | $6.84e^{-1}$ | $4.61e^{-1}$ | $8.30e^{-1}$ | $7.65e^{-1}$ | $6.72e^{-1}$ | $8.10e^{-1}$ | $5.66e^{-1}$ | $6.22e^{-1}$ | $6.30e^{-1}$ | $5.55e^{-1}$ | $5.35e^{-1}$ | 5.33 | 6.46 | C.former |
| | Rank | $2.62e^{1}$ | $2.16e^{1}$ | $1.99e^{1}$ | $2.43e^{1}$ | $2.55e^{1}$ | $1.36e^{1}$ | 5.90 | $1.24e^{1}$ | $1.70e^{1}$ | 5.14 | **5.71** | $1.81e^{1}$ | $1.41e^{1}$ | $2.10e^{1}$ | $1.93e^{1}$ | $1.63e^{1}$ | $1.41e^{1}$ | $1.41e^{1}$ | $1.56e^{1}$ | $1.50e^{1}$ | $1.42e^{1}$ | $1.00e^{1}$ | 8.86 | 8.62 | P.TST |
| Short | sMAPE | 1.12 | 1.00 | 1.09 | 1.04 | 1.27 | 1.02 | 1.18 | 1.00 | $9.70e^{-1}$ | 1.03 | 1.10 | 2.05 | 1.19 | 1.10 | 1.35 | $9.35e^{-1}$ | 1.02 | $9.22e^{-1}$ | $9.09e^{-1}$ | **$9.02e^{-1}$** | 1.03 | $9.66e^{-1}$ | $9.73e^{-1}$ | Chr.L |
| | MASE | 1.14 | 1.00 | $9.35e^{-1}$ | $9.55e^{-1}$ | $9.84e^{-1}$ | 1.20 | $8.83e^{-1}$ | 1.14 | $8.62e^{-1}$ | $8.32e^{-1}$ | $8.89e^{-1}$ | 1.02 | 3.57 | 1.07 | $9.93e^{-1}$ | 1.26 | $8.23e^{-1}$ | $8.71e^{-1}$ | $7.79e^{-1}$ | $7.68e^{-1}$ | **$7.61e^{-1}$** | $8.97e^{-1}$ | $8.19e^{-1}$ | $8.21e^{-1}$ | Chr.L |
| | ND | 1.08 | 1.00 | $9.12e^{-1}$ | $9.30e^{-1}$ | $9.52e^{-1}$ | 1.09 | $8.12e^{-1}$ | 1.03 | $8.47e^{-1}$ | $7.88e^{-1}$ | $8.42e^{-1}$ | $8.99e^{-1}$ | 4.54 | 1.01 | $9.31e^{-1}$ | 1.19 | $8.09e^{-1}$ | $8.50e^{-1}$ | $7.46e^{-1}$ | $7.31e^{-1}$ | **$7.25e^{-1}$** | $8.51e^{-1}$ | $7.73e^{-1}$ | $7.73e^{-1}$ | Chr.L |
| | MSE | 1.15 | 1.00 | $8.71e^{-1}$ | $7.38e^{-1}$ | $9.52e^{-1}$ | 1.08 | $6.73e^{-1}$ | $6.17e^{-1}$ | $7.88e^{-1}$ | $6.36e^{-1}$ | $7.22e^{-1}$ | $7.00e^{-1}$ | 2.40 | $8.30e^{-1}$ | 1.34 | $6.71e^{-1}$ | $6.88e^{-1}$ | $7.22e^{-1}$ | $7.09e^{-1}$ | $6.30e^{-1}$ | **$7.25e^{-1}$** | | | | Chr.L |
| | MAE | 1.08 | 1.00 | $9.12e^{-1}$ | $9.31e^{-1}$ | $9.52e^{-1}$ | 1.09 | $8.13e^{-1}$ | 1.03 | $8.47e^{-1}$ | $7.88e^{-1}$ | $8.42e^{-1}$ | $8.99e^{-1}$ | 1.94 | 1.01 | $9.31e^{-1}$ | 1.19 | $8.09e^{-1}$ | $8.50e^{-1}$ | $7.46e^{-1}$ | $7.31e^{-1}$ | **$7.25e^{-1}$** | $8.51e^{-1}$ | $7.73e^{-1}$ | $7.73e^{-1}$ | Chr.L |
| | CRPS | 1.09 | 1.00 | $7.35e^{-1}$ | $8.16e^{-1}$ | 1.35 | $7.95e^{-1}$ | $5.92e^{-1}$ | $7.95e^{-1}$ | $7.48e^{-1}$ | $5.71e^{-1}$ | $6.11e^{-1}$ | $7.94e^{-1}$ | 4.01 | $8.91e^{-1}$ | $8.22e^{-1}$ | $8.76e^{-1}$ | $5.77e^{-1}$ | $7.51e^{-1}$ | $5.52e^{-1}$ | $5.42e^{-1}$ | **5.38e-1** | $6.09e^{-1}$ | $5.48e^{-1}$ | $5.53e^{-1}$ | Chr.L |
| | Rank | $2.36e^{1}$ | $2.31e^{1}$ | $1.64e^{1}$ | $1.86e^{1}$ | $1.91e^{1}$ | $1.62e^{1}$ | $1.09e^{1}$ | $1.79e^{1}$ | $1.87e^{1}$ | 9.27 | $1.02e^{1}$ | $2.02e^{1}$ | $2.47e^{1}$ | $2.20e^{1}$ | $2.07e^{1}$ | $1.97e^{1}$ | 8.80 | $1.96e^{1}$ | 9.65 | 8.33 | 8.33 | $1.14e^{1}$ | 6.18 | **6.93** | Moi.B |

## F.4 ADDITIONAL QUALITATIVE EXAMPLES

In addition to the four examples shared in the main paper, we present three additional qualitative examples in Figure 3. Figure 3(a) illustrates the forecasts of foundation models on the *Bizitobs_l2c* dataset (hourly, medium-term). Similar to previous observations, `Chronos` forecasts tend to degrade over longer time horizons. Unlike in earlier scenarios, `Moirai` also shows poor performance on this dataset, missing all the regular peaks and troughs. In contrast, the `VisionTS` model provides the forecast closest to the ground truth. Figure 3(b) presents forecasts for a higher-frequency dataset: *Electricity* (15-minute intervals, long-term). Notably, `Chronos` excels on this dataset, showing better consistency than both `Moirai` and `VisionTS`. While `Moirai` performs reasonably well, it tends to predict some stationary changes (see the rightmost side of the upper plot) that are not aligned with the ground truth data. In contrast, `VisionTS` repeats a mistake observed in earlier datasets by predicting shifted peaks. The final plots in Figure 3(c) display forecasts from deep learning models on the same *Electricity* dataset (15-minute intervals, long-term). Compared to the foundation models, these deep learning models demonstrate poorer performance. Notably, the model that differs most by its forecast is `DeepAR`, which quickly flattens at the beginning of the prediction—a phenomenon also observed with another deep learning model, `N-BEATS`, in the *Solar* dataset example in Figure 2(b).

## F.5 INTER-VARIATE CORRELATION ANALYSIS FOR MULTIVARAITE DATASETS

R: [In this section, we present the results of our inter-variate correlation analysis for the selected multivariate datasets (*i.e.* ones with more than two variates) included in GIFT-Eval. These analyses aim to demonstrate that the multivariate datasets in GIFT-Eval exhibit strong correlations across different variates, making them suitable for evaluating multivariate forecasting capabilities. If these

Table 19: Results on GIFT-Eval with all models aggregated by frequency. The best results across each row are **bolded**, while second best results are underlined.

| Freq. | Metric | Nv. | S.Nv. | A.Ar. | A.Th. | A.ETS | D.AR | TFT | TiDE | N-B. | P.TST | iTr. | DLin. | C.former | Timer | TTM | L-Llama | T.FM | V.TS | Chr.$_S$ | Chr.$_B$ | Chr.$_L$ | Moi.$_S$ | Moi.$_B$ | Moi.$_L$ | Best |
|---|---|---|---|---|---|---|---|---|---|---|---|---|---|---|---|---|---|---|---|---|---|---|---|---|---|---|
| 10S | sMAPE | 1.57 | 1.00 | 1.00 | **4.49e⁻¹** | 1.19 | 9.12e⁻¹ | 1.46 | 8.48e⁻¹ | 7.45e⁻¹ | 6.59e⁻¹ | 6.47e⁻¹ | 1.08 | 1.53 | 1.34 | 1.67 | 1.34 | 1.80 | 7.21e⁻¹ | 1.21 | 1.19 | 1.16 | 1.34 | 1.78 | 1.24 | A.Th. |
| | MASE | 1.98 | 1.00 | 1.00 | **1.59e⁻¹** | 5.51e⁻¹ | 3.76e⁻¹ | 5.37e⁻¹ | 3.23e⁻¹ | 2.71e⁻¹ | 2.24e⁻¹ | 2.35e⁻¹ | 3.68e⁻¹ | 6.15e⁻¹ | 5.17e⁻¹ | 7.28e⁻¹ | 1.34 | 7.87e⁻¹ | 2.16e⁻¹ | 5.23e⁻¹ | 5.23e⁻¹ | 5.06e⁻¹ | 7.95e⁻¹ | 8.41e⁻¹ | 5.72e⁻¹ | A.Th. |
| | ND | 1.24 | 1.00 | 1.00 | **3.51e⁻¹** | 1.01 | 8.02e⁻¹ | 6.79e⁻¹ | 7.36e⁻¹ | 5.62e⁻¹ | 5.83e⁻¹ | 5.44e⁻¹ | 7.35e⁻¹ | 6.86e⁻¹ | 9.00e⁻¹ | 1.44 | 1.17 | 1.35 | 6.50e⁻¹ | 8.03e⁻¹ | 8.72e⁻¹ | 8.27e⁻¹ | 1.34 | 1.20 | 1.12 | A.Th. |
| | MSE | 1.29 | 1.00 | 1.00 | **1.03e⁻¹** | 1.06 | 5.01e⁻¹ | 4.60e⁻¹ | 4.31e⁻¹ | 3.02e⁻¹ | 3.59e⁻¹ | 2.66e⁻¹ | 9.92e⁻¹ | 3.96e⁻¹ | 6.00e⁻¹ | 1.08 | 1.15 | 1.17 | 4.44e⁻¹ | 5.85e⁻¹ | 6.02e⁻¹ | 5.63e⁻¹ | 1.46 | 1.06 | 1.07 | A.Th. |
| | MAE | 1.26 | 1.00 | 1.00 | **3.51e⁻¹** | 1.01 | 8.02e⁻¹ | 6.78e⁻¹ | 7.36e⁻¹ | 5.61e⁻¹ | 5.83e⁻¹ | 5.43e⁻¹ | 7.35e⁻¹ | 6.85e⁻¹ | 9.00e⁻¹ | 1.44 | 1.17 | 1.35 | 6.48e⁻¹ | 8.03e⁻¹ | 8.72e⁻¹ | 8.27e⁻¹ | 1.34 | 1.20 | 1.12 | A.Th. |
| | CRPS | 1.44 | 1.00 | 1.00 | **3.15e⁻¹** | 1.93 | 7.54e⁻¹ | 6.72e⁻¹ | 7.05e⁻¹ | 5.98e⁻¹ | 5.36e⁻¹ | 5.10e⁻¹ | 7.82e⁻¹ | 7.29e⁻¹ | 9.58e⁻¹ | 1.53 | 1.12 | 1.30 | 6.91e⁻¹ | 7.93e⁻¹ | 8.59e⁻¹ | 8.18e⁻¹ | 1.24 | 1.06 | 1.02 | A.Th. |
| | Rank | 1.93e¹ | 1.13e¹ | 1.03e¹ | 1.00 | 2.58e¹ | 1.23e¹ | 8.83 | 1.12e¹ | 7.17 | 5.00 | 2.50 | 1.28e¹ | 1.22e¹ | 1.65e¹ | 2.55e¹ | 1.92e¹ | 2.53e¹ | 1.08e¹ | 1.12e¹ | 1.33e¹ | 1.23e¹ | 2.26e¹ | 1.95e¹ | 1.78e¹ | A.Th. |
| 5T | sMAPE | **9.24e⁻¹** | 1.00 | 1.00 | 1.32 | 9.56e⁻¹ | 1.56 | 1.15 | 1.26 | 1.17 | 1.11 | 1.13 | 1.30 | 1.40 | 1.27 | 1.35 | 1.34 | 1.14 | 1.16 | 1.24 | 1.24 | 1.25 | 1.07 | 1.01 | 9.75e⁻¹ | Nv. |
| | MASE | 9.42e⁻¹ | 1.00 | 1.00 | 9.84e⁻¹ | 9.10e⁻¹ | 1.40 | 8.36e⁻¹ | 9.61e⁻¹ | 8.84e⁻¹ | 7.87e⁻¹ | 7.73e⁻¹ | 1.16 | 1.45 | 9.59e⁻¹ | 1.22 | 1.02 | 2.38 | 8.19e⁻¹ | 8.72e⁻¹ | 8.62e⁻¹ | 8.69e⁻¹ | 7.39e⁻¹ | 6.89e⁻¹ | **6.69e⁻¹** | Moi.$_L$ |
| | ND | 7.99e⁻¹ | 1.00 | 1.00 | 8.85e⁻¹ | 9.13e⁻¹ | 8.83e⁻¹ | 6.95e⁻¹ | 7.93e⁻¹ | 7.39e⁻¹ | 6.54e⁻¹ | 6.63e⁻¹ | 8.26e⁻¹ | 7.30e⁻¹ | 8.76e⁻¹ | 8.59e⁻¹ | 9.84e⁻¹ | 8.03e⁻¹ | 7.43e⁻¹ | 7.72e⁻¹ | 7.68e⁻¹ | 7.75e⁻¹ | 6.32e⁻¹ | **6.16e⁻¹** | Moi.$_L$ |
| | MSE | 7.23e⁻¹ | 1.00 | 1.00 | 7.41e⁻¹ | 8.79e⁻¹ | 7.85e⁻¹ | 5.12e⁻¹ | 5.76e⁻¹ | 5.65e⁻¹ | 4.81e⁻¹ | **4.56e⁻¹** | 5.55e⁻¹ | 5.49e⁻¹ | 6.30e⁻¹ | 6.12e⁻¹ | 9.02e⁻¹ | 6.38e⁻¹ | 5.36e⁻¹ | 6.90e⁻¹ | 6.99e⁻¹ | 7.04e⁻¹ | 5.00e⁻¹ | 5.09e⁻¹ | 4.69e⁻¹ | iTr. |
| | MAE | 7.99e⁻¹ | 1.00 | 1.00 | 8.84e⁻¹ | 9.13e⁻¹ | 8.82e⁻¹ | 6.96e⁻¹ | 7.93e⁻¹ | 7.38e⁻¹ | 6.54e⁻¹ | 6.63e⁻¹ | 8.26e⁻¹ | 8.68e⁻¹ | 8.75e⁻¹ | 8.59e⁻¹ | 9.84e⁻¹ | 8.03e⁻¹ | 7.43e⁻¹ | 7.71e⁻¹ | 7.69e⁻¹ | 7.75e⁻¹ | 6.31e⁻¹ | **6.16e⁻¹** | Moi.$_L$ |
| | CRPS | 1.19 | 1.00 | 1.00 | 9.48e⁻¹ | 9.58e⁻¹ | 7.49e⁻¹ | 5.36e⁻¹ | 6.31e⁻¹ | 6.99e⁻¹ | 5.22e⁻¹ | 5.22e⁻¹ | 7.81e⁻¹ | 6.90e⁻¹ | 8.28e⁻¹ | 8.12e⁻¹ | 7.95e⁻¹ | 6.73e⁻¹ | 7.02e⁻¹ | 6.82e⁻¹ | 6.83e⁻¹ | 6.87e⁻¹ | 4.96e⁻¹ | **4.61e⁻¹** | Moi.$_L$ |
| | Rank | 2.34e¹ | 2.39e¹ | 2.24e¹ | 2.28e¹ | 2.17e¹ | 1.77e¹ | 6.58 | 1.33¹ | 1.64¹ | 6.75 | 7.75 | 1.92¹ | 1.62¹ | 2.18¹ | 2.02¹ | 1.96¹ | 1.52¹ | 1.63¹ | 1.48¹ | 1.51¹ | 1.58¹ | 7.44 | 6.42 | 4.58 | Moi.$_L$ |
| 10T | sMAPE | 1.68 | 1.00 | 1.00 | 1.89 | 1.97 | 2.33 | 1.86 | 1.92 | 2.06 | 1.91 | 1.96 | 1.84 | 2.29 | 1.93 | 1.80 | 2.18 | 1.78 | 1.81 | 1.98 | 1.93 | 1.94 | 2.01 | 2.11 | 2.08 | S.Nv. |
| | MASE | 1.28 | 1.00 | 1.00 | 1.62 | 1.77 | 1.55 | 9.42e⁻¹ | 1.27 | 1.21 | 1.19 | 1.09 | 1.45 | 2.22 | 1.44 | 9.47e⁻¹ | 1.48 | 1.27 | **9.12e⁻¹** | 1.20 | 1.09 | 1.07 | 1.00 | 1.15 | 1.13 | V.TS |
| | ND | 1.51 | 1.00 | 1.00 | 2.11 | 1.87 | 1.39 | 1.01 | 1.28 | 1.55 | 1.16 | 1.21 | 1.35 | 1.29 | 1.75 | 1.03 | 1.61 | 1.17 | **9.95e⁻¹** | 1.41 | 1.22 | 1.21 | 1.20 | 1.31 | 1.34 | V.TS |
| | MSE | 1.57 | 1.00 | 1.00 | 1.67 | 2.99 | 1.36 | 7.35e⁻¹ | 1.40 | 1.55 | 9.35e⁻¹ | 1.17 | 8.25e⁻¹ | 1.23 | 1.44 | **6.88e⁻¹** | 1.77 | 1.06 | 8.02e⁻¹ | 1.50 | 1.19 | 1.18 | 1.29 | 1.51 | 1.48 | TTM |
| | MAE | 1.49 | 1.00 | 1.00 | 9.78e⁻¹ | 1.06 | 1.56 | 9.34e⁻¹ | 1.00 | 1.02 | 8.74e⁻¹ | **8.68e⁻¹** | 9.83e⁻¹ | 2.07 | 1.30 | 1.05 | 1.36 | 9.54e⁻¹ | 9.09e⁻¹ | 9.13e⁻¹ | 8.82e⁻¹ | 8.78e⁻¹ | 9.42e⁻¹ | 9.23e⁻¹ | 9.64e⁻¹ | iTr. |
| | CRPS | 2.20 | 1.00 | 1.00 | 9.52e⁻¹ | 1.51 | 1.19e² | 1.26 | 7.08e⁻¹ | 7.92e⁻¹ | 9.63e⁻¹ | 6.55e⁻¹ | **6.51e⁻¹** | 9.26e⁻¹ | 1.34 | 1.22 | 9.89e⁻¹ | 1.02 | 7.68e⁻¹ | 8.56e⁻¹ | 7.73e⁻¹ | 7.49e⁻¹ | 7.46e⁻¹ | 7.39e⁻¹ | 6.91e⁻¹ | 7.20e⁻¹ | iTr. |
| | Rank | 2.67e¹ | 2.22e¹ | 2.12e¹ | 2.80e¹ | 2.42e¹ | 1.47e¹ | 5.67 | 1.65e¹ | 1.82e¹ | 9.50 | 8.00 | 1.60¹ | 5.72e⁻¹ | 2.20¹ | 1.03e¹ | 1.77¹ | 1.00¹ | 9.33 | 1.55¹ | 1.06¹ | 9.67 | 1.10¹ | 1.28e¹ | 1.30e¹ | TFT |
| 15T | sMAPE | 1.41 | 1.00 | 9.78e⁻¹ | 1.11 | 1.27 | 1.66 | 1.01 | 1.06 | 1.06 | 9.18e⁻¹ | **9.15e⁻¹** | 1.02 | 1.81 | 1.23 | 1.07 | 1.23 | 9.98e⁻¹ | 9.52e⁻¹ | 1.02 | 9.68e⁻¹ | 9.62e⁻¹ | 9.87e⁻¹ | 9.72e⁻¹ | 1.00 | iTr. |
| | MASE | 1.52 | 1.00 | 9.78e⁻¹ | 1.03 | 1.61 | 1.76 | 9.66e⁻¹ | 1.02 | 1.02 | **8.77e⁻¹** | 9.92e⁻¹ | 2.76 | 1.24 | 1.02 | 1.27 | 9.56e⁻¹ | 9.05e⁻¹ | 9.20e⁻¹ | 8.87e⁻¹ | 8.85e⁻¹ | 9.49e⁻¹ | 9.25e⁻¹ | 9.77e⁻¹ | iTr. |
| | ND | 1.50 | 1.00 | 9.78e⁻¹ | 1.06 | 1.56 | 1.63 | 9.33e⁻¹ | 1.00 | 1.02 | **8.74e⁻¹** | **8.67e⁻¹** | 9.82e⁻¹ | 1.43 | 1.30 | 1.05 | 1.36 | 9.54e⁻¹ | 9.09e⁻¹ | 9.13e⁻¹ | 8.82e⁻¹ | 8.78e⁻¹ | 9.42e⁻¹ | 9.22e⁻¹ | 9.63e⁻¹ | iTr. |
| | MSE | 1.99 | 1.00 | 9.59e⁻¹ | 1.10 | 3.19 | 2.04 | 7.99e⁻¹ | 9.00e⁻¹ | 9.99e⁻¹ | 7.39e⁻¹ | **7.26e⁻¹** | 8.71e⁻¹ | 2.95 | 1.56 | 9.79e⁻¹ | 1.89 | 8.61e⁻¹ | 8.13e⁻¹ | 8.51e⁻¹ | 8.05e⁻¹ | 8.06e⁻¹ | 8.67e⁻¹ | 8.98e⁻¹ | 9.09e⁻¹ | iTr. |
| | MAE | 1.49 | 1.00 | 9.78e⁻¹ | 1.06 | 1.56 | 1.64 | 9.34e⁻¹ | 1.00 | 1.02 | **8.74e⁻¹** | **8.68e⁻¹** | 9.83e⁻¹ | 2.07 | 1.30 | 1.05 | 1.36 | 9.54e⁻¹ | 9.09e⁻¹ | 9.13e⁻¹ | 8.82e⁻¹ | 8.78e⁻¹ | 9.42e⁻¹ | 9.23e⁻¹ | 9.64e⁻¹ | iTr. |
| | CRPS | 2.20 | 1.00 | 9.52e⁻¹ | 1.51 | 1.19e² | 1.26 | 7.08e⁻¹ | 7.92e⁻¹ | 9.63e⁻¹ | **6.55e⁻¹** | **6.51e⁻¹** | 9.26e⁻¹ | 1.34 | 1.22 | 9.89e⁻¹ | 1.02 | 7.68e⁻¹ | 8.56e⁻¹ | 7.73e⁻¹ | 7.49e⁻¹ | 7.46e⁻¹ | 7.39e⁻¹ | 6.91e⁻¹ | 7.20e⁻¹ | iTr. |
| | Rank | 2.73e¹ | 2.03e¹ | 1.91e¹ | 2.38e¹ | 2.47e¹ | 1.97e¹ | 8.67 | 1.13e¹ | 1.08e¹ | 9.00 | 5.00 | 4.67 | 1.94e¹ | 2.36e¹ | 1.96e¹ | 1.78e¹ | 1.07e¹ | 1.73e¹ | 1.28e¹ | 1.06e¹ | 9.00 | 6.17 | 9.58 | iTr. |
| H | sMAPE | 1.46 | 1.00 | 1.21 | 1.40 | 1.32 | 1.34 | 1.06 | 1.13 | 1.08 | 1.00 | 1.03 | 1.15 | 1.61 | 1.13 | 1.06 | 1.12 | 9.88e⁻¹ | 1.01 | 9.74e⁻¹ | **9.65e⁻¹** | 9.66e⁻¹ | 1.09 | 9.91e⁻¹ | 9.84e⁻¹ | Chr.$_B$ |
| | MASE | 1.46 | 1.00 | 1.02 | 1.28 | 1.27 | 1.31 | 8.25e⁻¹ | 9.59e⁻¹ | 8.72e⁻¹ | 7.74e⁻¹ | 8.05e⁻¹ | 9.43e⁻¹ | 1.73 | 9.41e⁻¹ | 8.52e⁻¹ | 8.95e⁻¹ | 8.24e⁻¹ | 7.70e⁻¹ | 7.73e⁻¹ | **7.63e⁻¹** | 7.63e⁻¹ | 8.92e⁻¹ | 7.78e⁻¹ | 7.70e⁻¹ | Chr.$_B$ |
| | ND | 1.43 | 1.00 | 1.04 | 1.39 | 1.21 | 1.14 | 7.95e⁻¹ | 8.99e⁻¹ | 8.89e⁻¹ | **7.57e⁻¹** | 7.90e⁻¹ | 9.07e⁻¹ | 8.00e⁻¹ | 9.47e⁻¹ | 8.52e⁻¹ | 8.92e⁻¹ | 8.27e⁻¹ | 7.86e⁻¹ | 7.87e⁻¹ | 7.77e⁻¹ | 7.80e⁻¹ | 8.62e⁻¹ | 7.66e⁻¹ | 7.58e⁻¹ | P.TST |
| | MSE | 1.85 | 1.00 | 1.04 | 1.48 | 1.35 | 1.15 | 6.12e⁻¹ | 6.96e⁻¹ | 7.64e⁻¹ | **5.79e⁻¹** | 6.28e⁻¹ | 6.80e⁻¹ | 6.50e⁻¹ | 7.81e⁻¹ | 6.18e⁻¹ | 7.76e⁻¹ | 6.52e⁻¹ | 6.06e⁻¹ | 5.34e⁻¹ | 6.34e⁻¹ | 6.54e⁻¹ | 7.48e⁻¹ | 6.17e⁻¹ | 5.97e⁻¹ | P.TST |
| | MAE | 1.43 | 1.00 | 1.04 | 1.39 | 1.21 | 1.14 | 7.95e⁻¹ | 8.99e⁻¹ | 8.99e⁻¹ | **7.57e⁻¹** | 7.90e⁻¹ | 9.06e⁻¹ | 1.08 | 9.47e⁻¹ | 8.51e⁻¹ | 8.92e⁻¹ | 8.27e⁻¹ | 7.86e⁻¹ | 7.77e⁻¹ | 7.77e⁻¹ | 7.79e⁻¹ | 8.62e⁻¹ | 7.66e⁻¹ | 7.58e⁻¹ | P.TST |
| | CRPS | 1.67 | 1.00 | 7.43e⁻¹ | 1.57 | 5.01e⁻¹ | 6.23e⁻¹ | 4.28e⁻¹ | 5.11e⁻¹ | 6.00e⁻¹ | **4.07e⁻¹** | 4.24e⁻¹ | 6.06e⁻¹ | 5.34e⁻¹ | 6.33e⁻¹ | 5.69e⁻¹ | 4.89e⁻¹ | 4.69e⁻¹ | 5.25e⁻¹ | 4.68e⁻¹ | 4.62e⁻¹ | 4.64e⁻¹ | 5.13e⁻¹ | 4.13e⁻¹ | 4.07e⁻¹ | P.TST |
| | Rank | 2.75e¹ | 2.48e¹ | 2.20e¹ | 2.66e¹ | 2.64e¹ | 1.52e¹ | 8.77 | 1.44e¹ | 1.85e¹ | 6.97 | 8.32 | 1.94e¹ | 1.53e¹ | 1.99e¹ | 1.83e¹ | 1.36e¹ | 1.16e¹ | 1.64e¹ | 1.18e¹ | 1.10e¹ | 1.12e¹ | 1.13e¹ | 5.42 | 5.23 | Moi.$_L$ |
| D | sMAPE | 1.00 | 1.00 | 9.37e⁻¹ | 9.98e⁻¹ | 8.95e⁻¹ | 1.05 | 8.14e⁻¹ | 1.11 | 8.40e⁻¹ | 8.00e⁻¹ | 9.24e⁻¹ | 9.56e⁻¹ | 2.00 | 1.05 | 1.00 | 1.30 | 8.26e⁻¹ | 8.95e⁻¹ | 8.39e⁻¹ | **8.13e⁻¹** | 8.94e⁻¹ | 8.55e⁻¹ | 8.75e⁻¹ | Chr.$_B$ |
| | MASE | 1.00 | 1.00 | 8.35e⁻¹ | 9.98e⁻¹ | 8.05e⁻¹ | 9.06e⁻¹ | 7.25e⁻¹ | 1.15 | 7.75e⁻¹ | 7.49e⁻¹ | 8.31e⁻¹ | 8.87e⁻¹ | 4.83 | 9.82e⁻¹ | 9.44e⁻¹ | 1.19 | 7.46e⁻¹ | 8.22e⁻¹ | 7.37e⁻¹ | **7.14e⁻¹** | 7.83e⁻¹ | 7.47e⁻¹ | 7.66e⁻¹ | Chr.$_B$ |
| | ND | 1.00 | 1.00 | 8.24e⁻¹ | 9.08e⁻¹ | 8.33e⁻¹ | 9.12e⁻¹ | 6.87e⁻¹ | 1.09 | 7.96e⁻¹ | 7.29e⁻¹ | 8.10e⁻¹ | 8.25e⁻¹ | 5.46 | 9.68e⁻¹ | 8.91e⁻¹ | 1.18 | 7.77e⁻¹ | 7.66e⁻¹ | 7.19e⁻¹ | 6.81e⁻¹ | **6.79e⁻¹** | 7.49e⁻¹ | 7.34e⁻¹ | 7.46e⁻¹ | Chr.$_B$ |
| | MSE | 1.00 | 1.00 | 6.64e⁻¹ | 7.39e⁻¹ | 6.70e⁻¹ | 8.25e⁻¹ | 5.35e⁻¹ | 1.10 | 6.92e⁻¹ | 5.43e⁻¹ | 7.03e⁻¹ | 6.22e⁻¹ | 2.64 | 7.99e⁻¹ | 6.64e⁻¹ | 1.18 | 6.78e⁻¹ | 5.67e⁻¹ | 5.85e⁻¹ | **5.28e⁻¹** | 5.38e⁻¹ | 5.62e⁻¹ | 5.63e⁻¹ | 6.07e⁻¹ | Chr.$_B$ |
| | MAE | 1.00 | 1.00 | 8.23e⁻¹ | 9.06e⁻¹ | 8.32e⁻¹ | 9.11e⁻¹ | 6.87e⁻¹ | 1.08 | 7.95e⁻¹ | 7.29e⁻¹ | 8.10e⁻¹ | 8.24e⁻¹ | 2.15 | 9.67e⁻¹ | 8.91e⁻¹ | 1.18 | 7.76e⁻¹ | 7.65e⁻¹ | 7.18e⁻¹ | **6.82e⁻¹** | 6.78e⁻¹ | 7.49e⁻¹ | 7.34e⁻¹ | 7.45e⁻¹ | Chr.$_B$ |
| | CRPS | 7.94e⁻¹ | 1.00 | 4.69e⁻¹ | 5.43e⁻¹ | 9.07e⁻¹ | 4.91e⁻¹ | **3.70e⁻¹** | 6.51e⁻¹ | 5.24e⁻¹ | 3.92e⁻¹ | 4.38e⁻¹ | 5.43e⁻¹ | 3.60 | 6.37e⁻¹ | 5.86e⁻¹ | 6.44e⁻¹ | 4.13e⁻¹ | 5.04e⁻¹ | 3.97e⁻¹ | 3.78e⁻¹ | 3.72e⁻¹ | 3.97e⁻¹ | 3.86e⁻¹ | 3.96e⁻¹ | TFT |
| | Rank | 2.48e¹ | 2.67e¹ | 1.45e¹ | 1.91e¹ | 1.93e¹ | 1.49e¹ | 8.87 | 1.82e¹ | 1.94e¹ | 9.73 | 1.18e¹ | 2.03e¹ | 2.60e¹ | 2.21e¹ | 2.10e¹ | 2.09e¹ | 7.47 | 2.17e¹ | 9.20 | 9.07 | 9.10 | 7.13 | 8.27 | Moi.$_B$ |
| W | sMAPE | 1.00 | 1.00 | 1.00 | 1.09 | 9.61e⁻¹ | 1.55 | 9.36e⁻¹ | 1.34 | 1.16 | 9.71e⁻¹ | 1.34 | 1.14 | 2.74 | 1.19 | 1.14 | 1.52 | 8.74e⁻¹ | 1.08 | 7.83e⁻¹ | 7.85e⁻¹ | 7.59e⁻¹ | 1.00 | 9.28e⁻¹ | 9.81e⁻¹ | Chr.$_L$ |
| | MASE | 1.00 | 1.00 | 9.46e⁻¹ | 1.03 | 9.32e⁻¹ | 1.46 | 9.21e⁻¹ | 1.29 | 1.08 | 9.29e⁻¹ | 1.25 | 1.14 | 4.36 | 1.14 | 1.13 | 1.63 | 8.17e⁻¹ | 1.04 | 7.45e⁻¹ | 7.62e⁻¹ | **7.37e⁻¹** | 1.00 | 9.01e⁻¹ | 9.31e⁻¹ | Chr.$_L$ |
| | ND | 1.00 | 1.00 | 9.39e⁻¹ | 1.02 | 9.24e⁻¹ | 1.36 | 9.30e⁻¹ | 1.24 | 1.04 | 8.87e⁻¹ | 1.27 | 1.02 | 1.23e¹ | 1.05 | 1.14 | 1.55 | 8.17e⁻¹ | 1.01 | 6.98e⁻¹ | 7.08e⁻¹ | **6.87e⁻¹** | 9.40e⁻¹ | 8.56e⁻¹ | 8.53e⁻¹ | Chr.$_L$ |
| | MSE | 1.00 | 1.00 | 8.67e⁻¹ | 9.75e⁻¹ | 8.00e⁻¹ | 1.64 | 9.30e⁻¹ | 1.42 | 1.02 | 7.61e⁻¹ | 1.62 | 9.20e⁻¹ | 5.56 | 9.77e⁻¹ | 1.14 | 2.30 | 7.29e⁻¹ | 9.48e⁻¹ | 4.80e⁻¹ | 4.85e⁻¹ | **4.58e⁻¹** | 8.56e⁻¹ | 7.15e⁻¹ | 7.02e⁻¹ | Chr.$_L$ |
| | MAE | 1.00 | 1.00 | 9.39e⁻¹ | 1.02 | 9.24e⁻¹ | 1.36 | 9.31e⁻¹ | 1.24 | 1.04 | 8.87e⁻¹ | 1.27 | 1.02 | 2.87 | 1.05 | 1.14 | 1.55 | 8.18e⁻¹ | 1.01 | 6.99e⁻¹ | 7.08e⁻¹ | **6.87e⁻¹** | 9.40e⁻¹ | 8.56e⁻¹ | 6.34e⁻¹ | Chr.$_L$ |
| | CRPS | 8.74e⁻¹ | 1.00 | 7.31e⁻¹ | 7.87e⁻¹ | 7.74e⁻¹ | 9.94e⁻¹ | 7.26e⁻¹ | 9.56e⁻¹ | 9.71e⁻¹ | 6.66e⁻¹ | 9.56e⁻¹ | 9.48e⁻¹ | 1.18e¹ | 8.97e⁻¹ | 1.06 | 1.20 | 5.36e⁻¹ | 5.42e⁻¹ | **5.29e⁻¹** | 6.95e⁻¹ | 6.37e⁻¹ | 6.34e⁻¹ | Chr.$_L$ |
| | Rank | 1.81e¹ | 2.20e¹ | 1.32e¹ | 1.60e¹ | 1.46e¹ | 1.69e¹ | 7.91 | 1.49e¹ | 1.62e¹ | 1.02e¹ | 1.62¹ | 2.15e¹ | 2.88e¹ | 2.21e¹ | 2.29e¹ | 2.39e¹ | 6.12 | 2.10e¹ | 6.75 | 5.62 | 1.12e¹ | 6.88 | 6.88 | Chr.$_L$ |
| M | sMAPE | 1.21 | 1.00 | **8.35e⁻¹** | 9.59e⁻¹ | 8.62e⁻¹ | 1.30 | 1.03 | 1.08 | 9.46e⁻¹ | 9.70e⁻¹ | 9.86e⁻¹ | 1.08 | 1.72 | 1.35 | 1.15 | 1.46 | 9.04e⁻¹ | 9.65e⁻¹ | 9.21e⁻¹ | 9.76e⁻¹ | 9.27e⁻¹ | 1.13 | 9.15e⁻¹ | 1.52 | A.Ar. |
| | MASE | 1.20 | 1.00 | **7.59e⁻¹** | 9.32e⁻¹ | 8.21e⁻¹ | 1.22 | 9.01e⁻¹ | 1.10 | 8.51e⁻¹ | 8.59e⁻¹ | 9.07e⁻¹ | 9.97e⁻¹ | 2.95 | 1.21 | 1.18 | 1.43 | **8.00e⁻¹** | 9.15e⁻¹ | 8.27e⁻¹ | 8.57e⁻¹ | 8.12e⁻¹ | 1.04 | 8.07e⁻¹ | 8.17e⁻¹ | A.Ar. |
| | ND | 1.24 | 1.00 | **7.88e⁻¹** | 1.05 | 8.46e⁻¹ | 1.15 | 1.09 | 8.32e⁻¹ | 1.24 | 7.92e⁻¹ | 8.57e⁻¹ | 7.46e⁻¹ | 5.05 | 1.30 | 1.26 | 1.44 | 8.29e⁻¹ | 7.22e⁻¹ | 7.81e⁻¹ | 7.78e⁻¹ | 9.09e⁻¹ | 8.63e⁻¹ | 1.16 | 7.22e⁻¹ | 6.73e⁻¹ | A.Ar. |
| | MSE | 1.58 | 1.00 | **6.06e⁻¹** | 8.20e⁻¹ | 6.40e⁻¹ | 1.09 | 8.32e⁻¹ | 1.24 | 7.92e⁻¹ | 8.57e⁻¹ | 7.46e⁻¹ | 9.68e⁻¹ | 3.30 | 1.69 | 1.48 | 1.99 | 6.16e⁻¹ | 7.87e⁻¹ | 7.31e⁻¹ | 7.33e⁻¹ | 7.53e⁻¹ | 1.16 | 7.22e⁻¹ | 6.73e⁻¹ | A.Ar. |
| | MAE | 1.24 | 1.00 | **7.89e⁻¹** | 9.55e⁻¹ | 8.46e⁻¹ | 1.13 | 9.12e⁻¹ | 1.15 | 8.62e⁻¹ | 8.85e⁻¹ | 8.64e⁻¹ | 1.02 | 2.12 | 1.30 | 1.26 | 1.41 | 8.29e⁻¹ | 9.22e⁻¹ | 8.17e⁻¹ | 9.08e⁻¹ | 8.60e⁻¹ | 1.09 | 8.21e⁻¹ | 8.40e⁻¹ | A.Ar. |
| | CRPS | 1.52 | 1.00 | 7.59e⁻¹ | 8.73e⁻¹ | 7.70e⁻¹ | 1.03 | 8.40e⁻¹ | 1.16 | 9.62e⁻¹ | 8.32e⁻¹ | 9.03e⁻¹ | 1.17 | 5.64 | 1.45 | 1.41 | 1.35 | **7.33e⁻¹** | 1.03 | 8.18e⁻¹ | 8.49e⁻¹ | 8.07e⁻¹ | 9.93e⁻¹ | 7.51e⁻¹ | 7.25e⁻¹ | T.FM |
| | Rank | 2.52e¹ | 1.80e¹ | 8.60 | 1.16e¹ | 7.80 | 1.56e¹ | 1.02e¹ | 2.00e¹ | 1.44e¹ | 1.00e¹ | 7.40 | 2.06e¹ | 1.84e¹ | 2.24e¹ | 2.28e¹ | 1.90e¹ | 4.80 | 1.90e¹ | 1.06e¹ | 1.16e¹ | 1.04e¹ | 1.67e¹ | 4.20 | 7.00 | Moi.$_B$ |
| Q | sMAPE | 9.28e⁻¹ | 1.00 | 8.72e⁻¹ | 8.40e⁻¹ | 8.16e⁻¹ | 8.96e⁻¹ | 7.64e⁻¹ | 1.02 | 8.40e⁻¹ | 8.72e⁻¹ | 8.48e⁻¹ | 9.36e⁻¹ | 6.06 | 1.66 | 1.14 | 2.16 | 9.12e⁻¹ | 9.96e⁻¹ | 9.21e⁻¹ | 9.76e⁻¹ | 9.27e⁻¹ | 1.13 | 9.05e⁻¹ | 9.27e⁻¹ | Moi.$_B$ |
| | MASE | 9.28e⁻¹ | 1.00 | 8.00e⁻¹ | 7.44e⁻¹ | 7.25e⁻¹ | 9.08e⁻¹ | 8.12e⁻¹ | 1.05 | 7.56e⁻¹ | 8.25e⁻¹ | 7.69e⁻¹ | 9.12e⁻¹ | 1.59e¹ | 1.84 | 1.24 | 3.73 | 8.75e⁻¹ | 8.50e⁻¹ | 7.75e⁻¹ | 7.69e⁻¹ | 7.69e⁻¹ | 7.76e⁻¹ | 9.12e⁻¹ | **7.11e⁻¹** | Moi.$_B$ |
| | ND | 9.24e⁻¹ | 1.00 | 8.31e⁻¹ | 8.02e⁻¹ | 8.03e⁻¹ | 8.66e⁻¹ | 8.57e⁻¹ | 1.01 | 8.10e⁻¹ | 8.66e⁻¹ | 8.24e⁻¹ | 9.24e⁻¹ | 1.00e² | 1.49 | 1.08 | 2.66 | 8.82e⁻¹ | 8.74e⁻¹ | 8.27e⁻¹ | 8.21e⁻¹ | 8.22e⁻¹ | 8.25e⁻¹ | **7.71e⁻¹** | 7.71e⁻¹ | Moi.$_B$ |
| | MSE | 9.12e⁻¹ | 1.00 | 8.76e⁻¹ | 7.25e⁻¹ | 8.39e⁻¹ | 8.18e⁻¹ | 7.67e⁻¹ | 8.31e⁻¹ | 9.28e⁻¹ | 7.55e⁻¹ | 7.79e⁻¹ | 7.47e⁻¹ | 8.19e⁻¹ | 1.63 | 8.71e⁻¹ | 4.46 | 8.55e⁻¹ | 7.99e⁻¹ | 8.11e⁻¹ | 7.99e⁻¹ | 7.62e⁻¹ | 7.52e⁻¹ | **6.93e⁻¹** | 6.93e⁻¹ | Moi.$_B$ |
| | MAE | 9.25e⁻¹ | 1.00 | 8.34e⁻¹ | 8.04e⁻¹ | 8.05e⁻¹ | 8.67e⁻¹ | 8.62e⁻¹ | 1.02 | 8.12e⁻¹ | 8.89e⁻¹ | 8.24e⁻¹ | 9.27e⁻¹ | 6.30 | 1.50 | 1.08 | 2.67 | 8.84e⁻¹ | 8.79e⁻¹ | 8.28e⁻¹ | 8.24e⁻¹ | 8.20e⁻¹ | 8.27e⁻¹ | **7.74e⁻¹** | 7.74e⁻¹ | Moi.$_B$ |
| | CRPS | 9.51e⁻¹ | 1.00 | 8.23e⁻¹ | 7.97e⁻¹ | 7.96e⁻¹ | 8.41e⁻¹ | 8.37e⁻¹ | 1.02 | 9.72e⁻¹ | 8.35e⁻¹ | 7.79e⁻¹ | 7.11 | 1.20e² | 1.78 | 1.29 | 2.52 | 8.53e⁻¹ | 1.05 | 8.46e⁻¹ | 8.39e⁻¹ | 8.40e⁻¹ | 7.94e⁻¹ | **7.40e⁻¹** | 7.40e⁻¹ | Moi.$_B$ |
| | Rank | 1.80e¹ | 2.00e¹ | 9.00 | 6.00 | 8.00 | 1.40e¹ | 1.10e¹ | 2.10e¹ | 1.90e¹ | 1.00e¹ | 7.00 | 2.30e¹ | 3.00e¹ | 2.60e¹ | 2.50e¹ | 2.70e¹ | 1.60e¹ | 2.20e¹ | 1.50e¹ | 1.20e¹ | 1.30e¹ | 4.50 | 1.00 | 2.00 | Moi.$_B$ |
| A | sMAPE | 1.00 | 1.00 | 9.88e⁻¹ | 8.84e⁻¹ | 8.54e⁻¹ | 8.66e⁻¹ | 8.60e⁻¹ | 1.17 | 8.54e⁻¹ | 8.72e⁻¹ | 8.78e⁻¹ | 1.04 | 4.30 | 3.12 | 1.24 | 1.98 | 9.09e⁻¹ | 1.00 | 1.03 | 1.00 | 1.00 | **8.18e⁻¹** | 8.19e⁻¹ | 8.11e⁻¹ | Moi.$_L$ |
| | MASE | 1.00 | 1.00 | 9.35e⁻¹ | 7.83e⁻¹ | 7.76e⁻¹ | 8.36e⁻¹ | 8.56e⁻¹ | 1.16 | 7.93e⁻¹ | 8.29e⁻¹ | 8.49e⁻¹ | 1.05 | 7.91 | 2.90 | 1.29 | 2.41 | 9.65e⁻¹ | 9.55e⁻¹ | 9.75e⁻¹ | 9.51e⁻¹ | 9.51e⁻¹ | 7.51e⁻¹ | 7.58e⁻¹ | **7.49e⁻¹** | Moi.$_L$ |
| | ND | 1.00 | 1.00 | 9.65e⁻¹ | 8.40e⁻¹ | 8.33e⁻¹ | 8.52e⁻¹ | 9.43e⁻¹ | 1.14 | 8.27e⁻¹ | 8.77e⁻¹ | 8.53e⁻¹ | 1.04 | 8.77e⁻¹ | 2.49 | 1.19 | 2.06 | 8.89e⁻¹ | 9.61e⁻¹ | 9.75e⁻¹ | 9.51e⁻¹ | 9.51e⁻¹ | **8.05e⁻¹** | 7.99e⁻¹ | 7.99e⁻¹ | Moi.$_L$ |
| | MSE | 1.00 | 1.00 | 1.10 | 8.23e⁻¹ | 8.38e⁻¹ | 8.23e⁻¹ | 8.43e⁻¹ | 1.02 | 7.99e⁻¹ | 9.14e⁻¹ | 8.72e⁻¹ | 1.03 | 7.91 | 3.93 | 1.19 | 2.92 | 9.19e⁻¹ | 1.01 | 1.01 | 9.78e⁻¹ | 9.80e⁻¹ | 7.62e⁻¹ | 7.69e⁻¹ | **7.53e⁻¹** | Moi.$_L$ |
| | MAE | 1.00 | 1.00 | 9.62e⁻¹ | 8.41e⁻¹ | 8.33e⁻¹ | 8.50e⁻¹ | 8.36e⁻¹ | 1.14 | 8.30e⁻¹ | 8.78e⁻¹ | 8.84e⁻¹ | 1.04 | 4.10 | 2.50 | 1.19 | 2.05 | 8.86e⁻¹ | 9.83e⁻¹ | 9.77e⁻¹ | 9.51e⁻¹ | 9.53e⁻¹ | **8.05e⁻¹** | 8.06e⁻¹ | 7.99e⁻¹ | Moi.$_L$ |
| | CRPS | 9.93e⁻¹ | 1.00 | 9.42e⁻¹ | 8.33e⁻¹ | 8.04e⁻¹ | 8.19e⁻¹ | 7.97e⁻¹ | 1.12 | 9.71e⁻¹ | 8.48e⁻¹ | 8.48e⁻¹ | 1.22 | 1.03e² | 2.92 | 1.39 | 2.04 | 8.46e⁻¹ | 1.15 | 1.01 | 9.78e⁻¹ | 9.78e⁻¹ | 7.64e⁻¹ | **7.62e⁻¹** | 7.57e⁻¹ | Moi.$_L$ |
| | Rank | 1.90e¹ | 2.00e¹ | 1.40e¹ | 1.00e¹ | 7.00 | 8.00 | 6.00 | 2.20e¹ | 1.60e¹ | 1.20e¹ | 1.10e¹ | 2.40e¹ | 3.00e¹ | 2.70e¹ | 2.50e¹ | 2.60e¹ | 1.30e¹ | 2.30e¹ | 2.10e¹ | 1.70e¹ | 1.80e¹ | 3.50 | 2.00 | 1.00 | Moi.$_L$ |

Table 20: Results on GIFT-Eval aggregated by number of variates. The best results across each row are **bolded**, while second best results are underlined.

| Num. Var. | Metric | Nv. | S.Nv. | A.Ar. | A.Th. | A.ETS | D.AR | TFT | TiDE | N-B. | P.TST | iTr. | DLin. | C.former | Timer | TTM | L-Llama | T.FM | V.TS | Chr.$_S$ | Chr.$_B$ | Chr.$_L$ | Moi.$_S$ | Moi.$_B$ | Moi.$_L$ | Best |
|---|---|---|---|---|---|---|---|---|---|---|---|---|---|---|---|---|---|---|---|---|---|---|---|---|---|---|
| False | sMAPE | 1.15 | 1.00 | 1.12 | 1.14 | 1.20 | 1.74 | 1.23 | 1.29 | 1.13 | 1.05 | 1.12 | 1.24 | 1.39 | 1.28 | 1.25 | 1.34 | 1.20 | 1.08 | 1.16 | 1.15 | 1.18 | 1.06 | 1.06 | 1.30 | S.Nv. |
| | MASE | 1.15 | 1.00 | 1.03 | 8.01e⁻¹ | 1.05 | 1.50 | 8.40e⁻¹ | 1.01 | 7.82e⁻¹ | 7.11e⁻¹ | 7.37e⁻¹ | 9.63e⁻¹ | 1.16 | 8.95e⁻¹ | 9.30e⁻¹ | 9.57e⁻¹ | 1.17 | **6.05e⁻¹** | 8.04e⁻¹ | 7.94e⁻¹ | 7.88e⁻¹ | 8.44e⁻¹ | 8.31e⁻¹ | 8.11e⁻¹ | V.TS |
| | ND | 1.05 | 1.00 | 9.56e⁻¹ | 1.10 | 1.30 | 8.19e⁻¹ | 1.03 | 8.78e⁻¹ | **7.45e⁻¹** | 7.93e⁻¹ | 9.08e⁻¹ | 9.17e⁻¹ | 1.06 | 9.23e⁻¹ | 9.51e⁻¹ | 1.06 | 9.23e⁻¹ | 8.01e⁻¹ | 8.45e⁻¹ | 8.43e⁻¹ | 8.36e⁻¹ | 8.72e⁻¹ | 8.74e⁻¹ | 8.89e⁻¹ | P.TST |
| | MSE | 1.08 | 1.00 | 1.06 | 7.87e⁻¹ | 1.28 | 1.42 | 6.42e⁻¹ | 8.52e⁻¹ | 7.18e⁻¹ | **5.60e⁻¹** | 6.24e⁻¹ | 6.38e⁻¹ | 6.68e⁻¹ | 7.96e⁻¹ | 7.24e⁻¹ | 1.04 | 7.88e⁻¹ | 6.20e⁻¹ | 7.55e⁻¹ | 7.45e⁻¹ | 7.36e⁻¹ | 7.92e⁻¹ | 7.80e⁻¹ | 8.08e⁻¹ | P.TST |
| | MAE | 1.05 | 1.00 | 1.04 | 9.57e⁻¹ | 1.10 | 1.30 | 8.20e⁻¹ | 1.03 | 8.77e⁻¹ | **7.45e⁻¹** | 7.92e⁻¹ | 9.08e⁻¹ | 9.17e⁻¹ | 1.06 | 9.48e⁻¹ | 1.06 | 9.24e⁻¹ | 8.01e⁻¹ | 8.45e⁻¹ | 8.36e⁻¹ | 8.36e⁻¹ | 8.72e⁻¹ | 8.53e⁻¹ | 8.89e⁻¹ | P.TST |
| | CRPS | 1.26 | 1.00 | 8.37e⁻¹ | 9.26e⁻¹ | 4.05 | 4.92e⁻¹ | 6.59e⁻¹ | 6.41e⁻¹ | 6.41e⁻¹ | **4.51e⁻¹** | 4.78e⁻¹ | 6.18e⁻¹ | 9.17e⁻¹ | 7.16e⁻¹ | 6.94e⁻¹ | 6.47e⁻¹ | 5.82e⁻¹ | 5.85e⁻¹ | 5.55e⁻¹ | 5.55e⁻¹ | 5.44e⁻¹ | 5.15e⁻¹ | 5.25e⁻¹ | P.TST |
| | Rank | 2.40e¹ | 2.26e¹ | 1.95e¹ | 2.08e¹ | 2.37e¹ | 1.90e¹ | 8.95 | 1.55¹ | 1.69¹ | 6.56 | 7.05 | 1.80¹ | 1.73¹ | 2.00¹ | 1.91¹ | 1.70¹ | 1.37¹ | 1.53¹ | 1.24¹ | 1.23¹ | 1.25¹ | 9.94 | 8.63 | 8.91 | P.TST |
| True | sMAPE | 1.33 | 1.00 | 9.89e⁻¹ | 1.17 | 1.10 | 1.13 | 9.44e⁻¹ | 1.07 | 1.01 | 9.32e⁻¹ | 9.78e⁻¹ | 1.06 | 2.26 | 1.13 | 1.10 | 1.30 | 9.23e⁻¹ | 9.70e⁻¹ | 9.16e⁻¹ | 9.01e⁻¹ | 8.94e⁻¹ | 9.88e⁻¹ | 9.10e⁻¹ | **8.97e⁻¹** | Chr.$_L$ |
| | MASE | 1.36 | 1.00 | 9.12e⁻¹ | 1.15 | 1.11 | 1.02 | 8.08e⁻¹ | 9.59e⁻¹ | 8.92e⁻¹ | 8.05e⁻¹ | 8.57e⁻¹ | 9.44e⁻¹ | 4.00 | 1.13 | 1.00 | 1.23 | 8.29e⁻¹ | 8.45e⁻¹ | 7.79e⁻¹ | **7.80e⁻¹** | 7.75e⁻¹ | 8.95e⁻¹ | 7.96e⁻¹ | 7.86e⁻¹ | Chr.$_L$ |
| | ND | 1.35 | 1.00 | 9.02e⁻¹ | 1.15 | 1.10 | 9.88e⁻¹ | 7.85e⁻¹ | 9.33e⁻¹ | 9.92e⁻¹ | 7.93e⁻¹ | 8.38e⁻¹ | 9.26e⁻¹ | 3.01 | 1.12 | 9.81e⁻¹ | 1.22 | 8.09e⁻¹ | 8.16e⁻¹ | 7.72e⁻¹ | 6.68e⁻¹ | 6.33e⁻¹ | 7.49e⁻¹ | 6.96e⁻¹ | **6.04e⁻¹** | Chr.$_L$ |
| | MSE | 1.63 | 1.00 | 8.06e⁻¹ | 1.06 | 1.15 | 8.91e⁻¹ | 6.33e⁻¹ | 7.96e⁻¹ | 7.93e⁻¹ | 6.49e⁻¹ | 7.20e⁻¹ | 7.45e⁻¹ | 2.28 | 1.04 | 7.99e⁻¹ | 1.40 | 6.95e⁻¹ | 6.72e⁻¹ | 6.08e⁻¹ | 6.23e⁻¹ | 7.49e⁻¹ | 6.36e⁻¹ | **6.04e⁻¹** | Chr.$_L$ |
| | MAE | 1.35 | 1.00 | 9.01e⁻¹ | 1.15 | 1.10 | 9.88e⁻¹ | 7.86e⁻¹ | 9.32e⁻¹ | 8.92e⁻¹ | 8.06e⁻¹ | 8.50e⁻¹ | 9.26e⁻¹ | 2.15 | 1.12 | 9.81e⁻¹ | 1.22 | 8.38e⁻¹ | 8.35e⁻¹ | 7.95e⁻¹ | 7.71e⁻¹ | **7.65e⁻¹** | 8.73e⁻¹ | 7.83e⁻¹ | 7.66e⁻¹ | Chr.$_L$ |
| | CRPS | 1.49 | 1.00 | 7.21e⁻¹ | 1.16 | 9.02 | 6.62e⁻¹ | 5.24e⁻¹ | 6.46e⁻¹ | 7.30e⁻¹ | 5.35e⁻¹ | 5.64e⁻¹ | 7.58e⁻¹ | 2.46 | 2.23e¹ | 8.03e⁻¹ | 8.31e⁻¹ | 5.69e⁻¹ | 6.83e⁻¹ | 5.64e⁻¹ | 5.47e⁻¹ | 5.43e⁻¹ | 5.98e⁻¹ | 5.16e⁻¹ | **5.08e⁻¹** | Moi.$_L$ |
| | Rank | 2.56e¹ | 2.29e¹ | 1.70e¹ | 2.13e¹ | 2.07e¹ | 1.34¹ | 8.76 | 1.52¹ | 1.85¹ | 8.56 | 9.80 | 2.01¹ | 1.99¹ | 2.23e¹ | 2.04e¹ | 1.88e¹ | 9.46 | 1.81e¹ | 1.19e¹ | 9.94 | 9.69 | 1.17e¹ | 6.07 | 6.50 | Moi.$_B$ |

Table 21: Results on GIFT-Eval with all models aggregated by all datasets. The best results across each row are **bolded**, while second best results are underlined.

| Metric | Nv. | S.Nv. | A.Ar. | A.Th. | A.ETS | D.AR | TFT | TiDE | N-B. | P.TST | iTr. | DLin. | C.former | Timer | TTM | L-Llama | T.FM | V.TS | Chr.$_S$ | Chr.$_B$ | Chr.$_L$ | Moi.$_S$ | Moi.$_B$ | Moi.$_L$ | Best |
|---|---|---|---|---|---|---|---|---|---|---|---|---|---|---|---|---|---|---|---|---|---|---|---|---|---|
| sMAPE | 1.25 | 1.00 | 1.05 | 1.14 | 1.37 | 1.06 | 1.16 | 1.16 | 1.09 | **9.83e⁻¹** | 1.04 | 1.12 | 1.82 | 1.23 | 1.17 | 1.32 | 1.04 | 1.02 | 1.01 | 1.00 | 1.06 | 1.04 | 1.02 | P.TST |
| MASE | 1.26 | 1.00 | 9.64e⁻¹ | 9.78e⁻¹ | 1.09 | 1.21 | 8.22e⁻¹ | 9.80e⁻¹ | 8.42e⁻¹ | **7.62e⁻¹** | 8.02e⁻¹ | 9.52e⁻¹ | 2.31 | 1.02 | 9.69e⁻¹ | 1.10 | 9.67e⁻¹ | 7.75e⁻¹ | 8.00e⁻¹ | 7.86e⁻¹ | 7.81e⁻¹ | 8.74e⁻¹ | 8.11e⁻¹ | 7.97e⁻¹ | P.TST |
| ND | 1.20 | 1.00 | 9.60e⁻¹ | 1.06 | 1.10 | 1.03 | 8.37e⁻¹ | 9.73e⁻¹ | 8.86e⁻¹ | **7.79e⁻¹** | 8.24e⁻¹ | 9.16e⁻¹ | 1.78 | 9.22e⁻¹ | 7.65e⁻¹ | 1.22 | 7.35e⁻¹ | 6.48e⁻¹ | 7.05e⁻¹ | 6.80e⁻¹ | 6.72e⁻¹ | 7.67e⁻¹ | 6.96e⁻¹ | 6.87e⁻¹ | P.TST |
| MSE | 1.36 | 1.00 | 9.11e⁻¹ | 9.30e⁻¹ | 1.21 | 1.10 | 6.37e⁻¹ | 8.20e⁻¹ | 7.58e⁻¹ | **6.08e⁻¹** | 6.76e⁻¹ | 6.95e⁻¹ | 1.32 | 9.22e⁻¹ | 7.65e⁻¹ | 1.22 | 7.35e⁻¹ | 6.48e⁻¹ | 7.05e⁻¹ | 6.72e⁻¹ | 7.67e⁻¹ | 6.96e⁻¹ | 6.87e⁻¹ | P.TST |
| MAE | 1.20 | 1.00 | 9.60e⁻¹ | 1.06 | 1.10 | 1.11 | 8.01e⁻¹ | 9.73e⁻¹ | 8.86e⁻¹ | **7.78e⁻¹** | 8.24e⁻¹ | 9.18e⁻¹ | 1.47 | 1.05 | 9.68e⁻¹ | 1.15 | 8.75e⁻¹ | 8.20e⁻¹ | 8.17e⁻¹ | 8.02e⁻¹ | **7.96e⁻¹** | 8.73e⁻¹ | 8.22e⁻¹ | 8.18e⁻¹ | P.TST |
| CRPS | 1.38 | 1.00 | 7.70e⁻¹ | 1.05 | 6.33 | 7.21e⁻¹ | 5.11e⁻¹ | 6.52e⁻¹ | 6.89e⁻¹ | **4.96e⁻¹** | 5.24e⁻¹ | 7.14e⁻¹ | 1.38 | 8.20e⁻¹ | 7.53e⁻¹ | 7.44e⁻¹ | 5.75e⁻¹ | 6.38e⁻¹ | 5.60e⁻¹ | 5.51e⁻¹ | 5.47e⁻¹ | 5.76e⁻¹ | 5.16e⁻¹ | 5.15e⁻¹ | P.TST |
| Rank | 2.49e¹ | 2.28e¹ | 1.81e¹ | 2.11e¹ | 2.20e¹ | 1.59e¹ | 8.85 | 1.53¹ | 1.78¹ | 7.67 | 8.58 | 1.92¹ | 1.88¹ | 2.13e¹ | 1.98e¹ | 1.80e¹ | 1.13e¹ | 1.69e¹ | 1.21e¹ | 1.10e¹ | 1.09e¹ | 1.10e¹ | 7.21 | 7.57 | Moi.$_B$ |

1458
1459
1460
1461
1462
1463
1464
1465
1466
1467
1468
1469
1470

Table 22: Results on all dataset configs for GIFT-Eval | Table 1/3. The best results across each row are **bolded**, while second best results are underlined.

1471
1472
...
1511

Table 23: Results on all dataset configs for GIFT-Eval | Table 2/3. The best results across each row are **bolded**, while second best results are underlined.

| Dataset, term, frequency | Metric | Nv. | S.Nv. | A.Ar. | A.Th. | A.ETS | D.AR | TFT | TiDE | N-B. | P.TST | iTr. | DLin. | C.former | Timer | TTM | L-Llama | T.FM | V.TS | Chr.$_S$ | Chr.$_B$ | Chr.$_L$ | Moi.$_S$ | Moi.$_B$ | Moi.$_L$ | Best |
|---|---|---|---|---|---|---|---|---|---|---|---|---|---|---|---|---|---|---|---|---|---|---|---|---|---|---|

Table 24: Results on all dataset configs for GIFT-Eval | Table 3/3. The best results across each row are **bolded**, while second best results are underlined.

| Dataset, term, frequency | Metric | Nv. | S.Nv. | A.Ar. | A.Th. | A.ETS | D.AR | TFT | TiDE | N-B. | P.TST | iTr. | DLin. | C.former | Timer | TTM | L-Llama | T.FM | V.TS | Chr.S | Chr.B | Chr.L | Moi.S | Moi.B | Moi.L | Best |
|---|---|---|---|---|---|---|---|---|---|---|---|---|---|---|---|---|---|---|---|---|---|---|---|---|---|---|
| loop_seattle, short, 5T | MASE | 1.17 | 1.00 | 1.00 | 1.02 | 1.10 | 1.05 | 9.59e-1 | 1.01 | 9.65e-1 | 9.76e-1 | 9.65e-1 | 1.17 | 2.10 | 1.05 | 1.17 | 1.17 | 1.10 | 1.04 | 1.01 | 1.00 | 1.00 | 7.86e-1 | 6.90e-1 | **6.40e-1** | Moi.L |
| loop_seattle, short, 5T | CRPS | 1.44 | 1.00 | 1.00 | 1.01 | 1.03 | 8.91e-1 | 8.01e-1 | 9.01e-1 | 9.89e-1 | 8.12e-1 | 8.12e-1 | 1.22 | 5.64 | 1.10 | 1.22 | 1.02 | 9.29e-1 | 1.08 | 8.66e-1 | 8.61e-1 | 8.64e-1 | 6.29e-1 | 5.58e-1 | **5.14e-1** | Moi.L |
| loop_seattle, short, 5T | Rank | 2.90e1 | 1.90e1 | 1.80e1 | 2.00e1 | 2.20e1 | 1.40e1 | 8.00 | 1.50e1 | 1.70e1 | 1.00e1 | 9.00 | 2.60e1 | 3.00e1 | 2.40e1 | 2.50e1 | 2.10e1 | 1.60e1 | 2.30e1 | 1.30e1 | 1.10e1 | 1.20e1 | 3.50 | 2.00 | 1.00 | Moi.L |
| loop_seattle, short, D | MASE | 1.00 | 1.00 | 8.61e-1 | 8.03e-1 | 7.23e-1 | 6.24e-1 | 5.62e-1 | 5.35e-1 | 5.19e-1 | 5.40e-1 | 5.38e-1 | 5.20e-1 | 9.36e-1 | 8.67e-1 | 9.48e-1 | 9.31e-1 | **5.09e-1** | 5.49e-1 | 5.78e-1 | 5.27e-1 | 5.25e-1 | 5.55e-1 | 5.22e-1 | 5.29e-1 | T.FM |
| loop_seattle, short, D | CRPS | 7.86e-1 | 1.00 | 5.93e-1 | 5.48e-1 | 4.46e-1 | 3.97e-1 | 3.68e-1 | 3.37e-1 | 4.08e-1 | 3.53e-1 | 3.48e-1 | 4.08e-1 | 2.77 | 6.69e-1 | 7.24e-1 | 6.43e-1 | **3.24e-1** | 4.29e-1 | 3.68e-1 | 3.40e-1 | 3.40e-1 | 3.52e-1 | 3.38e-1 | 3.45e-1 | T.FM |
| loop_seattle, short, D | Rank | 2.70e1 | 2.80e1 | 2.20e1 | 2.10e1 | 2.00e1 | 1.60e1 | 1.50e1 | 3.00 | 1.70e1 | 1.10e1 | 8.00 | 1.80e1 | 2.90e1 | 2.50e1 | 2.60e1 | 2.40e1 | 1.00 | 1.90e1 | 1.40e1 | 5.00 | 6.00 | 9.50 | 4.00 | 7.00 | T.FM |
| loop_seattle, short, H | MASE | 1.23 | 1.00 | 1.00 | 1.09 | 1.20 | 7.29e-1 | 8.06e-1 | 8.76e-1 | 7.98e-1 | 8.29e-1 | 8.22e-1 | 8.76e-1 | **5.71e-1** | 9.30e-1 | 8.99e-1 | 8.29e-1 | 8.91e-1 | 8.60e-1 | 7.50e-1 | 7.18e-1 | 7.18e-1 | 9.65e-1 | 8.19e-1 | 7.40e-1 | C.former |
| loop_seattle, short, H | CRPS | 1.69 | 1.00 | 1.00 | 1.53 | 1.45 | **6.07e-1** | 6.74e-1 | 7.35e-1 | 8.33e-1 | 7.07e-1 | 6.91e-1 | 9.14e-1 | 1.44 | 9.72e-1 | 9.44e-1 | 7.19e-1 | 7.59e-1 | 8.96e-1 | 6.41e-1 | 6.11e-1 | 6.10e-1 | 8.17e-1 | 6.84e-1 | 6.20e-1 | D.AR |
| loop_seattle, short, H | Rank | 3.00e1 | 2.40e1 | 2.30e1 | 2.90e1 | 2.80e1 | 1.00 | 7.00 | 1.30e1 | 1.80e1 | 1.00e1 | 9.00 | 2.00e1 | 2.70e1 | 2.20e1 | 2.10e1 | 1.10e1 | 1.40e1 | 1.90e1 | 5.00 | 3.00 | 2.00 | 1.65e1 | 8.00 | 4.00 | D.AR |
| m4_daily, short, D | MASE | 1.00 | 1.00 | 9.94e-1 | 1.02 | 9.88e-1 | 1.40 | 1.00 | 1.05 | 1.02 | 9.82e-1 | 9.94e-1 | 1.04 | 6.04e1 | 1.33 | 1.34 | 2.93 | 9.97e-1 | 1.22 | **9.63e-1** | 9.70e-1 | 9.70e-1 | 1.22 | 1.90 | 2.80 | Chr.S |
| m4_daily, short, D | CRPS | 9.21e-1 | 1.00 | 8.72e-1 | 8.87e-1 | 1.08 | 1.15 | 8.60e-1 | 1.21 | 1.08 | 8.60e-1 | 8.45e-1 | 1.09 | 2.76e2 | 1.26 | 1.26 | 2.03 | **8.04e-1** | 1.25 | 8.11e-1 | 8.34e-1 | 8.42e-1 | 9.57e-1 | 1.87 | 2.72 | T.FM |
| m4_daily, short, D | Rank | 1.20e1 | 1.50e1 | 9.00 | 1.00e1 | 1.60e1 | 1.90e1 | 8.00 | 2.00e1 | 1.70e1 | 7.00 | 6.00 | 1.80e1 | 3.00e1 | 2.30e1 | 2.40e1 | 2.60e1 | 1.00 | 2.20e1 | 2.00 | 4.00 | 5.00 | 1.35e1 | 2.50e1 | 2.70e1 | T.FM |
| m4_hourly, short, H | MASE | 9.75 | 1.00 | 8.66e-1 | 2.07 | 1.35 | 2.97 | 2.08 | 3.29 | 1.13 | 1.18 | 1.22 | 1.42 | 1.38e2 | 2.92 | 1.92 | 2.95 | 6.45e-1 | 7.77e-1 | 6.21e-1 | **5.82e-1** | 5.83e-1 | 1.58 | 8.17e-1 | 7.44e-1 | Chr.B |
| m4_hourly, short, H | CRPS | 3.44 | 1.00 | 8.53e-1 | 1.05 | 1.76 | 3.37 | 1.02 | 1.21 | 1.27 | 9.87e-1 | 1.04 | 1.40 | 1.07e2 | 1.13 | 9.49e-1 | 1.56 | 5.32e-1 | 9.97e-1 | 6.41e-1 | 6.18e-1 | 6.53e-1 | 7.16e-1 | 5.66e-1 | **5.03e-1** | Moi.L |
| m4_hourly, short, H | Rank | 2.70e1 | 1.60e1 | 1.20e1 | 1.90e1 | 2.50e1 | 2.60e1 | 1.70e1 | 2.10e1 | 2.20e1 | 1.40e1 | 1.80e1 | 2.30e1 | 3.00e1 | 2.00e1 | 1.30e1 | 2.40e1 | 2.00 | 1.50e1 | 6.00 | 5.00 | 7.00 | 1.05e1 | 4.00 | 1.00 | Moi.L |
| m4_monthly, short, M | MASE | 9.60e-1 | 1.00 | 7.75e-1 | 7.67e-1 | 7.65e-1 | 2.52 | 9.60e-1 | 1.39 | 8.33e-1 | 8.41e-1 | 1.28 | 8.97e-1 | 1.61e1 | 1.22 | 1.14 | 2.48 | 7.60e-1 | 9.21e-1 | 7.79e-1 | 7.72e-1 | 7.71e-1 | 8.76e-1 | **7.56e-1** | 7.75e-1 | Moi.B |
| m4_monthly, short, M | CRPS | 1.02 | 1.00 | 7.77e-1 | 7.78e-1 | 7.92e-1 | 1.46 | 8.97e-1 | 1.07 | 9.68e-1 | 8.10e-1 | 9.05e-1 | 1.02 | 4.09e1 | 1.28 | 1.21 | 1.67 | 7.69e-1 | 1.10 | 8.17e-1 | 8.25e-1 | 8.17e-1 | 8.22e-1 | **7.45e-1** | 7.55e-1 | Moi.B |
| m4_monthly, short, M | Rank | 1.90e1 | 1.70e1 | 4.00 | 5.00 | 6.00 | 2.60e1 | 1.40e1 | 2.10e1 | 1.60e1 | 8.00 | 1.50e1 | 1.80e1 | 3.00e1 | 2.40e1 | 2.30e1 | 2.70e1 | 3.00 | 2.20e1 | 1.00e1 | 1.30e1 | 9.00 | 1.15e1 | 1.00 | 2.00 | Moi.B |
| m4_quarterly, short, Q | MASE | 9.25e-1 | 1.00 | 8.00e-1 | 7.44e-1 | 7.25e-1 | 9.00e-1 | 8.12e-1 | 1.05 | 7.56e-1 | 8.25e-1 | 7.69e-1 | 9.12e-1 | 1.59e1 | 1.84 | 1.24 | 3.73 | 8.75e-1 | 8.50e-1 | 7.75e-1 | 7.69e-1 | 7.69e-1 | 7.76e-1 | 7.11e-1 | **7.11e-1** | Moi.L |
| m4_quarterly, short, Q | CRPS | 9.51e-1 | 1.00 | 8.23e-1 | 7.97e-1 | 7.98e-1 | 8.41e-1 | 8.37e-1 | 1.02 | 9.72e-1 | 8.35e-1 | 7.97e-1 | 1.11 | 1.20e2 | 1.78 | 1.29 | 2.52 | 8.53e-1 | 1.05 | 8.46e-1 | 8.40e-1 | 8.40e-1 | 7.94e-1 | **7.40e-1** | 7.40e-1 | Moi.B |
| m4_quarterly, short, Q | Rank | 1.80e1 | 2.00e1 | 9.00 | 6.00 | 8.00 | 1.40e1 | 1.10e1 | 2.10e1 | 1.90e1 | 1.00e1 | 7.00 | 2.30e1 | 3.00e1 | 2.60e1 | 2.50e1 | 2.70e1 | 1.60e1 | 2.20e1 | 1.50e1 | 1.20e1 | 1.30e1 | 4.50 | 1.00 | 2.00 | Moi.B |
| m4_weekly, short, W | MASE | 1.00 | 1.00 | 8.49e-1 | 9.57e-1 | 9.17e-1 | 1.66 | 9.64e-1 | 1.83 | **7.09e-1** | 8.42e-1 | 9.28e-1 | 1.67 | 3.78e1 | 1.39 | 1.02 | 3.15 | 8.71e-1 | 9.10e-1 | 7.52e-1 | 7.48e-1 | 7.48e-1 | 1.29 | 1.00 | 9.27e-1 | N-B. |
| m4_weekly, short, W | CRPS | 8.38e-1 | 1.00 | 6.86e-1 | 7.35e-1 | 7.13e-1 | 8.51e-1 | 6.80e-1 | 9.05e-1 | 6.51e-1 | 5.53e-1 | 6.93e-1 | 9.57e-1 | 1.14e2 | 1.09 | 8.57e-1 | 1.42 | 5.68e-1 | 7.87e-1 | 5.43e-1 | **5.14e-1** | 5.16e-1 | 7.48e-1 | 6.65e-1 | 6.38e-1 | Chr.B |
| m4_weekly, short, W | Rank | 1.90e1 | 2.50e1 | 1.10e1 | 1.40e1 | 1.30e1 | 2.00e1 | 1.00e1 | 2.20e1 | 7.00 | 4.00 | 1.20e1 | 2.40e1 | 3.00e1 | 2.60e1 | 2.10e1 | 2.70e1 | 5.00 | 1.80e1 | 3.00 | 1.00 | 2.00 | 1.65e1 | 9.00 | 6.00 | Chr.B |
| m4_yearly, short, A | MASE | 1.00 | 1.00 | 9.35e-1 | 7.83e-1 | 7.76e-1 | 8.56e-1 | 7.78e-1 | 1.26 | 7.93e-1 | 8.29e-1 | 8.49e-1 | 1.05 | 7.91 | 2.90 | 1.29 | 2.41 | 8.44e-1 | 9.65e-1 | 9.42e-1 | 9.17e-1 | 9.17e-1 | 7.51e-1 | 7.58e-1 | **7.49e-1** | Moi.L |
| m4_yearly, short, A | CRPS | 9.93e-1 | 1.00 | 9.42e-1 | 8.33e-1 | 8.04e-1 | 8.19e-1 | 7.97e-1 | 1.12 | 9.71e-1 | 8.48e-1 | 8.48e-1 | 1.22 | 1.03e2 | 2.92 | 1.39 | 2.04 | 8.48e-1 | 1.15 | 1.01 | 9.78e-1 | 9.78e-1 | 7.64e-1 | 7.62e-1 | **7.57e-1** | Moi.L |
| m4_yearly, short, A | Rank | 1.90e1 | 2.00e1 | 1.40e1 | 1.00e1 | 7.00 | 8.00 | 6.00 | 2.20e1 | 1.60e1 | 1.20e1 | 1.10e1 | 2.40e1 | 3.00e1 | 2.70e1 | 2.50e1 | 2.60e1 | 1.30e1 | 2.30e1 | 2.10e1 | 1.70e1 | 1.80e1 | 3.50 | 2.00 | 1.00 | Moi.L |
| m_dense, long, H | MASE | 1.93 | 1.00 | 8.18e-1 | 1.55 | 1.07 | 5.44e-1 | 4.89e-1 | 6.53e-1 | 7.97e-1 | 4.99e-1 | 5.63e-1 | 8.38e-1 | 2.95 | 9.73e-1 | 7.09e-1 | 7.23e-1 | 5.32e-1 | 5.55e-1 | 4.98e-1 | 5.22e-1 | 5.32e-1 | 6.97e-1 | 4.95e-1 | **4.60e-1** | Moi.L |
| m_dense, long, H | CRPS | 2.61 | 1.00 | 4.89e-1 | 2.59 | 1.07 | 2.36e-1 | 2.08e-1 | 2.93e-1 | 4.40e-1 | 2.17e-1 | 2.45e-1 | 4.69e-1 | **1.15e-1** | 5.94e-1 | 3.97e-1 | 3.44e-1 | 2.45e-1 | 3.01e-1 | 2.41e-1 | 2.45e-1 | 2.52e-1 | 3.15e-1 | 2.21e-1 | 2.00e-1 | C.former |
| m_dense, long, H | Rank | 2.90e1 | 2.60e1 | 2.20e1 | 2.80e1 | 2.70e1 | 6.00 | 3.00 | 1.30e1 | 1.90e1 | 4.00 | 9.00 | 2.10e1 | 1.00 | 2.30e1 | 1.80e1 | 1.70e1 | 1.00e1 | 1.40e1 | 7.00 | 8.00 | 1.10e1 | 1.50e1 | 5.00 | 2.00 | C.former |
| m_dense, medium, H | MASE | 1.75 | 1.00 | 8.09e-1 | 1.11 | 9.49e-1 | 4.70e-1 | 4.66e-1 | 6.32e-1 | 5.67e-1 | 4.82e-1 | 5.33e-1 | 5.92e-1 | 2.47 | 8.03e-1 | 6.43e-1 | 6.34e-1 | 4.87e-1 | 5.68e-1 | 4.54e-1 | 4.82e-1 | 4.82e-1 | 6.65e-1 | 4.67e-1 | **4.38e-1** | Moi.L |
| m_dense, medium, H | CRPS | 2.51 | 1.00 | 5.32e-1 | 2.53 | 9.56e-1 | 2.46e-1 | **2.38e-1** | 3.51e-1 | 3.84e-1 | 2.65e-1 | 2.86e-1 | 3.99e-1 | 2.78e-1 | 5.91e-1 | 4.41e-1 | 3.65e-1 | 2.76e-1 | 3.78e-1 | 2.67e-1 | 2.84e-1 | 2.80e-1 | 3.65e-1 | 2.57e-1 | 2.38e-1 | TFT |
| m_dense, medium, H | Rank | 2.80e1 | 2.50e1 | 2.20e1 | 2.90e1 | 2.40e1 | 3.00 | 1.00 | 1.40e1 | 1.80e1 | 5.00 | 1.10e1 | 1.90e1 | 8.00 | 2.30e1 | 2.00e1 | 1.60e1 | 7.00 | 1.70e1 | 6.00 | 1.00e1 | 9.00 | 1.50e1 | 4.00 | 2.00 | TFT |
| m_dense, short, D | MASE | 1.00 | 1.00 | 8.02e-1 | 7.31e-1 | 7.13e-1 | 4.75e-1 | 4.78e-1 | 5.46e-1 | 4.23e-1 | 4.38e-1 | 4.35e-1 | 6.05e-1 | 8.44 | 7.19e-1 | 7.19e-1 | 7.72e-1 | **4.20e-1** | 4.72e-1 | 4.99e-1 | 4.26e-1 | 4.34e-1 | 4.98e-1 | 6.57e-1 | 5.79e-1 | T.FM |
| m_dense, short, D | CRPS | 7.72e-1 | 1.00 | 4.59e-1 | 4.29e-1 | 4.18e-1 | 2.60e-1 | 2.61e-1 | 3.09e-1 | 2.98e-1 | **2.38e-1** | 2.45e-1 | 4.18e-1 | 1.03e1 | 5.17e-1 | 5.10e-1 | 4.29e-1 | 2.39e-1 | 3.20e-1 | 2.95e-1 | 2.55e-1 | 2.54e-1 | 2.83e-1 | 3.57e-1 | 3.26e-1 | P.TST |
| m_dense, short, D | Rank | 2.80e1 | 2.90e1 | 2.30e1 | 2.10e1 | 1.90e1 | 6.00 | 7.00 | 1.40e1 | 1.30e1 | 1.00 | 3.00 | 2.00e1 | 3.00e1 | 2.50e1 | 2.40e1 | 2.20e1 | 2.00 | 1.60e1 | 1.20e1 | 5.00 | 4.00 | 1.05e1 | 1.80e1 | 1.70e1 | P.TST |
| m_dense, short, H | MASE | 1.82 | 1.00 | 1.00 | 1.13 | 9.80e-1 | 5.34e-1 | 5.89e-1 | 6.60e-1 | 6.14e-1 | 6.91e-1 | 6.00e-1 | 6.98e-1 | 3.14 | 7.65e-1 | 7.18e-1 | 6.28e-1 | 5.70e-1 | 7.65e-1 | 5.39e-1 | 5.37e-1 | 5.38e-1 | 7.54e-1 | 5.44e-1 | **5.21e-1** | Moi.L |
| m_dense, short, H | CRPS | 1.92 | 1.00 | 1.00 | 1.95 | 8.61e-1 | 4.56e-1 | 4.95e-1 | 5.98e-1 | 6.76e-1 | 6.16e-1 | 5.34e-1 | 7.62e-1 | 4.48 | 8.79e-1 | 7.94e-1 | 5.66e-1 | 4.98e-1 | 8.58e-1 | 4.98e-1 | 4.88e-1 | 4.88e-1 | 6.70e-1 | 4.84e-1 | **4.54e-1** | Moi.L |
| m_dense, short, H | Rank | 2.60e1 | 2.50e1 | 2.40e1 | 2.90e1 | 2.20e1 | 2.00 | 6.00 | 1.30e1 | 1.80e1 | 1.50e1 | 9.00 | 1.90e1 | 3.00e1 | 2.30e1 | 2.00e1 | 1.10e1 | 8.00 | 2.10e1 | 7.00 | 4.00 | 5.00 | 1.65e1 | 3.00 | 1.00 | Moi.L |
| restaurant, short, D | MASE | 1.00 | 1.00 | 9.20e-1 | 8.35e-1 | 8.52e-1 | 7.06e-1 | 7.43e-1 | 7.50e-1 | 7.05e-1 | 6.83e-1 | **6.81e-1** | 6.99e-1 | 1.96 | 8.28e-1 | 8.22e-1 | 1.16 | 6.97e-1 | 7.36e-1 | 7.50e-1 | 7.21e-1 | 7.23e-1 | 7.33e-1 | 6.98e-1 | 7.07e-1 | iTr. |
| restaurant, short, D | CRPS | 7.43e-1 | 1.00 | 3.99e-1 | 3.63e-1 | 1.21e2 | 2.98e-1 | 3.13e-1 | 3.27e-1 | 3.77e-1 | 2.89e-1 | **2.87e-1** | 3.75e-1 | 3.97 | 4.55e-1 | 4.49e-1 | 5.31e-1 | 2.92e-1 | 3.95e-1 | 3.22e-1 | 3.08e-1 | 3.08e-1 | 3.11e-1 | 2.94e-1 | 2.97e-1 | iTr. |
| restaurant, short, D | Rank | 2.70e1 | 2.80e1 | 2.10e1 | 1.70e1 | 3.00e1 | 6.00 | 1.40e1 | 1.60e1 | 1.90e1 | 2.00 | 1.00 | 1.80e1 | 2.90e1 | 2.30e1 | 2.20e1 | 2.50e1 | 3.00 | 2.00e1 | 1.50e1 | 9.00 | 1.00e1 | 1.15e1 | 4.00 | 5.00 | iTr. |
| saugeen, short, D | MASE | 1.00 | 1.00 | 1.10 | 1.06 | 1.14 | 1.26 | 9.44e-1 | 9.27e-1 | 9.62e-1 | 9.62e-1 | 9.82e-1 | 1.23 | 1.39 | 1.32 | 1.18 | 1.23 | 9.68e-1 | 1.17 | 8.74e-1 | 9.65e-1 | 9.33e-1 | 1.01 | **8.56e-1** | 9.56e-1 | Moi.B |
| saugeen, short, D | CRPS | 7.76e-1 | 1.00 | 7.48e-1 | 8.87e-1 | 7.90e-1 | 7.59e-1 | 5.56e-1 | 5.27e-1 | 6.34e-1 | 5.41e-1 | 5.54e-1 | 8.13e-1 | 6.13e-1 | 8.69e-1 | 7.81e-1 | 7.06e-1 | 5.53e-1 | 7.72e-1 | 5.13e-1 | 5.73e-1 | 5.57e-1 | 5.56e-1 | **4.68e-1** | 5.21e-1 | Moi.B |
| saugeen, short, D | Rank | 2.20e1 | 2.80e1 | 1.90e1 | 2.70e1 | 2.40e1 | 2.00e1 | 1.10e1 | 6.00 | 1.70e1 | 8.00 | 1.00e1 | 2.50e1 | 1.60e1 | 2.60e1 | 2.30e1 | 1.80e1 | 9.00 | 2.10e1 | 3.00 | 1.50e1 | 1.40e1 | 1.25e1 | 1.00 | 5.00 | Moi.B |
| saugeen, short, M | MASE | 1.26 | 1.00 | 7.43e-1 | 9.34e-1 | 7.43e-1 | 1.67 | 8.86e-1 | 1.03 | 7.77e-1 | 9.15e-1 | 8.03e-1 | 9.78e-1 | **7.30e-1** | 8.78e-1 | 7.75e-1 | 7.92e-1 | 8.34e-1 | 8.20e-1 | 1.02 | 8.75e-1 | 8.90e-1 | 1.20 | 8.57e-1 | 7.85e-1 | C.former |
| saugeen, short, M | CRPS | 1.73 | 1.00 | 7.33e-1 | 8.38e-1 | 7.24e-1 | 1.55 | 7.64e-1 | 9.53e-1 | 8.72e-1 | 8.36e-1 | 7.10e-1 | 1.10 | **4.79e-1** | 9.87e-1 | 8.70e-1 | 7.10e-1 | 7.37e-1 | 9.21e-1 | 1.04 | 9.17e-1 | 9.33e-1 | 1.12 | 7.96e-1 | 7.38e-1 | C.former |
| saugeen, short, M | Rank | 3.00e1 | 2.00e1 | 5.00 | 1.10e1 | 4.00 | 2.80e1 | 8.00 | 1.80e1 | 1.30e1 | 1.00e1 | 2.00 | 2.20e1 | 1.00 | 1.90e1 | 1.20e1 | 3.00 | 6.00 | 1.60e1 | 2.10e1 | 1.50e1 | 1.70e1 | 2.35e1 | 9.00 | 7.00 | C.former |
| saugeen, short, W | MASE | 1.00 | 1.00 | 7.79e-1 | 1.07 | 1.00 | 6.58e-1 | 7.79e-1 | 7.89e-1 | 7.74e-1 | 7.79e-1 | 7.49e-1 | 9.10e-1 | 1.37 | 7.89e-1 | 8.99e-1 | 8.24e-1 | **6.43e-1** | 8.14e-1 | 6.88e-1 | 6.78e-1 | 6.73e-1 | 7.57e-1 | 7.12e-1 | 6.88e-1 | T.FM |
| saugeen, short, W | CRPS | 8.58e-1 | 1.00 | 6.42e-1 | 8.58e-1 | 1.05 | 4.64e-1 | 5.74e-1 | 6.12e-1 | 6.71e-1 | 5.66e-1 | 5.45e-1 | 7.87e-1 | 2.97 | 6.81e-1 | 7.80e-1 | 6.34e-1 | **4.47e-1** | 7.05e-1 | 5.64e-1 | 5.53e-1 | 5.51e-1 | 5.49e-1 | 4.99e-1 | 5.05e-1 | T.FM |
| saugeen, short, W | Rank | 2.70e1 | 2.80e1 | 1.80e1 | 2.60e1 | 2.90e1 | 2.00 | 1.50e1 | 1.60e1 | 1.90e1 | 1.40e1 | 7.00 | 2.30e1 | 3.00e1 | 2.00e1 | 2.20e1 | 1.70e1 | 1.00 | 2.10e1 | 1.30e1 | 1.10e1 | 1.00e1 | 8.50 | 3.00 | 4.00 | T.FM |
| solar, long, 10T | MASE | 2.32 | 1.00 | 1.00 | 5.20 | 2.32 | 1.47 | 1.15 | 1.09 | 2.33 | 1.05 | 1.33 | 1.35 | **4.58e-1** | 3.20 | 1.40 | 3.27 | 1.81 | 1.31 | 2.27 | 1.91 | 1.91 | 2.30 | 2.32 | 2.32 | C.former |
| solar, long, 10T | CRPS | 1.97 | 1.00 | 1.00 | 8.45 | 3.73 | 6.98e-1 | 4.82e-1 | 4.54e-1 | 1.28 | 4.31e-1 | 5.47e-1 | 7.44e-1 | **3.92e-2** | 1.76 | 7.67e-1 | 1.60 | 8.94e-1 | 7.15e-1 | 1.15 | 9.52e-1 | 9.50e-1 | 1.03 | 1.15 | 1.12 | C.former |
| solar, long, 10T | Rank | 2.70e1 | 1.60e1 | 1.50e1 | 2.90e1 | 2.80e1 | 7.00 | 4.00 | 3.00 | 2.20e1 | 2.00 | 5.00 | 9.00 | 1.00 | 2.40e1 | 1.10e1 | 2.30e1 | 1.20e1 | 8.00 | 2.00e1 | 1.40e1 | 1.30e1 | 1.70e1 | 2.10e1 | 1.90e1 | C.former |
| solar, long, H | MASE | 2.06 | 1.00 | 9.30e-1 | 4.90 | 1.45 | 9.08e-1 | 1.05 | 1.06 | 2.07 | 9.14e-1 | 1.01 | 1.26 | **6.58e-2** | 1.26 | 9.81e-1 | 1.35 | 1.35 | 8.93e-1 | 9.44e-1 | 1.00 | 9.53e-1 | 1.17 | 9.46e-1 | 9.28e-1 | C.former |
| solar, long, H | CRPS | 2.35 | 1.00 | 4.13e-1 | 4.98 | 1.17 | 2.59e-1 | 2.73e-1 | 2.92e-1 | 6.80e-1 | 2.40e-1 | 2.50e-1 | 4.16e-1 | **3.30e-3** | 4.18e-1 | 3.26e-1 | 3.56e-1 | 3.89e-1 | 2.96e-1 | 3.00e-1 | 3.16e-1 | 3.05e-1 | 2.87e-1 | 2.36e-1 | 2.32e-1 | C.former |
| solar, long, H | Rank | 2.80e1 | 2.60e1 | 2.00e1 | 2.90e1 | 2.70e1 | 9.00 | 1.00e1 | 1.20e1 | 2.30e1 | 5.00 | 7.00 | 2.10e1 | 1.00 | 2.20e1 | 1.70e1 | 1.80e1 | 1.90e1 | 1.30e1 | 1.40e1 | 1.60e1 | 1.50e1 | 1.10e1 | 4.00 | 3.00 | C.former |
| solar, medium, 10T | MASE | 2.76 | 1.00 | 1.00 | 2.90 | 1.28e1 | 1.31 | 1.00 | 1.05 | 2.14 | 9.85e-1 | 1.49 | 1.17 | 2.18 | 2.96 | 1.28 | 2.82 | 1.49 | **9.60e-1** | 1.93 | 1.66 | 1.66 | 2.02 | 2.05 | 2.06 | V.TS |
| solar, medium, 10T | CRPS | 1.78 | 1.00 | 1.00 | 7.35 | 5.95 | 6.29e-1 | 4.70e-1 | 5.12e-1 | 1.30 | 4.62e-1 | 7.12e-1 | 7.16e-1 | **3.50e-1** | 1.79 | 7.83e-1 | 1.53 | 8.08e-1 | 5.85e-1 | 1.03 | 8.90e-1 | 8.83e-1 | 9.99e-1 | 1.10 | 1.09 | C.former |
| solar, medium, 10T | Rank | 2.40e1 | 1.80e1 | 1.70e1 | 2.90e1 | 2.80e1 | 7.00 | 3.00 | 4.00 | 2.20e1 | 2.00 | 8.00 | 9.00 | 1.00 | 2.50e1 | 1.10e1 | 2.30e1 | 1.20e1 | 6.00 | 1.90e1 | 1.40e1 | 1.30e1 | 1.60e1 | 2.10e1 | 2.00e1 | C.former |
| solar, medium, H | MASE | 2.27 | 1.00 | 9.07e-1 | 3.07 | 1.89 | 1.06 | 9.45e-1 | 1.00 | 2.27 | 1.03 | 1.14 | 1.25 | **3.94e-1** | 1.38 | 1.02 | 1.42 | 1.11 | 9.51e-1 | 8.72e-1 | 8.62e-1 | 8.62e-1 | 1.23 | 9.37e-1 | 9.82e-1 | C.former |
| solar, medium, H | CRPS | 2.35 | 1.00 | 4.39e-1 | 4.83 | 1.67 | 2.77e-1 | 2.60e-1 | 2.71e-1 | 7.87e-1 | 2.71e-1 | 2.95e-1 | 4.35e-1 | **3.11e-2** | 4.77e-1 | 3.53e-1 | 4.01e-1 | 3.35e-1 | 3.29e-1 | 2.84e-1 | 2.80e-1 | 2.80e-1 | 3.23e-1 | 2.58e-1 | 2.66e-1 | C.former |
| solar, medium, H | Rank | 2.80e1 | 2.40e1 | 2.10e1 | 2.90e1 | 2.70e1 | 8.00 | 4.00 | 7.00 | 2.30e1 | 6.00 | 1.40e1 | 2.00e1 | 1.00 | 2.20e1 | 1.80e1 | 1.90e1 | 1.70e1 | 1.60e1 | 1.20e1 | 1.00e1 | 9.00 | 1.50e1 | 3.00 | 5.00 | C.former |
| solar, short, 10T | MASE | 1.31 | 1.00 | 1.00 | 1.62 | 1.10 | 1.32 | 1.00 | 1.37 | **7.64e-1** | 1.98 | 2.05 | 1.12 | 1.54 | 1.53 | 1.05 | 1.27 | 1.34 | 1.00 | 1.10 | 1.00 | 9.91e-1 | 1.07 | 1.07 | 1.08 | N-B. |
| solar, short, 10T | CRPS | 1.10 | 1.00 | 1.00 | 2.74 | 1.01 | 1.08 | 7.19e-1 | 1.10 | 6.70e-1 | 1.59 | 1.59 | 9.58e-1 | 3.42 | 1.30 | 9.00e-1 | 9.09e-1 | 1.01 | 8.56e-1 | 7.38e-1 | 6.73e-1 | **6.69e-1** | 7.27e-1 | 7.00e-1 | 7.23e-1 | Chr.L |
| solar, short, 10T | Rank | 2.20e1 | 1.80e1 | 1.70e1 | 2.90e1 | 1.90e1 | 2.10e1 | 6.00 | 2.30e1 | 2.00 | 2.60e1 | 2.50e1 | 1.60e1 | 3.00e1 | 2.40e1 | 1.40e1 | 1.50e1 | 2.00e1 | 1.30e1 | 1.00e1 | 3.00 | 1.00 | 8.50 | 5.00 | 7.00 | Chr.L |
| solar, short, D | MASE | 1.00 | 1.00 | 8.71e-1 | 9.05e-1 | 8.79e-1 | 2.15 | 8.61e-1 | 9.74e-1 | 1.04 | **8.29e-1** | 2.70 | 8.88e-1 | 1.30 | 8.79e-1 | 8.52e-1 | 1.50 | 8.53e-1 | 1.00 | 9.31e-1 | 8.71e-1 | 8.71e-1 | 9.51e-1 | 8.79e-1 | 8.52e-1 | P.TST |
| solar, short, D | CRPS | 7.38e-1 | 1.00 | 3.73e-1 | 3.78e-1 | 3.71e-1 | 9.01e-1 | **3.66e-1** | 4.65e-1 | 5.94e-1 | 3.79e-1 | 1.15 | 5.06e-1 | 2.81 | 4.98e-1 | 4.83e-1 | 6.74e-1 | 3.80e-1 | 5.67e-1 | 4.45e-1 | 4.31e-1 | 4.33e-1 | 3.94e-1 | 3.89e-1 | 3.86e-1 | TFT |
| solar, short, D | Rank | 2.60e1 | 2.80e1 | 3.00 | 5.00 | 2.00 | 2.70e1 | 1.00 | 1.70e1 | 2.30e1 | 8.00 | 2.90e1 | 2.00e1 | 3.00e1 | 1.90e1 | 1.80e1 | 2.40e1 | 9.00 | 2.20e1 | 1.60e1 | 1.40e1 | 1.50e1 | 1.25e1 | 1.10e1 | 1.00e1 | TFT |
| solar, short, H | MASE | 2.20 | 1.00 | 1.00 | 2.15 | 1.89 | 1.07 | 1.01 | 1.05 | 1.09 | 1.00 | 1.19 | 1.11 | **1.88e-1** | 1.19 | 9.83e-1 | 1.14 | 1.09 | 9.15e-1 | 8.98e-1 | 8.69e-1 | 8.72e-1 | 1.16 | 9.39e-1 | 9.19e-1 | C.former |
| solar, short, H | CRPS | 2.12 | 1.00 | 1.00 | 3.71 | 1.72 | 6.19e-1 | 5.84e-1 | 6.05e-1 | 7.93e-1 | 5.41e-1 | 6.78e-1 | 8.07e-1 | **3.31e-1** | 8.57e-1 | 7.12e-1 | 6.82e-1 | 6.42e-1 | 6.61e-1 | 5.49e-1 | 5.32e-1 | 5.37e-1 | 6.29e-1 | 5.38e-1 | 5.30e-1 | C.former |
| solar, short, H | Rank | 2.80e1 | 2.50e1 | 2.40e1 | 3.00e1 | 2.60e1 | 1.30e1 | 1.00e1 | 1.20e1 | 2.10e1 | 7.00 | 1.80e1 | 2.20e1 | 1.00 | 2.30e1 | 2.00e1 | 1.90e1 | 1.60e1 | 1.70e1 | 8.00 | 3.00 | 4.00 | 1.45e1 | 5.00 | 2.00 | C.former |
| solar, short, W | MASE | 1.00 | 1.00 | 7.62e-1 | 7.82e-1 | 6.80e-1 | 1.15 | **4.70e-1** | 9.12e-1 | 1.59 | 7.48e-1 | 9.18e-1 | 7.69e-1 | 2.30 | 7.62e-1 | 1.84 | 2.31 | 7.82e-1 | 1.46 | 5.97e-1 | 7.82e-1 | 7.82e-1 | 1.06 | 1.13 | 1.05 | TFT |
| solar, short, W | CRPS | 8.90e-1 | 1.00 | 6.44e-1 | 6.57e-1 | 5.89e-1 | 1.03 | **4.83e-1** | 7.92e-1 | 1.83 | 6.86e-1 | 8.60e-1 | 8.90e-1 | 3.58e1 | 8.90e-1 | 2.12 | 2.25 | 6.65e-1 | 1.67 | 5.25e-1 | 6.82e-1 | 6.86e-1 | 9.36e-1 | 9.94e-1 | 9.06e-1 | TFT |
| solar, short, W | Rank | 1.40e1 | 2.10e1 | 4.00 | 5.00 | 3.00 | 2.20e1 | 1.00 | 1.00e1 | 2.60e1 | 9.00 | 1.20e1 | 1.30e1 | 3.00e1 | 1.50e1 | 2.70e1 | 2.80e1 | 6.00 | 2.50e1 | 2.00 | 7.00 | 8.00 | 1.75e1 | 2.00e1 | 1.60e1 | TFT |
| sz_taxi, long, 15T | MASE | 1.07 | 1.00 | 8.65e-1 | 1.10 | 3.55 | 1.06 | **7.74e-1** | 1.25 | 9.64e-1 | 1.10 | 1.13 | 1.22 | 2.59 | 8.02e-1 | 7.87e-1 | 8.25e-1 | 7.74e-1 | 8.05e-1 | 8.45e-1 | 8.21e-1 | 8.22e-1 | 7.75e-1 | 7.77e-1 | 8.02e-1 | TFT |
| sz_taxi, long, 15T | CRPS | 4.33 | 1.00 | 7.18e-1 | 1.14 | 9.33e14 | 5.16e-1 | 4.35e-1 | 5.69e-1 | 5.83e-1 | 5.07e-1 | 5.09e-1 | 7.15e-1 | **2.20e-1** | 4.87e-1 | 4.80e-1 | 3.97e-1 | 4.30e-1 | 4.93e-1 | 4.96e-1 | 4.78e-1 | 4.78e-1 | 3.77e-1 | 3.77e-1 | 3.85e-1 | C.former |
| sz_taxi, long, 15T | Rank | 2.80e1 | 2.60e1 | 2.50e1 | 2.70e1 | 2.90e1 | 2.10e1 | 1.00e1 | 2.20e1 | 2.30e1 | 1.90e1 | 2.00e1 | 2.40e1 | 1.00 | 1.60e1 | 1.40e1 | 8.00 | 9.00 | 1.70e1 | 1.80e1 | 1.10e1 | 1.20e1 | 5.00 | 3.00 | 6.00 | C.former |
| sz_taxi, medium, 15T | MASE | 1.09 | 1.00 | 8.86e-1 | 1.00 | 2.69 | 7.83e-1 | **7.69e-1** | 9.21e-1 | 9.28e-1 | 8.25e-1 | 7.69e-1 | 8.82e-1 | 2.72 | 8.22e-1 | 8.16e-1 | 8.30e-1 | 7.83e-1 | 7.87e-1 | 8.67e-1 | 8.37e-1 | 8.32e-1 | 7.87e-1 | 7.83e-1 | 7.99e-1 | TFT |
| sz_taxi, medium, 15T | CRPS | 4.27 | 1.00 | 7.73e-1 | 1.17 | 1.74e8 | 4.63e-1 | 4.54e-1 | 5.57e-1 | 6.92e-1 | 4.85e-1 | 4.58e-1 | 6.52e-1 | **4.27e-1** | 6.15e-1 | 6.10e-1 | 4.96e-1 | 5.13e-1 | 5.90e-1 | 6.15e-1 | 5.90e-1 | 5.88e-1 | 4.67e-1 | 4.65e-1 | 4.73e-1 | C.former |
| sz_taxi, medium, 15T | Rank | 2.80e1 | 2.60e1 | 2.50e1 | 2.70e1 | 2.90e1 | 5.00 | 2.00 | 1.40e1 | 2.40e1 | 1.10e1 | 3.00 | 2.30e1 | 1.00 | 2.20e1 | 2.00e1 | 1.20e1 | 1.30e1 | 1.70e1 | 2.10e1 | 1.60e1 | 1.50e1 | 7.00 | 6.00 | 1.00e1 | C.former |
| sz_taxi, short, 15T | MASE | 1.03 | 1.00 | 1.00 | 8.49e-1 | 9.11e-1 | 7.88e-1 | 7.89e-1 | 8.08e-1 | 7.91e-1 | 7.33e-1 | 7.50e-1 | 7.62e-1 | 2.42 | 7.74e-1 | 7.64e-1 | 7.98e-1 | **7.30e-1** | 8.19e-1 | 7.83e-1 | 7.71e-1 | 7.68e-1 | 7.53e-1 | 7.48e-1 | 7.61e-1 | T.FM |
| sz_taxi, short, 15T | CRPS | 2.14 | 1.00 | 1.00 | 9.32e-1 | 9.42e-1 | 7.09e-1 | 7.18e-1 | 7.80e-1 | 9.09e-1 | 6.70e-1 | 6.93e-1 | 8.77e-1 | 6.67 | 8.93e-1 | 8.83e-1 | 7.35e-1 | **6.67e-1** | 9.45e-1 | 7.80e-1 | 7.64e-1 | 7.64e-1 | 6.88e-1 | 6.85e-1 | 6.95e-1 | T.FM |
| sz_taxi, short, 15T | Rank | 2.90e1 | 2.80e1 | 2.70e1 | 2.40e1 | 2.50e1 | 1.10e1 | 1.20e1 | 1.70e1 | 2.30e1 | 2.00 | 8.00 | 1.80e1 | 3.00e1 | 2.00e1 | 1.90e1 | 1.30e1 | 1.00 | 2.60e1 | 1.60e1 | 1.40e1 | 1.50e1 | 6.50 | 3.00 | 9.00 | T.FM |
| sz_taxi, short, H | MASE | 1.13 | 1.00 | 8.46e-1 | 9.36e-1 | 1.15 | 7.80e-1 | 8.06e-1 | 8.97e-1 | 8.46e-1 | 8.01e-1 | 7.80e-1 | 9.04e-1 | 4.19 | 8.33e-1 | 8.16e-1 | 9.66e-1 | **7.67e-1** | 8.05e-1 | 7.85e-1 | 7.80e-1 | 7.79e-1 | 8.48e-1 | 7.99e-1 | 8.14e-1 | T.FM |
| sz_taxi, short, H | CRPS | 2.12 | 1.00 | 7.42e-1 | 1.01 | 7.07 | 6.07e-1 | 6.29e-1 | 8.52e-1 | 8.30e-1 | 6.29e-1 | 6.16e-1 | 8.78e-1 | 9.21 | 8.21e-1 | 8.03e-1 | 7.64e-1 | **5.98e-1** | 7.95e-1 | 6.51e-1 | 6.51e-1 | 6.46e-1 | 6.57e-1 | 6.25e-1 | 6.37e-1 | T.FM |
| sz_taxi, short, H | Rank | 2.80e1 | 2.40e1 | 1.60e1 | 2.50e1 | 2.90e1 | 2.00 | 6.00 | 2.20e1 | 2.10e1 | 5.00 | 3.00 | 2.30e1 | 3.00e1 | 2.00e1 | 1.90e1 | 1.70e1 | 1.00 | 1.80e1 | 1.20e1 | 1.10e1 | 1.00e1 | 1.45e1 | 4.00 | 7.00 | T.FM |
| temperature_rain, short, D | MASE | 1.00 | 1.00 | 8.51e-1 | 9.60e-1 | 9.80e-1 | 8.56e-1 | 7.16e-1 | 8.96e-1 | 9.45e-1 | 7.51e-1 | 7.16e-1 | 9.10e-1 | 1.31e1 | 8.36e-1 | 8.16e-1 | 7.51e-1 | 7.06e-1 | 7.96e-1 | 7.16e-1 | 7.01e-1 | 7.01e-1 | 6.66e-1 | 6.54e-1 | **5.97e-1** | Moi.L |
| temperature_rain, short, D | CRPS | 7.79e-1 | 1.00 | 4.26e-1 | 4.67e-1 | 4.63e-1 | 4.18e-1 | 3.63e-1 | 4.28e-1 | 5.15e-1 | 3.95e-1 | 3.88e-1 | 5.09e-1 | 3.72 | 5.01e-1 | 4.79e-1 | 4.02e-1 | 3.56e-1 | 4.79e-1 | 3.85e-1 | 3.74e-1 | 3.74e-1 | 3.39e-1 | 3.28e-1 | **2.94e-1** | Moi.L |
| temperature_rain, short, D | Rank | 2.80e1 | 2.90e1 | 1.70e1 | 2.00e1 | 1.90e1 | 1.60e1 | 9.00 | 1.80e1 | 2.50e1 | 1.40e1 | 1.30e1 | 2.40e1 | 3.00e1 | 2.30e1 | 2.10e1 | 1.50e1 | 8.00 | 2.20e1 | 1.20e1 | 1.10e1 | 1.00e1 | 3.50 | 2.00 | 1.00 | Moi.L |
| us_births, short, D | MASE | 1.00 | 1.00 | 8.49e-1 | 8.76e-1 | 8.60e-1 | 2.88e-1 | **1.69e-1** | 3.35e-1 | 2.45e-1 | 2.62e-1 | 2.81e-1 | 3.47e-1 | 8.87e-1 | 8.49e-1 | 8.60e-1 | 7.63e-1 | 2.97e-1 | 2.99e-1 | 2.34e-1 | 2.26e-1 | 2.33e-1 | 5.81e-1 | 2.70e-1 | 2.70e-1 | TFT |
| us_births, short, D | CRPS | 8.33e-1 | 1.00 | 5.13e-1 | 5.19e-1 | 5.17e-1 | 1.91e-1 | **1.12e-1** | 2.15e-1 | 2.02e-1 | 1.73e-1 | 1.85e-1 | 2.85e-1 | 4.85e-1 | 6.94e-1 | 7.08e-1 | 4.81e-1 | 1.98e-1 | 2.47e-1 | 1.61e-1 | 1.53e-1 | 1.57e-1 | 3.95e-1 | 1.84e-1 | 1.85e-1 | TFT |
| us_births, short, D | Rank | 2.90e1 | 3.00e1 | 2.20e1 | 2.40e1 | 2.30e1 | 9.00 | 1.00 | 1.30e1 | 1.20e1 | 5.00 | 7.00 | 1.70e1 | 2.10e1 | 2.50e1 | 2.60e1 | 2.00e1 | 1.10e1 | 1.50e1 | 4.00 | 2.00 | 3.00 | 1.85e1 | 6.00 | 8.00 | TFT |
| us_births, short, M | MASE | 1.97 | 1.00 | 6.12e-1 | 1.16 | 7.73e-1 | 9.99e-1 | 1.14 | 1.03 | 1.22 | 1.03 | 1.08 | 1.54 | **5.58e-1** | 2.09 | 2.16 | 2.88 | 8.17e-1 | 9.55e-1 | 7.52e-1 | 1.02 | 7.69e-1 | 1.61 | 9.06e-1 | 9.56e-1 | C.former |
| us_births, short, M | CRPS | 2.32 | 1.00 | 6.19e-1 | 1.13 | 7.14e-1 | 9.35e-1 | 1.27 | 1.38 | 1.51 | 9.94e-1 | 1.08 | 1.92 | **1.15e-1** | 2.61 | 2.68 | 2.89 | 7.50e-1 | 1.18 | 7.98e-1 | 1.05 | 7.98e-1 | 1.68 | 8.55e-1 | 8.94e-1 | C.former |
| us_births, short, M | Rank | 2.50e1 | 1.10e1 | 2.00 | 1.40e1 | 3.00 | 9.00 | 1.70e1 | 1.80e1 | 1.90e1 | 1.00e1 | 1.30e1 | 2.30e1 | 1.00 | 2.60e1 | 2.70e1 | 2.80e1 | 4.00 | 1.50e1 | 6.00 | 1.20e1 | 5.00 | 2.05e1 | 7.00 | 8.00 | C.former |
| us_births, short, W | MASE | 1.00 | 1.00 | 9.49e-1 | 9.55e-1 | 9.55e-1 | 9.29e-1 | 1.02 | 7.56e-1 | 8.97e-1 | 7.88e-1 | 1.52 | 9.36e-1 | 2.16 | 1.05 | 1.06 | 1.33 | 6.79e-1 | 8.91e-1 | 5.90e-1 | 5.97e-1 | **5.29e-1** | 1.07 | 9.25e-1 | 9.38e-1 | Chr.L |
| us_births, short, W | CRPS | 8.94e-1 | 1.00 | 8.33e-1 | 8.47e-1 | 8.47e-1 | 7.87e-1 | 8.98e-1 | 7.59e-1 | 9.72e-1 | 6.71e-1 | 1.26 | 1.01 | 4.08 | 1.13 | 1.14 | 1.25 | 5.79e-1 | 9.58e-1 | 5.32e-1 | 5.14e-1 | **4.59e-1** | 9.14e-1 | 8.00e-1 | 8.26e-1 | Chr.L |
| us_births, short, W | Rank | 1.40e1 | 2.10e1 | 1.10e1 | 1.30e1 | 1.20e1 | 8.00 | 1.50e1 | 7.00 | 2.00e1 | 5.00 | 2.70e1 | 2.20e1 | 3.00e1 | 2.40e1 | 2.50e1 | 2.60e1 | 4.00 | 1.90e1 | 3.00 | 2.00 | 1.00 | 1.75e1 | 9.00 | 1.00e1 | Chr.L |

Table 25: `Moirai` vs `Moirai-Leakage` results on datasets that LOTSA collection and our GIFT-Eval has in common.

| Dataset | Model | Short | | Medium | | Long | |
|---|---|---|---|---|---|---|---|
| | | MAPE | CRPS | MAPE | CRPS | MAPE | CRPS |
| hierarchical_sales, D | Moi Leak.$_B$ | 0.51 | 0.24 | NA | NA | NA | NA |
| hierarchical_sales, D | Moi.$_B$ | 0.49 | 0.25 | NA | NA | NA | NA |
| hierarchical_sales, D | Moi Leak.$_L$ | 0.53 | 0.25 | NA | NA | NA | NA |
| hierarchical_sales, D | Moi.$_L$ | 0.52 | 0.24 | NA | NA | NA | NA |
| hierarchical_sale, D | Moi Leak.$_S$ | 0.50 | 0.25 | NA | NA | NA | NA |
| hierarchical_sales, D | Moi.$_S$ | 0.51 | 0.25 | NA | NA | NA | NA |
| loop_seattle, 5T | Moi Leak.$_B$ | 0.67 | 0.57 | 0.42 | 0.37 | 0.50 | 0.38 |
| loop_seattle, 5T | Moi.$_B$ | 0.84 | 0.64 | 0.83 | 0.62 | 0.78 | 0.57 |
| loop_seattle, 5T | Moi Leak.$_L$ | 0.66 | 0.51 | 0.33 | 0.31 | 0.46 | 0.36 |
| loop_seattle, 5T | Moi.$_L$ | 0.83 | 0.65 | 0.85 | 0.65 | 0.81 | 0.59 |
| loop_seattle, 5T | Moi Leak.$_S$ | 0.84 | 0.69 | 0.75 | 0.57 | 0.70 | 0.53 |
| loop_seattle, 5T | Moi.$_S$ | 0.87 | 0.65 | 0.77 | 0.61 | 0.75 | 0.57 |
| loop_seattle, D | Moi Leak.$_B$ | 0.50 | 0.34 | NA | NA | NA | NA |
| loop_seattle, D | Moi.$_B$ | 0.52 | 0.35 | NA | NA | NA | NA |
| loop_seattle, D | Moi Leak.$_L$ | 0.51 | 0.35 | NA | NA | NA | NA |
| loop_seattle, D | Moi.$_L$ | 0.49 | 0.33 | NA | NA | NA | NA |
| loop_seattle, D | Moi Leak.$_S$ | 0.53 | 0.35 | NA | NA | NA | NA |
| loop_seattle, D | Moi.$_S$ | 0.54 | 0.35 | NA | NA | NA | NA |
| loop_seattle, H | Moi Leak.$_B$ | 0.96 | 0.68 | 0.54 | 0.50 | 0.49 | 0.26 |
| loop_seattle, H | Moi.$_B$ | 1.08 | 0.72 | 0.65 | 0.55 | 0.59 | 0.30 |
| loop_seattle, H | Moi Leak.$_L$ | 0.84 | 0.61 | 0.53 | 0.45 | 0.47 | 0.23 |
| loop_seattle, H | Moi.$_L$ | 0.89 | 0.65 | 0.71 | 0.59 | 1.18 | 0.45 |
| loop_seattle, H | Moi Leak.$_S$ | 1.22 | 0.80 | 0.73 | 0.64 | 0.70 | 0.35 |
| loop_seattle, H | Moi.$_S$ | 1.19 | 0.78 | 0.70 | 0.60 | 0.71 | 0.33 |
| m_dense, D | Moi Leak.$_B$ | 0.78 | 0.35 | NA | NA | NA | NA |
| m_dense, D | Moi.$_B$ | 0.55 | 0.27 | NA | NA | NA | NA |
| m_dense, D | Moi Leak.$_L$ | 0.67 | 0.32 | NA | NA | NA | NA |
| m_dense, D | Moi.$_L$ | 0.63 | 0.31 | NA | NA | NA | NA |
| m_dense, D | Moi Leak.$_S$ | 0.58 | 0.28 | NA | NA | NA | NA |
| m_dense, D | Moi.$_S$ | 0.53 | 0.26 | NA | NA | NA | NA |
| m_dense, H | Moi Leak.$_B$ | 0.54 | 0.50 | 0.49 | 0.26 | 0.53 | 0.22 |
| m_dense, H | Moi.$_B$ | 0.65 | 0.55 | 0.59 | 0.30 | 0.61 | 0.26 |
| m_dense, H | Moi Leak.$_L$ | 0.53 | 0.45 | 0.47 | 0.23 | 0.49 | 0.21 |
| m_dense, H | Moi.$_L$ | 0.71 | 0.59 | 1.18 | 0.45 | 1.69 | 0.45 |
| m_dense, H | Moi Leak.$_S$ | 0.73 | 0.64 | 0.70 | 0.35 | 0.71 | 0.31 |
| m_dense, H | Moi.$_S$ | 0.70 | 0.60 | 0.71 | 0.33 | 0.91 | 0.33 |
| restaurant | Moi Leak.$_B$ | 0.70 | 0.29 | NA | NA | NA | NA |
| restaurant | Moi.$_B$ | 0.72 | 0.31 | NA | NA | NA | NA |
| restaurant | Moi Leak.$_L$ | 0.75 | 0.30 | NA | NA | NA | NA |
| restaurant | Moi.$_L$ | 0.76 | 0.30 | NA | NA | NA | NA |
| restaurant | Moi Leak.$_S$ | 0.74 | 0.31 | NA | NA | NA | NA |
| restaurant | Moi.$_S$ | 0.74 | 0.31 | NA | NA | NA | NA |
| sz_taxi, 15T | Moi Leak.$_B$ | 0.90 | 0.69 | 0.71 | 0.33 | 2.16 | 0.38 |
| sz_taxi, 15T | Moi.$_B$ | 0.84 | 0.69 | 0.64 | 0.46 | 2.42 | 0.38 |
| sz_taxi, 15T | Moi Leak.$_L$ | 0.78 | 0.69 | 0.71 | 0.46 | 2.14 | 0.38 |
| sz_taxi, 15T | Moi.$_L$ | 0.82 | 0.69 | 0.60 | 0.47 | 2.24 | 0.38 |
| sz_taxi, 15T | Moi Leak.$_S$ | 0.95 | 0.69 | 0.65 | 0.47 | 2.12 | 0.38 |
| sz_taxi, 15T | Moi.$_S$ | 1.11 | 0.70 | 0.60 | 0.47 | 2.30 | 0.39 |
| sz_taxi, H | Moi Leak.$_B$ | 0.64 | 0.62 | NA | NA | NA | NA |
| sz_taxi, H | Moi.$_B$ | 0.72 | 0.64 | NA | NA | NA | NA |
| sz_taxi, H | Moi Leak.$_L$ | 0.63 | 0.64 | NA | NA | NA | NA |
| sz_taxi, H | Moi.$_L$ | 0.70 | 0.64 | NA | NA | NA | NA |
| sz_taxi, H | Moi Leak.$_S$ | 0.65 | 0.66 | NA | NA | NA | NA |
| sz_taxi, H | Moi.$_S$ | 0.77 | 0.65 | NA | NA | NA | NA |

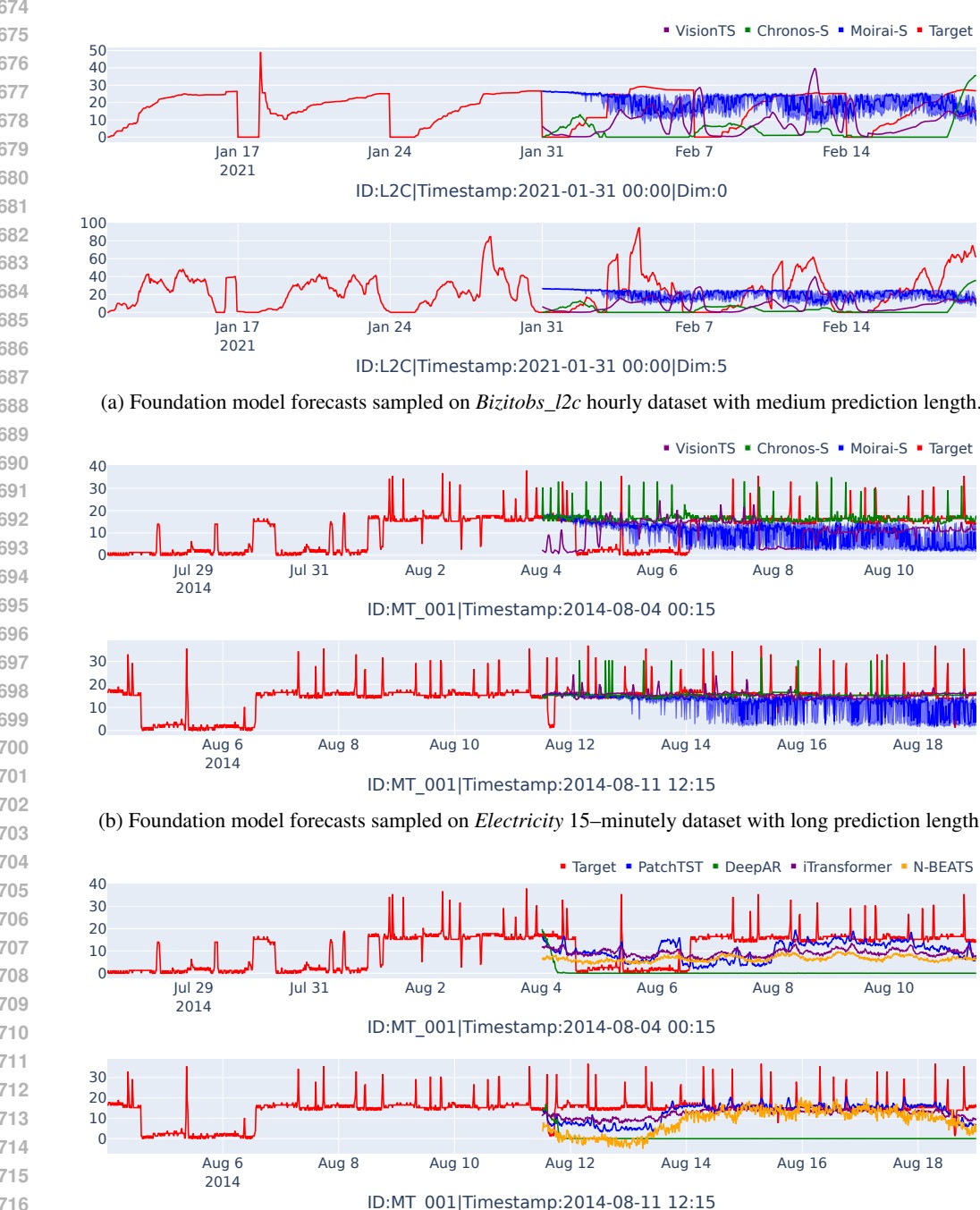

(a) Foundation model forecasts sampled on *Bizitobs_l2c* hourly dataset with medium prediction length.

(b) Foundation model forecasts sampled on *Electricity* 15–minutely dataset with long prediction length.

(c) Deep learning model forecasts sampled on *Electricity* 15–minutely dataset with long prediction length.

Figure 3: Qualitative plots showing forecasts from various deep learning and foundation models on several time series forecasting datasets.

datasets had entirely uncorrelated variates, there would be little justification for using multivariate models, as univariate models could predict each variate independently with comparable effectiveness.]

R: [ Figure 4 illustrates the correlation matrices for each multivariate dataset, highlighting the degree of inter-variate correlation. Specific statistics for each dataset are provided in the respective figure captions to offer additional insights into their characteristics.]

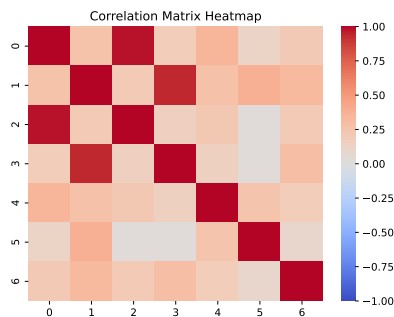

(a) Correlation matrix for Dataset ETT1. Mean: 0.38, Median: 0.25, Std: 0.33.

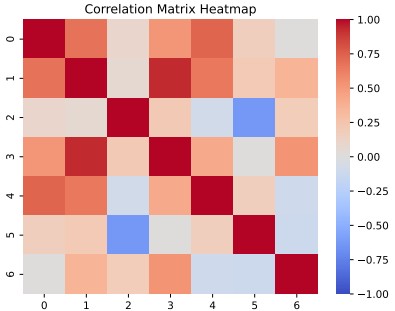

(b) Correlation matrix for Dataset ETT2. Mean: 0.34, Median: 0.21, Std: 0.41.

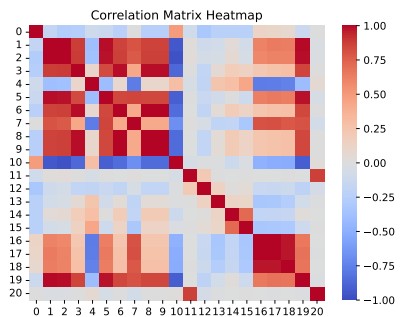

(c) Correlation matrix for Dataset Jena Weather. Mean: 0.18, Median: 0.03, Std: 0.50.

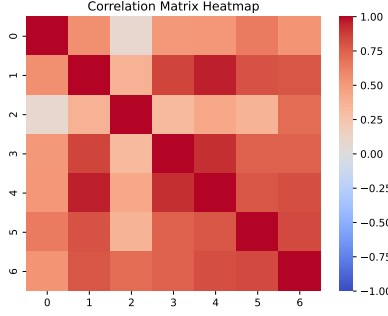

(d) Correlation matrix for Dataset Bizitobs_l2c. Mean: 0.68, Median: 0.74, Std: 0.24.

Figure 4: R: [Inter-variate correlation matrices for selected multivariate datasets in GIFT-Eval. Each heatmap visualizes the correlation across variates for a specific dataset, highlighting the strength and distribution of inter-variate dependencies. Descriptive statistics (mean, median, standard deviation) are provided in the subcaptions for further insight.]

