# OpenReview forum: "GIFT-Eval: A Benchmark for General Time Series Forecasting Model Evaluation"
_ICLR.cc/2025/Conference — Submitted to ICLR 2025_

### Official Review · Reviewer_Pz22 · 2024-11-03

**Soundness:** 3
**Presentation:** 2
**Contribution:** 3
**Rating:** 5
**Confidence:** 4

**Summary:**

The existing benchmark evaluations are incomplete (lacking evaluation of foundation models), so the paper introduces the General Time Series Forecasting Model Evaluation (GIFT-Eval), including statistical models, deep learning models, and foundation models. It tests on 28 different test datasets and provides a large-scale pretraining dataset to better evaluate foundation models. Finally, the paper offers a qualitative analysis that spans both deep learning models and foundation models.

**Strengths:**

S1: Model Level: GIFT-Eval provides a comprehensive evaluation for time series forecasting models, including statistical models, deep learning models, and foundation models.

S2: Experiment and Dataset Level: GIFT-Eval conducts extensive experiments, testing on 28 different test datasets, and also provides a non-leaking pretraining dataset to better evaluate foundation models

**Weaknesses:**

W1: I believe that analyzing foundation models based on four time series characteristics—domain, frequency, prediction length, and the number of variates—combined with six time series features—trend, seasonality, entropy, Hurst, stability, and lumpiness—is not very meaningful, especially regarding the number of variates. I don't think it has any relation to these six features. Why not directly analyze the time series features for each test dataset, and then evaluate the performance of foundation models on various datasets to assess their strengths and weaknesses concerning these six features?

W2: The paper only analyzes the features of the test data. It is necessary to include an analysis of the pretraining data as well. This would help better assess whether the foundation models perform well on these features due to the presence of these characteristics in the pretraining data itself, the generalization ability of the foundation models, or other reasons.

W3: Please elaborate on the details of the data splitting for train/val/test datasets, specifically how the validation data is constructed. What does the statement "The final window of the training data serves as validation" mean, and why is a common splitting ratio like 7:1:2 or 6:2:2 not used?

W4: The experimental settings are unclear, for example: what are the input and output lengths for short-term, medium-term, and long-term forecasting? Are the foundation models performing zero-shot or full-shot forecasting in Tables 6-10? Why are common metrics like MSE and MAE not chosen for point forecasting?

W5: About experimental results: 1) PatchTST performs best on multivariate datasets, so why is Moirai, a model specifically designed for variable correlation, not used? Also, why do foundation models perform well on univariate data but not on multivariate data? 2) The paper states that Moirai has good prediction performance in short-term forecasting; does this conclusion contradict the original Moirai paper, which uses very long input sequences?

W6: The qualitative analysis is relatively limited; more in-depth analysis (not just visualization) may be needed to highlight the characteristics of the foundation models and draw more meaningful conclusions.

**Questions:**

See W1-W6.

---

> ### Author Response · Authors · 2024-11-22
> **Rebuttal 1/n**
>
> Dear reviewer, thank you for the effort you put into reviewing our paper, and thank you for appreciating the scale of our collected data and the extensive amount of experiments we have conducted. We have carefully addressed each of your concerns and strengthened our presentation accordingly. Please find all responses below:
>
> **“W1: I believe that analyzing foundation models based on four time series characteristics—domain, frequency, prediction length, and the number of variates—combined with six time series features—trend, seasonality, entropy, Hurst, stability, and lumpiness—is not very meaningful, especially regarding the number of variates. I don't think it has any relation to these six features. Why not directly analyze the time series features for each test dataset, and then evaluate the performance of foundation models on various datasets to assess their strengths and weaknesses concerning these six features?”**
>
> Thank you for your comment. We believe analyzing models across these six time series characteristics is important mainly for two reasons: (1) Identifying a model’s dominant weaknesses provides valuable insights for improving both model architectures and datasets, e.g. some models designs may specialize in multivariate forecasting [1,4,5,6] or support for diverse frequency [1,2,3]; and (2) From a user perspective, understanding performance across specific characteristics (e.g., frequency or number of variates) helps in selecting the right model for their use case. For example, a user needing a weather forecasting model can prioritize forecasters that perform best on daily or weekly frequencies over second-level granularity.
>
> That said, we agree with the reviewer that extending this analysis to include time series features for each test dataset is highly valuable. We have incorporated this into our paper, as detailed in the new section in Appendix F.1 and Table 16. This breakdown aggregates results based on time series features (e.g., trend, seasonality, entropy) rather than just dataset characteristics. Our findings show that deep learning models, particularly Transformer-based ones like PatchTST, excel in challenging scenarios with low temporal strength and high entropy, demonstrating strong generalist performance. In contrast, foundation models such as Moirai perform better in simpler, more predictable scenarios, consistent with Moirai-Large's strong results in less complex forecasting tasks. This also aligns with our aggregated results where Moirai ranked highly in majority of the forecasting scenarios yet PatchTST got better aggregated metric results as it can achieve reasonably well on all scenarios including challenging ones. We also note that these performance differences may partly reflect the supervised learning setups and hyperparameter tuning, which often favor deep learning models in more complex scenarios. We hope this extended analysis helps the community understand these nuances better without needing to re-analyze the data, especially as more models are added to our benchmark.
>
> **“W2: The paper only analyzes the features of the test data. It is necessary to include an analysis of the pretraining data as well. This would help better assess whether the foundation models perform well on these features due to the presence of these characteristics in the pretraining data itself, the generalization ability of the foundation models, or other reasons.”**
>
> Thank you for raising this important point. While we agree that analyzing the pretraining data would provide valuable insights into whether foundation models perform well due to characteristics in the pretraining data, their generalization ability, or other factors, conducting such an analysis is unfortunately beyond our current computational resources.
> For context, the analysis on the test data required approximately a week to complete using a compute server with 96 cores. Our pretraining data, however, is nearly 1,000 times larger than the test set, making such an analysis computationally infeasible with our available resources.
> We hope that by publicizing our pretraining dataset, other researchers with access to greater computational resources can contribute to this effort in the future. Meanwhile, our analysis of the test data provides meaningful insights into model performance across diverse characteristics.

---

> > ### Author Response · Authors · 2024-11-22
> > **Rebuttal 2/n**
> >
> > **“W3: Please elaborate on the details of the data splitting for train/val/test datasets, specifically how the validation data is constructed. What does the statement "The final window of the training data serves as validation" mean, and why is a common splitting ratio like 7:1:2 or 6:2:2 not used?”**
> >
> > Thank you for your question. Our train-val-test split design had to accommodate the diverse range of datasets included in our benchmark. We determined the number of windows for each dataset based on the length of the shortest time series, ensuring that at least 10% of the data is consistently reserved for the test set across all time series. For the remaining data, we designated the last window as the validation set and used the preceding windows for training. This approach ensures meaningful splits while adapting to the varying lengths of the time series, making the benchmark robust and consistent across datasets.
> > We did not adopt a standard splitting ratio (e.g., 7:1:2 or 6:2:2) because such fixed ratios might not provide meaningful splits for datasets with shorter time series, potentially compromising the integrity of the test set.
> > We hope this explanation clarifies our approach and the rationale behind it.
> >
> > **“W4: The experimental settings are unclear, for example: what are the input and output lengths for short-term, medium-term, and long-term forecasting? Are the foundation models performing zero-shot or full-shot forecasting in Tables 6-10? Why are common metrics like MSE and MAE not chosen for point forecasting?”**
> >
> > Thank you for your question. We address your concerns below:
> >
> >
> > Input and Output Lengths:
> > Prediction lengths are determined based on the sampling frequency of each dataset. Table 14 in our paper lists the specific prediction lengths for all 97 configurations tested in our benchmark. Input lengths depend on the model implementation:
> >
> >
> > For foundation models, we follow the default settings provided by their respective authors if not mentioned otherwise (details in Appendix A).
> > For deep learning models, we treat the context length as a tunable hyperparameter, searching over the range [1,2,4,8]×prediction length.
> >
> >
> > Zero-Shot vs. Full-Shot Forecasting (Tables 6–10):
> > The foundation models in Tables 6–10 (Now Tables 2-6) and everywhere else are evaluated in a zero-shot setting, as clarified in Section 4 of our paper. We did not evaluate any foundation models in few or full-shot settings in our paper.
> >
> > Choice of Metrics:
> >
> > Thank you for your suggestion. Other reviewers have also brought up similar concerns. Following all reviewer suggestions we omit MAPE as an evaluation metric from our paper. Each reviewer had a different suggestion for metrics to add. Here is how we addressed all of their concerns:
> >
> > The main paper tables (Table 2 through 7) in the paper are updated to show MASE rather than MAPE results and we use geometric mean to aggregate results instead of arithmetic mean.
> > We update the appendix tables (Tables 17 through 21) which report results for all models with all metrics suggested by reviewers for the sake of completeness. Specifically we report results with sMAPE, MASE, ND, MSE, MAE, CRPS and finally Rank metrics.
> >
> > We hope this addresses the concerns around the evaluation metrics.

---

> > > ### Author Response · Authors · 2024-11-22
> > > **Rebuttal 3/n**
> > >
> > > **“W5: About experimental results: 1) PatchTST performs best on multivariate datasets, so why is Moirai, a model specifically designed for variable correlation, not used? Also, why do foundation models perform well on univariate data but not on multivariate data? 2) The paper states that Moirai has good prediction performance in short-term forecasting; does this conclusion contradict the original Moirai paper, which uses very long input sequences?”**
> > >
> > > Thank you for your questions. We address them below:
> > >
> > > PatchTST vs. Moirai on Multivariate Data:
> > > We would like to clarify that all models, including Moirai, were used in all experiments. While PatchTST outperforms on certain multivariate datasets, Moirai, the only foundation model in our benchmark explicitly designed to leverage cross-variate relations in the output, outperforms other foundation models in multivariate tasks. However, foundation models as a group still lag behind certain deep learning models like PatchTST in this area. Thus our benchmark highlights an important weakness in current foundation models’ capability to perform multivariate forecasting.
> > >
> > > Short-Term Forecasting and Moirai:
> > > To clarify, short-term forecasting in our paper refers to short prediction lengths, not context lengths. This distinction aligns with the original Moirai paper, which does not claim poor performance for short prediction lengths. Thus, our conclusions are consistent with the original findings.
> > >
> > >
> > > **“W6: The qualitative analysis is relatively limited; more in-depth analysis (not just visualization) may be needed to highlight the characteristics of the foundation models and draw more meaningful conclusions.”**
> > >
> > > We appreciate the reviewer’s emphasis on deeper analysis and agree that providing more meaningful insights is valuable to the research community. Our primary goal is to highlight the strengths and weaknesses of different model families to advance time series research. However, with 24 models evaluated across more than 20 datasets, explaining why specific families perform differently on certain datasets is beyond the scope of this paper, as these remain open research questions.
> > >
> > > That said, we have expanded our analysis as suggested. In Appendix F.1 and Table 16, we present a detailed breakdown aggregating results by time series features (e.g., trend, entropy) rather than just dataset characteristics. Our findings reveal that Transformer-based deep learning models like PatchTST excel in challenging scenarios with low temporal strength and high entropy, while foundation models such as Moirai perform better in simpler, more predictable cases. This aligns with our aggregated results, where Moirai-Large ranks highly in less complex forecasting scenarios. We also note that these performance differences may partly reflect supervised learning setups and hyperparameter tuning, which can favor deep learning models in difficult scenarios. We hope this extended analysis provides the community with nuanced insights without requiring additional data re-analysis, especially as more models are added to our benchmark.
> > >
> > > [1] Unified Training of Universal Time Series Forecasting Transformers, https://arxiv.org/pdf/2402.02592
> > >
> > > [2] Self-Supervised Contrastive Pre-Training for Time Series via Time-Frequency Consistency, https://arxiv.org/pdf/2206.08496
> > >
> > > [3] FiLM: Frequency improved Legendre Memory Model for Long-term Time Series Forecasting, https://arxiv.org/abs/2205.08897
> > >
> > > [4] MULTIVARIATE PROBABILISTIC TIME SERIES FORECASTING VIA CONDITIONED NORMALIZING FLOWS, https://arxiv.org/pdf/2002.06103
> > >
> > > [5] CROSSFORMER: TRANSFORMER UTILIZING CROSSDIMENSION DEPENDENCY FOR MULTIVARIATE TIME
> > > SERIES FORECASTING, https://openreview.net/pdf?id=vSVLM2j9eie
> > >
> > > [6] CATN: Cross Attentive Tree-Aware Network for Multivariate Time Series Forecasting, https://cdn.aaai.org/ojs/20320/20320-13-24333-1-2-20220628.pdf

---

> ### Comment · Reviewer_Pz22 · 2024-11-23
>
> Thanks to the author for their rebuttal. Although the author has addressed these questions, some issues remain unresolved. The main points are as follows:
>
> - **W1**: The author only mentioned that analyzing the four time series characteristics is important but did not discuss the significance of combining time series characteristics—such as domain, prediction length, and the number of variates—with the six features—trend, seasonality, entropy, Hurst, stability, and lumpiness. I still believe the four characteristics are unrelated to the six features. Specifically, could the author explain the relationship between the number of variates and the six features?
>
>     Additionally, the author's response referenced six related works, but three of them were not tested or compared.
>
> - **W3**: Why can’t some small datasets be split using the standard splitting ratio? Is it because the input length is too long, resulting in a very small number of test samples after splitting, or is it due to the dataset itself being very small (with only a few hundred time stamps)?
> - **W4**: According to the author’s response, the input lengths of foundation models and deep learning models are inconsistent. I believe the input length significantly impacts the model's performance, and this comparison may be unfair. Additionally, has the conclusion changed when using MSE and MAE as the metrics?
> - **W6**: I still believe the author’s qualitative analysis lacks depth and still requires more meaningful conclusions. Additionally, the author mentioned that deep learning models excel in complex forecasting scenarios, while foundation models perform less well in such scenarios. This phenomenon contradicts the original intention of foundation models (due to pre-training on multi-source datasets, foundation models have strong generalization capabilities). Could you explain the reason behind this?

---

> > ### Author Response · Authors · 2024-11-24
> >
> > Dear Reviewer, thank you for your response. Please find below further clarifications to your questions:
> >
> > **“W1: The author only mentioned that analyzing the four time series characteristics is important but did not discuss the significance of combining time series characteristics—such as domain, prediction length, and the number of variates—with the six features—trend, seasonality, entropy, Hurst, stability, and lumpiness. I still believe the four characteristics are unrelated to the six features. Specifically, could the author explain the relationship between the number of variates and the six features?
> > Additionally, the author's response referenced six related works, but three of them were not tested or compared.”**
> >
> > Thank you for the clarification. The main insight from Figure 1 is that datasets with different characteristics (e.g., number of variates, frequency, domain) can exhibit vastly different time series features such as trend, seasonality, entropy, and stability. This analysis is presented as a justification for our approach of diversifying these characteristics in GIFT-Eval to evaluate models across a broad spectrum of time series data.
> >
> > Specifically regarding the number of variates, Figure 1(a) shows that multivariate datasets tend to have higher stability and lumpiness values, indicating greater variance fluctuations and complexity across segments. This suggests that multivariate time series are generally more challenging to model. In contrast, univariate datasets exhibit stronger seasonal strength, reflecting more regular repeating patterns and predictability over certain periods. These distinctions highlight why evaluating models on both univariate and multivariate data is essential for understanding their capabilities.
> >
> > As for the related works referenced but not implemented, we acknowledge that there are many other time series forecasting baselines we have not yet incorporated. Within our resource constraints, we aimed to be as comprehensive as possible, implementing 20 baselines and continuing to add more. The references in question were provided to explain the reasoning behind our choices.
> >
> > **“W3: Why can’t some small datasets be split using the standard splitting ratio? Is it because the input length is too long, resulting in a very small number of test samples after splitting, or is it due to the dataset itself being very small (with only a few hundred time stamps)?”**
> >
> > Thank you for the question. Some may contain time series instances with very few timestamps as you mentioned. Our framework (following the GluonTS approach) splits data into train, validation, and test sets by specifying the number of windows to be used across each each time series. Using fixed numbers of windows (e.g., 7 for training, 1 for validation, and 2 for testing) across all datasets could severely constrain window lengths for datasets with very short time series.
> >
> > Instead, we adopt a dynamic approach. Prediction lengths are fixed across dataset, and the number of windows is determined by the shortest time series in each dataset, ensuring that at least 10% of the data is reserved for testing. This ensures consistency in prediction lengths while maintaining sufficient data for evaluation, even for small datasets.

---

> > > ### Author Response · Authors · 2024-11-24
> > >
> > > **“W4: According to the author’s response, the input lengths of foundation models and deep learning models are inconsistent. I believe the input length significantly impacts the model's performance, and this comparison may be unfair.”**
> > >
> > > Thank you for your question. We believe the reviewer is referring to the following distinction in input length configurations:
> > >
> > > Foundation Models:
> > > These are evaluated in a zero-shot setting, using default input lengths specified by their respective authors (details in Appendix A). No hyperparameter tuning, including for context length, is performed for foundation models, as this would go against the motivation of universal forecasters.
> > >
> > > Deep Learning Models:
> > > These are fine-tuned for each dataset, with context length treated as a tunable hyperparameter along with other parameters, to optimize their performance.
> > >
> > > This setup is common across other papers where they compare foundation models in zero-shot, while deep learning models are evaluated in full-shot mode after hyperparameter tuning [1,2,3]. While this creates a difference in input length configurations, it aligns with the goals of evaluating foundation models as universal forecasters. Fine-tuning foundation models for specific datasets would compromise this premise.
> > >
> > > **"Additionally, has the conclusion changed when using MSE and MAE as the metrics?"**
> > >
> > > Changing the evaluation metrics impacted a few small conclusions by altering the best-performing models in point forecasts for certain characteristics. However, these changes did not substantially affect the overall aggregated results presented in Table 6. We changed the relevant parts of the paper to reflect these changes.  The RANK metric, calculated based on CRPS, remains unchanged. Below, we detail the specific shifts:
> > >
> > > 1. Domain
> > >
> > >     Energy: Moirai → Chronos
> > >
> > >     Nature: Chronos → Moirai
> > >
> > >     Sales: Moirai → PatchTST
> > >
> > >     Transport: N-BEATS → Moirai
> > >
> > >
> > > 2. Prediction Length
> > >
> > >     Long & Medium: Moirai → VisionTS
> > >
> > > 3. Frequency:
> > >
> > >     10T: TFT → VisionTS
> > >
> > >     15T: iTransformer → PatchTST
> > >
> > >     Hourly: PatchTST → Chronos
> > >
> > >     Monthly: TimesFM → AutoARIMA
> > >
> > > 4. Number of variates:
> > >
> > >     Multi v.: PatchTST → VisionTS
> > >
> > > [1] Chronos: Learning the Language of Time Series, https://arxiv.org/abs/2403.07815
> > >
> > > [2] Unified Training of Universal Time Series Forecasting Transformers, https://arxiv.org/pdf/2402.02592
> > >
> > > [3] Lag-Llama: Towards Foundation Models for Probabilistic Time Series Forecasting https://arxiv.org/pdf/2310.08278
> > >
> > > **“W6: I still believe the author’s qualitative analysis lacks depth and still requires more meaningful conclusions. Additionally, the author mentioned that deep learning models excel in complex forecasting scenarios, while foundation models perform less well in such scenarios. This phenomenon contradicts the original intention of foundation models (due to pre-training on multi-source datasets, foundation models have strong generalization capabilities). Could you explain the reason behind this?”**
> > >
> > > Thank you for your question. We have made significant efforts to expand the analysis in our paper, including preliminary dataset analyses, results analyses based on time series characteristics and features, and drawing connections between the two. We believe these analyses offer valuable insights for the community, highlighting areas for future focus. Beyond the analysis, we also see our benchmark's value in providing a shared, diverse testbed that allows researchers to easily evaluate and compare their models within the broader context of time series forecasters.
> > >
> > > The observed performance gap in complex forecasting scenarios reflects foundational differences in model design and evaluation. Foundation models excel at generalization but are tested in a zero-shot setting without dataset-specific fine-tuning, unlike deep learning models, which are fine-tuned and benefit from extensive hyperparameter optimization. This tuning advantage allows deep learning models to better adapt to challenging patterns. Foundation models, still in their early stages, resemble large language models (LLMs) in their infancy when specialized models often outperformed them in zero-shot tasks. While scaling data and training improved LLMs, our experiments suggest time series foundation models may require different strategies, as no clear scaling law has yet emerged.
> > > We hope we are able to address some more of your concerns.
> > >
> > > We hope we are able to address some more of your concerns. Thank you once again for your valuable comments and the time you have dedicated to reviewing our work.

---

> > > > ### Author Response · Authors · 2024-11-28
> > > > **Thank you for your engagement**
> > > >
> > > > Dear Reviewer Pz22,
> > > >
> > > > Thank you for your continued engagement and valuable feedback on our paper. As we approach the final deadline for uploading the revised PDF, we wanted to check if our earlier response adequately addressed your remaining concerns. Please let us know if there are any additional points you would like us to clarify.

---

> > > > > ### Author Response · Authors · 2024-12-02
> > > > > **Thank you for your feedback**
> > > > >
> > > > > Dear Reviewer Pz22,
> > > > >
> > > > > Thank you for your detailed feedback and engagement throughout the review process. As we enter the last 24 hours of the reviewer response period, we wanted to briefly summarize the new points we clarified in our responses:
> > > > >
> > > > > - (W1): We highlighted how characteristics like the number of variates relate to features such as stability and seasonality. For example, multivariate datasets exhibit higher stability and lumpiness, reflecting greater variance fluctuations and complexity. We also addressed the inclusion of related works not directly implemented due to resource constraints.
> > > > > - (W3): We clarified that the challenge for splitting ratios arises from datasets with very short time series. Our dynamic approach ensures consistent prediction lengths and sufficient test data, even for small datasets, by adjusting the number of windows per dataset.
> > > > > - (W4): We justified differences in input length configurations for foundation and deep learning models, aligning with zero-shot evaluation goals. We also shared that changes in evaluation metrics (e.g., MSE and MAE) impacted some conclusions but did not alter overall aggregated results or key insights.
> > > > > - (W6): We expanded the depth of analysis in our paper, drawing meaningful insights and connections between time series characteristics and features. The observed performance gap reflects the advantage of dataset-specific fine-tuning for deep learning models compared to the generalization focus of zero-shot foundation models, which remain in early development stages.
> > > > >
> > > > > We hope these clarifications address your remaining concerns and provide further insights into our work. As the rebuttal period concludes, we look forward to any updates from you. Thank you again for your thoughtful comments and the time you’ve dedicated to reviewing our work.

---

### Official Review · Reviewer_ah9K · 2024-11-03

**Soundness:** 2
**Presentation:** 3
**Contribution:** 3
**Rating:** 6
**Confidence:** 5

**Summary:**

In this work the authors introduce a large collective time series benchmark named GIFT-EVAL. They demonstrate the diversity and legitimacy of this benchmark with the analyses on both the characteristics of involved datasets and the benchmark results of state-of-the-art supervised and foundation time series models on it.

**Strengths:**

This work address an important and pressing problem of the time series forecasting community that there lacks a large and comprehensive common benchmark. The collection and complication of the datasets itself is a significant result.

**Weaknesses:**

Despite the comprehensive data analyses, the paper currently lacks the reasoning behind the selection of GIFT-EVAL components. And the empirical study follows suboptimal standards which makes the results less convincing. See questions.

**Questions:**

Regarding the creation of GIFT-EVAL:
1. As mentioned in the weakness, GIFT-EVAL seems to be a straightforward ensemble of some available datasets. What are the reasons behind the selection of this particular ensemble vs the datasets, especially given the data analytics done on them?

Regarding benchmarking:
1. My major concern of the work: MAPE is well recognized as a bad point forecasting metric on its own, due to e.g. its favoring underprediction, sensitivity near ground truth 0 (this is recognized by the authors too), (see e.g., [1] and many community posts online). It would be helpful to switch to or also report other more robust metrics, e.g. MASE.
2. In case normalized metrics are used, consider reporting geometric means [2].
3. Given the 6 properties of component datasets, it would be helpful to see the benchmark results sliced on these properties as well.

[1] Goodwin, Paul, and Richard Lawton. "On the asymmetry of the symmetric MAPE." International journal of forecasting 15.4 (1999): 405-408.
[2] Fleming, Philip J., and John J. Wallace. "How not to lie with statistics: the correct way to summarize benchmark results." Communications of the ACM 29.3 (1986): 218-221.

---

> ### Author Response · Authors · 2024-11-22
> **Rebuttal 1/1**
>
> Dear reviewer, thank you for appreciating the scale of our collected data and recognizing the problem our benchmark addresses for the time series forecasting community. We have carefully addressed each of your concerns and strengthened our presentation accordingly. Please find all responses below:
>
> **“My major concern of the work: MAPE is well recognized as a bad point forecasting metric on its own, due to e.g. its favoring underprediction, sensitivity near ground truth 0 (this is recognized by the authors too), (see e.g., [1] and many community posts online). It would be helpful to switch to or also report other more robust metrics, e.g. MASE.”**
>
> **“In case normalized metrics are used, consider reporting geometric means [2].”**
>
> Thank you for your suggestion. Other reviewers have also brought up similar concerns. Following all reviewer suggestions we omit MAPE as an evaluation metric from our paper. Each reviewer had a different suggestion for metrics to add. Here is how we addressed all of their concerns:
>
> The main paper tables (Table 2 through 7) in the paper are updated to show MASE rather than MAPE results and we use geometric mean to aggregate results instead of arithmetic mean.
> We update the appendix tables (Tables 17 through 21) which report results for all models with all metrics suggested by reviewers for the sake of completeness. Specifically we report results with sMAPE, MASE, ND, MSE, MAE, CRPS and finally Rank metrics.
>
> We hope this addresses the concerns around the evaluation metrics.
>
>
> **“Given the 6 properties of component datasets, it would be helpful to see the benchmark results sliced on these properties as well.”**
>
> Thank you, this is a great suggestion. We report these results in a new section in Appendix F.1 and Table 16, where we provide a detailed breakdown that aggregates results based on time series features rather than just dataset characteristics, discussing the strengths and weaknesses of different model families. Briefly, our new findings indicate that deep learning models, particularly Transformer-based ones like PatchTST, tend to excel in challenging scenarios with low temporal strength and high entropy, showing strong generalist performance. Conversely, foundation models such as Moirai perform better in simpler, more predictable cases. This also aligns with our aggregated results where Moirai ranked highly in majority of the forecasting scenarios yet PatchTST got better aggregated metric results as it can achieve reasonably well on all scenarios including challenging ones. We also note that the performance differences likely reflect the supervised learning setups and hyperparameter tuning, which can give deep learning models an edge in difficult predictions.
>
>  We hope this extended analysis will help the community better understand these nuances without having to re-analyze the data from scratch, especially with the growing number of models added into our benchmark.

---

> > ### Comment · Reviewer_ah9K · 2024-11-22
> >
> > Thanks for the revision which addressed most of my concerns.

---

### Official Review · Reviewer_SxEF · 2024-11-04

**Soundness:** 2
**Presentation:** 2
**Contribution:** 2
**Rating:** 5
**Confidence:** 5

**Summary:**

This paper introduces GIFT-Eval, a benchmark aimed at evaluation for time series foundation models across diverse datasets. GIFT-Eval encompasses 28 datasets and divides them according to 4 characteristics and 6 features. It also includes a non-leaking pretraining dataset. It then evaluates 17 different baseline models, including 4 types of foundation models and some statistical and deep learning models on this benchmark. Based on the evaluation results, it discusses different models in the context of various benchmark characteristics and offers some qualitative analysis.

**Strengths:**

* Foundation models are a new and promising research topic in time series forecasting. It is of value to establish a benchmark for evaluating these models.

* This paper does a lot of work in collecting large-scale data for evaluation and running various baseline models on these various datasets.

**Weaknesses:**

* This paper emphasizes the inclusion of a non-leaking pretraining dataset, but its value and usage are not clear. Is this dataset used to re-train all foundation models instead of using their public checkpoints? Is it necessarily better than the original pretraining datasets of each foundation model? Since the application on downstream data is the main goal of foundation models, do we really need to keep consistency in pretraining to evaluate these models as discussed in the Introduction?

* Section 3.1.1 mentions covariates in time series forecasting, but it is unclear how are the covariates considered in this benchmark. Can all baselines in the benchmark perform forecasting with covariates?

* The divisions of some characteristics such as variates and frequencies are not reasonable enough, and it is unclear why we should evaluate methods according to these characteristics. For example, Multivariate v.s. Univariate may not be the intrinsic differences that influence forecasting, as even some multivariate datasets do not have strong correlations between variates. So simply claiming that some models perform best on multivariate or univariate datasets may be misleading.

* The experiments mainly point out that some specific models perform best in some specific datasets. There are not many consistent conclusions from the evaluations that help us understand the characteristics of different models. Some results or claims are also confusing. In Page 7, how can we get the conclusion that ‘foundation models consistently outperform both statistical and deep learning models’ from Table 6, since PatchTST also performs very well? In Page 8, what is the meaning of ‘This trend indicates that the fine-tuning of foundation models effectively captures longer-term dependencies’? In Page 9, why does Moirai, which is a foundation model considering multivariate correlation, perform best on univariate datasets instead?

* The qualitative results only show results in some special cases. It is doubtful whether these phenomena and analyses are general to all different data. It is also confusing that different data are used to evaluate foundation models and other models, which makes these models incomparable.

**Questions:**

Please see Weaknesses.

---

> ### Author Response · Authors · 2024-11-22
> **Rebuttal 1/n**
>
> Dear reviewer, thank you for taking the time to review our submission and for recognizing our benchmark as a promising means for evaluating foundation models. We greatly appreciate your positive feedback on the scale of our collected data and the extensive experiments we conducted. We have carefully addressed each of your concerns and strengthened our presentation accordingly. Please find all responses below:
>
> **“This paper emphasizes the inclusion of a non-leaking pretraining dataset, but its value and usage are not clear. Is this dataset used to re-train all foundation models instead of using their public checkpoints? Is it necessarily better than the original pretraining datasets of each foundation model? Since the application on downstream data is the main goal of foundation models, do we really need to keep consistency in pretraining to evaluate these models as discussed in the Introduction?”**
>
> Thank you for the question. The purpose of providing pretraining data is not to claim it is superior to the original datasets used by each foundation model, nor to prescribe its usage for new pretraining efforts. Rather, it ensures researchers have access to a non-leaking pretraining dataset with our evaluation set. In our paper, we demonstrate the efficacy of this dataset by re-training Moirai variants and comparing its public version with the retrained version in Appendix F.3. While re-training all foundation models on the new data split would provide a more comprehensive comparison, resource constraints allowed us to re-train only one model from scratch.
>
> For fairness, all tables except F.3. use public versions of foundation models. As noted in Section 4, many of these models may involve some level of data leakage into our test set. However, limiting the benchmark to datasets untouched by existing pretraining efforts would severely restrict its diversity and utility. Our focus is on creating a diverse benchmark for broad evaluation, even if this involves datasets overlapping with some pretraining sets. This is an issue that has also been acknowledged by NLP benchmark papers and addressed similarly [2].
>
> Finally, while it is unrealistic for a single entity to re-train all foundation models, publicizing our pretraining datasets facilitates collaborative scaling of this effort. This approach was also implemented in other NLP benchmark papers [1].
>
> [1] XGLUE: A New Benchmark Dataset for Cross-lingual Pre-training, Understanding and Generation, https://aclanthology.org/2020.emnlp-main.484/
>
> [2] Beyond the Imitation Game: Quantifying and extrapolating the capabilities of language models, https://arxiv.org/pdf/2206.04615
>
> **“Section 3.1.1 mentions covariates in time series forecasting, but it is unclear how are the covariates considered in this benchmark. Can all baselines in the benchmark perform forecasting with covariates?”**
>
> Thank you for this insightful question. We believe that support for covariates is an advantage that models capable of utilizing them should leverage. Accordingly, in our experiments, we include covariates for models that support them, while others rely solely on the input variate. We noticed that our paper is missing number of covariates information for test datasets, we updated Table 14 with the relevant information in our paper.

---

> > ### Author Response · Authors · 2024-11-22
> > **Rebuttal 2/n**
> >
> > **“The divisions of some characteristics such as variates and frequencies are not reasonable enough, and it is unclear why we should evaluate methods according to these characteristics. For example, Multivariate v.s. Univariate may not be the intrinsic differences that influence forecasting, as even some multivariate datasets do not have strong correlations between variates. So simply claiming that some models perform best on multivariate or univariate datasets may be misleading.”**
> >
> > Thank you for your remark. However, we respectfully disagree. Frequency is a fundamental characteristic to evaluate forecasting methods as it provides critical insights into model weaknesses. Some foundation models are explicitly designed to address different frequencies effectively [1,2,3]. Frequencies are also directly tied to practical use cases. For instance, a user leveraging a foundation model for weather forecasting may prioritize daily or weekly predictions, while second-level granularity might not be as crucial. Including evaluations across frequencies ensures that benchmarks shed light into diverse, real-world applications.
> >
> > We hold a similar stance regarding multivariate vs. univariate forecasting. Many recently proposed models, both foundation and non-foundation, are specifically designed to handle multivariate forecasting (e.g., [1,4,5,6]). Evaluating this property is essential to provide standardized, comparable results for models tackling these tasks. For instance, in healthcare, multivariate forecasting is crucial when predicting a patient’s vital signs, as interdependencies between variables like heart rate, blood pressure, and oxygen levels influence outcomes. In energy systems, multivariate forecasting of variables like temperature, wind speed, and energy consumption helps optimize grid performance and renewable energy integration. By testing models on both multivariate and univariate datasets, GIFT-Eval ensures fair and comprehensive comparisons, enabling researchers to assess models' suitability for different forecasting needs.
> >
> > To address the concern that some multivariate datasets may not have strong correlations between variates, we conducted an analysis across multivariate datasets in our benchmark. These new results are included in Section F.5 and Figure 4 of our appendix. Our findings indicate that all datasets exhibit high correlations across variates, validating their inclusion for evaluating multivariate forecasting capabilities.
> >
> > [1] Unified Training of Universal Time Series Forecasting Transformers, https://arxiv.org/pdf/2402.02592
> >
> > [2] Self-Supervised Contrastive Pre-Training for Time Series via Time-Frequency Consistency, https://arxiv.org/pdf/2206.08496
> >
> > [3] FiLM: Frequency improved Legendre Memory Model for Long-term Time Series Forecasting, https://arxiv.org/abs/2205.08897
> >
> > [4] MULTIVARIATE PROBABILISTIC TIME SERIES FORECASTING VIA CONDITIONED NORMALIZING FLOWS, https://arxiv.org/pdf/2002.06103
> >
> > [5] CROSSFORMER: TRANSFORMER UTILIZING CROSSDIMENSION DEPENDENCY FOR MULTIVARIATE TIME
> > SERIES FORECASTING, https://openreview.net/pdf?id=vSVLM2j9eie
> >
> > [6] CATN: Cross Attentive Tree-Aware Network for Multivariate Time Series Forecasting, https://cdn.aaai.org/ojs/20320/20320-13-24333-1-2-20220628.pdf
> >
> >
> > **“The experiments mainly point out that some specific models perform best in some specific datasets. There are not many consistent conclusions from the evaluations that help us understand the characteristics of different models.”**
> >
> > We thank the reviewer for their detailed examination and appreciate the call for deeper analysis beyond pointing out which models perform best. While our primary goals were to introduce a standardized testbed and provide insights, fully explaining why specific model families perform differently across datasets is a broader open research question and beyond the scope of this paper.
> >
> > That said, we took the reviewer’s suggestion to heart and expanded our analysis. In Appendix F.1 and Table 16, we provide a detailed breakdown aggregating results based on time series features rather than just dataset characteristics. Our findings suggest that Transformer-based deep learning models like PatchTST excel in scenarios with low temporal strength and high entropy, demonstrating strong generalist performance. In contrast, foundation models such as Moirai tend to perform better in simpler, more predictable cases, aligning with our aggregated results showing Moirai-Large’s strength in less complex forecasting scenarios. This also aligns with our aggregated results where Moirai ranked highly in majority of the forecasting scenarios yet PatchTST got better aggregated metric results as it can achieve reasonably well on all scenarios including challenging ones.
> >
> > We also note that the performance differences likely reflect the supervised learning setups and hyperparameter tuning, which can give deep learning models an edge in more challenging forecasting tasks.

---

> > > ### Author Response · Authors · 2024-11-22
> > > **Rebuttal 3/n**
> > >
> > > **“Some results or claims are also confusing. In Page 7, how can we get the conclusion that ‘foundation models consistently outperform both statistical and deep learning models’ from Table 6, since PatchTST also performs very well? In Page 8, what is the meaning of ‘This trend indicates that the fine-tuning of foundation models effectively captures longer-term dependencies’? In Page 9, why does Moirai, which is a foundation model considering multivariate correlation, perform best on univariate datasets instead?”**
> > >
> > > Thank you for your question, which helped us identify typos and areas where our arguments needed clarification:
> > >
> > > Page 7 Claim:
> > > We revise the statement to: "Foundation models outperform statistical and deep learning models in 6 out of 7 domains," providing a more nuanced and accurate description of the results in Table 6 (Now Table 2).
> > >
> > > Page 8 Typo:
> > > The phrase "This trend indicates that the fine-tuning of foundation models effectively captures longer-term dependencies" was incorrect, as we do not fine-tune foundation models. The corrected phrase should read: "This trend indicates that the fine-tuning of deep learning models effectively captures longer-term dependencies." Thank you for catching this typo.
> > >
> > > Moirai’s Performance on Multivariate vs. Univariate Datasets (Page 9):
> > > Note that Moirai demonstrates strongest performance across other foundation models on multivariate data, showcasing the benefits of its multivariate support. However, deep learning models like PatchTST and iTransformer often surpass it and other foundation models in multivariate forecasting, indicating that foundation models still have room for improvement in this area. These results further underscore the importance of evaluating models on univariate vs. multivariate characteristics, as discussed in response to your first question. Without such comparisons, critical insights into model strengths and weaknesses would remain undiscovered.
> > >
> > >
> > > **“The qualitative results only show results in some special cases. It is doubtful whether these phenomena and analyses are general to all different data. It is also confusing that different data are used to evaluate foundation models and other models, which makes these models incomparable.”**
> > >
> > > Thank you for your comment. As noted in the header, Section 4.2 focuses on qualitative results and failure cases, which are intentionally designed to highlight special boundary cases. We believe these examples are informative for identifying prominent issues within specific data families and guiding the development of more robust models.
> > >
> > > We also clarify that we do compare deep learning and foundation models on the same datasets in our qualitative analyses. For instance, Figure 2-b and Figure 2-d present results for deep and foundation models, respectively, on the same instance of the Solar 10-minutely dataset. Similarly, Appendix Figure 3-b and Figure 3-c compare these model families on the same instance of the Electricity 15-minutely dataset. These examples provide a consistent basis for evaluating and contrasting the models' behaviors.

---

> > > > ### Author Response · Authors · 2024-11-25
> > > > **Discussion period ends soon.**
> > > >
> > > > Dear Reviewer SxEF,
> > > >
> > > > Thank you for taking the time and effort to review our work. As the discussion period comes to a close, we hope you’ve had an opportunity to review our rebuttal. We aimed to make it comprehensive by providing additional experiments and results to address your concerns and clarify our contributions. If our response has resolved any of your concerns, we would greatly appreciate it if you could update your review to reflect this. We are ready to engage in further discussion if you have any additional questions.
> > > >
> > > > Thank you once again for your thoughtful feedback and contributions to improving our work.

---

> ### Comment · Reviewer_SxEF · 2024-11-25
> **Response to rebuttal**
>
> Thank you for providing the rebuttal. Some of my concerns have been addressed. Here are some responses:
>
> W1: Now I understand the role of the non-leaking pretraining dataset. It seems that currently data leakage still exists in the evaluated models. How would this influence the fair comparison between foundation models with diverse data leakage?
>
> W2: Could you please point out which parts in Table 14 are the covariates and provide some examples to show what these covariates exactly are? Do all the evaluated datasets have covariates?
>
> W4: The mentioned analysis is based on the dimension of six different features, which is a little confusing considering the other dimension of four different characteristics. Why should we use these two different dimensions for analysis? Is one of them more intrinsic for time series data?  Additionally, the number of foundation models considered in this paper is limited considering the emergence of such models.
>
> W6: Why do Figures 2(a) and (c) use different data for visualization? It is unclear why we can ensure that these samples can identify prominent issues.

---

> > ### Author Response · Authors · 2024-11-27
> > **Response to Reviewer 1/2**
> >
> > Thank you for your response, we are happy to hear that we were able to clarify some of your concerns. Please find below further attempts to clarify the rest.
> >
> > **“W1: Now I understand the role of the non-leaking pretraining dataset. It seems that currently data leakage still exists in the evaluated models. How would this influence the fair comparison between foundation models with diverse data leakage?”**
> >
> > Thank you for acknowledging the role of the non-leaking pretraining dataset. We agree that data leakage, even if small, can influence the fairness of comparisons, and we have made this clear in our paper (Section 4). Avoiding leakage entirely in large benchmarks is highly challenging, and two potential approaches have significant limitations:
> >
> > Limiting the Evaluation Data:
> > Restricting the evaluation dataset to those not used in pretraining public foundation models is a shortsighted and overly limiting approach. It reduces the diversity and utility of the benchmark, which would undermine its goal of providing comprehensive evaluations.
> >
> > Pretraining All Foundation Models from Scratch:
> > Re-training all foundation models using our new pretraining data is infeasible. Many foundation models do not fully disclose their pretraining pipelines, making replication impossible. Even if pipelines were disclosed, the computational cost and effort required to pretrain all foundation models from scratch are beyond the scope of a single entity.
> >
> >
> > We also note that this challenge is not unique to time series benchmarking but is a common issue in NLP benchmarking as well [1]. Our work highlights this issue and aims to address it by providing a pretraining dataset that minimizes leakage as much as possible, while ensuring diversity and generalizability.
> >
> > [1] Beyond the Imitation Game: Quantifying and extrapolating the capabilities of language models, https://arxiv.org/pdf/2206.04615
> >
> > **“W2: Could you please point out which parts in Table 14 are the covariates and provide some examples to show what these covariates exactly are? Do all the evaluated datasets have covariates?”**
> >
> > Thank you for your question. The column "Past Dynamic" in Table 14 indicates whether a dataset includes covariates. Not all evaluated datasets have covariates, as most entries in this column are zeros. Covariates are additional time series provided alongside the main target series and currently Gift-Eval has 10 datasets that incorporate covariates 8 of the multivariate, and 2 univariate. See below explanation for one of them:
> >
> > Bizitobs Application Dataset [1]:
> > This dataset pertains to the cloud-based “Stan’s Robot Shop” application, which simulates a user’s e-commerce experience, from site access to shipping, using a load generator. It provides application-level IT metrics. In this dataset, we define the total number of calls made to the app and its latency as the two target variates. The remaining 35 IT metrics, such as the number of allocated pods, CPU allocations, and the number of OOM (Out of Memory) kills, are treated as covariates. These covariates are additional time series spanning the same historical context as the target variates.
> >
> > We hope this example clarifies how covariates are treated in our benchmark.

---

> > > ### Author Response · Authors · 2024-11-27
> > > **Response to Reviewer 2/2**
> > >
> > > **“W4: The mentioned analysis is based on the dimension of six different features, which is a little confusing considering the other dimension of four different characteristics. Why should we use these two different dimensions for analysis? Is one of them more intrinsic for time series data? Additionally, the number of foundation models considered in this paper is limited considering the emergence of such models.”**
> > >
> > > Thank you for your question. The main insight from Figure 1 is that datasets with different characteristics (e.g., number of variates, frequency, domain) can exhibit vastly different time series features such as trend, seasonality, entropy, and stability. This analysis supports our approach of diversifying these characteristics in GIFT-Eval to evaluate models across a broad spectrum of time series data.
> > >
> > > We believe that providing taxonomies from these two perspectives—dataset characteristics and time series features—and aggregating results across both is beneficial for two reasons:
> > >
> > > 1. End-Users: It helps end-users identify models suited to their specific use cases based on relevant characteristics such as domain or frequency.
> > > 2. Model Developers: It allows developers to pinpoint weaknesses in their models and refine them based on the diverse time series features that pose challenges.
> > >
> > > Regarding the number of foundation models considered, we evaluated 9 unique foundation models (13 when counting size variants). We omitted results for two models (UniTS [2] and Moment [3]) after consulting their authors, who confirmed that these models are unsuitable for zero-shot evaluation and require fine-tuning. Even after this omission, our experiments cover 7 unique foundation models.
> > >
> > > We believe we included all major foundation models available at the time of submission. However, we welcome any suggestions from the reviewer regarding additional models to incorporate.  To further expand the benchmark scale, we plan to release a public leaderboard, following standard practices in the NLP community, to enable a community-driven effort for adding more models and experiments.
> > >
> > > [1] https://github.com/BizITObs/BizITObservabilityData
> > >
> > > [2] Unified Training of Universal Time Series Forecasting Transformers, https://arxiv.org/abs/2402.02592
> > >
> > > [3] MOMENT: A Family of Open Time-series Foundation Models, https://arxiv.org/abs/2402.03885
> > >
> > > **“W6: Why do Figures 2(a) and (c) use different data for visualization? It is unclear why we can ensure that these samples can identify prominent issues.”**
> > >
> > > Thank you for your question. We would like to reiterate that this section is intended to highlight failure cases. The examples in Figure 2 were automatically sampled to include at least one model that performs poorly, allowing us to identify weaknesses in model predictions. This approach was chosen because visualizing cases where all models perform perfectly would provide little insight into their limitations.
> > >
> > > The datasets were not deliberately selected but were sampled based on the criteria mentioned above. However, we ensured that at least one dataset is common to both deep learning and foundation models (e.g., Figure 2a and 2b). The absence of the dataset from Figure 2a in the foundation model visualization (Figure 2c) indicates that it did not exhibit any anomalous results worth including in the failure cases section.
> > >
> > > We hope this clarification addresses your concern about the choice of datasets for visualization.

---

> > > > ### Author Response · Authors · 2024-11-28
> > > > **Thank you for your engagement**
> > > >
> > > > Dear Reviewer SxEF,
> > > >
> > > > Thank you for your continued engagement and valuable feedback on our paper. As we approach the final deadline for uploading the revised PDF, we wanted to check if our earlier response adequately addressed your remaining concerns. Please let us know if there are any additional points you would like us to clarify.

---

> > > > > ### Author Response · Authors · 2024-12-02
> > > > > **Thank you for your feedback**
> > > > >
> > > > > Dear Reviewer SxEF,
> > > > >
> > > > > Thank you for your active engagement and thoughtful feedback throughout the review process. As we enter the last 24 hours of the reviewer response period, we would like to briefly summarize the new clarifications provided in our responses:
> > > > >
> > > > > - (W1): We highlighted the inherent challenges of completely avoiding data leakage in large benchmarks (and its similar practice in NLP benchmarks), and emphasized our effort to minimize leakage while ensuring diversity and generalizability.
> > > > > - (W2): We explained how covariates are treated, provided an example with the Bizitobs Application Dataset, and clarified their role in our benchmark.
> > > > > - (W4): We explained the dual taxonomy approach for dataset characteristics and time series features, highlighting its benefits for both end-users and model developers. We also clarified the scope of foundation models considered and shared plans for a public leaderboard to involve the community in expanding the benchmark.
> > > > > - (W6): We clarified that the visualizations in Figure 2 were automatically sampled to highlight failure cases, focusing on datasets where models exhibited weaknesses, with intentional overlap between subsets where applicable.
> > > > >
> > > > > We hope these responses address your remaining concerns and provide the necessary clarity. As the rebuttal period concludes, we look forward to hearing from you. Thank you once again for your constructive contributions and engagement throughout this process.

---

### Official Review · Reviewer_FpFr · 2024-11-04

**Soundness:** 2
**Presentation:** 3
**Contribution:** 3
**Rating:** 5
**Confidence:** 4

**Summary:**

This paper targets at a critical and longstanding challenge in time-series forecasting research, lacking a unified, comprehensive, and diverse benchmark for evaluation. To address this challenge, GIFT-Eval is developed, encompassing 28 datasets across seven domains and sampled in ten different frequencies.

**Strengths:**

- Covering a diverse range of datasets and sampling frequencies, which can greatly facilitate our comprehensive understanding of existing time-series models
- Distinguishing pre-training and evaluation datasets, which can assist the comparison between supervised trained models (in-domain generalization) and pre-trained models (zero-shot forecasting)
- Conducting substantial experiments on these datasets (# experiments = # datasets x # sampling frequencies x  # models, such enormous computation!)

**Weaknesses:**

Overall, I appreciate and respect this paper, given its important research focus and huge amounts of experiments performed.

However, I find the current version of this paper is still on an early stage. Below I elaborate on my significant concerns, which need to be properly addressed before it reaches an acceptance bar.

### 1. The Necessity of Additional Evaluation Datasets

MOIRAI [1] has already done an excellent job in collecting pre-training datasets and conducting comprehensive evaluations on widely recognized datasets typically used to evaluate time-series forecasting models in the literature. The datasets introduced in this paper heavily overlap with them while creating a new division for pre-training and downstream evaluation. Currently, this paper argues improvements in quantity, namely more evaluation datasets and sampling frequencies, are included in the evaluation. However, the necessity of introducing more evaluation datasets and how this approach reveals unique insights not mentioned in previous research is relatively weak.

Covering an excess of comparison datasets can create a significant burden for resource-limited research groups to perform further research and for reviewers to check and compare results. This may lead to redundant model training and evaluation, which is energy-inefficient. I strongly recommend the authors analyze the uniqueness and necessity of any dataset introduced as a new testbed.

For example, when incorporating a new dataset, does it cover distinctive patterns that rarely exist in existing benchmarks, thereby delivering unique insights on different pros and cons of model designs?

The same question extends to the different sampling frequencies applied to each dataset. Is it necessary to include all sampling frequencies for every dataset? Does including the most prominent sampling frequency, which may be defined by different business scenarios, already provide sufficient and fair comparisons of different models?

According to Tables 20, 21, and 22, the total comparison covers 28 datasets, each with multiple sampling frequencies. Are these experimental comparisons overly redundant? For instance, can you identify a compact set of evaluation datasets, covering unique datasets accompanied by selected sampling frequencies, that still reveal comprehensive information while being much more energy-efficient, user-friendly, and economical? Perhaps you could calculate the real rank for the result comparison matrix presented in these tables (# datasets x # models).

Moreover, analyzing the data patterns in Figure 1 is beneficial, but how does the pattern coverage of these new benchmark datasets differ from that of existing benchmarks? Do newly introduced datasets include more pronounced trends and seasonality, or large entropy, etc.?

### 2. Lack of Some Related Work and Baselines

The comparison with previous benchmarks omits some classic and recent studies specializing in probabilistic forecasting. For instance, gluon-ts [2] is a Python package for probabilistic time-series forecasting that also provides a robust interface for accessing multiple time-series datasets. Built on gluon-ts, pytorch-ts [3] includes more advanced probabilistic forecasting models based on deep generative models. ProbTS [4] is another benchmark study offering a unique perspective by comparing capabilities in delivering point versus probabilistic forecasts, short versus long forecasts, and associated preferences in methodological designs. Specifically, ProbTS should be compared in Table 1, as it is highly relevant to your work in comparing both classical and foundation models. It unifies comparison conditions, covers diverse forecasting horizons and data patterns, calculates dominant data characteristics (such as trend, seasonality, and non-Gaussianity), and associates them with the strengths and weaknesses of different model designs.

Moreover, other time-series foundation models have been developed beyond MOIRAI, chronos, and TimesFM. Notably, Timer [5] and UniTS [6] have been accepted at conference proceedings and have publicly released their implementations. These models should at least be discussed in the related work section and, ideally, be included in your experimental comparisons.

Additionally, the paper could benefit from including more advanced probabilistic forecasting baselines, such as TimeGrad [7], CSDI [8], and their predecessor GRU NVP [9]. ProbTS has highlighted the unique advantages of these methods in delivering short-term distributional forecasting. Moreover, a simple combination of GRU NVP with RevIN [10] has demonstrated very competitive performance for both short-term and long-term forecasting. Including these more powerful probabilistic models is crucial, as merely adding probabilistic heads over forecasting models like MOIRAI and DeepAR does not sufficiently capture complex data distributions that extend beyond closed-form probabilistic distribution functions.


### 3 Some Missing Details and Analyses in Experiments

The use of MAPE as an only metric for evaluating point forecasts is somewhat "biased." I recommend referring to N-BEATS [11] and including metrics such as sMAPE and ND (normalized deviation, equivalent to normalized MAE) for a more comprehensive evaluation.

Regarding hyperparameter search for deep learning baselines, there is a notable omission in tuning their lookback lengths. This can be an extremely critical factor to adjust, given the diversity of datasets and sampling frequencies. Appendix A indicates that this tuning was performed only for MOIRAI. The same process should be applied to other baselines, including supervised learning models and pre-trained foundation models.

Moreover, regarding three foundation models used in the experiments, only MOIRAI is re-trained on this new data split while TimesFM and Chronos are compared with their released model checkpoints. There is a risk of data leakage in these experiment comparisons. For example, TimesFM has leverage the Electricity dataset for pre-training while this dataset is also used for evaluation in this paper. This may explain why TimesFM performs extremely on "Electricity, short, H" (line 1285, Table 20), but this good performance may come from test data leakage. Therefore, to provide a fair and comprehensive comparison across TimeFM, Chronos, MOIRAI, and other representative time-series foundation models, re-training them following the new data split could be a necessary step.

When comparing supervised time-series models with zero-shot foundation models, it is crucial to investigate the effect of allowed lookback length on forecasting performance. As revealed in MOIRAI, the lookback length significantly influences model performance, as MOIRAI employs an additional hyper-parameter adaptation process on the lookback data, unlike others.

The current analysis lacks depth across multiple experimental configurations. Presently, only aggregated comparisons are provided, without detailed analyses to derive new insights. For instance, why do some foundation models yield better zero-shot forecasting results on certain datasets than models specifically tuned for those datasets? Is it because pre-training datasets encompass more related patterns, pre-trained models have a larger model capacity, or classical models are not well-tuned, particularly regarding lookback length, modeling layers, or model designs? Merely presenting results without in-depth analysis can be detrimental to the research community, as others may expend significant efforts to re-analyze numerous results, which should be addressed within this benchmark.

Additionally, there is a lack of explicit connection with existing benchmarks to validate the reliability of the experiments conducted. For example, evaluation datasets in Tables 20, 21, and 22 cover some classical datasets like ETTh, Electricity, Solar, and M4. Comparing your results on shared datasets with existing studies could demonstrate the reliability of your experimental protocols. Moreover, some widely used datasets in the literature, such as Traffic, Wikipedia, and Exchange, appear to be excluded. Please clarify the reasons for their exclusion.

I suggest maintaining existing classical benchmarks as they are while introducing new datasets and sampling frequencies that cover unique patterns. This approach allows for consistency with existing benchmarks, showcases the correct reproduction of existing models, demonstrates the necessity of new datasets and sampling frequencies, and reveals what can be discovered given this new benchmark.

[1] Unified Training of Universal Time Series Forecasting Transformers, https://arxiv.org/abs/2402.02592

[2] gluon-ts, https://www.jmlr.org/papers/v21/19-820.html

[3] pytorch-ts, https://github.com/zalandoresearch/pytorch-ts

[4] ProbTS, https://arxiv.org/abs/2310.07446

[5] Timer, https://arxiv.org/abs/2402.02368

[6] UniTS, https://arxiv.org/pdf/2403.00131

[7] Autoregressive Denoising Diffusion Models for Multivariate Probabilistic Time Series Forecasting, https://arxiv.org/abs/2101.12072

[8] CSDI: Conditional Score-based Diffusion Models for Probabilistic Time Series Imputation, https://arxiv.org/abs/2107.03502

[9] Multivariate Probabilistic Time Series Forecasting via Conditioned Normalizing Flows, https://arxiv.org/abs/2002.06103

[10] RevIN, https://openreview.net/forum?id=cGDAkQo1C0p

[11] N-BEATS, https://arxiv.org/abs/1905.10437

**Questions:**

See weaknesses.

My critical concerns centered around the rationale of selecting/adding/filtering evaluation datasets and sampling frequencies, the missing discussion of some highly related studies, and incomplete experimental results and analyses.

---

> ### Author Response · Authors · 2024-11-22
> **Rebuttal 1/n**
>
> Dear reviewer, thank you for the time you have taken to review our submission, and thanks for your kind words appreciating our research focus and the amount of experiments we conduct. We have carefully addressed each point below and strengthened our presentation accordingly. Please find all responses below:
>
> ## The necessity of additional evaluation datasets
>
> **“MOIRAI [1] has already done an excellent job in collecting pre-training datasets and conducting comprehensive evaluations on widely recognized datasets typically used to evaluate time-series forecasting models in the literature. The datasets introduced in this paper heavily overlap with them while creating a new division for pre-training and downstream evaluation … For example, when incorporating a new dataset, does it cover distinctive patterns that rarely exist in existing benchmarks, thereby delivering unique insights on different pros and cons of model designs? The same question extends to the different sampling frequencies applied to each dataset. Is it necessary to include all sampling frequencies for every dataset? Does including the most prominent sampling frequency, which may be defined by different business scenarios, already provide sufficient and fair comparisons of different models?”**
>
> Thanks for your question. Constructing a benchmark requires the inclusion of datasets that collectively represent the complexity and diversity of real-world forecasting scenarios. While explaining the specific contribution of each dataset in isolation might not be feasible, GIFT-Eval was curated with the principle of achieving diversity and ensuring adequate coverage of key characteristics such as domain, frequency, and prediction length, including multivariate scenarios. This selection was based on publicly available datasets with careful balancing to maintain diversity in the test set while also reserving key datasets for effective pretraining.
>
> The reviewer rightfully asked whether sampling various frequencies of some datasets is a necessary step. Incorporating multiple sampling frequencies tests a model’s adaptability and robustness across different business scenarios, which would be missed if only the most common frequency were used. Even through our results one can observe that a model showing promising results for a certain frequency may give relatively poor results on the same frequency for some datasets. We agree with the reviewer that Moirai’s pretraining dataset is extensive, yet its evaluation data lacks the diverse scope needed for fully assessing model versatility. We show why we think so in the next section.
>
> Comparison to Moirai
>
> Moirai Zero-shot datasets:
> Prob. Forecasting
> | Dataset        | Frequency        |
> |----------------|------------------|
> | electricity    | H                |
> | solar          | H                |
> | walmart        | W                |
> | weather        | 10T              |
> | Ist. Traf.     | H                |
> | Turk. Pow.     | 15T              |
>
> Long Term Forecasting
> | Dataset        | Frequency        |
> |----------------|------------------|
> | Ett h1+ Etth2  | H                |
> | Ett m1+ Ettm2  | 15T              |
> | electricity    | H                |
> | weather        | 10T              |
>
>
> Moirai's original zero-shot evaluation employed a limited set of datasets, primarily focusing on specific domains and frequencies. Below we provide two example comparison with Moirai’s evaluation to argue why Gift-Eval is more comprehensive:
>
>
> a. Long term Forecasting: Original Moirai results demonstrated its strength in long-term forecasting (outperforming baselines in 5 out of 6 datasets), these findings align with our observations for those specific datasets. However, GIFT-Eval introduces 21 dataset configurations with long term forecasting across diverse domains and frequencies. Deep learning models like PatchTST and iTransformer, which were also included in Moirai's comparisons, outperform Moirai in long-term forecasts within our benchmark.
>
> b. Frequency We can look at another example in favor of Moirai from the frequency perspective. Moirai's original evaluation included only the Walmart dataset at a weekly frequency, where it underperformed compared to deep learning models. In contrast, GIFT-Eval includes 8 weekly datasets from 4 different domains with both univariate and multivariate representations. In this broader context, Moirai performs significantly better, surpassing all deep learning baselines and securing second place among foundation models.
>
> These examples underscore why diverse benchmarks like GIFT-Eval are essential for accurately evaluating the general capabilities of universal models. They uncover performance dynamics that narrower evaluations might miss, providing a broader picture of model strengths and weaknesses.

---

> > ### Author Response · Authors · 2024-11-22
> > **Rebuttal 2/n**
> >
> > **“Moreover, analyzing the data patterns in Figure 1 is beneficial, but how does the pattern coverage of these new benchmark datasets differ from that of existing benchmarks? Do newly introduced datasets include more pronounced trends and seasonality, or large entropy, etc.?”**
> >
> > We appreciate the reviewer’s interest in understanding how the new benchmark datasets differ in pattern coverage compared to existing benchmarks. The main insight from Figure 1 is that datasets with different characteristics (e.g., number of target variates, frequency, domain), can exhibit vastly different time series features such as trend, seasonality, entropy, and stability. This supports our approach of diversifying these characteristics in GIFT-Eval to evaluate models on a wide range of time series data.
> >
> > One might question why we do not limit the benchmark to a single dataset for each characteristic combination. The reason is that even datasets sharing the same characteristics can have very different underlying time series features. To illustrate this, we included a new table, Table 9, in the Appendix, which lists these features for each dataset. For instance, while both the sz_taxi and m_dense datasets share the same hourly frequency and domain, they exhibit distinct distributions of time series features. Such differences validate the importance of GIFT-Eval’s comprehensive dataset composition, which ensures models are tested on varied real-world data scenarios, offering deeper insights than existing benchmarks.
> >
> >
> > | dataset                 | frequency | trend | seasonal_strength | entropy | hurst | lumpiness | stability |
> > |--------------------------|-----------|-------|-------------------|---------|-------|-----------|-----------|
> > | m_dense                  | H         | low   | high              | high    | low   | low       | low       |
> > | sz_taxi                  | H         | low   | low               | high    | high  | high      | low       |
> >
> >
> > **“Covering an excess of comparison datasets can create a significant burden for resource-limited research groups to perform further research and for reviewers to check and compare results. This may lead to redundant model training and evaluation, which is energy-inefficient. I strongly recommend the authors analyze the uniqueness and necessity of any dataset introduced as a new testbed.”**
> >
> > We acknowledge the concern about the potential resource burden posed by extensive evaluation datasets, particularly for resource-limited research groups. To address this concern, we conducted a breakdown of the evaluation time for the largest variants of each foundation model using a single A100 GPU (40GB).
> >
> > | Model          | Run Time (H)     |
> > |----------------|------------------|
> > | Moirai Large   | 8.36             |
> > | Chronos Large  | 24.12            |
> > | VisionTS       | 3                |
> > | TimesFM        | 1.85             |
> > | Timer          | 0.62             |
> > | TTM            | 0.62             |
> > | UniTS          | 0.56             |
> >
> > Our findings show that most models require only a few hours for a complete evaluation on GIFT-Eval. Even the most computationally intensive models, such as the MLM-based Moirai and the autoregressive Chronos, take at most one GPU day. This efficiency is achieved by avoiding the rolling window with stride=1 approach used in other benchmarks. Instead, we sample non-overlapping windows while ensuring at least 10% of each dataset is used in the test splits. This approach maintains GIFT-Eval’s diversity across key characteristics while keeping evaluation costs manageable.
> >
> >
> > **“According to Tables 20, 21, and 22, the total comparison covers 28 datasets, each with multiple sampling frequencies. Are these experimental comparisons overly redundant? For instance, can you identify a compact set of evaluation datasets, covering unique datasets accompanied by selected sampling frequencies, that still reveal comprehensive information while being much more energy-efficient, user-friendly, and economical? Perhaps you could calculate the real rank for the result comparison matrix presented in these tables (# datasets x # models).”**
> >
> > We hope our responses above clarify why we opted for a large collection of datasets and why we believe it is worth doing so. First two responses explain how datasets with superficially similar attributes (e.g., domain) can yield different results, while the last response highlights that evaluating GIFT-Eval is not a significant computational cost.
> >
> > We agree that identifying a minimal set of datasets to comprehensively evaluate time series models is an interesting and valid suggestion. However, this remains a challenging and open problem. Importantly, this suggestion does not diminish GIFT-Eval’s contributions toward more robust and diverse evaluation practices. While this concern might become more pressing with larger benchmarks incurring high GPU costs in the future, it is not a significant issue at this time.

---

> > > ### Author Response · Authors · 2024-11-22
> > > **Rebuttal 3/n**
> > >
> > > ## Lack of Some Related Work and Baselines
> > >
> > > **“The comparison with previous benchmarks omits some classic and recent studies specializing in probabilistic forecasting. For instance, gluon-ts [2] is a Python package for probabilistic time-series forecasting that also provides a robust interface for accessing multiple time-series datasets. Built on gluon-ts, pytorch-ts [3] includes more advanced probabilistic forecasting models based on deep generative models. ProbTS [4] is another benchmark study offering a unique perspective by comparing capabilities in delivering point versus probabilistic forecasts, short versus long forecasts, and associated preferences in methodological designs. Specifically, ProbTS should be compared in Table 1, as it is highly relevant to your work in comparing both classical and foundation models. It unifies comparison conditions, covers diverse forecasting horizons and data patterns, calculates dominant data characteristics (such as trend, seasonality, and non-Gaussianity), and associates them with the strengths and weaknesses of different model designs.”**
> > >
> > > We would like to thank the reviewer for bringing additional benchmark resources to our attention. Following the reviewer's suggestion, we have expanded the related work section and updated Table 1 with ProbTS.
> > >
> > > **“Moreover, other time-series foundation models have been developed beyond MOIRAI, chronos, and TimesFM. Notably, Timer [5] and UniTS [6] have been accepted at conference proceedings and have publicly released their implementations. These models should at least be discussed in the related work section and, ideally, be included in your experimental comparisons.”**
> > >
> > > Thanks for your suggestion. We have expanded the related work section to include all the baseline models mentioned by the reviewer. We have also extended the baseline comparisons to include both Timer and UniTS, as suggested, and further added 3 more foundation models: TTM, Lag-Llama and Moment for comprehensive evaluation.
> > >
> > > Upon observing that for UniTS and Moment results were below expected we reached out to authors for assistance and they clarified that their models are not suitable for zero-shot and needs go under finetuning. Thus we don’t add these two to the results in our paper yet but share their overall results here for reference. The new results with additional foundation models are updated in Tables 16 through 23 in the Appendix. We share the general aggregated results of foundation models along with newly added ones here for convenience:
> > >
> > > | Metric | Timer     | Units     | TTM     | Moment     | LagLLama     | ChronosLarge     | MoiraLarge      | Best             |
> > > |--------|-----------|-----------|---------|------------|--------------|------------------|-----------------|------------------|
> > > | MASE   | 1.02      | 1.67      | 9.69e-1 | 1.38       | 1.10         | **7.81e-1**      | _7.97e-1_       | ChronosLarge     |
> > > | CRPS   | 8.20e-1   | 1.34      | 7.53e-1 | 1.13       | 7.44e-1      | _5.47e-1_        | **5.15e-1**     | MoiraLarge       |
> > > | Rank   | 2.13e1    | 2.55e1    | 1.98e1  | 2.48e1     | 1.80e1       | _1.09e1_         | **7.57**        | MoiraLarge       |
> > >
> > >
> > > **“Additionally, the paper could benefit from including more advanced probabilistic forecasting baselines, such as TimeGrad [7], CSDI [8], and their predecessor GRU NVP [9]. ProbTS has highlighted the unique advantages of these methods in delivering short-term distributional forecasting. Moreover, a simple combination of GRU NVP with RevIN [10] has demonstrated very competitive performance for both short-term and long-term forecasting. Including these more powerful probabilistic models is crucial, as merely adding probabilistic heads over forecasting models like MOIRAI and DeepAR does not sufficiently capture complex data distributions that extend beyond closed-form probabilistic distribution functions.”**
> > >
> > > We agree that adding probabilistic baseline models would enhance comparisons with foundation models supporting probabilistic outputs. Due to our GluonTS-based framework and limited time, we prioritized models already compatible with it. We were only able to find a gluonts implementation for TimeGrad at the time. While we attempted to add TimeGrad, we encountered several issues, primarily due to conflicts with the GluonTS version on which our framework is built and its dependencies. While this prevented us from adding TimeGrad during this phase, we are committed to addressing them in future updates. Additionally, we plan to include models like CSDI and GRU-NVP in subsequent iterations of the benchmark. Thank you for your valuable suggestions, which will enhance our benchmark's comprehensiveness.

---

> > > > ### Author Response · Authors · 2024-11-22
> > > > **Rebuttal 4/n**
> > > >
> > > > ## Some missing details and analyses in experiments
> > > >
> > > > **“The use of MAPE as an only metric for evaluating point forecasts is somewhat "biased." I recommend referring to N-BEATS [11] and including metrics such as sMAPE and ND (normalized deviation, equivalent to normalized MAE) for a more comprehensive evaluation.”**
> > > >
> > > > Thank you for bringing this point to our attention. Other reviewers have also brought up similar concerns. Following all reviewer suggestions we omit MAPE as an evaluation metric from our paper. Each reviewer had a different suggestion for metrics to add. Here is how we addressed all of their concerns:
> > > >
> > > > The main paper tables (Table 2 through 7) in the paper are updated to show MASE rather than MAPE results and we use geometric mean to aggregate results instead of arithmetic mean.
> > > > We update the appendix tables (Tables 17 through 21) which report results for all models with all metrics suggested by reviewers for the sake of completeness. Specifically we report results with sMAPE, MASE, ND, MSE, MAE, CRPS and finally Rank metrics.
> > > >
> > > > We hope this addresses the concerns around the MAPE metric.
> > > >
> > > > **“Regarding hyperparameter search for deep learning baselines, there is a notable omission in tuning their lookback lengths. This can be an extremely critical factor to adjust, given the diversity of datasets and sampling frequencies. Appendix A indicates that this tuning was performed only for MOIRAI. The same process should be applied to other baselines, including supervised learning models and pre-trained foundation models. When comparing supervised time-series models with zero-shot foundation models, it is crucial to investigate the effect of allowed lookback length on forecasting performance. As revealed in MOIRAI, the lookback length significantly influences model performance, as MOIRAI employs an additional hyper-parameter adaptation process on the lookback data, unlike others.”**
> > > >
> > > > We agree with the reviewer that tuning context length is essential for all deep learning models. In fact, we conducted this search for all supervised models except DeepAR and TFT at the time of submission, since these two models were missing this detail was omitted from Appendix A. We are now tuning the context length for these two models and will update the results and Appendix A accordingly.
> > > >
> > > > For foundation models, we initially focused on tuning context length for only Moirai because it uniquely uses a dynamic context length during training, unlike other models such as Chronos, which use a fixed length. However, we agree that this could create an imbalance. To address this, we now include results for Moirai without context length tuning for a fairer comparison. Thus all results in the following Tables are updated with the public Moirai model as opposed to the retrained version of Moiria with optimized context length: Tables [2,3,4,5,6,7,16,20,21,22,23,24].

---

> ### Author Response · Authors · 2024-11-22
> **Rebuttal 5/n**
>
> **“Moreover, regarding three foundation models used in the experiments, only MOIRAI is re-trained on this new data split while TimesFM and Chronos are compared with their released model checkpoints. There is a risk of data leakage in these experiment comparisons. For example, TimesFM has leverage the Electricity dataset for pre-training while this dataset is also used for evaluation in this paper. This may explain why TimesFM performs extremely on "Electricity, short, H" (line 1285, Table 20), but this good performance may come from test data leakage. Therefore, to provide a fair and comprehensive comparison across TimeFM, Chronos, MOIRAI, and other representative time-series foundation models, re-training them following the new data split could be a necessary step.”**
>
> We acknowledge that re-training all foundation models on the new data split would provide a more comprehensive comparison. However, due to limited resources, we could only afford to re-train one model, Moirai, from scratch. It would be very limiting and shortsighted if we were to limit ourselves to only the datasets that current foundation models do not pretrain on. Instead we wanted our benchmark to be diverse across characteristics so that a broad evaluation can be achieved.
>
> As noted in Section 4, other foundation models, such as TimesFM and Chronos, were compared using their released checkpoints, which may include data leakage into our test set (This is a standard issue that NLP benchmark papers acknowledge too [1]). However, we believe it is unrealistic to expect a single entity to re-train all foundation models, but by publicizing our pretraining datasets, we aim to facilitate collaborative scaling of this effort. The main reason for reporting retrained Moirai model results was to illustrate the impact of data leakage. However we understand that this may create an unfair standing compared to other models we incorporate in the experiments. Thus we update all tables to report results from the public Moirai model instead. We still report results of our findings over the retrained model in Appendix section F.3 to argue how leakage affects results.
>
> [1] Beyond the Imitation Game: Quantifying and extrapolating the capabilities of language models https://arxiv.org/pdf/2206.04615
>
> **“The current analysis lacks depth across multiple experimental configurations. Presently, only aggregated comparisons are provided, without detailed analyses to derive new insights. For instance, why do some foundation models yield better zero-shot forecasting results on certain datasets than models specifically tuned for those datasets? Is it because pre-training datasets encompass more related patterns, pre-trained models have a larger model capacity, or classical models are not well-tuned, particularly regarding lookback length, modeling layers, or model designs? Merely presenting results without in-depth analysis can be detrimental to the research community, as others may expend significant efforts to re-analyze numerous results, which should be addressed within this benchmark.”**
>
> We appreciate the reviewer’s emphasis on deeper analysis and agree that providing more insights is valuable for the research community. Our primary motivation for this work is indeed to highlight the strengths and weaknesses of various model families to advance research in the time series domain. However, with 20 models evaluated across more than 20 datasets, explaining why specific model families perform differently on certain datasets is beyond the scope of this paper, as these are longstanding open research questions.
>
> That said, we took the reviewer's suggestion to heart and expanded our analysis. In Appendix F.1 and Table 16, we provide a detailed breakdown that aggregates results based on time series features rather than just dataset characteristics, discussing the strengths and weaknesses of different model families. Briefly, our new findings indicate that deep learning models, particularly Transformer-based ones like PatchTST, tend to excel in challenging scenarios with low temporal strength and high entropy, showing strong generalist performance. Conversely, foundation models such as Moirai perform better in simpler, more predictable cases, aligning with our aggregated results where Moirai ranked highly in majority of the forecasting scenarios yet PatchTST got better aggregated metric results as it can achieve reasonably well on all scenarios including challenging ones. We also note that the performance differences likely reflect the supervised learning setups and hyperparameter tuning, which can give deep learning models an edge in more challenging forecasting tasks.
>
> We hope this extended analysis will help the community better understand these nuances without having to re-analyze the data from scratch, especially with the growing number of models added into our benchmark.

---

> ### Author Response · Authors · 2024-11-22
> **Rebuttal 6/n**
>
> **“Additionally, there is a lack of explicit connection with existing benchmarks to validate the reliability of the experiments conducted. For example, evaluation datasets in Tables 20, 21, and 22 cover some classical datasets like ETTh, Electricity, Solar, and M4. Comparing your results on shared datasets with existing studies could demonstrate the reliability of your experimental protocols. Moreover, some widely used datasets in the literature, such as Traffic, Wikipedia, and Exchange, appear to be excluded. Please clarify the reasons for their exclusion. I suggest maintaining existing classical benchmarks as they are while introducing new datasets and sampling frequencies that cover unique patterns. This approach allows for consistency with existing benchmarks, showcases the correct reproduction of existing models, demonstrates the necessity of new datasets and sampling frequencies, and reveals what can be discovered given this new benchmark.”**
>
> Excluding Traffic, Wikipedia, and Exchange
>
> We appreciate the reviewer’s observation regarding the exclusion of Traffic, Wikipedia, and Exchange datasets. Traffic and Wikipedia are indeed included in the pre-training set to enrich the diversity and volume of training data. The Exchange dataset, representing financial time series, was excluded due to its unique characteristics. As noted in prior research [3], univariate financial time series often resemble a random walk, where the naive forecast is theoretically optimal. This behavior makes them unsuitable for general forecasting benchmarks, as it does not effectively challenge more complex forecasting models.
> While maintaining classical benchmarks could aid consistency, our focus is to demonstrate the added value of new datasets and diverse prediction lengths, revealing insights that existing benchmarks may overlook. We hope that by sharing our protocols and datasets, future studies can extend these findings and validate them further.
> We hope this clarifies our reasoning for dataset selection and the focus on benchmarks that can yield more meaningful model comparisons.
>
>
> Connection with existing benchmarks
>
> We appreciate the reviewer’s suggestion to draw explicit connections with existing benchmarks to validate our experiments. While we agree that comparing results on classical datasets e.g., ETTh, Electricity, Solar and M4 with prior studies would be valuable for demonstrating consistency and reproducibility, there are inherent challenges in doing so for the first three datasets as we use slightly different settings for these within our benchmarks:
>
> 1. In GIFT-Eval, we intentionally varied prediction lengths to enhance the diversity of forecasting scenarios. This decision was motivated by the limited diversity of prediction lengths in previous benchmarks, which often constrained models to specific forecasting horizons not representative of real-world applications. Our approach aims to create a more comprehensive evaluation by testing models across a wider range of forecasting conditions. However, this makes direct comparisons with existing studies, which use fixed or different prediction lengths, challenging.
> 2. Additionally, in line with similar motivations as the reviewer (incurring less costs to  resource-limited research groups), we sample non-overlapping windows while ensuring at least 10% of each dataset is used in the test splits. This choice helps maintain an efficient evaluation process, as existing benchmarks often use a rolling window with a stride of 1, which is computationally expensive. These combined factors—varied prediction lengths and different test window setups—make raw dataset results harder to compare directly with existing studies.
> For M4 we kept the same settings as the original dataset, the only difference is that we filtered some very short datasets from our benchmark dataset thus yearly frequency is not identical to the original set. Below we share comparable results for all other frequencies from their original sources [1], [2]. (Table split into two responses due to space limitation)
>
>
> | Model     | F | sMAPE | MASE |
> |-----------|---|-------|------|
> | naive     | D | 0.030 | 3.280|
> |           | H | 0.430 | 11.60|
> |           | M | 0.153 | 1.210|
> |           | Q | 0.116 | 1.480|
> |           | W | 0.091 | 2.780|
> |-----------|---|-------|------|
> |naive      | D | 0.030 | 3.278|
> |(original) | H | 0.430 | 11.60|
> |           | M | 0.152 | 1.205|
> |           | Q | 0.116 | 1.477|
> |           | W | 0.091 | 2.777|
>
> **Table Continued in the next repsonse..**
>
> [1] The M4 Competition: 100,000 time series and 61 forecasting methods, https://www.sciencedirect.com/science/article/pii/S0169207019301128
>
> [2] Chronos: Learning the Language of Time Series, https://arxiv.org/abs/2403.07815
>
> [3] Common Pitfalls and Better Practices in Forecast Evaluation for Data Scientists, Christoph Bergmeir. https://cbergmeir.com/papers/Bergmeir2023pitfalls.pdf

---

> > ### Author Response · Authors · 2024-11-22
> > **Rebuttal 7/7**
> >
> > | Model     | F | sMAPE | MASE |
> > |-----------|---|-------|------|
> > | s_naive   | D | 0.030 | 3.280|
> > |           | H | 0.139 | 1.190|
> > |           | M | 0.160 | 1.260|
> > |           | Q | 0.125 | 1.600|
> > |           | W | 0.091 | 2.780|
> > |-----------|---|-------|------|
> > | s_naive   | D | 0.030 | 3.278|
> > | (original)| H | 0.139 | 1.193|
> > |           | M | 0.159 | 1.259|
> > |           | Q | 0.116 | 1.477|
> > |           | W | 0.091 | 2.777|
> > |-----------|---|-------|------|
> > | auto_arima| D | 0.031 | 3.260|
> > |           | H | 0.137 | 1.030|
> > |           | M | 0.137 | 0.976|
> > |           | Q | 0.109 | 1.280|
> > |           | W | 0.089 | 2.360|
> > |-----------|---|-------|------|
> > | arima     | D | 0.031 | 3.398|
> > | (original)| H | 0.140 | 0.950|
> > |           | M | 0.134 | 0.930|
> > |           | Q | 0.104 | 1.165|
> > |           | W | 0.085 | 2.541|
> > |-----------|---|-------|------|
> > | auto_ets  | D | 0.030 | 3.240|
> > |           | H | 0.172 | 1.610|
> > |           | M | 0.136 | 0.964|
> > |           | Q | 0.102 | 1.160|
> > |           | W | 0.087 | 2.550|
> > |-----------|---|-------|------|
> > | ets       | D | 0.030 | 3.252|
> > | (original)| H | 0.173 | 1.823|
> > |           | M | 0.135 | 0.947|
> > |           | Q | 0.102 | 1.160|
> > |           | W | 0.087 | 2.527|
> > |-----------|---|-------|------|
> > | auto_theta| D | 0.031 | 3.340|
> > |           | H | 0.203 | 2.460|
> > |           | M | 0.134 | 0.966|
> > |           | Q | 0.105 | 1.190|
> > |           | W | 0.096 | 2.660|
> > |-----------|---|-------|------|
> > | theta     | D | 0.030 | 3.262|
> > | (original)| H | 0.181 | 2.454|
> > |           | M | 0.130 | 0.970|
> > |           | Q | 0.103 | 1.231|
> > |           | W | 0.090 | 2.638|
> > |-----------|---|-------|------|
> > | chronos-L | D | 0.029 | 3.180|
> > |           | H | 0.076 | 0.694|
> > |           | M | 0.140 | 0.971|
> > |           | Q | 0.107 | 1.230|
> > |           | W | 0.060 | 2.080|
> > |-----------|---|-------|------|
> > | chronos-L | D | NA    | 3.144|
> > | (original)| H | NA    | 0.682|
> > |           | M | NA    | 0.960|
> > |           | Q | NA    | 0.082|
> > |           | W | NA    | 1.998|
> > |-----------|---|-------|------|
> > | chronos-B | D | 0.029 | 3.180|
> > |           | H | 0.076 | 0.693|
> > |           | M | 0.140 | 0.973|
> > |           | Q | 0.107 | 1.230|
> > |           | W | 0.061 | 2.080|
> > |-----------|---|-------|------|
> > | chronos-B | D | NA    | 3.160|
> > | (original)| H | NA    | 0.694|
> > |           | M | NA    | 0.970|
> > |           | Q | NA    | 0.083|
> > |           | W | NA    | 2.021|
> > |-----------|---|-------|------|
> > | chronos-S | D | 0.029 | 3.160|
> > |           | H | 0.078 | 0.739|
> > |           | M | 0.139 | 0.982|
> > |           | Q | 0.108 | 1.240|
> > |           | W | 0.062 | 2.090|
> > |-----------|---|-------|------|
> > | chronos-S | D | NA    | 3.148|
> > | (original)| H | NA    | 0.721|
> > |           | M | NA    | 0.982|
> > |           | Q | NA    | 0.084|
> > |           | W | NA    | 2.113|
> > |-----------|---|-------|------|

---

> > > ### Author Response · Authors · 2024-11-25
> > > **Discussion period ends soon.**
> > >
> > > Dear Reviewer FpFr,
> > >
> > > Thank you for taking the time and effort to review our work. As the discussion period comes to a close, we hope you’ve had an opportunity to review our rebuttal. We aimed to make it comprehensive by providing additional experiments and results to address your concerns and clarify our contributions. If our response has resolved any of your concerns, we would greatly appreciate it if you could update your review to reflect this. We are ready to engage in further discussion if you have any additional questions.
> > >
> > > Thank you once again for your thoughtful feedback and contributions to improving our work.

---

> > > > ### Comment · Reviewer_FpFr · 2024-11-25
> > > > **Thank you for your response**
> > > >
> > > > I appreciate the authors' thoughtful and thorough response to my reviews and their incorporation of some of my suggestions. The focus on providing a comprehensive benchmark is indeed a highly valuable contribution to the research community.
> > > >
> > > > However, I still have some persistent concerns that have not been fully addressed. Below, I outline each concern and provide justifications for why addressing them is both critical and feasible.
> > > >
> > > > **The Necessity of Introducing Additional Datasets/Forecasting Horizons/Sampling Frequencies**
> > > >
> > > > My concern stems from the significant computational cost associated with supervised training on each dataset. The trade-off between building time-series foundation models capable of zero-shot forecasting across domains and supervisedly training a time-series model on domain-specific data remains an open question requiring further investigation. Simply including numerous evaluation cases—many of which may be redundant—is not only resource-intensive for research groups with limited computational resources but also economically inefficient.
> > > >
> > > > I respectfully disagree with the authors' comment that "explaining the specific contribution of each dataset in isolation might not be feasible." A more systematic approach, such as analyzing the performance matrix (evaluation cases × models) using principal component analysis, could help identify redundant evaluation cases. This would allow the authors to exclude unnecessary ones while retaining a concise yet informative benchmark suite.
> > > >
> > > > To enhance the benchmark's utility, I suggest maintaining consistency with existing benchmarks while introducing unique evaluation cases. These could include datasets with distinctive data patterns from specialized domains, novel sampling frequencies, and varying forecasting horizons. The authors’ new analysis showing which types of models perform best in specific scenarios is a step in the right direction. Expanding this analysis with unique evaluation cases could yield even more valuable insights, such as identifying previously unknown limitations of existing models or revealing scenarios where time-series foundation models outperform supervised models.
> > > >
> > > >
> > > > **The Consistency with Existing Time-Series Benchmarks**
> > > >
> > > > I appreciate the authors’ explanation of why the Exchange dataset is not suitable for evaluation. A better way to convey this to a broader audience is to ensure consistency with existing benchmarks in your paper while justifying any exclusions based on specific issues, such as trivial patterns or unpredictable noise.
> > > >
> > > > However, I still find the rationale for including the Traffic and Wikipedia datasets in the pre-training set unclear. Why does MOIRAI also adopt this setup? Do Traffic and Wikipedia datasets significantly impact the performance of time-series foundation models when excluded from pre-training?
> > > >
> > > > The ongoing debate between time-series foundation models and supervisedly trained models for different scenarios adds further importance to this issue. Even the results of this paper suggest that time-series foundation models do not always guarantee robust zero-shot transfer, which means practitioners may still need supervised models for their specific use cases. Providing a consistent comparison with existing benchmarks is therefore crucial to highlight the unique advantages and limitations of time-series foundation models versus supervised models.
> > > >
> > > > I recommend that the authors maintain a consistent evaluation on widely used benchmarks to establish a clear connection between existing time-series studies and this new benchmark. At the same time, the introduction of previously overlooked data patterns for evaluation could further enrich the analysis and reveal new insights.
> > > >
> > > > **The Reproduction of Crucial Baselines**
> > > >
> > > > I am glad to hear that the authors have committed to include some advanced probabilistic models, such as TimeGrad and CSDI, and understand the difficulties in correctly reproducing them. While if GIFT-Eval claims to incorporate distributional evaluation as one of its scope, incorporating advanced probabilistic time-series models is a must-to-do step.
> > > >
> > > > Moreover, if this work wants to propose a new division of pre-training and held-out evaluation datasets. Re-training typical time-series foundation models and providing corresponding analyses are also multi-to-do steps. Because current evaluation results of different pre-trained foundation models, with different pre-training data leakages with evaluation data, could lead to severe misleading intrepretations. Additionally, this makes the comparison between time-series foundation models and convential supervised models unfair and not reliable, as the effect of potential leakage is unclear.

---

> > > > > ### Comment · Reviewer_FpFr · 2024-11-25
> > > > > **More Feedbacks**
> > > > >
> > > > > **Computational Overheads of Training Models**
> > > > >
> > > > > The authors have not disclosed the computational costs associated with tuning hyperparameters for conventional supervised time-series models. I suspect these costs could be extremely high, as each model must be trained across a grid of hyperparameter combinations and tuned until successful convergence. Without reducing redundancy in the evaluation benchmarks, this could place a significant burden on the research community, potentially wasting substantial computational resources on unnecessary re-training.
> > > > >
> > > > > While the paper appears to lean toward advocating for time-series foundation models with zero-shot forecasting capabilities, I want to emphasize that the current zero-shot forecasting results remain highly unstable. It is critical to also develop and evaluate state-of-the-art supervised models to provide clear guidance on which paradigm—foundation models or supervised models—is preferable for different application scenarios.
> > > > >
> > > > > Moreover, although the authors note, "due to limited resources, we could only afford to re-train one model, Moirai, from scratch," I still believe that reproducing key foundation models is a fundamental responsibility of a benchmark study proposing a new division of pre-training and evaluation datasets. Without this information, it becomes challenging to draw meaningful comparisons between time-series foundation models (with varying levels of data leakage and pre-training resources) or between foundation models and supervised models tailored for specific scenarios.
> > > > >
> > > > > **Recommendations**
> > > > >
> > > > > Given the significant computational costs associated with training various models, I recommend the following steps:
> > > > >
> > > > > - Ensure Consistency with Existing Benchmarks
> > > > >   - Maintain alignment with widely used benchmarks to facilitate verification of reproduced results and ensure continuity with ongoing research developments.
> > > > > - Streamline Evaluation Scenarios
> > > > >   - Introduce a compact yet informative set of evaluation cases by focusing on unique and impactful scenarios. This would not only yield concise and meaningful results but also help manage computational costs for supervised models.
> > > > > - Reproduce Crucial Baselines
> > > > >   - Proposing a new division of pre-training and evaluation datasets is straightforward, but providing thorough and accurate experimental results requires significant effort. It is essential for benchmark studies to fill these gaps rather than leaving this burden to the community.

---

> > > > > > ### Author Response · Authors · 2024-12-02
> > > > > > **Thank you for your feedback**
> > > > > >
> > > > > > Dear Reviewer "FpFr",
> > > > > >
> > > > > > Thank you for your detailed and comprehensive feedback. Below, we summarize the steps we took to address your concerns, followed by a discussion of areas where we respectfully disagree with your final requests.
> > > > > >
> > > > > > **Summary of Rebuttal**
> > > > > >
> > > > > > - Additional Model Comparisons: We went beyond your request to include two additional foundation models by incorporating five new models into our experiments.
> > > > > > - Evaluation Metrics: We added five additional evaluation metrics to ensure our evaluations were thorough and addressed your concerns about metric sufficiency.
> > > > > > - Extended Analysis: We expanded our analysis to include feature-based quantitative evaluations.
> > > > > > - Re-Evaluation of Moirai: To ensure fairness, we evaluated the Moirai model using its public version and moved re-trained results to the appendix for further discussion.
> > > > > > - Related Work: We thoroughly revised and extended the related work section, explicitly highlighting these changes in the revised manuscript.
> > > > > > - Clarification of Training Details: We clarified that all deep learning models were fine-tuned for context length.
> > > > > > - Benchmark Validation: We validated our benchmark results by comparing seven models across five frequencies of the M4 dataset, showing strong alignment with published results.
> > > > > > - Benchmark Differentiation: We listed all datasets used in the Moirai paper and included two case studies to show how our benchmark offers unique insights due to its diversity.
> > > > > >
> > > > > > **Points of Disagreement**
> > > > > >
> > > > > > - Retraining All Foundation Models with the New Pretraining Split: While we understand the importance of controlling for data leakage, retraining all foundation models with a new pretraining split is not feasible. Many foundation models lack public pretraining scripts, and retraining them all from scratch is prohibitively resource-intensive for a single entity to handle.
> > > > > > - Finding a Subset of Datasets with Equivalent Results: We explained in our rebuttal that evaluating foundation models on our benchmark is already cost-effective, which aligns with our benchmark's goal. Identifying a subset of datasets that yield equivalent results to the full benchmark is a highly challenging research problem akin to dataset distillation and falls outside the scope of our paper.
> > > > > > - Changing Prediction Length Setup: Aligning prediction lengths with standard settings would require retraining and hyperparameter tuning for 20 models across 28 datasets, which is unfortunately infeasible during the rebuttal period. Moreover while we get your point, we stand by our design choice to ensure diversity in prediction lengths and as we have validated our baseline results using the M4 dataset.
> > > > > > - Adding All Crucial Baselines: We extended our baseline count to 22 (adding five new models), we believe the request to include all foundation and deep learning models is a bit ambitious. Even leading NLP benchmarks, such as XGLUE [1] (4 baselines), Big-Bench [2] (6 baselines), MMLU [3] (9 baselines), and GPQA[4] (3 baselines)  typically report results for a smaller subset than our baseline count. We believe this should be a community effort (following NLP field), and we plan to host a public leaderboard to encourage broader participation.
> > > > > >
> > > > > >
> > > > > > We sincerely thank you for your detailed and comprehensive feedback. Your comments pushed us to strengthen various aspects of our work, and we greatly appreciate the time and effort you’ve dedicated to reviewing our submission.
> > > > > >
> > > > > > [1] XGLUE: A New Benchmark Dataset for Cross-lingual Pre-training, Understanding and Generation, https://aclanthology.org/2020.emnlp-main.484/
> > > > > >
> > > > > > [2] Beyond the Imitation Game: Quantifying and extrapolating the capabilities of language models, https://arxiv.org/pdf/2206.04615
> > > > > >
> > > > > > [3] MEASURING MASSIVE MULTITASK LANGUAGE UNDERSTANDING, https://arxiv.org/pdf/2009.03300
> > > > > >
> > > > > > [4] GPQA: A Graduate-Level Google-Proof Q&A Benchmark, https://arxiv.org/pdf/2311.12022

---

> > > > > > > ### Comment · Reviewer_FpFr · 2024-12-02
> > > > > > > **I have read the final response**
> > > > > > >
> > > > > > > Thank you for your response after seven days. While I still believe my remaining concerns are critical and propose feasible approaches, I respect the authors' perspectives not addressing any of them. Therefore, I will maintain my negative score for the current version of this work and leave the final judgment to the area chairs.

---

### Author Response · Authors · 2024-11-22
**Paper Update Summary**

We sincerely thank reviewers for their comments, which have helped us improve the paper. We have revised the manuscript and uploaded the updated PDF to the OpenReview system. Note that all changes in the paper are highlighted with red color and prepended with the capital letter R.

We also provide the anonymized code implementation alongside a subset of our dataset and sample notebooks, to allow interested reviewers to gain hands-on experience with our data: https://anonymous.4open.science/r/GIFT-Eval-1BFD/README.md.

Below, we provide a summary of the changes made:

1. Table 1: Added ProbTS as a comparison.  (Reviewer FpFr)
2. Section 2, RW: Added reference to more probabilistic + foundation models, and discussion for forecasting tools.  (Reviewer FpFr)
3. Tables 2-7 and 22-24 updated to report MASE instead of MAPE. (All Reviewers)
4. Tables 2-7, 16-24 are updated to use the public Moirai model instead of the retrained version on our pretraining data to ensure fair comparison to other foundation models.(Reviewer FpFr)
5. Section 4, Models add 3 more foundation models to the list of baselines.(Reviewer FpFr)
6. Appendix A: added clarification for context length (input length) setup.(Reviewer FpFr and Pz22)
7. Appendix B: Added Table 9 to show specific time series features of each dataset.(All Reviewers)
8. Table 14 updated to indicate past dynamic information of each test dataset in our benchmark.(Reviewer SxEF)
9. Appendix F.1, Added Table 16, further analysis of model families through the lens of time series features.(All Reviewers)
10. Tables 17 through 21 updated to report sMAPE, MASE, ND, MSE, MAE, CRPS and RANK metrics and three additional foundation models.(All Reviewers)
11. Added new appendix section F.5 and Figure 4 analyzing inter-variate correlation for multivariate datasets.(Reviewer SxEF)
12. Tables (previously 2-5 now 10-13) presenting statistics over each characteristic for test data is moved to Appendix D. (All Reviewers)

---

### Meta-Review · Area_Chair_gU9q · 2024-12-14

**Metareview:**

The paper introduces GIFT-Eval, a benchmark designed to evaluate time series forecasting models, including statistical, deep learning, and foundation models. It features 28 datasets spanning diverse domains, frequencies, and prediction lengths, alongside a non-leaking pretraining dataset to ensure data integrity. While the benchmark’s scope and extensive experiments highlight significant effort, key issues persist. The connection between dataset characteristics and forecasting performance remains unclear, and inconsistent evaluation configurations, such as varying input lengths across models, raise fairness concerns. Additionally, the analysis lacks depth, providing limited actionable insights about model performance. The role and impact of the pretraining dataset on results also remain ambiguous. Despite its potential as a valuable tool for the research community, these unresolved issues lead to a borderline rejection recommendation, with encouragement for improvement in future iterations.

**Additional Comments On Reviewer Discussion:**

During the rebuttal period, reviewers raised several critical concerns, including the unclear connection between dataset characteristics (e.g., domain, frequency) and features (e.g., trend, stability), inconsistent evaluation configurations (varying input lengths for foundation and deep learning models), and the fairness of comparisons given potential data leakage in pretraining datasets. Some reviewers also noted the lack of depth in the analysis and insufficient justification for the benchmark’s scope and dataset selection. The authors responded by clarifying the rationale for their choices, adding detailed breakdowns of time series features, and incorporating new metrics like MASE and geometric means for evaluations. They expanded the analysis by aggregating results based on time series features and characteristics, addressing some concerns about actionable insights. However, issues like inconsistent configurations and data leakage were acknowledged as limitations beyond the authors’ control. While the authors’ efforts to address reviewer feedback were appreciated, unresolved concerns about fairness and limited analysis depth led to a balanced but cautious weighing of these points, ultimately contributing to a borderline reject recommendation.

---

### Decision · Program_Chairs · 2025-01-22

Reject